# Origin and evolutionary trajectories of brown algal sex chromosomes

Josué Barrera-Redondo [1,9,11], Agnieszka P. Lipinska[1,2,11], Pengfei Liu [1], Erica Dinatale [1], Guillaume Cossard[1], Kenny Bogaert [1], Masakazu Hoshino [1,10], Rory J. Craig[1], Komlan Avia [3], Goncalo Leiria [1], Elena Avdievich[1], Daniel Liesner [1], Rémy Luthringer[1], Olivier Godfroy [2], Svenja Heesch [2], Zofia Nehr [2], Loraine Brillet-Guéguen [2,4], Akira F. Peters [5], Galice Hoarau[6], Gareth Pearson [7], Jean-Marc Aury [8], Patrick Wincker [8], France Denoeud [8,12], J. Mark Cock [2,12], Fabian B. Haas [1,12] & Susana M. Coelho [1,12] ✉

Research on the biology and evolution of sex chromosomes has primarily focused on diploid XX/XY and ZW/ZZ systems. In contrast, the rise, evolution and demise of U/V systems has remained an enigma. Here we analyse genomes of nine brown algal species with different sexual systems to determine the history of their sex determination. U/V sex chromosomes emerged between 450 and 224 million years ago, when a region containing the pivotal male-determinant *MIN* ceased recombining. Seven ancestral genes within the sex-determining region show remarkable conservation over this vast evolutionary time, although nested inversions caused expansions of the sex locus, independently in each lineage. We evaluate whether these expansions are associated with increased morphological complexity and sexual differentiation, and show that taxonomically restricted genes evolve unexpectedly often in U and V chromosomes. We also investigate two situations in which U/V-linked regions have changed. First, we demonstrate that convergent evolution of two monoicous species occurred by ancestral males acquiring U-specific genes. Second, the *Fucus* dioecious system involves new sex-determining gene(s), acting upstream of formerly V-specific genes during development. Both situations have led to the demise of U and V chromosomes and erosion of their specific genomic characteristics.

The mechanisms controlling the development of male or female identities, or co-sexuality, when individuals express both sex functions, vary widely across different organisms[1,2]. In species with separate sexes, sex chromosomes may be present, carrying a sex-determining region (SDR)[3] that encodes factors directing sex identity and which often does not undergo recombination in the heterogametic sex (XY or ZW)[4] of diploid species (dioecious), or in the diploid stage of haploid-dominant (dioicous) species. Sex chromosomes have independently evolved from autosomes multiple times and may be subject to specific evolutionary forces, including differential selection between sexes, asymmetrical expression of deleterious mutations and hemizygosity, meiotic silencing and dosage compensation[3].

Research on the biology and evolution of sex chromosomes has primarily focused on diploid XX/XY and ZW/ZZ systems in mammals, birds, fish, *Drosophila* and diploid plants[4,5]. U/V haploid sex-determination systems, such as those of bryophytes and algae[6,7], have been less explored. In U/V systems, sex is not determined at fertilization but during meiosis, when haploid spores inherit either a

U chromosome, and will develop into a female gametophyte, or a V chromosome, controlling male gametophyte formation[8]. These fundamental inheritance differences between U/V and XX/XY or ZW/ZZ systems have broad evolutionary and genomic implications[9,10]. However, so far, only the U/V systems of the brown alga *Ectocarpus* and the U/Vs of four distantly related bryophyte taxa[11–14] have been fully sequenced and assembled into chromosomes. While these studies helped understand the genomic structure of bryophyte U- and V-linked regions, the species involved diverged ~500 Ma (million years ago) and do not share homologous U/V chromosomes[15]. As a result, we still lack a broad comparative view across multiple homologous U/V systems that would inform a reconstruction of their evolutionary history. Brown algae represent exceptional models for studying sex chromosome evolution because they display diverse reproductive systems, life cycles and sex chromosome systems in a single lineage[16]. Their ancestral state probably involved separate sexes[16], suggesting that their sex chromosomes could share a common origin. Here we study the origin, evolution and demise of U/V sex chromosomes in the brown algae.

## Results

### The origin of brown algal sex chromosomes

We focused on species covering the phylogenetic, morphological and reproductive diversity of the brown algal clade[17] and their closest extant outgroup, *Schizocladia ischiensis*[17,18]. We substantially improved the brown algal genome datasets available[18–20] to reach chromosome or near-chromosome-level genome assemblies (Extended Data Table 1 and Supplementary Table 1). This revealed that brown algae have 27–33 chromosomes and largely conserved macrosynteny (Fig. 1a).

We identified the female (U) and male (V) sex-determining regions (SDRs) in the dioicous species using a combination of bioinformatic and experimental approaches (see 'Discovery of the U/V sex determination regions' in Methods; Supplementary Figs. 1–5). All U/V species share the same, albeit highly rearranged, ancestral sex chromosome, showing remarkable stability despite the large evolutionary time (Fig. 1b,c and Extended Data Table 1). The recombination suppression event leading to the birth of U/V sex chromosomes occurred after the split of *S. ischiensis* and *Dictyota dichotoma*, ~450–224 Ma[21] (Fig. 1a). The male-determining gene *MIN*[22] is the only V-specific gene consistently present in all V-SDRs of the dioicous species. We note that one dioecious (*Fucus serratus*) and two monoicous (haploid, co-sexual *Chordaria linearis* and *Desmarestia dudresnayi*) species lack U/V sex chromosomes but still retain *MIN* on a chromosome homologous to the ancestral U/V ('U/V-homologue' hereafter). The outgroup *S. ischiensis* has low synteny with the brown algae and exhibits putative fusion-with-mixing events[23] (Fig. 1a and Extended Data Fig. 1).

We next examined the U/V-SDRs by comparing male and female genome assemblies (Methods). The SDRs contain a small number of genes overall (between 18 and 52), and compared with the pseudo-autosomal regions (PARs) (between 229 and 904), with considerable variation in gene content and size across species, the smallest being found in the Ectocarpales (*Ectocarpus* sp. 7, *Ectocarpus crouaniorum*, *Scytosiphon promiscuus*; Fig. 1b,c, Extended Data Fig. 2a and Extended Data Table 1). SDR size differences across species are strongly correlated with the number of genes ($R^2 = 0.97$; Extended Data Fig. 2b) and the repeat content ($R^2 = 0.99$; Extended Data Fig. 2c) inside these regions. Many genes located in the PARs of the Ectocarpales are within the V-SDRs of *Undaria pinnatifida*, *Desmarestia herbacea* and *D. dichotoma*, indicating that the SDR boundaries have changed across species. The boundary differences coincide with extensive structural rearrangements, particularly inversions, even among closely related taxa (Fig. 1b,c and Extended Data Fig. 3). Note that the centromere in the V chromosome of *Ectocarpus* is found within the SDR[19], so we cannot exclude that a centromere-related suppression of recombination may have preceded the inversion events found on the SDR[24].

Together, our results indicate that the brown algal U/V sex chromosomes evolved between 450–224 Ma, via suppressed recombination in a genomic region that contained *MIN* (henceforth male-determining locus). The presence of *MIN* in distantly related lineages could push the age of the U/V chromosomes further back in time, but more evidence would be required to establish that dioicy existed in these organisms.

### The evolution of the SDRs involved boundary expansions and gene gains

The brown algal U- and V-SDRs carry homologous genes (gametologue pairs), indicating descent from a common ancestral region (Supplementary Table 2). *Ectocarpus* sp. 7 and *D. herbacea* show similar ratios of gametologues and U- or V-specific genes (16/14 and 11/7 gametologue/sex-specific genes in *Ectocarpus* and *D. herbacea*, respectively; Supplementary Table 2). Only ten genes share SDR orthologues between both species, while the rest were mostly acquired independently in the SDR of each species, with one gene that was retained as a gametologue pair in *Ectocarpus* sp. 7 but lost both copies in *D. herbacea* (Supplementary Table 2). Five gametologue pairs conserved both copies in the two species, while another three gametologue pairs lost either the male or the female copy in *D. herbacea* (Supplementary Table 2). In addition, *MIN* and a U-specific gene are also conserved between species (Supplementary Table 2). Although the total number of U/V-SDR genes differs between *Ectocarpus* sp. 7 (18 genes) and *D. herbacea* (30 genes), each species shows an equal number of gametologues and sex-specific genes in its U- and V-SDRs (Supplementary Table 2). This intraspecies symmetry supports the idea that the U and V chromosomes may have undergone parallel evolutionary changes within each lineage[10,25,26]. The V-SDR of *D. herbacea* contains 20 additional genes that belong to endogenous viral elements, which are common across brown algal genomes[18].

Diploid sex chromosome in animals and plants exhibit evolutionary strata representing different recombination suppression events over time. Strata are identified by analysing synonymous substitutions ($K_s$) between male/female gametologue pairs[27] whose locations in fully X or Z-linked regions are known. However, detecting evolutionary strata in U/V systems is difficult because neither of these fully sex-linked regions recombines and gene movements and chromosome rearrangements disrupt collinearity of both chromosomes between species[25,28,29]. Moreover, in the absence of a recombining outgroup (which does not exist in brown algae), the ancestral gene order cannot be reliably inferred. In both *Ectocarpus* sp. 7 and *D. herbacea*, the V- and U-SDR rearrangements differ by inversions (Fig. 2a,b and Extended Data Fig. 4), consistent with the idea that inversions may lead to suppressed recombination between sex chromosomes. An analysis of gametologue pair divergence revealed saturated levels of $K_s$ values (Fig. 2b,c and Supplementary Table 3), further limiting the inference of evolutionary strata across brown algal SDRs. Nonetheless, the gametologue $K_s$ values are broadly consistent between orthologues in *Ectocarpus* sp. 7 and *D. herbacea*, where shared SDR gametologues between species have higher $K_s$ values and probably spent more evolutionary time diverging than the gametologues that are not shared between species (Supplementary Table 3). Furthermore, the location of gametologues with the lowest $K_s$ values in the U-SDR of *D. herbacea*, relative to the PAR genes in *Ectocarpus* sp. 7, suggests that inversions involving the entire U-SDR and adjacent PAR segments probably contributed to the expansion of the U/V-SDR boundaries in *D. herbacea*, in a process we term 'engulfment' (Extended Data Fig. 4). The expansion of the SDR boundaries in *D. herbacea* led to the engulfment of a region containing four genes in the PAR1 of *Ectocarpus* sp. 7, and a second region with 13 genes located on the PAR2 (Fig. 2d, Supplementary Table 4 and Extended Data Fig. 4). Twelve of these engulfed genes into the SDR of *D. herbacea* were retained as gametologues. These observations support a scenario where expansions in the SDR boundaries of brown algae

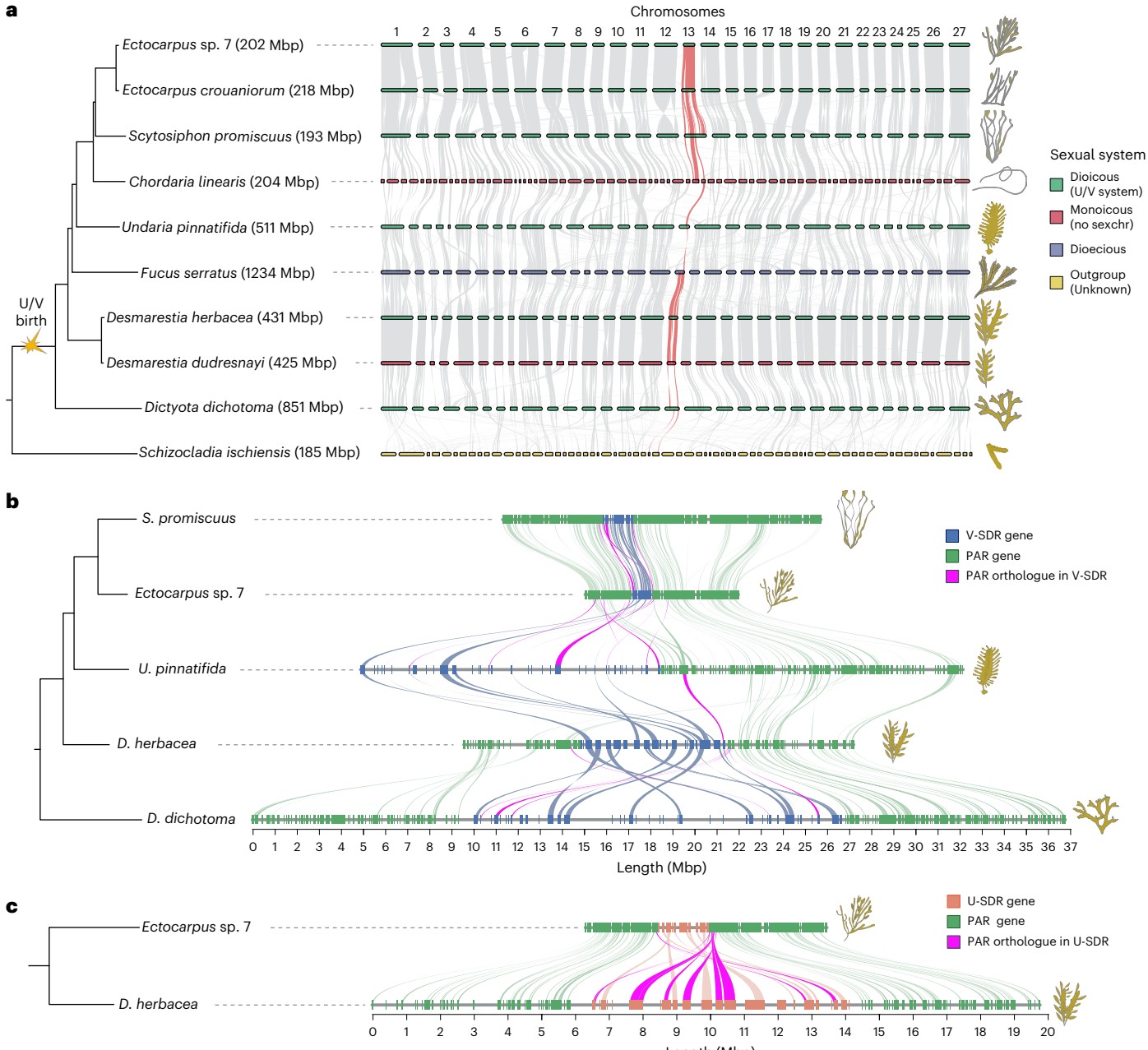

**Fig. 1 | Origins of U/V sex chromosomes in brown algae. a**, Macrosynteny plot comparing genomes of six dioicous (green), two monoicous (red), one dioecious (blue) and one outgroup species (yellow). The chromosomes were originally numbered by their physical size in the *Ectocarpus* v2 genome[141]. Note that the dioecious species *F. serratus* has a fully diploid life cycle (without gametophytes[7]). Syntenic blocks of the V sex chromosome are highlighted in red, with the emergence of U/V chromosomes shown in the phylogeny. Genome sizes are indicated in brackets. **b**, Microsynteny plot of V chromosomes in five dioicous species, highlighting the male sex-determining regions (blue) and the PARs (green). The PAR genes whose orthologues are found within the SDR of other species are highlighted in purple. **c**, Microsynteny plot of U chromosomes in two dioicous species, highlighting the female sex-determining regions (peach) and the PARs (green). The PAR genes whose orthologues are found within the SDR of other species are highlighted in purple. Note that the genome assemblies for *C. linearis* and *S. ischiensis* are not chromosome level, leading to a high number of contigs.

occur through nested inversions. Two chromatin-related transcription factors in the *Ectocarpus* PARs were independently incorporated into the SDRs of four other dioicous species (Supplementary Tables 4 and 5).

The observation of greater V-SDR gene content in early diverging lineages (such as *D. dichotoma*) than in the later-diverging Ectocarpales (Extended Data Table 1) could reflect either gene loss in the V-SDRs of Ectocarpales or independent gene gains in the V-SDRs of each lineage (as predicted in ref. 10), from an ancestral state with low V-SDR gene content that is retained in Ectocarpales. To distinguish between these possibilities, we reconstructed the ancestral

SDR gene content (Supplementary Table 5), focusing on the V chromosome, as the genomic data are of better quality (Supplementary Table 1), and assuming parallel U/V-SDR evolution[10,25,26] as seen in *Ectocarpus* sp. 7 and *D. herbacea* (Fig. 2d and Supplementary Table 2). This analysis revealed that brown algal V-SDR evolution occurred via lineage-specific gene gains rather than gene loss in the Ectocarpales (Fig. 2e,f). Gene gains were caused by a combination of three processes: expansions of the SDR boundaries into the PARs, translocation of autosomal genes into the SDR and lineage-specific gene birth events within the SDR (Fig. 2e and Supplementary Table 5). Consistently, ancestral

V-SDR genes were associated with higher gametologue $K_s$ values, while independently acquired gametologues in *D. herbacea* had lower $K_s$ values (Supplementary Table 2 and Fig. 2c).

The seven genes in the ancestral V-SDR (Fig. 2f and Extended Data Fig. 2d) include the male-determinant *MIN*[22] and six V gametologues of genes that are also carried on the U chromosome (Fig. 2f and Supplementary Table 6). As predicted by early models of U/V-SDR evolution[10], all seven genes are probably related to sex determination processes. Gametologue pairs include putative transmembrane proteins that may play a role in gamete recognition[30], STE20 serine/threonine kinase gametologues probably involved in pheromone pathways[31], and a casein kinase, a MEMO-like domain protein and a GTPase-activating protein which may act in signal transduction (Supplementary Table 6). All of these genes are gametologue pairs in *Ectocarpus* sp. 7, but in *D. herbacea* the casein kinase was lost from the U-SDR and the putative transmembrane receptor was lost in both sexes. We noticed that these ancestral V-SDR genes remain in the U/V-homologue of the species that have lost their U/V system (Fig. 2f and Supplementary Table 6), emphasizing their importance for pathways in sex organ development even in the absence of sex chromosomes.

The V-SDR size appears to be associated with the level of sexual dimorphism (Fig. 2e and Extended Data Fig. 2), but the small sample size is insufficient for formal statistical analysis. Species with low sexual dimorphism (anisogamous) retained the ancestral V-SDR genes with very few gene gains, further suggesting that they may represent the V-SDR ancestral state. Although the number of SDR changes is small, oogamous species each independently gained diverse V-SDR genes, and one gene (ATP-dependent RNA helicase) was convergently acquired in all (OG0003211 in Extended Data Fig. 2d). All the detected autosomal translocations into the V-SDRs of *Ectocarpus* sp. 7 and *D. herbacea* (Fig. 2e) also involve sex-specific genes (Supplementary Tables 2 and 5), consistent with a model where sexual antagonism in autosomal loci may be solved by gaining sex linkage[32]. In contrast, we found no correlation between autosomal sex-biased gene (SBG) expression and sexual dimorphism level (false discovery rate (FDR)-corrected $P > 0.01$; Supplementary Table 7), supporting previous studies[33] (Extended Data Fig. 5a). However, we observed an enrichment of male-biased genes on the PARs in all species (chi-square test $P < 0.01$) except *D. dichotoma* (Extended Data Fig. 5b).

Most U/V-SDR genes were prominently expressed in fertile haploid gametophytes, consistent with gene preservation via haploid purifying selection (Supplementary Table 8). Gametologues had typically higher expression levels than sex-specific genes (present in only one of the SDRs) (Wilcoxon test, $P = 0.00075$ in *D. herbacea*; $P = 0.08843$ in *Ectocarpus* sp. 7) (Fig. 2d). A comparative analysis in fertile gametophytes between SDR genes and their autosomal counterparts in other species showed that newly acquired genes on the SDR had similar expression levels to their autosomal counterparts (Extended Data Fig. 6), suggesting either a co-option of autosomal biological activity into male-specific functions in the V-SDR or the general importance of these genes for gametophyte development. Examining expression levels across multiple tissues in *Ectocarpus* sp. 7 revealed that activity of U/V-SDR genes is not confined to fertile gametophytes (Fig. 2d).

Therefore, the SDRs contain not only genes involved in sex determination and gametophyte fertility but also genes playing a broader role in development.

Altogether, our analyses illustrate how brown algal U/V-SDRs undergo structural changes, evolving mainly by lineage-specific gene gains associated with increasing levels of sexual dimorphism. We identified a set of conservatively sex-linked genes in dioicous brown algae, suggesting their role in sex determination and/or differentiation, along with genes potentially involved in other developmental pathways.

## Structural features and evolutionary dynamics of brown algal U/V sex chromosomes

We next examined the structural features that differentiate the entire U/V sex chromosomes (V-SDR and PARs) from the rest of the genome (Fig. 3a and Supplementary Fig. 12a). As expected for non-recombining regions[25,34], all V sex chromosomes are repeat rich and gene poor (Wilcoxon rank-sum test, FDR-corrected $P < 0.01$; Fig. 3a, Extended Data Fig. 7 and Supplementary Tables 9–11). V-SDRs have significantly higher repeat density than the PARs or the autosomes (permutation test, FDR-corrected $P < 0.001$; Extended Data Fig. 8). This low gene density is not influenced by the presence of centromeres within the SDRs, as the coding density in the *Ectocarpus* sp. 7 V centromere (3.51%) is slightly higher than in the rest of the V-SDR (2.85%), presumably due to the small size of the centromere (153 kbp)[19]. The PARs were also significantly enriched in repeats when compared with the autosomes, although less so than the V-SDRs (permutation test, FDR-corrected $P < 0.001$; Extended Data Fig. 8). Among repetitive elements, 'unclassified' transposable elements (TEs) were enriched in the PARs and SDRs of the Ectocarpales (permutation test, FDR-corrected $P < 0.01$), while the V-SDRs of species that underwent genome expansion (for example, *U. pinnatifida*, *D. herbacea*, *D. dichotoma*) predominantly accumulated long terminal repeat (LTR) elements (Supplementary Fig. 6 and Supplementary Table 9).

Moreover, sex chromosomes had fewer orthologues conserved between species compared with the autosomes (chi-square test, $P < 10^{-4}$, Supplementary Table 12), possibly reflecting increased numbers of taxonomically restricted genes (TRGs; that is, genes that are not detectable outside of a defined taxonomic group). Phylostratigraphy analyses[35,36] confirmed an enrichment of TRGs in the sex chromosomes of all dioicous species (Wilcoxon rank-sum test, FDR-corrected $P < 0.01$; Fig. 3b and Supplementary Tables 13 and 14). TRG enrichment was localized in the PARs of the Ectocarpales, but this pattern extended to the entire sex chromosome, including the SDRs, in species with larger V-SDRs (permutation test, FDR-corrected $P < 0.001$; Fig. 3b). Importantly, sex chromosomes have statistically younger TRGs than the last common ancestor of the five dioicous species (same Order or broader taxonomic groups), indicating that TRG enrichment arose independently in each species (Pearson standardized residuals >2.4; Supplementary Figs. 7–11).

We previously proposed a theoretical model where generation-antagonistic selection may favour the retention of young sporophyte-beneficial loci in the PARs of *Ectocarpus* sp. 7 U/Vs[37]. Consistent with

**Fig. 2 | Lineage-specific U/V-SDR expansion from an ancestral SDR and its association with sexual dimorphism. a**, Microsynteny plot between the U and V chromosomes of *Ectocarpus* sp. 7 and *D. herbacea*. **b**, Synteny between the U and V gametologues within the V-SDRs of both species, coloured by synonymous substitutions per site ($K_s$). **c**, Identification of ancestral SDR gametologues (red squares) and independently acquired gametologues (black dots) with respect to their gametologue $K_s$ values and their position in the V-SDR. **d**, Circos plot linking gametologue pairs in each species with expression levels of all SDR genes ($\log_2(\text{TPM} + 1)$) across different life stages in *Ectocarpus* sp.7 and mature gametophytes (matGA) in *D. herbacea*. Gametologues are highlighted in dark colours, sex-specific genes are highlighted in light colours, viral insertions

are marked in grey and asterisks denote conserved SDR genes (also in **f**).
**e**, The ancestral state reconstruction of V-SDR gene content across brown algae, showing the expected number of genes in the SDR (white circles), gene retention (blue numbers), gene gain through expansion of the SDR boundaries (orange), gene gain through autosomal translocation (purple), gene birth event inside the SDR (yellow) and gene loss (red) along with changes in gamete dimorphism[16].
**f**, Schematic of the seven ancestral V-SDR gene orthogroups (OGs), with genomic locations marked: retained in the V-SDR (blue), found in the U/V-homologue of non-dioicous species (green), translocated from the V-SDR to an autosome (yellow), present in a non-scaffolded contig (red), and lost (grey). Bold: *MIN*. See Supplementary Tables 5 and 6.

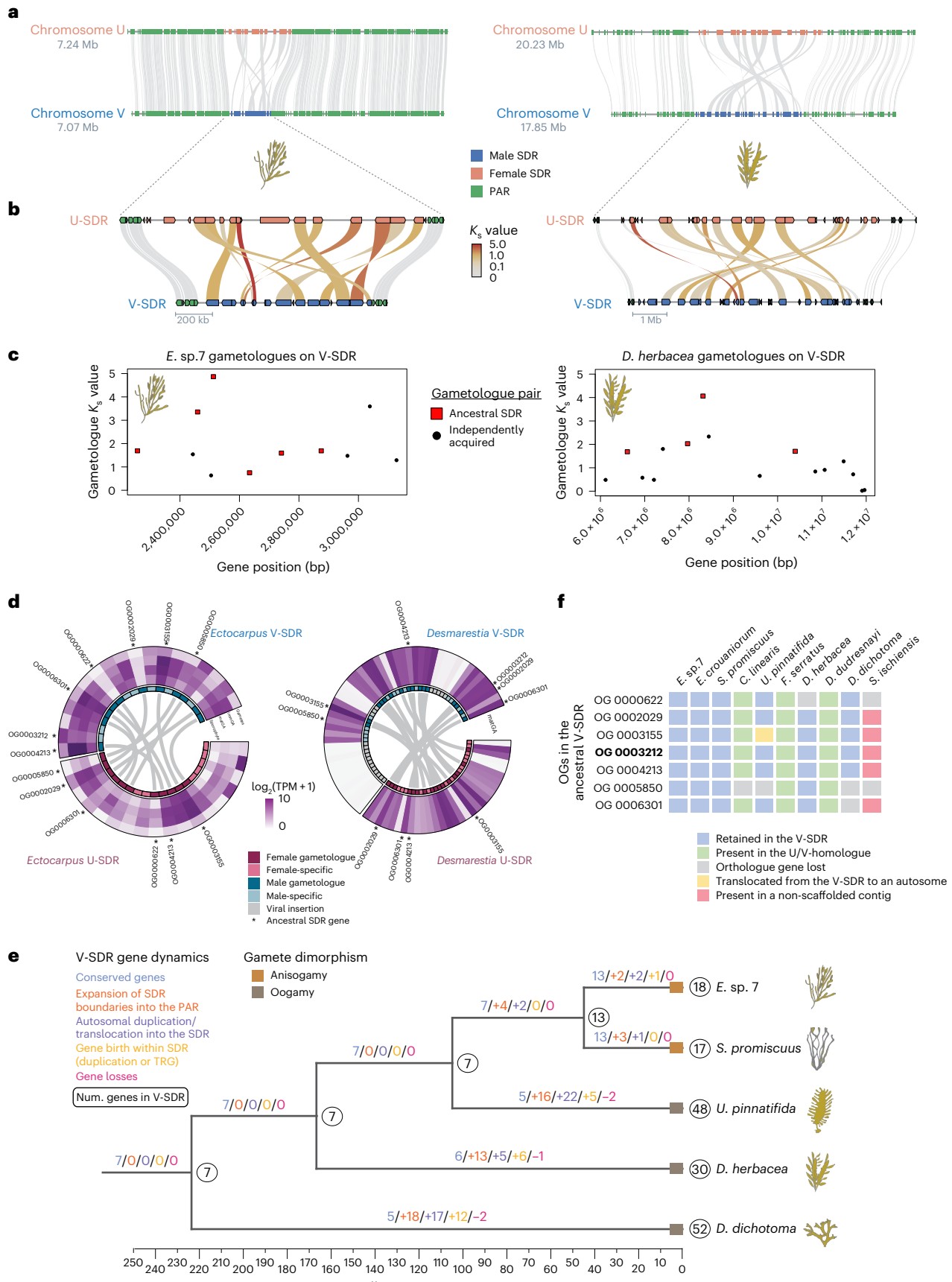

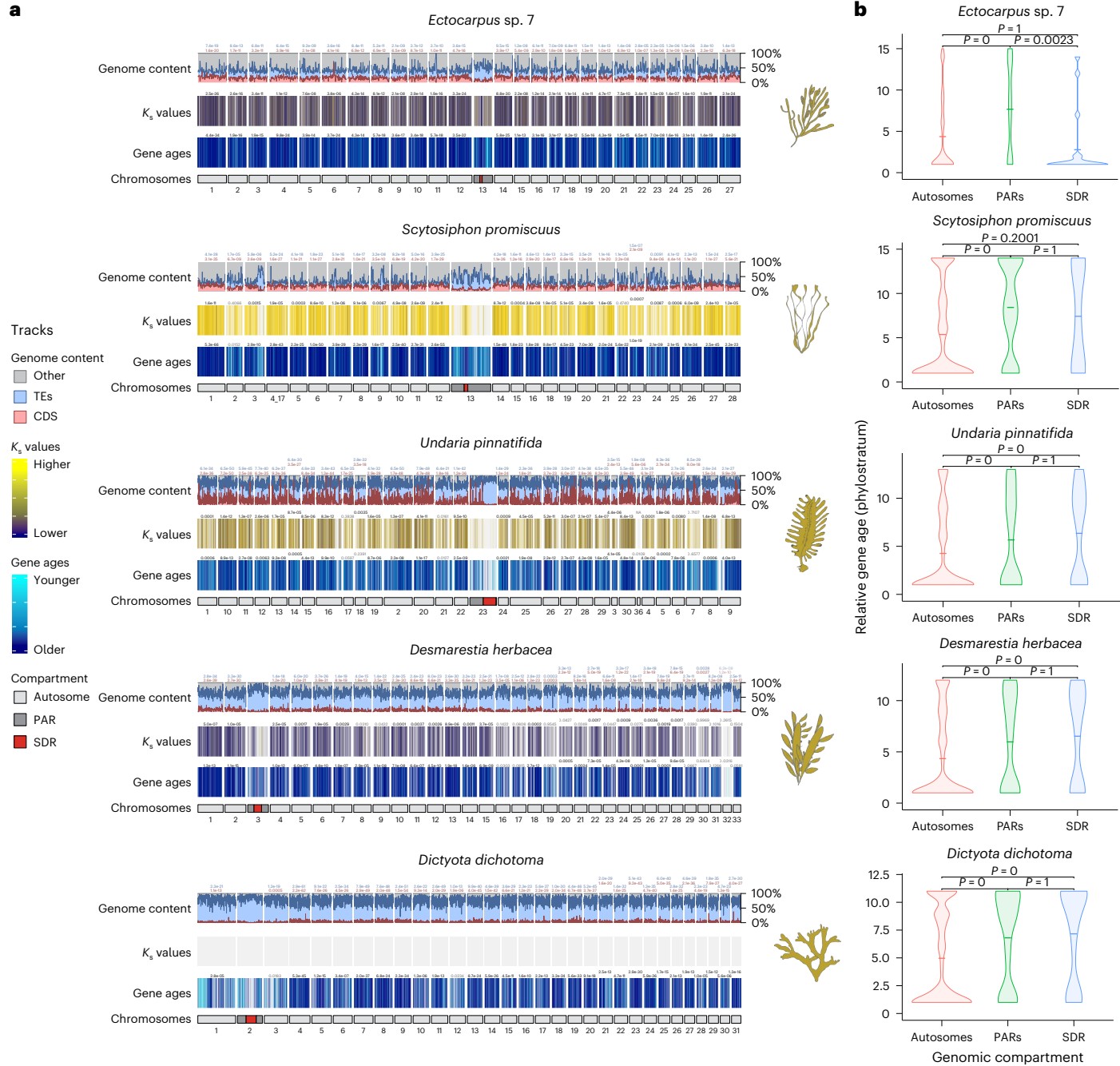

**Fig. 3 | The U/V sex chromosomes are enriched in taxonomically restricted genes. a**, Karyoplots for five dioicous species showing the following features from bottom to top: chromosome compartments (autosomes, PARs and SDR), relative gene ages, interspecies $K_s$ values, and proportion of coding (CDS, red) and repeat (TEs, blue) density. Statistically significant differences for each feature between each autosome and the V chromosome are depicted on top of the track for that autosome (FDR-corrected two-sided Wilcoxon rank-sum test; values indicated with solid colours when $P < 0.01$ for the tested hypothesis). The precise range of gene age categories and interspecies $K_s$ values for each species can be found in Supplementary Figs. 7–11. **b**, Violin plots for five dioicous species showing the relative gene age ranks (higher ranks equate to younger ages) of the TRGs across chromosome compartments (autosomes, PARs and SDR). Statistically significant differences in mean values of gene ages (centre line) were assessed using FDR-corrected two-sided permutation tests.

this model, sporophyte-biased genes are indeed enriched in the sex chromosomes of *Ectocarpus* sp. 7 and *U. pinnatifida* (fold change >2, adjusted $P < 0.05$), but less so in *D. dichotoma* and *S. promiscuus* (Supplementary Table 15). Moreover, we explored additional mechanisms underlying TRG emergence by estimating interspecies $K_s$ values between orthologues in closely related species (Supplementary Table 16) and comparing these values between chromosomes and genomic compartments (V-SDR, PAR, autosomes). If synonymous mutations behave neutrally[38,39], then interspecies $K_s$ can be used as a proxy for mutation rates[40,41]. Consistently, we found higher interspecies $K_s$ values in the V sex chromosomes compared with autosomes across all dioicous species (Wilcoxon rank-sum test, FDR-corrected $P < 0.01$; Fig. 3a and Supplementary Tables 16 and 17), suggesting higher mutation rates relative to autosomes. Higher interspecies $K_s$ values are also localized in the PARs, mirroring the pattern observed with the TRGs (Supplementary Figs. 7–10). Therefore, the enrichment of TRGs in the U and V is associated with both enrichment of sporophyte-biased genes and higher synonymous substitution rates.

To test the generality of the pattern of TRG enrichment on U and V chromosomes, we applied the same approach in other organisms with haploid sex determination, the plants *Ceratodon purpureum*, *Sphagnum angustifolium*, *Marchantia polymorpha*[11,12,42] and the fungus *Cryptococcus neoformans*[43]. We observed a clear enrichment of TRGs in the V chromosomes of *C. purpureum* and *S. angustifolium* (Wilcoxon rank-sum test, FDR-corrected $P < 0.01$; Supplementary Figs. 12 and 13 and Supplementary Tables 13 and 14), but not in the U/V chromosomes of *M. polymorpha* or the mating-type chromosome of *C. neoformans* (Supplementary Figs. 14 and 15 and Supplementary Tables 13 and 14).

### Fate of U/V sex chromosomes following loss of dioicy

We studied the evolutionary trajectory of brown algal genomes after the loss of the U/V system, by exploring two independent transitions to monoicy in *C. linearis* and *D. dudresnayi* that undergo sexual reproduction and develop male and female gametangia[33]. Most genes in the 'ex'-sex chromosomes (U/V-homologues) of both monoicous species are male derived, indicating that monoicy emerged from a male background (Fig. 4a,b). The U/V-homologue of *C. linearis* contains several rearrangements spanning the regions that are homologous to the PAR and SDR (SDR-homologue), with 11 V-SDR-derived and 2 U-SDR-derived orthologues located within the SDR-homologue, with an additional V-SDR-derived gene that was translocated elsewhere in the U/V-homologue (Fig. 4a and Supplementary Table 18). Likewise, *D. dudresnayi* underwent at least two inversion events within the SDR-homologue after splitting from *D. herbacea* (Fig. 4b), containing 20 V-SDR-derived genes and 4 U-SDR-derived genes (Supplementary Table 19).

Both monoicous species retained mostly male and a few female copies for most of the U/V-SDR-derived gametologues (91% in *C. linearis* and 100% in *D. dudresnayi*), whereas several U- and V-specific orthologues were lost in these species (10 and 17 sex-specific genes in *C. linearis* and *D. dudresnayi*, respectively). Of these lost orthologues, 60% and 43% present closely related autosomal paralogues in *C. linearis* and *D. dudresnayi*, respectively (Supplementary Tables 18 and 19), although it is unclear whether the expression of these autosomal paralogues is compensating the activity of the lost genes. The only three U-SDR-derived orthologues in *C. linearis* are flanked by PAR orthologues translocated at the end of the V-SDR-derived region, suggesting that the U/V-homologue (contig 12) of *C. linearis* acquired its U-SDR-derived genes through two translocations (Extended Data Fig. 9). Three U-SDR-derived genes in *D. dudresnayi* are dispersed across the V-SDR-derived region of the U/V-homologue, suggesting independent U-SDR translocations into the V-SDR, while the fourth U-SDR-derived gene was translocated to an autosome (Extended Data Fig. 9).

The seven ancestral V-SDR genes are transcriptionally active ($\log_2(\text{TPM} + 1) > 2$) during reproductive stages of both monoicous species (Supplementary Table 20), emphasizing their role in reproduction despite the absence of a U/V system, particularly *MIN*[22] which is retained in both species. While most U-SDR-derived genes are absent in monoicous species, a single intracellular cholesterol transporter gene was convergently preserved in both monoicous genomes (Supplementary Table 20) and actively expressed during fertility in both *Ectocarpus* sp. 7 and *D. herbacea* (Supplementary Table 8).

The U/V-homologue of *D. dudresnayi* retains some vestiges of its past as a U/V chromosome, such as low coding density, high repeat density and enrichment of TRGs (Wilcoxon rank-sum test, FDR-corrected $P < 0.01$), although we found non-significant differences in interspecies $K_s$ values across the genome (Fig. 4c, Extended Data Fig. 7, Supplementary Fig. 16 and Supplementary Tables 9–14, 16 and 17).

Finally, we examined the transition from haploid to diploid sex determination, which has remained unstudied in eukaryotes. Although the ancestral state for the brown algae is a U/V sexual system, the Fucales recently transitioned to a diploid life cycle[44], with many species, such as *F. serratus*[45], exhibiting diploid separate sexes (dioecy)[46]. Dioecy probably evolved from monoecy (both sexes in the same diploid individual) in the last common ancestor of the *Fucus* genus[47] (25–5 Ma[18]), consistent with a young sex chromosome in *F. serratus*. Our extensive bioinformatic analysis and PCR sex-linkage testing for candidate genes such as *MIN* (Methods) failed to identify sex-linked sequences in *F. serratus* (Supplementary Fig. 17), suggesting that the SDR is small and undifferentiated. However, male *F. serratus* conserves the *MIN* gene and all the ancestral V-SDR genes in its U/V-homologue (Figs. 2f and 5a). Importantly, although none of the U/V-SDR-derived genes are sex-linked in *F. serratus*, *MIN* and four other ancestral V-SDR genes are exclusively expressed in males (fully silenced in females) (fold change (FC) > 2, $p_{adj} < 0.05$; Fig. 5a and Supplementary Table 21). This pattern is consistent in three other Fucales species (Supplementary Table 22). Therefore, the ancestral V-SDR genes probably still play roles in male sex determination or differentiation pathways.

Contrary to the observations in *D. dudresnayi*, the U/V-homologue of *F. serratus* lacks the TRG enrichment pattern and all the other distinctive features of the U/V chromosomes (Wilcoxon rank-sum test, FDR-corrected $P > 0.01$). Thus, this 'ex'-sex chromosome has lost all the evolutionary vestiges of its past as a U/V chromosome (Fig. 5b, Extended Data Fig. 7, Supplementary Fig. 18 and Supplementary Tables 9–14, 16 and 17).

## Discussion

Here we characterized the evolutionary trajectory of brown algal sex chromosomes (Fig. 6). Brown algal sex chromosomes date back 450–244 Ma[21], at the origin of brown algae. We propose that the male-determining gene *MIN* underlies the birth of this U/V system[22]. The ancestral cassette with seven V-SDR genes suggests a very early evolution of these genes into a non-recombining locus during the evolution of the U/V-SDRs. The ancestral V-SDR genes probably contribute to reproduction, but may also be involved in broader developmental functions. Despite their old age, brown algal U/V chromosomes retain large PARs bordering the SDR, unlike haploid systems in non-vascular plants that mostly lack detectable PARs[11–14].

Brown algal genomes have a high degree of synteny conservation. The U/V-SDR is however prone to accumulate structural rearrangements, including inversions that may have caused the initial recombination suppression event in proto-sex chromosomes and the later expansion of the U/V-SDRs into the PARs. Similar to other haploid systems[48,49], TEs conspicuously accumulated in the SDRs following recombination suppression[50], possibly causing further rearrangements through TE-mediated inversions[51].

**Fig. 4 | Fate of sex chromosomes during transitions from dioicy to co-sexuality (monoicy). a**, Comparison of the U/V-homologue in *C. linearis* against the U and V chromosomes of *Ectocarpus* sp. 7. **b**, Comparison of the U/V-homologue in *D. dudresnayi* against the U and V chromosomes of *D. herbacea*. The colour code represents the identity of the genes alongside the chromosomes, while the shapes represent the evolutionary fate of each SDR gene in the monoicous genome. The matching shades between the SDRs and the U/V-homologue are either colour coded by their ancestral background or they appear as transparent dotted shades if the gametologue of the other sex was retained. **c**, Karyoplot of *D. dudresnayi* showing the following features from bottom to top: chromosome compartment (autosomes and U/V-homologue), relative gene ages, interspecies $K_s$ values, proportion of coding (CDS, red) and repeat (TEs, blue) density. Statistically significant differences for each feature between each autosome and the U/V-homologue are depicted on top of the track for that autosome (FDR-corrected two-sided Wilcoxon rank-sum test; values indicated with solid colours when $P < 0.01$ for the tested hypothesis). **d**, Violin plot showing the relative gene age ranks (higher ranks equate to younger ages) of the TRGs between the autosomes and the U/V-homologue of *D. dudresnayi*. Statistically significant difference in mean values of gene ages (centre line) was assessed using an FDR-corrected two-sided permutation test.

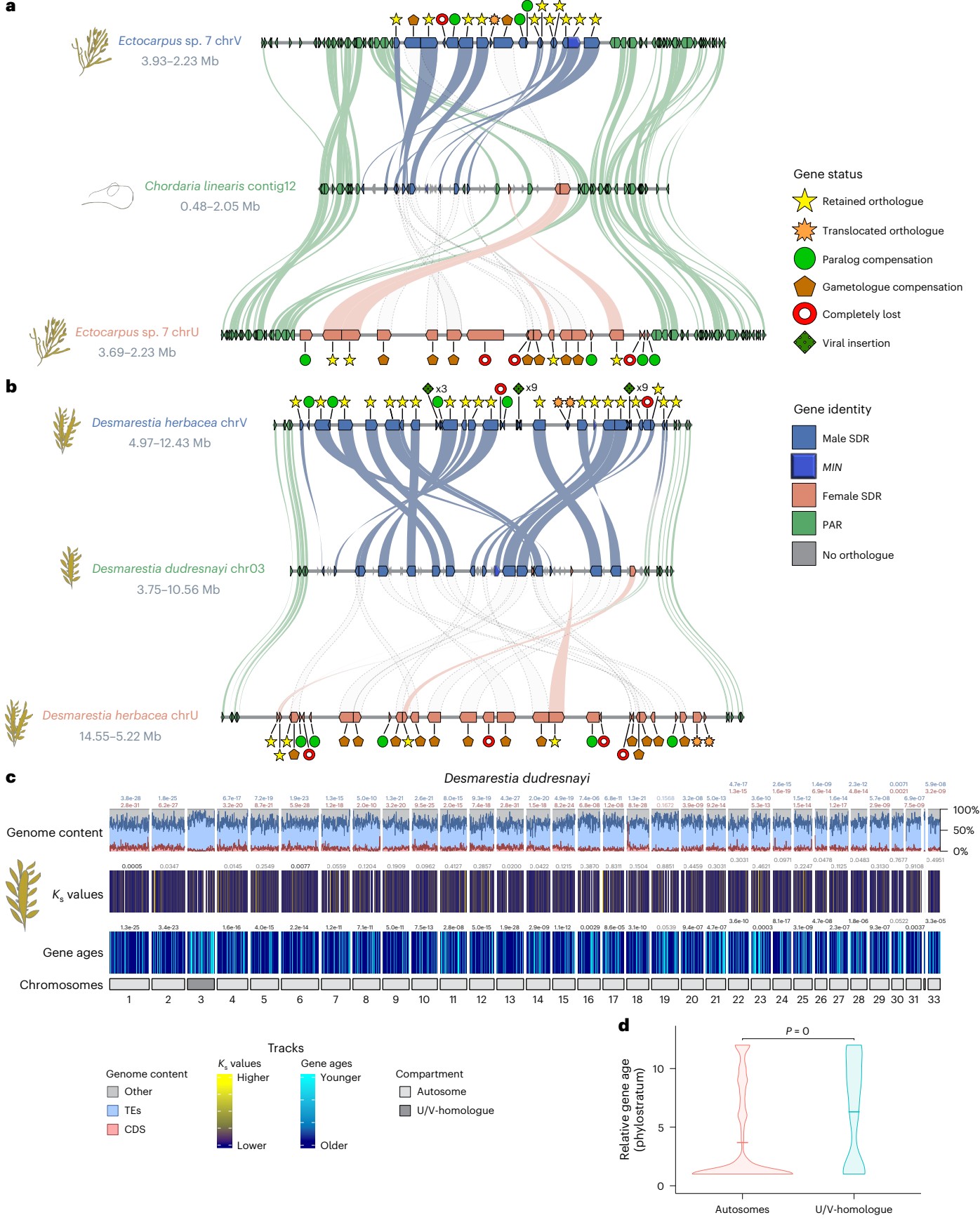

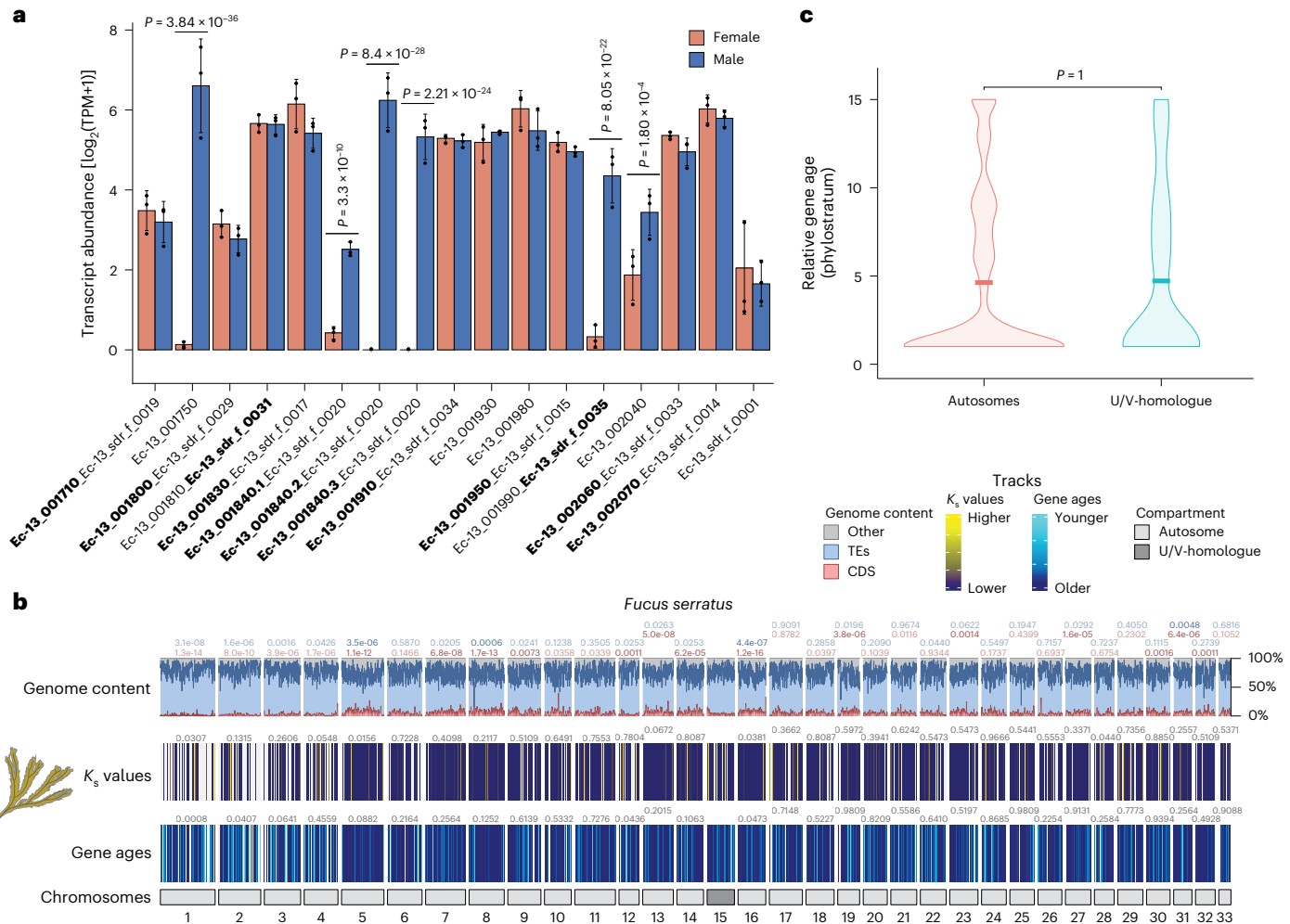

**Fig. 5 | Transition from haploid to diploid sex determination. a**, Expression of ancestral U/V-SDR genes in the diplontic species *F. serratus*. Gene expression of mature algae (using 3 males and 3 females, see Methods) is given as log₂(TPM + 1) and bars represent standard deviation of the mean. *P* values shown on the plot are derived from differential expression analysis performed using DESeq2, which applies a two-sided Wald test with Benjamini–Hochberg correction for multiple testing. Bold text represents whether the gene in *F. serratus* corresponds to an ancestral male or the female gametologue. **b**, Karyoplot of *F. serratus* showing the following features from bottom to top: chromosome compartment (autosomes and U/V-homologue), relative gene ages, interspecies *K*ₛ values (between

0.00079 and 6.838, with an average value of 0.148), proportion of coding (CDS, red) and repeat (TEs, blue) density. Statistically significant differences for each feature between each autosome and the U/V-homologue are depicted on top of the track for that autosome (FDR-corrected two-sided Wilcoxon rank-sum test; values indicated with solid colours when *P* < 0.01 for the tested hypothesis). **c**, Violin plot showing the relative gene age ranks (higher ranks equate to younger ages) of the TRGs between the autosomes and the U/V-homologue of *F. serratus*. The mean values of gene ages (centre line) are not significantly different (FDR-corrected two-sided permutation test).

Models of XX/XY and ZW/ZZ sex chromosome evolution suggest that sexually antagonistic selection may lead to SDR expansions, making former PAR genes fully sex linked and gaining new genes[52,53]. We show that the three brown algal species with greater sexual dimorphism in gametes (oogamy) indeed have enlarged U/V-SDR gene contents, although statistical testing is not possible given the small sample size. Oogamy is proposed to be ancestral in the brown algae[16,21], but this trait seems to be highly labile and our analyses suggest multiple independent transitions to oogamy from a less dimorphic ancestor with a small ancestral SDR, accompanied by SDR gene gain. Anisogamous species only experienced few gene gains in their SDRs, as predicted in ref. 10. Similar to *M. polymorpha*, the relatively simple gene content of the U/V-SDRs in the brown algae could regulate an autosomal effector gene network controlling sexual dimorphism, including differences in somatic development between male and female gametophytes[54]. Accordingly, we observed substantial sex-biased expression of autosomal genes between mature male and female gametophytes carrying sex organs. This contrasts with *C. purpureus*,

where sex chromosomes carry thousands of genes and few sex-biased autosomal genes[11]. Consistent with other studies[33,55], we found no correlation between levels of sex-biased expression and sexual dimorphism in brown algae. Sexual dimorphism in this lineage may be controlled by a relatively small subset of genes, and most sex-biased genes may influence gametophyte physiology or vegetative development.

In diploid sexual systems, recombination suppression between sex chromosomes explains Y and W degeneration[56] (but see refs. 57–59). Differences in the evolution of U/V systems are predicted, since sex determination occurs in the haploid stage, in which deleterious mutations are more efficiently removed by selection than in diploid systems[29,60] and some of the predicted differences have been documented in bryophytes[29]. In brown algae, as in diploid systems, the SDRs lost recombination, leading to TE accumulation and consequent reduction in gene density. However, unlike the degeneration documented in some XX/XY and ZW/ZZ systems, our ancestral state reconstruction revealed more gene gains than losses or gene movements out of the V-SDR during evolution of brown algal lineages.

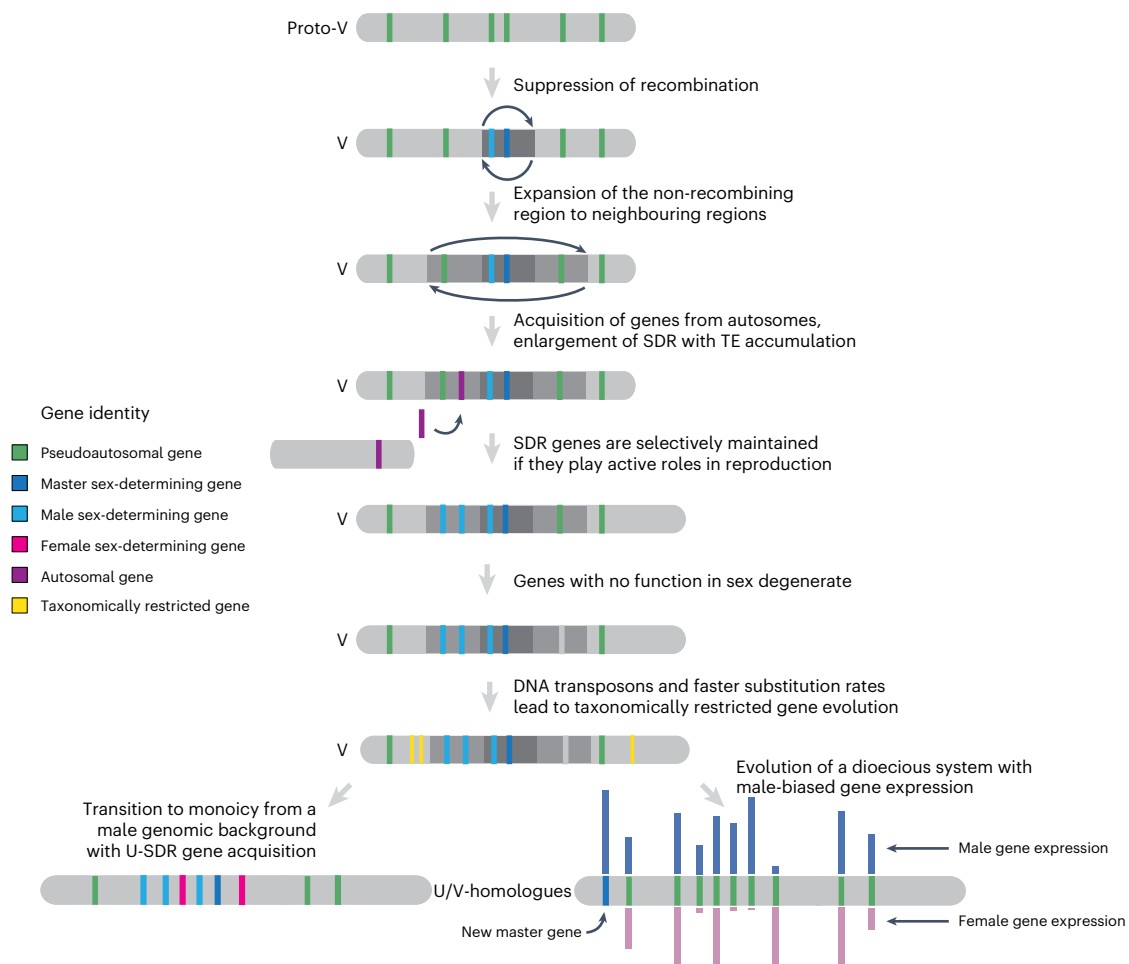

**Fig. 6 | Hypothetical model for U/V sex chromosome evolution.**
U/V sex chromosomes arose from an ancestral autosome, via suppression of recombination that probably occurred via an inversion. The SDR boundaries expanded into neighbouring PAR via inversions, but also by recruitment of genes from autosomes; expansion occurred in a lineage-specific fashion, concomitant with increased sexual dimorphism of the different species. SDR genes are maintained within the SDR if they have roles in sex, whereas genes with no role in sex are lost. Faster substitution rates, probably a consequence of the heterochromatic context of the sex chromosome, may promote the rise of taxonomically restricted genes, which are selectively maintained on the sex chromosome if they have advantages to the sporophyte generation. In species that switch to a diploid life cycle, the U/V system disappears, but the genes that are in the V-specific region retain roles in sex, although they are no longer masters. Transition from U/V separate sexes to co-sexuality (monoicy) occurred when a male haploid individual acquired female-specific genes via translocations. During the demise of the U/V sex chromosomes, their structural and evolutionary footprints slowly disappear.

Unclassified repeats accumulate in the Ectocarpales sex chromosomes, including the PARs, but not in species with larger genomes, where LTR retroelements become dominant. DNA transposons are over-represented among unclassified repeats[61], and they often insert near the progenitor locus in a process called local hopping[62]. We propose that the U/V-SDR may thus act as a source of DNA transposons that hop to the PARs, thus increasing their repeat density, whereas increased colonization of LTR elements obscures this pattern in larger genomes.

Brown algal U/V chromosomes display an excess of TRGs. What mechanisms underlie this pattern? The SDR and the PARs of U/V chromosomes are enriched in heterochromatin[63], involved in repressing TEs[64], and heterochromatic regions tend to have higher mutation rates due to reduced access of the DNA repair machinery during replication[65]. Accordingly, we consistently observe higher interspecies $K_s$ values in U/Vs, particularly in the PARs, since these regions recombine between sexes and thus have a higher rate of neutral fixation than the SDR, which experiences the Hill–Robertson effect that reduces fixation probabilities of neutral mutations due to its linkage with other sites under selection[66]. We speculate that this feature could facilitate the evolution of TRGs. Alternatively, the high density of DNA transposons within the U/V could also promote the co-option of their regulatory motifs and enable de novo transcript birth, as seen in *Drosophila*[67]. Note that the pattern could be reinforced through generation-antagonistic selection[37], but DNA transposons and higher mutation rates may be sufficient to initiate this pattern in species lacking sporophyte-biased gene expression. Importantly, the TRG enrichment is unique to U/V systems and gradually disappears when these systems are lost. This pattern extends beyond brown algae to other eukaryotes with U/V systems, such as *C. purpureus*[11,68] and *S. angustifolium*[12]. However, we could not detect enrichment of TRGs on the small U/V sex chromosomes of *M. polymorpha*[42,69], or on the mating-type chromosome of *C. neoformans*[43]. Mosses such as *C. purpureus* and *S. angustifolium* display relatively more complex sporophyte body plans than liverworts such as *M. polymorpha*[70], which could underlie stronger generation-antagonistic selection[37]. Unlike brown algae, most of the sex chromosomes in bryophytes are sex linked, with minimal space for PARs[29]. In this context, the sex chromosome of *C. purpureus* expanded its SDR very recently in evolutionary time through fusions of autosomes with earlier established U and V chromosomes[71], retaining its evolutionary footprints as PARs,

while the sex-linked region in *M. polymorpha* is much older[69], which could also limit the formation of TRGs that are predominantly observed in the PARs of the brown algae. The pattern thus appears to be specific to U/V systems where chromosomal degeneration is mild and linked to haploid–diploid life cycles where the sporophyte stage is sufficiently complex, highlighting a key role for generation selection[37]. Therefore, our study hints at a unique interplay between complex life cycles, heterochromatic landscape, DNA transposons and higher mutation rates that may lead to TRG enrichment in U/V chromosomes, and this process is pervasive across distant, independently evolved eukaryotic kingdoms.

Monoicous brown algae have transcriptomic profiles resembling ancestral females[33]. However, our results show that monoicy arose at least twice from a male ancestor that acquired female genes. The male pathway requires *MIN*[22,25,72], which could have facilitated the evolution of monoicy from males, as also seen in the green lineage[73,74]. We note the presence of U-SDR-derived gene(s) in all monoicous species, particularly a cholesterol transporter gene that is found in all U-SDRs, suggesting that the U-SDR contains femaleness-promoting factor(s), consistent with reports in kelps[75]. We thus propose that monoicy evolved via translocation events adding essential femaleness-promoting genes to a male genetic background. Since the resulting monoicous individuals were capable of producing both male and female reproductive structures, individuals with U chromosomes were no longer essential for sexual reproduction, ultimately leading to the loss of the U-SDR in monoicous species. Although the combination of key female and male genes is essential for this evolutionary transition, the retention of a sex chromosome is not. For example, in *Volvox africanus*, monoicy required the retention of female SDR-like regions, while most male SDR genes were lost except for a multicopy array of the male-determining gene *MID*[76].

The evolution of a dioecious system in *F. serratus* is associated with an irreversible transition to diploidy in Fucales[16,47]. The U/V to dioecy transition has remained elusive[2,47], but our data in brown algae imply that it involved an intermediate monoicous stage, supporting previous predictions from ancestral state reconstruction analyses[16]. A small, undifferentiated Y-specific region consistent with a young XY system may explain why the sex chromosome in *F. serratus* was undetectable. Nonetheless, all ancestral V-SDR genes are found in the U/V-homologue of *F. serratus*, several showing a male-biased gene expression across Fucales species, particularly *MIN*[22]. Our findings imply that *MIN* and possibly other ancestral V-SDR genes are still involved in male differentiation, but shifted downwards in the sex determination cascade. These results thus support and extend the 'bottom-up' hypothesis of sex determination, where downstream components of sex differentiation are conserved across taxa, and new master sex regulators can replace older ones[77].

## Methods

### Biological material

*Scytosiphon promiscuus, Dictyota dichotoma, Undaria pinnatifida* and *Desmarestia dudresnayi* haploid gametophytes were cultivated under laboratory conditions as in ref. 78. We cultivated the gametophytes at 14 °C with a photoperiod of 12:12 h light:dark at an irradiance of 25 µmol photons $m^{-2} s^{-1}$. The media consisted of filtered natural seawater (NSW), which was autoclaved and enriched with half-strength Provasoli nutrient solution (Provasoli-enriched seawater; PES)[78]. We grew the first biomass in 140 mm Petri dishes and the gametophytes were later transferred to a 1 l flask with gentle aeration. The gametophytes were fragmented once a month and the media were changed every 2 weeks to promote biomass production. Before freezing, gametophytes were treated with antibiotics for 3 days with gentle agitation and under the same culture conditions. The first day, gametophytes were treated with a mix of streptomycin (2 g $l^{-1}$ of PES), penicillin G (0.5 g $l^{-1}$ of PES) and chloramphenicol (0.1 g $l^{-1}$ of PES); the next day with ampicilin (1 g $l^{-1}$ of PES), and on the last day with kanamycin (1 g $l^{-1}$

of PES). Between each day of treatment and before freezing, gametophytes were rinsed with 500 ml of NSW to remove traces of antibiotic.

Samples of fucoid algae sexual and vegetative tissue were collected in the intertidal zone during low tides in June 2012 from Viana do Castelo (*F. vesiculosus, A. nodosum*) and Caminha (Rio Minho; *F. ceranoides*), northern Portugal. Sexual phenotypes were verified in the field by sectioning and observing receptacles under a field microscope. Tissue samples were flash frozen in liquid nitrogen on the shore and transported to the laboratory in a cryoshipper, after which they were lyophilised and stored dry at room temperature on silica crystals (see Supplementary Table 1 for a list of strains used in this study).

### DNA and RNA extraction and sequencing

Genomic DNA was isolated from algal tissue (~100 mg) by grinding into fine powder under liquid nitrogen and subsequent cell lysis in 500 µl of Genomic Lysis Buffer (OMNIPREP for plant kit) for 1 h at 60 °C. The lysate was cleaned up with 200 µl of chloroform and DNA was precipitated in ethanol. The DNA pellet was digested in CF buffer (Macherey–Nagel) for 45 min at 65 °C and purified using NucleoBond AXG20 Mini columns according to the user manual (Macherey–Nagel). Final high molecular weight genomic DNA was quantified (Qubit), analysed for purity (Nanodrop) and checked for size distribution (Femto Pulse System) before preparing the sequencing libraries. We sequenced the libraries using an Oxford Nanopore Technologies (ONT) MinION Mk1B. We prepared the ONT libraries using an SQK-LSK110 library preparation kit for R9.4.1 flow cells and an SQK-LSK114 library preparation kit for R10.4.1 flow cells. Two libraries were sequenced for *D. dudresnayi* on R9.4.1 flow cells and a third library was sequenced on an R10.4.1 flow cell.

RNA was isolated from mature gametophytes of *U. pinnatifida* and *S. promiscuus* following modified procedure of the Qiagen RNAeasy kit, and the TruSeq RNA Library Prep Kit v.2 was used to sequence the transcriptomes in an Illumina NextSeq 2000 platform (150 bp, PE reads). Extraction of total RNA from fucoid algae (*F. vesiculosus, A. nodosum* and *F. ceranoides*) was performed following ref. 79 and RNA libraries were sequenced on an Illumina HiSeq 2000 machine (100 bp, PE reads).

### Genome assembly and annotation

High-quality, chromosome-level assemblies of brown algal genomes have been notoriously difficult to obtain due to technical challenges in extracting nucleic acids. Whole-genome assemblies and annotations of *S. promiscuus* male, *D. dichotoma* male, *D. herbacea* male and female, *E. crouanorum* male, *C. linearis, S. ischiensis* and *F. serratus* male were obtained from ref. 18. We also downloaded the genome of *Ectocarpus* sp. 7 (ref. 19) and the male genome of *U. pinnatifida*[20], which were already assembled at a chromosome level. For *D. dudresnayi*, we performed genome sequencing, de novo genome assembly and ab initio gene annotation. Base calling was done using ONT Guppy[80] with the configuration files 'dna_r9.4.1_450bps_sup.cfg' and 'dna_r10.4.1_e8.2_400bps_sup.cfg' and the options '−trim_adapters −trim_primers', yielding 17.4 Gbp of data in 2,871,152 reads. We merged all the reads and analysed them using Kraken (v.2.1.2)[81] and the bacteria database (August 2022) to remove potential contaminant sequences. All data classified as bacterial reads by Kraken were screened using blastN (v.2.13.0+)[82] (-evalue 0.001 -num_alignments 20) against the NCBI genbank bacterial database (downloaded November 2023). The blastN output was visualized in MEGAN (v.6.23.4)[83], and all reads that were declared as bacterial were extracted and removed from further analyses. We obtained 1,908,772 decontaminated reads with an average length of 5.1 Kbp (9.8 Gbp of data, 20× coverage), which were deposited on the NCBI Sequence Read Archive (Supplementary Table 1).

The decontaminated reads were assembled de novo using flye (v.2.9.1-b1780)[84] with the options '−nano-raw -g 450 m -t 28 -i 3 −scaffold'. The draft assembly consisted of 1,032 contigs with a total size of 425 Mbp, an N50 of 4.6 Mbp and an L50 of 29 contigs. We used TransposonPSI (http://transposonpsi.sourceforge.net/) to predict

the TEs and RepeatScout (v.1.0.6)[85] to predict the simple repeats in the genome assembly. Both predictions were combined to soft mask the repetitive content in the genome assembly using bedtools maskfasta (v.2.27.1)[86]. We mapped the RNA-seq data of *D. dudresnayi* from the PhaeoExplorer database[18] to the soft-masked genome assembly using STAR (v.2.7.1a)[87]. We used BRAKER v.2.1.6 alongside the RNA-seq data[88] to predict the protein-coding genes in the soft-masked genome assembly.

### Hi-C library preparation and sequencing for chromosome-level assemblies

We generated Hi-C libraries for three male genomes (*S. promiscuus*, *D. herbacea* and *D. dichotoma*) and two female genomes (*Ectocarpus* sp. 7 and *D. herbacea*). Fresh algal tissue was cross-linked for 20 min at room temperature in a solution of 2% formaldehyde with filtered NSW and then transferred into a 400 mM glycine solution with filtered NSW for 5 min to quench the formaldehyde. The samples were then stored at −80 °C until use. The Hi-C libraries were prepared as follows. The samples were de-frosted in 1 ml of 1× *Dpn*II buffer with protease inhibitors (Roche cOmplete), transferred to Precellys VK05 lysis tubes (Bertin) and disrupted using the Precellys apparatus with five grinding cycles of 30 s at 7,800 r.p.m., followed by 20 s pauses. SDS was added to the lysate at 0.5% final concentration and samples were incubated for 10 min at 62 °C, followed by the addition of Triton X-100 to a final concentration of 1% and 10 min of incubation at 37 °C under gentle shaking. We added 500 U of *Dpn*II to 4.6 ml of the digestion mixture and incubated the samples for 2 h at 37 °C under gentle shaking (180 r.p.m. in an inclined rack to prevent sedimentation), followed by the addition of another 500 U of *Dpn*II and an overnight incubation under the same conditions. The digested samples were centrifuged at 4 °C for 20 min at 16,000 × *g*. The supernatant was discarded and the pellet was incubated for biotinylation at 37 °C for 1 h under a constant shaking (300 r.p.m.) in a 500 ml biotinylation mix with a concentration of 1× ligation buffer, 0.09 mM dATP-dGTP-dTTP, 0.03 mM biotin-14-dCTP and 0.64 U ml⁻¹ Klenow fragments. After biotinylation, the samples were incubated for 3 h at room temperature in a 1.2 ml ligation reaction with a concentration of 1× ligation buffer, 100 mg ml⁻¹ BSA, 1 mM ATP and 0.4 U ml⁻¹ T4 DNA Ligase. The samples were then incubated overnight at 65 °C after adding 20 μl 0.5 M EDTA, 80 μl 10% SDS and 1.6 mg Proteinase K. DNA was extracted with 1 volume of phenol/cholorform/isoamyl alcohol (24:24:1), followed by 30 s of vortexing at top speed and a 5-min centrifugation at top speed. We precipitated the DNA by adding 1/10 volume of 3 M NaAC pH 5 and two volumes of 100% cold ethanol, followed by a 30-min incubation at −80 °C and a 20-min centrifugation at 14,000 × *g* and 4 °C. The DNA pellet was washed with 1 ml 70% ethanol, then dried at 37 °C for 10 min and resuspended in 100 μl 1× TE buffer with 1 mg ml⁻¹ RNase. DNA was sheared to 250–500 bp fragments using a Covaris S220 ultrasonicator, purified with AMPure beads (0.6×) (Beckman) and eluted in 20 μl 10 mM Tris pH 8.0. Biotinylated but not ligated DNA fragments were first removed by T4 DNA polymerase treatment (final concentration, 300 U per pellet; NEB), and the biotin-labelled fragments were selectively captured by Dynabeads MyOne Streptavidin C1 beads (Invitrogen). The libraries were prepared using the NEB Ultra II library preparation system and sequenced on the NextSeq 2000 Illumina platform (2 × 150 bp) (Supplementary Table 1).

We scaffolded the genomes from ref. 18 into chromosome-level assemblies using the Hi-C data. We filtered the low-quality Hi-C reads using Trimmomatic (v.0.39)[89] (ILLUMINACLIP:2:30:10 LEADING:25 TRAILING:25 SLIDINGWINDOW:4:15 MINLEN:75 AVGQUAL:28). We mapped the Hi-C reads against each genome assembly using BWA-mem v.0.7.17-r1188d in the Juicer v.1.6 pipeline[90] to generate a contact map, which was then fed to 3D-DNA v190716 (ref. 91) to scaffold the genomes into chromosomes. The obtained scaffolds were manually inspected against the contact maps to solve the limits of each chromosome using Juicebox (v.1.11.08)[92]. The PhaeoExplorer gene annotations[18] were lifted into the new assemblies using Liftoff (v.1.6.1)[93], while the annotation

of TEs was performed using RepeatModeler2 (ref. 94). We scaffolded the genomes of *E. crouaniorum* and *D. dudresnayi* into chromosomes using a reference-guided assembly with RagTag (v.2.0.1)[95] against the chromosome-level assemblies of *Ectocarpus* sp. 7 and *D. herbacea*, respectively. All genes within the SDRs in the brown algal species studied (see below) were manually curated to exclude any TE-related genes from the annotation.

### Discovery of the U/V sex determination regions

Male sex-determining regions (V-SDR) in *S. promiscuus*, *U. pinnatifida*, *D. herbacea* and *D. dichotoma*, as well as female sex-determining region (U-SDR) in *D. herbacea* were analysed following two complementary methods: (1) a *k*-mer-based YGS approach, originally designed to detected Y-linked sequences in heterogametic systems, developed in ref. 96 and (2) genomic coverage analysis, designed to identify sex-linked regions through differences in read depth between male and female individuals[97]. These methods are well suited for organisms with divergent sex chromosomes, such as brown algae, where U and V haplotypes have diverged over extended evolutionary time.

The YGS method principle is to identify male or female sex-linked scaffolds by comparing *k*-mer frequencies between reference genome assembly and *k*-mers generated from DNA-seq reads of the opposite sex. Regions in the male reference genome that contain *k*-mers that are absent in female reads will indicate candidate male SDR sequences; similarly, female genomic scaffolds with low coverage in male *k*-mers will denote female SDR region. For each species, 15-base-pair *k*-mer sequences were generated separately from male and female Illumina reads (see Supplementary Table 1 for data accession numbers) using Jellyfish v.2.3.0 count (-m 15 -s 10 G -C –quality-start=33 –min-quality=20) and converted to fasta format with Jellyfish dump (–lower-count=5)[98]. Next, non-overlapping 500-kb sliding windows (*Desmarestia*, *Dictyota* and *Undaria*) or 200-kb sliding windows (*Scytosiphon*) of the reference chromosome genomes (from the sex whose SDR was to be identified) were created using seqkit (v.2.3.1)[99] and used as input for the YGS.pl script[96] together with the fasta *k*-mer files produced in the previous step. Each window was then analysed to calculate the proportion of *k*-mers in the reference window that are not present in the opposite-sex *k*-mer database. Genomic windows with a minimum of ≥50% of unmatched single-copy *k*-mers were then retained as candidate male or female SDR sequences. Because the borders of the SDRs cannot be precisely defined at the single-nucleotide level with the available data, we focused on genes within these regions and defined the SDR boundaries on the basis of the flanking genes located at the transition to pseudoautosomal regions (PARs).

Candidate SDR regions identified by YGS were further validated by analysing sex-specific differences in read coverage. In detail, the short Illumina reads coming from males and females of each investigated species were trimmed with Trimmomatic[89] (see above) and mapped to the reference genome for which the SDR was to be studied, using HISAT2 (ref. 100) (default settings). Bam files produced by HISAT2 were used as input for Mosdepth[101] to calculate coverage in 10-kb windows along the genome sequence (-m -n -b 10000 –fast-mode -Q 30). Read mapping depth in genomic windows was normalized by the genome-wide mean for each sex, and the coverage in genomic intervals was then compared between males and females. Because V-SDR-linked sequences are present only in males, we expect them to have similar read coverage as autosomal regions in males, but little or no coverage in females (and conversely for the U-SDR sequences in *D. herbacea*). The comparison focused on regions within male reference genomes where the coverage in males fell within the range of 75–125% of the genome average, while the coverage in females remained below 50% of the genome average.

Both coverage and *k*-mer analysis identified identical genomic regions, providing high-confidence candidate SDRs (Extended Data Table 1 and Supplementary Figs. 1–5). In *D. herbacea*, where both male and female chromosome-level genome assemblies were available,

we directly compared U and V chromosomes to further confirm the SDR borders by analysing the collinearity of pseudoautosomal regions flanking the SDRs. The SDR scaffolds for all studied species were further validated by PCR amplification (Supplementary Table 1) using 4 males and 4 females.

## Genetic mapping and search for the sex chromosome in *F. serratus*

Three different sets of materials were used in this study: (1) 12 male and 12 female field samples, hereafter denoted the 24-individual natural population; (2) 157 sporophyte progeny population derived from a cross between one male sample and one female sample collected from the field and (3) 3 male and 3 female samples collected from the field for whole-genome sequencing. The 157-progeny population and 24-individual natural population were genotyped using the double digest RAD sequencing approach (ddRAD-seq). Briefly, individual genomic DNA was digested with the restricted enzymes PstI and HhaI to obtain fragments that were size selected between 400 and 800 bp before sequencing on in Illumina HiSeq 2500 platform (paired-end 2 × 125 bp). See ref. [102] for detailed protocol of the ddRAD-seq.

We performed whole-genome sequencing on an Illumina HiSeq 2500 system (2 × 150 bp paired-end) for the 3 male and 3 female samples. For ddRAD-seq data, raw reads were cleaned and trimmed with Trimmomatic as above and mapped to the draft genome of *F. serratus* male. For the progeny population, genotypes were called from the obtained bam files using the Stacks pipeline (v.2.5)[103]. The obtained vcf files were filtered with VCFtools (v.0.1.16)[104] and bcftools[105] (max missing per locus: 30%, max missing per sample: 40%, max mean coverage:30, minQG:20).

The filtered vcf file of the progeny population was used to construct a genetic map with Lep-MAP3 (ref. [106]). Briefly, the ParentCall2 module was used to call parental genotypes, the SeparateChromosomes2 module was used to split the markers into linkage groups and the OrderMarkers2 module was used to order the markers within each linkage group using 30 iterations per group and finally computing genetic distances. Phased data were converted to informative genotypes with the script map2genotypes.awk.

## We used different approaches to identify the SDR in *F. serratus*.

*Coverage analysis*. We combined whole-genome sequence data from the 3 males and 3 females alongside the ddRAD-seq data of the 24-individual natural population, mapping both datasets to the *F. serratus* male genome assembly using bwa-mem[107]. Coverage analyses was done in several ways:

- Using SATC (sex assignment through coverage)[108], a method that uses sequencing depth distribution across scaffolds to jointly identify: (1) male and female individuals and (2) sex-linked scaffolds. This identification was achieved by projecting the scaffold depths into a low-dimensional space using principal component analysis and subsequent Gaussian mixture clustering. Male and female whole-genome sequences were used for this analysis.
- Using the method SexChrCov described in ref. [109] with the 24-individual natural population.
- Using the method DifCover[110] which identifies regions in a reference genome for which the read coverage of one sample is significantly different from the read coverage of another sample when aligned to a common reference genome. The 24-individual natural population was used for this analysis.
- Using soap.coverage (v.2.7.9)[111] to calculate the coverage (number of times each site was sequenced divided by the total number of sequenced sites) of each scaffold in each sample. For each scaffold, the male to female (M:F) fold change coverage was calculated as $\log_2$(average male coverage) − $\log_2$(average female coverage). The 24-individual natural population was used for this analysis.

*Fixation index ($F_{ST}$) and sex-biased heterozygosity*. This approach has been previously used to find sex-linked genomic regions in several studies[112,113]. Using the 24-individual natural population, $F_{ST}$ was calculated using vcftools[104]. Sex-biased heterozygosity was defined as the $\log_{10}$ of the male heterozygosity:female heterozygosity ratio, where heterozygosity was measured as the fraction of sites that are heterozygous. This ratio is expected to be zero for autosomal scaffolds and elevated on young sex scaffolds due to excess heterozygosity in males.

*Identification of eventual female scaffolds that failed to map to the male reference genome*. Vcftools and bedtools were used to extract female regions that did not map to the reference genome consistently in the 3 resequenced female samples.

All candidate contigs were tested by PCR in 4 males and 4 females.

## Synteny analyses, $K_s$ analysis and transitions to co-sexuality

Whole-genome synteny comparisons were performed for each pair of chromosome-level assemblies using MCscan (v.1.2.14)[114], both between different species, between sex chromosomes in the same species and between monoicous species and their closest relatives with U/V chromosomes. The putative gametologues between sex chromosomes that were predicted with MCscan were reassessed using OrthoFinder (v.2.5.4)[115] and best reciprocal DIAMOND (v.2.1.8.162)[116] hits.

We calculated the number of synonymous substitutions per synonymous site ($K_s$) for each pair of male and female gametologues as a proxy to assess the relative time at which both genes diverged from each other. The amino acid sequences of each pair of gametologues were aligned with MAFFT (v.7.520)[117] and subsequently aligned into codons using pal2nal (v.14)[118]. The gametologue $K_s$ values were calculated using the model in ref. [119] as implemented in KaKs_calculator (v.2.0)[120].

We evaluated the male or female identity of the genes in the monoicous species whose orthologues were found within the SDR in their closest dioicous relatives. For this, we compared the results obtained with MCscan[114] against the orthogroup prediction performed with OrthoFinder[115], with best reciprocal DIAMOND[116] hits and by calculating gene trees for each orthogroup using an amino acid alignment with MAFFT[117] and gene tree reconstructions using FastTree (v.2.1.11)[121].

## Ancestral reconstruction of the male SDR

The brown algal phylogeny was obtained from ref. [18]. The species tree is based on 32 single-copy nuclear genes whose protein sequences were aligned manually using AliView[122], and whose best-fit substitution models were assessed independently using the Akaike information criterion. The tree was generated using a maximum likelihood approach implemented in RAxML bootstraps and the gamma model. Every node in the phylogeny has 99–100% bootstrap support values. Divergence times were subsequently calculated using MCMCtree[123] and three calibration points. The MCMC chains were run for 1.5 million generations and the first 200,000 MCMC chains were discarded as burn-in.

We searched for orthologue genes within the V-SDR of five species (*Ectocarpus* sp. 7, *S. promiscuus*, *U. pinnatifida*, *D. herbacea* and *D. dichotoma*) in our OrthoFinder results. For each V-SDR gene, we coded its orthologue in the other species as 'present' (1) if it is also sex linked in the V-SDR, whereas it was coded as 'absent' (0) if the orthologue resides in the PARs, in an autosome or if there is no detectable orthologue in that species. Once we generated this presence/absence matrix with the evolutionary relationship of the genes within the V-SDR (Supplementary Table 5), we used it as the input file for the software Count (v.10.04)[124] to estimate the ancestral content of the V-SDR throughout a phylogeny and determine the most likely scenario of V-SDR evolution in the brown algae. We employed posterior probabilities under a phylogenetic birth-and-death model with independent gain and loss rates across each branch in the phylogeny. We modelled the independent gain and loss rates through 10 gamma categories and performed 1,000 optimization rounds with a convergence threshold

on the likelihood >0.1 to find the best fitting model for the data. The branch lengths in the tree that were used for the ancestral state reconstruction were retrieved from the molecular clock analysis performed in ref. 16. We distinguished between conserved V-SDR genes that are ancestral and parallel acquisitions of the same gene in the V-SDR by analysing gene trees between male and female genomes, in addition to female transcriptome assemblies of *D. dichotoma* and *U. pinnatifida*. Sequence alignments were done using MAFFT[117] with default settings and uploaded to the https://www.phylogeny.fr/ platform. Alignments were further curated using Gblocks (v.0.91b)[125] (min. seq. for flank pos.: 85%, max. contig. nonconserved pos.: 8, min. block length: 10). Trees were produced using PhyML (v.3.1l)[126] with the default model and visualized in TreeDyn (v.198.3)[127]. The approximate likelihood-ratio test was chosen as the statistical test for branch support. We inferred the function of the ancestral V-SDR genes through the annotation of genes in *Ectocarpus* sp. 7 belonging to that orthogroup. The most likely acquisition mechanism of each SDR gene in each species was assessed on the basis of the position of each orthologue in the other species (pseudoautosomal, autosomic or missing; Supplementary Table 5).

### Genomic content across chromosomes

We used closely related genome assemblies available in the PhaeoExplorer database[18] to assess the depletion of orthologues in the sex chromosome. We predicted one-to-one orthologues using OrthoFinder[115] between the following species pairs: *Ectocarpus* sp. 7 with *Ectocarpus siliculosus*, *S. promiscuus* with *C. linearis*, *U. pinnatifida* with *Saccharina japonica*, *F. serratus* with *Fucus distichus*, *D. herbacea* with *D. dudresnayi*, and *D. dichotoma* with *Halopteris paniculata* (Supplementary Table 16). We calculated the expected number of detectable orthologues for each chromosome and compared it against the observed number of detected orthologues using chi-square tests. We performed Benjamini–Hochberg corrections to the *P* values of the chi-square tests to control the FDR in the analysis[128].

GenEra[36] was used by running DIAMOND in ultra-sensitive mode[116] against the NCBI NR database and all the PhaeoExplorer proteins[18] to perform a phylostratigraphic analysis (*e*-value threshold of $10^{-5}$) and calculate the relative ages of each gene in each genome (Supplementary Table 13). Phylostratigraphy is a genetic statistical method developed to date the putative origin of all the genes contained in the genome of a target species by detecting homologues across species at different evolutionary distances (all the way from species within the same genus to species from different domains of life). Finding the most distant homologues of each gene can link them to their founder events (that is, the first instance where a gene homologue is found in the history of that lineage), allowing us to then determine their relative ages, coded as the taxonomic group where that gene is detected[35,36,129]. The gene age categories outside of the brown algae and *S. ischiensis* were based on the taxonomic classification of each species within the NCBI Taxonomy database[130], while the gene ages within the brown algae were manually assessed to reflect the evolutionary relationships obtained in the PhaeoExplorer maximum likelihood tree[18]. We performed Wilcoxon rank-sum tests in R (v.4.3.1)[131] to assess non-random differences in gene age distributions between pairs of chromosomes (Supplementary Table 14). We performed Benjamini–Hochberg corrections to the *P* values of the Wilcoxon rank-sum tests to control the FDR in the analysis[128]. The gene ages responsible for these differences were found by evaluating the standardized residuals using mosaic plots.

We used the interspecies $K_s$ values between pairs of species as a proxy for neutral mutation rates across six of the seven chromosome-level assemblies by using the most closely related genome assemblies available in the PhaeoExplorer database[18]. We used the same set of one-to-one orthologues detected between species pairs as for the orthologue-depletion test (Supplementary Table 16). However, the evolutionary distance between *D. dichotoma* and *H. paniculata* prevented us from calculating reliable interspecies $K_s$ values for this species since synonymous substitutions reached the point of saturation. The amino acid sequences of each pair of orthologues were aligned with MAFFT[117] and subsequently aligned into codons using pal2nal[118]. The interspecies $K_s$ values were calculated using the model in ref. 119 as implemented in KaKs_calculator (v.2.0)[120]. We also evaluated the difference in interspecies $K_s$ values between the autosomes and the sex chromosomes through FDR-corrected Wilcoxon rank-sum tests (Supplementary Table 17). We calculated the protein-coding density, the density of TEs and the taxonomic identity of these TEs within 100-kb non-overlapping windows across each chromosome using bedtools[86] (Supplementary Table 9). The differences in protein-coding space and repeat content between the autosomes and the sex chromosomes were also tested using FDR-corrected Wilcoxon rank-sum tests (Supplementary Tables 10 and 11). The differences in repeat density, percentage of unclassified repeats and gene ages across genomic compartments (SDR, PARs and autosomes) were tested using FDR-corrected permutation tests with 10,000 permutations. All genomic features were plotted using karyoploteR (v.1.20.3)[132].

### Gene expression analysis

We used kallisto (v.0.44.0)[133] to calculate gene expression levels using 31-base-pair-long *k*-mers and 1,000 bootstraps. Transcript abundances were then summed within genes using the tximport v.3.19 package[134] to obtain the expression level for each gene in transcripts per million (TPM). Differential expression analysis was done in the DESeq2 v.3.19 package[135] in R v.4.3.1, applying FC ≥ 2 and $p_{adj}$ < 0.05 cut-offs. Sex-biased gene expression analysis in *Ectocarpus* sp. 7, *S. promiscuus*, *U. pinnatifida*, *D. herbacea* and *D. dichotoma* was performed between mature male and female gametophytes (gametophytes bearing reproductive structures). To discover genes with sporophyte-biased expression in *Ectocarpus* sp.7, *S. promiscuus*, *U.pinnatifida* and *D. dichotoma*, we first calculated the differential expression between male gametophytes and sporophytes, as well as female gametophytes and sporophytes. Genes that showed significant sporophyte-biased expression (FC ≥ 2, $p_{adj}$ < 0.05) in both comparisons were considered sporophyte biased.

A total of 314.2 M RNA-seq reads from *F. vesiculosus* male, female and vegetative tissue were assembled de novo with rnaSPAdes[136] using *k*-mer values of 33 and 49. Assembly quality was assessed by (pseudo) mapping reads back onto the resulting assembly and retaining 'good' contigs as defined using TransRate (v.1.0.3)[137] with default settings. The resulting 159,108 contigs were aligned with BLASTx[82] against a database of Stramenopile proteins, and those with top hits against brown algae (Phaeophyceae) were retained as the final curated reference transcriptome (36,394 contigs, N50 = 1,770 bp). Transcript expression levels were determined by mapping the reads from all samples against the reference transcriptome using Bowtie2 (ref. 138) and the RSEM-EBSeq (v.1.3.3)[139] pipeline, and relative expression values were recorded as TPM. All samples used in the gene expression analysis can be found in Supplementary Table 1.

### Reporting summary

Further information on research design is available in the Nature Portfolio Reporting Summary linked to this article.

## Data availability

The accession numbers and download links for all the chromosome-level genome assemblies and annotations that were generated and used in this study are available in Supplementary Table 1 and in the Edmond Repository[140] at https://doi.org/10.17617/3.OOWB2Y. The raw sequence reads for the Oxford Nanopore data, Hi-C libraries and RNA-seq libraries are available in the Sequence Read Archive under BioProject accession number PRJNA1059008. All genome assemblies and annotations are also accessible through the Phaeoexplorer database (https://phaeoexplorer.sb-roscoff.fr/) for comparative genomics

analyses. This paper does not report original data. Further information and requests for resources and reagents should be directed to and will be fulfilled by S.M.C. (susana.coelho@tuebingen.mpg.de).

## Code availability

This paper does not report original code.

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

## Acknowledgements

This work was supported by the MPG, the CNRS, Sorbonne University, the ERC (grant no. 864038 and 638240 to S.M.C.), the France Génomique National infrastructure project Phaeoexplorer (ANR-10-INBS-09), the JSPS Overseas Research Fellowships (to M.H.), the BMBF-funded de.NBI Cloud within the German Network for Bioinformatics Infrastructure (de.NBI) (031A532B, 031A533A, 031A533B, 031A534A, 031A535A, 031A537A, 031A537B, 031A537C, 031A537D, 031A538A), the Investissements d'Avenir project Idealg (ANR-10-BTBR-04-01), the European BG-01 Blue Growth H2020 project Genialg (727892) and the ANR project Epicycle (ANR-19-CE20-0028-01). S.M.C. was supported by the Moore Foundation (GBMF11489) and the Bettencourt-Schueller Foundation. J.B.-R. was supported by a Humboldt Research Fellowship for postdoctoral researchers from the Alexander von Humboldt Foundation. We thank the members of the Phaeoexplorer consortium, in particular C. Jolivet, L. Mest and D. Scornet for assistance with algae cultures, C. Cruaud for help with sequencing libraries preparation, E. Corre and A. Le Bars for support with the Phaeoexplorer database and A. Couloux for the genome assemblies and annotations. We also thank the Roscoff Bioinformatics platform ABiMS (http://abims.sb-roscoff.fr), part of the Institut Français de Bioinformatique (ANR-11-INBS-0013) and Biogenouest network, for providing computing and storage resources.

## Author contributions

J.B.-R and A.P.L. conducted investigation (equal) and formal analysis (equal), designed the methodology (equal), performed visualization (equal), wrote the original draft (equal), and reviewed and edited the manuscript (equal). P.L. conducted investigation (supporting) and formal analysis (supporting). E.D., G.C., O.G., K.B., M.H., R.J.C., K.A., G.L., E.A., D.L., R.L., O.G., S.H., Z.N., L.B.-G. and A.F.P. conducted investigation (supporting). G.H., J.-M.A,. G.P., P.W., F.D. and J.M.C. performed data curation (supporting) and data acquisition (supporting). F.B.H. conducted investigation (supporting), designed the methodology (equal), and performed data curation (equal) and formal analysis (supporting). S.M.C. conceptualized the project (lead), acquired funding (lead), administered and supervised the roject (lead), designed the methodology (equal), performed visualization (supporting), wrote the original draft (equal), and reviewed and edited the manuscript (lead).

## Funding

## Competing interests

The authors declare no competing interests.

## Additional information

**Extended data** is available for this paper at https://doi.org/10.1038/s41559-025-02838-w.

**Correspondence and requests for materials** should be addressed to Susana M. Coelho.

¹Department of Algal Development and Evolution, Max Planck Institute for Biology Tübingen, Tübingen, Germany. ²Sorbonne Université, CNRS, Integrative Biology of Marine Models Laboratory, Station Biologique de Roscoff, Roscoff, France. ³INRAE, Université de Strasbourg, UMR SVQV, Colmar, France. ⁴CNRS, Sorbonne Université, FR2424, ABiMS-IFB, Station Biologique, Roscoff, France. ⁵Bezhin Rosko, Santeg, France. ⁶Faculty of Biosciences and Aquaculture, Nord University, Bodø, Norway. ⁷Universidade do Algarve, UALG·Centro de Ciências do Mar (CCMAR), Montenegro, Portugal. ⁸Génomique Métabolique, Genoscope, Institut François Jacob, CEA, CNRS, Univ Evry, Université Paris-Saclay, Evry, France. ⁹Present address: Departamento de Biotecnología y Bioquímica, Centro de Investigación y de Estudios Avanzados del Instituto Politécnico Nacional, Unidad Irapuato, Irapuato, Mexico. ¹⁰Present address: Research Center for Inland Seas, Kobe University, Nadaku, Kobe, Japan. ¹¹These authors contributed equally: Josué Barrera-Redondo, Agnieszka P. Lipinska. ¹²These authors jointly supervised this work: France Denoeud, J. Mark Cock, Fabian B. Haas, Susana M. Coelho. ✉e-mail: susana.coelho@tuebingen.mpg.de

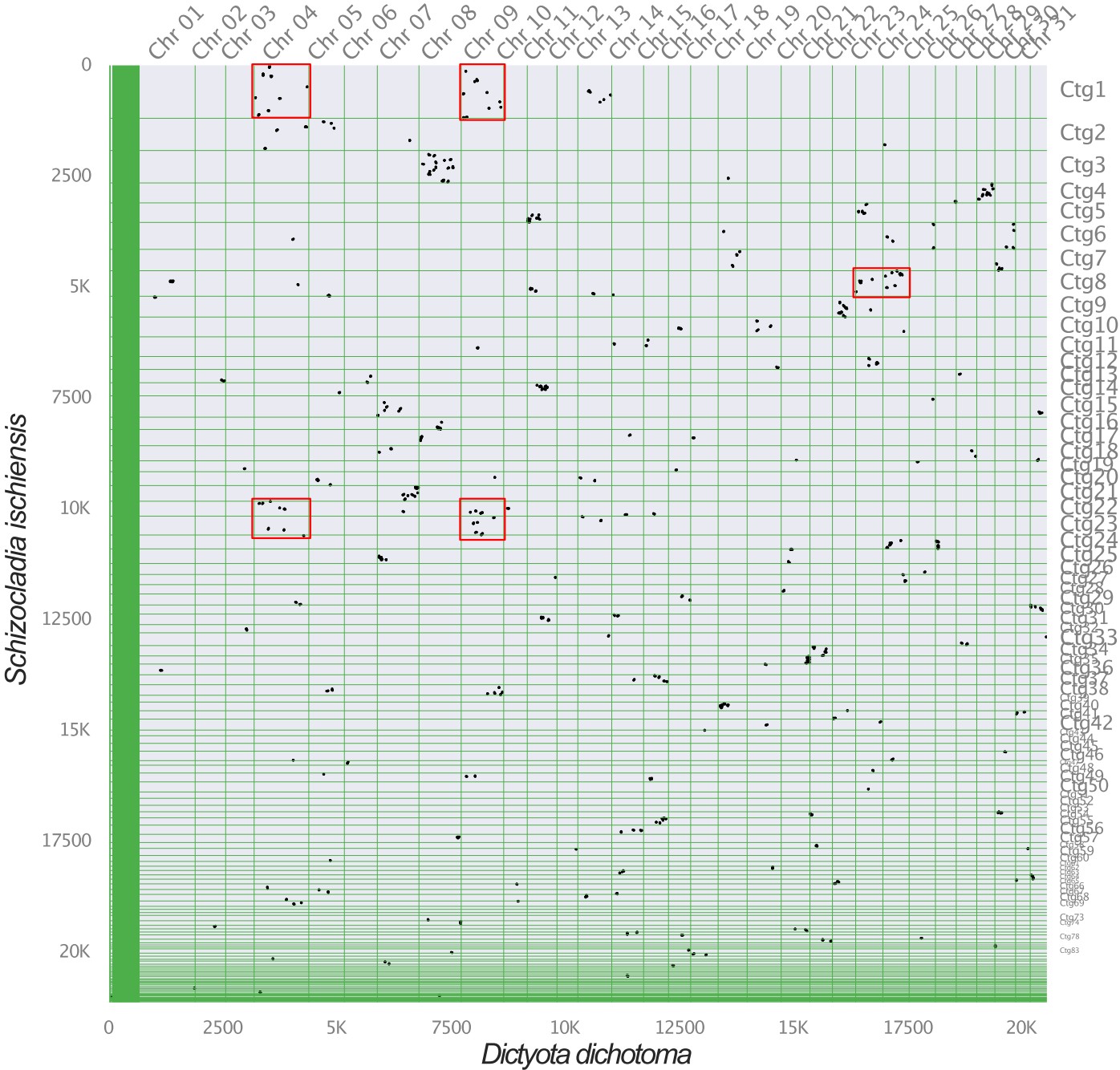

**Extended Data Fig. 1 | Macrosynteny plot between *Schizocladia ischiensis* and *Dictyota dichotoma* using 1,828 orthologs.** We highlight two fusion-with-mixing events (red squares) between chromosomes 4 and 9, and between chromosomes 23 and 24 in *D. dichotoma*.

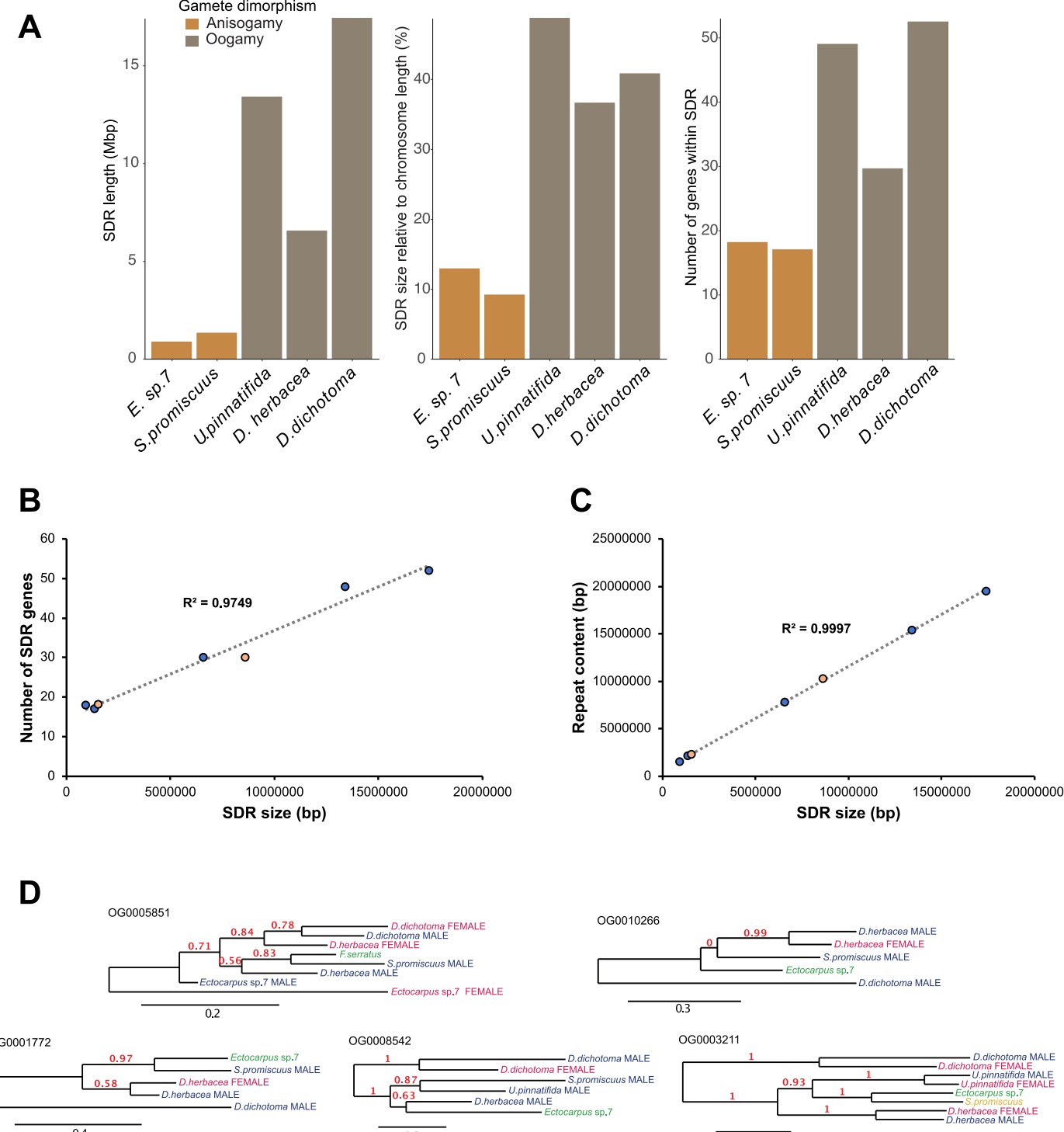

**Extended Data Fig. 2 | SDR size differences and detection of independently-acquired V-SDR genes across species.** (**a**) Differences in the size of the male SDR between brown algal species based on the total sequence length, the relative size of the SDR compared to the length of the V chromosome and the number of protein-coding genes retained within the SDR. The bars are colored accord-ing to the level of gamete dimorphism in each species (based on 16). (**b**) Correlation between the V (blue) and U (pink) SDR sizes and the SDR gene content across species. (**c**) Correlation between the V (blue) and U (pink) SDR sizes and the SDR repeat content across species. (**d**) Gene trees showing the independent acquisition of SDR gametologues across species that were previously interpreted as part of the ancestral male SDR genes.

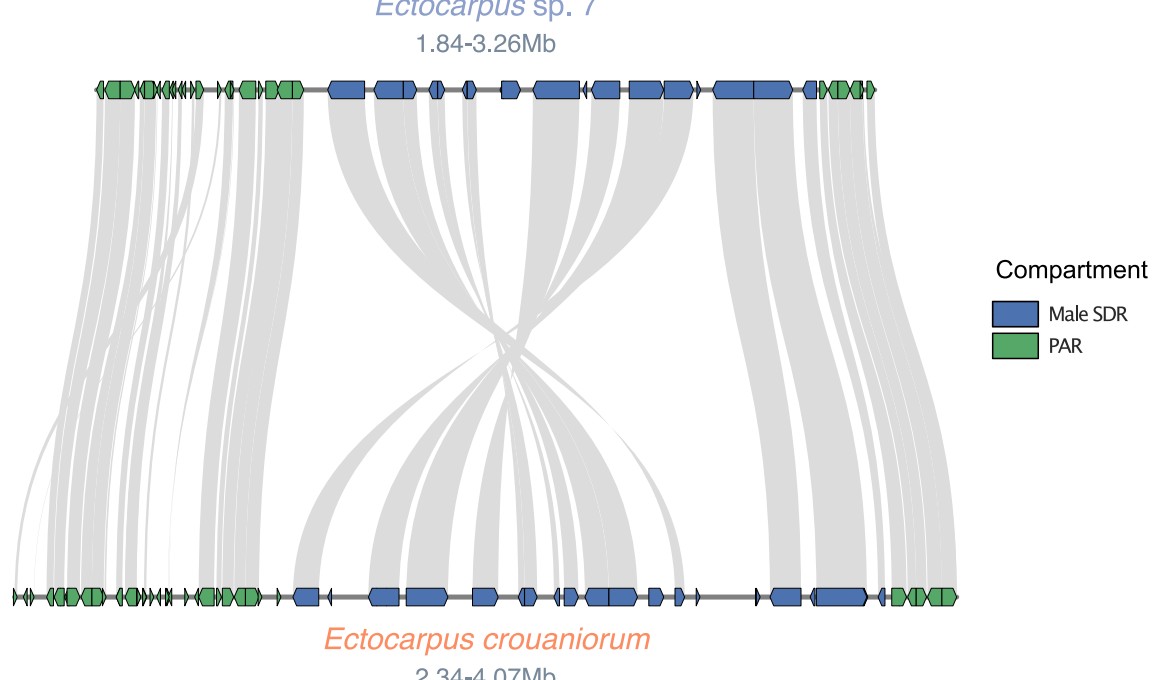

*Ectocarpus* sp. 7
1.84-3.26Mb

*Ectocarpus crouaniorum*
2.34-4.07Mb

**Compartment**
- Male SDR
- PAR

**Extended Data Fig. 3 | Male SDR synteny between *Ectocarpus* sp. 7 and *Ectocarpus crouaniorum*.** One of the species underwent a recent inversion event within the SDR. The arrows in the boxes represent the orientation of each gene within the chromosome.

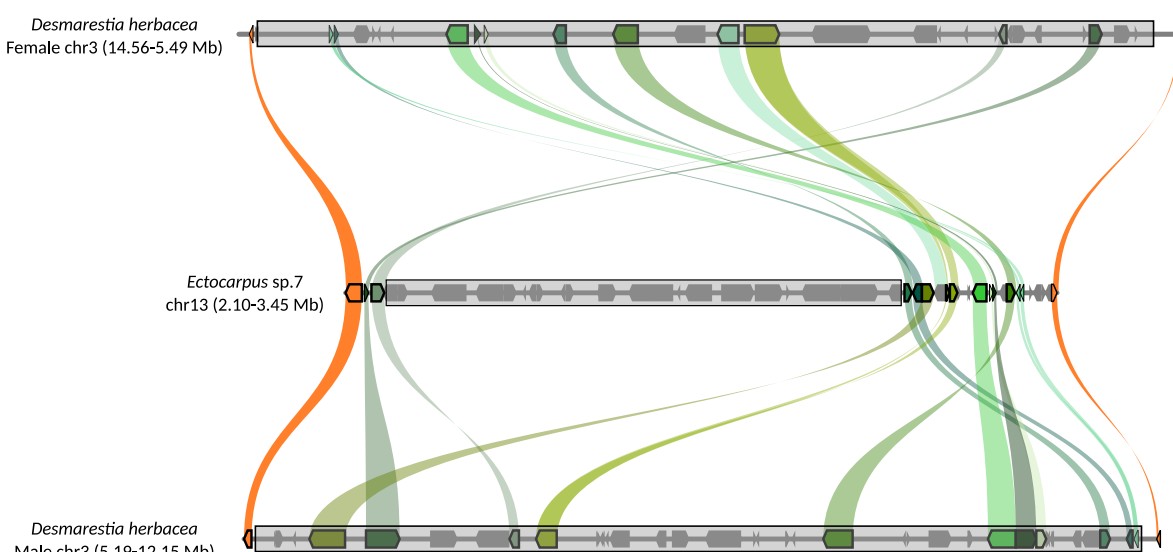

**Extended Data Fig. 4 | Synteny analysis plot illustrating the expansion of the *Desmarestia herbacea* U- and V-sex-determining regions (SDRs) into the surrounding pseudoautosomal region (PAR).** The *Ectocarpus* sp. 7 V-chromosome is shown in the middle as a reference, with the SDR regions outlined by grey boxes. Green lines trace syntenic relationships between *Ectocarpus* PAR genes and the recently-acquired *Desmarestia* SDR genes, with each gene pair represented in a distinct shade of green. This demonstrates that nearly all PAR genes from *Ectocarpus*, which have entered the expanded SDR in *Desmarestia*, are retained as gametologues. Orange lines highlight the PAR boundary genes in *Desmarestia*, which remain within the PAR of *Ectocarpus*.

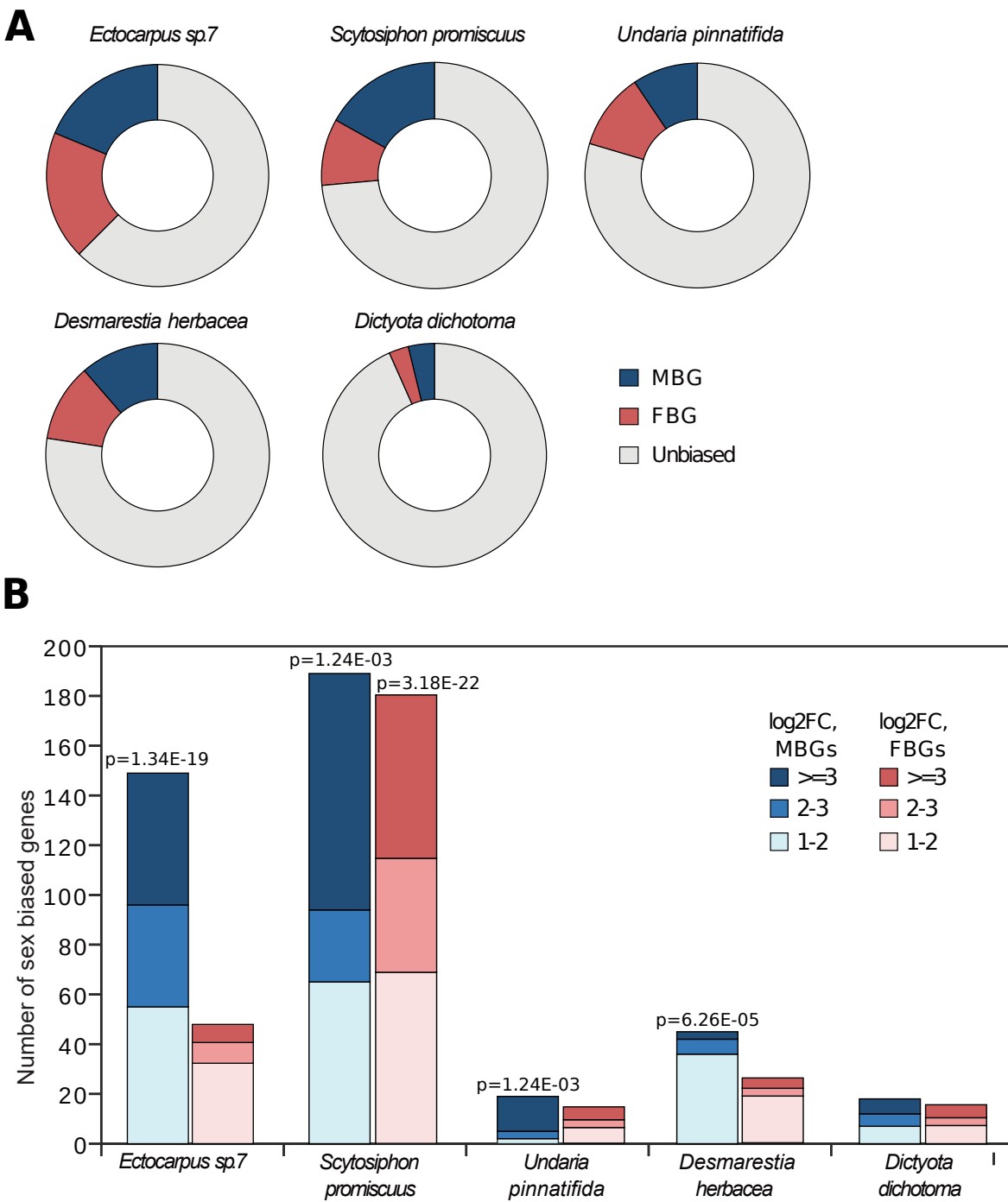

**Extended Data Fig. 5 | Sex-biased gene expression per dioicous species.**
**a**) Proportion of sex biased genes in each of the five dioicous species. MBG: male-biased genes; FBG: female biased genes. (**b**) Number of sex-biased genes in the pseudoautosomal regions of sex chromosomes (U-V-SDRs excluded),

male-biased genes are shown in blue and female-biased genes in red. Exact p-values above the bars mark significant enrichment of the sex-biased genes on the PAR (Chi-square test).

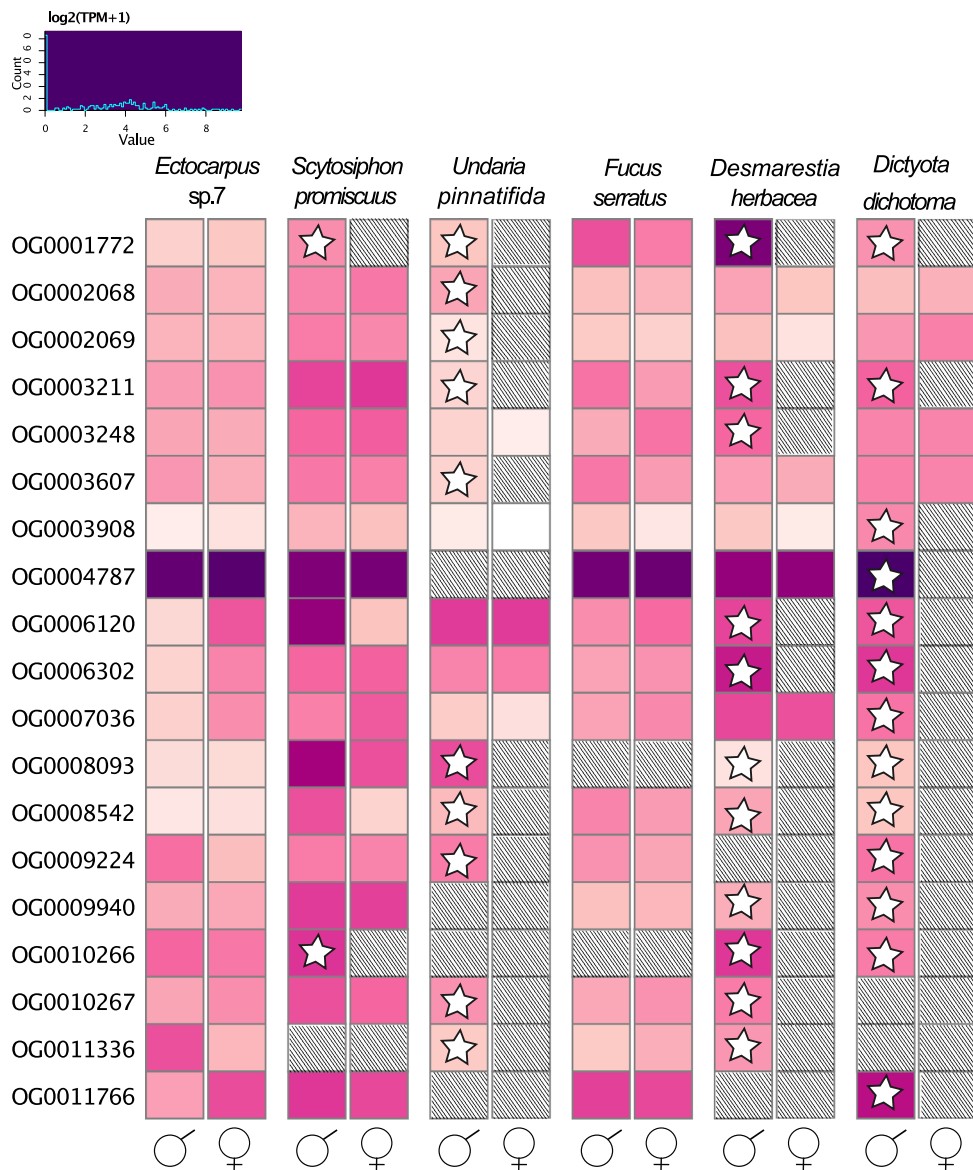

**Extended Data Fig. 6 | Expression of genes (log2(TPM + 1)) that entered the SDR independently in different species.** Expression is measured in mature male and female gametophytes, hashing marks missing orthologues, stars inside the cells indicate that the gene is inside the male non-recombining region (V-SDR). Orthogroups containing orthologues in less than three species or with multicopy genes were excluded from this analysis. M: male; F: female.

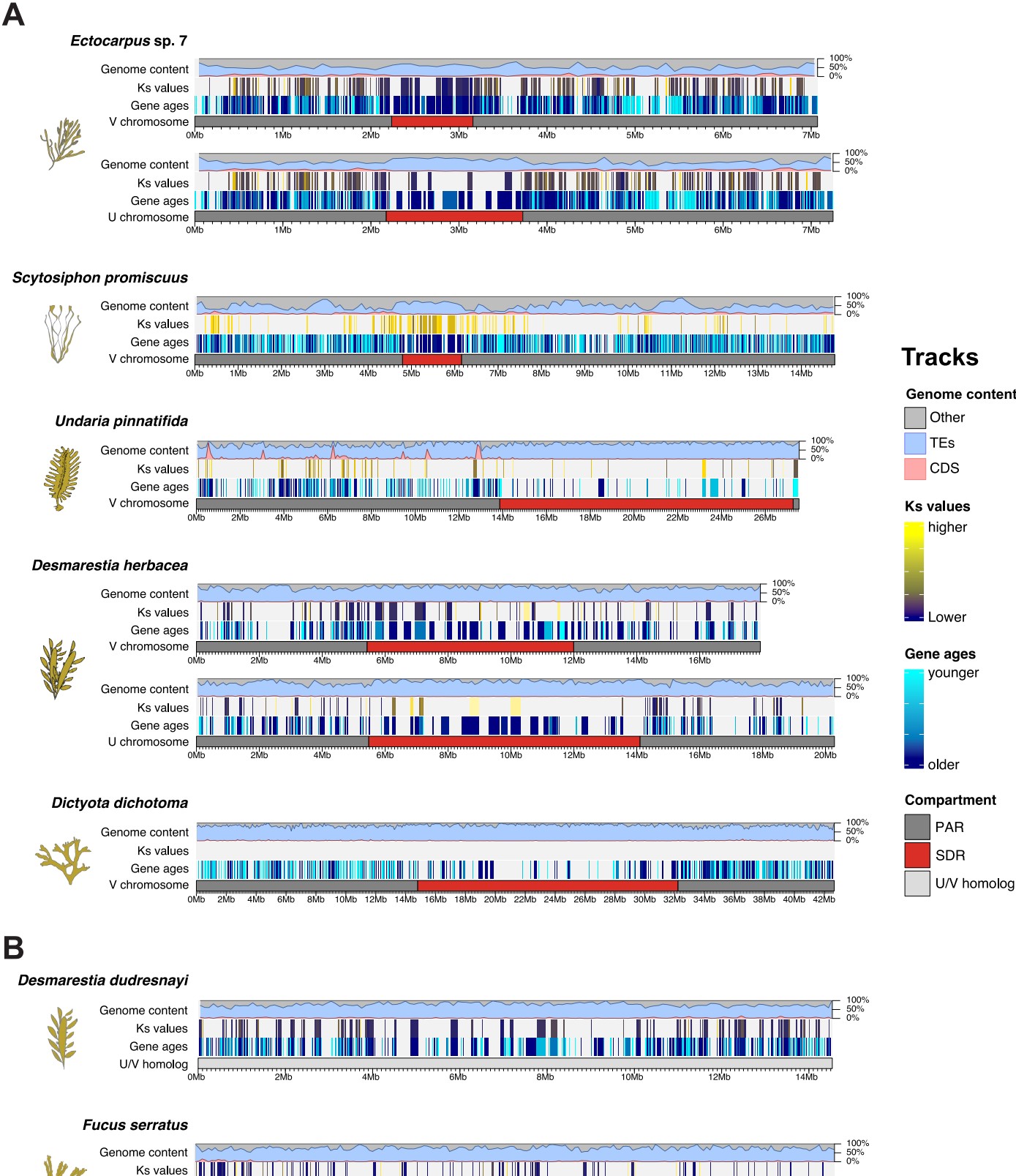

**Extended Data Fig. 7 | Structural features across the sex chromosomes and U/V-homologues of brown algae. (a)** V and U chromosomes of *Ectocarpus* sp. 7, *Scytosiphon promiscuus*, *Undaria pinnatifida*, *Desmarestia herbacea* and *Dictyota dichotoma*. (**b**) U/V-homologues of *Desmarestia dudresnayi* and *Fucus serratus*. Features displayed from bottom to top: chromosome compartments (PARs, SDR, U/V-homologue); relative gene ages, inter-species *Ks* values, and proportion of coding (CDS, red) and repeat (TEs, blue) density.

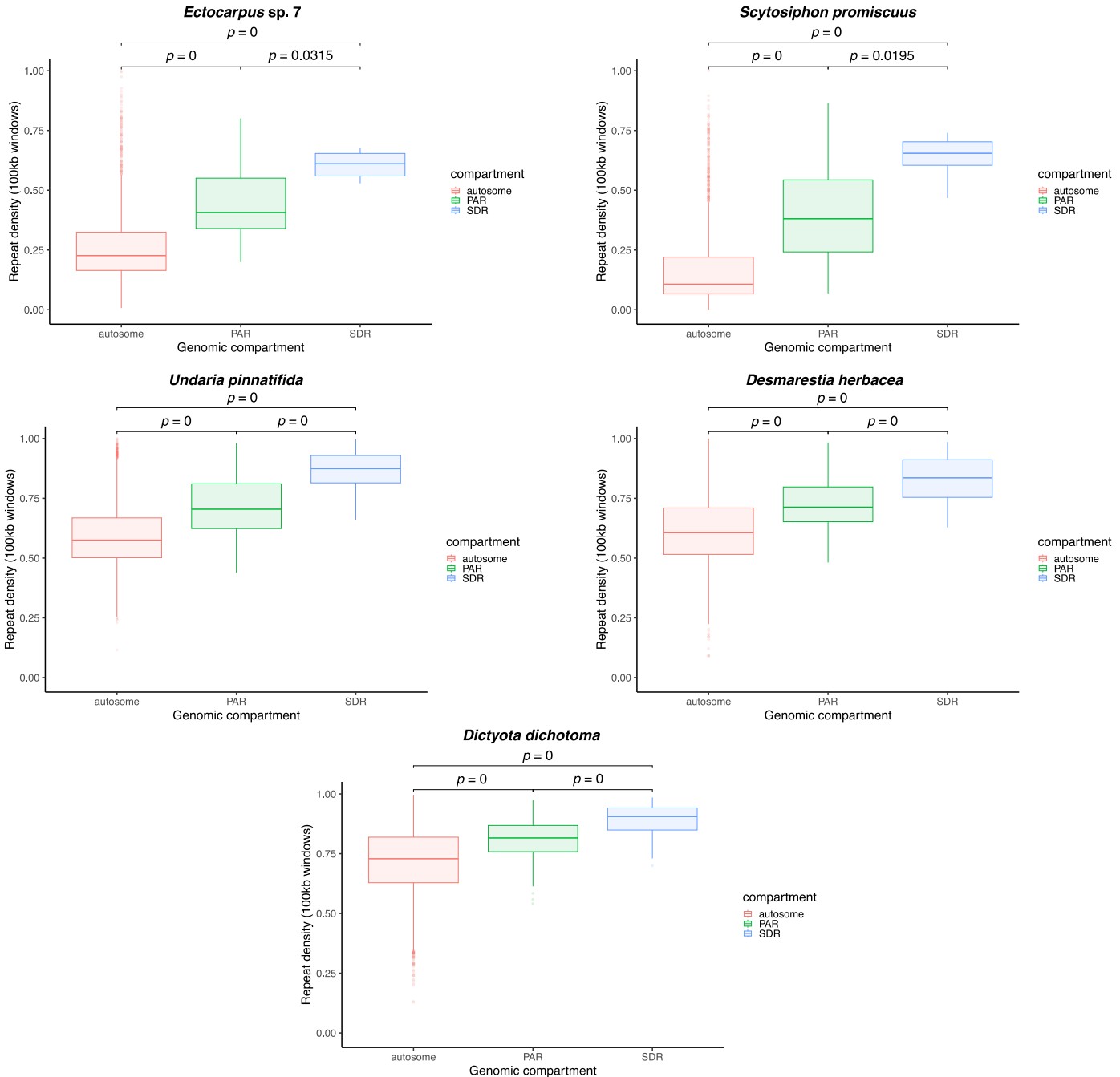

**Extended Data Fig. 8 | Accumulation of repetitive elements in the V-SDRs and PARs of five dioicous species.** Statistically significant differences in median values of repeat density (center line) were assessed using FDR-corrected permutation tests.

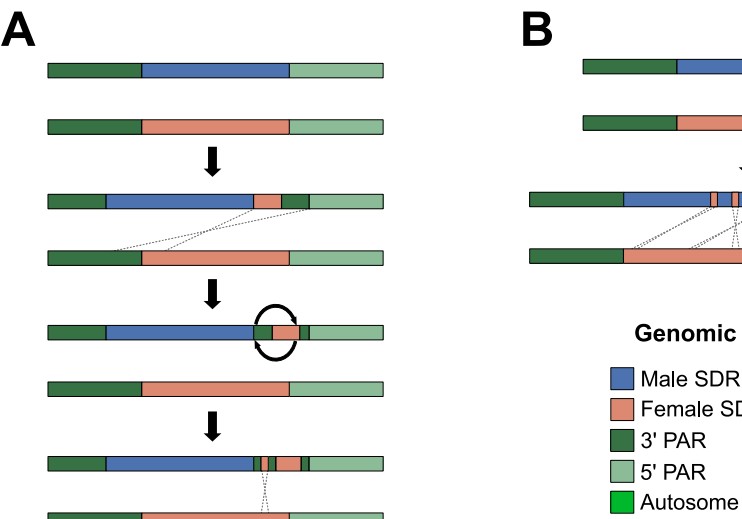

**Extended Data Fig. 9 | Proposed scenarios for the transition from dioicy to monoicy in *Chordaria linearis* and *Desmarestia dudresnayi*.** (**a**) The ancestor of *Chordaria linearis* likely underwent an initial translocation event from the U chromosome to the V chromosome, inserting part of the U-SDR and a piece of the 3′ PAR towards the 5′ end of the V-SDR potentially through an ectopic recombination event. A subsequent inversion within this translocation spread the 3′PAR genes to both sides of the U-SDR insertion. Finally, a second translocation led to the insertion of an additional piece of the U-SDR within the 3′ PAR translocation. (**b**) The ancestor of *Desmarestia dudresnayi* underwent three translocations of U-SDR genes into the V-SDR. Additionally, a fourth translocation event happened between the U-SDR and an autosome (chr_04).

**Extended Data Table 1 | General characteristics of the genomes and sex chromosomes in representative brown algal species and an outgroup (*Schizocladia ischiensis*)**

| Sexual system | Dioicous (V) | | | | | | Dioicous (U) | | Dioecious | Monoicous | | Unknown |
|---|---|---|---|---|---|---|---|---|---|---|---|---|
| Species | *E. sp. 7* | *E.crou* | *S.prom* | *U. pinn* | *D. herb* | *D. dich* | *E. sp. 7* | *D. herb* | *F. serr* | *C. lin* | *D. dud* | *S. isch* |
| Genome size (Mbp) | 200.170 | 218.478 | 193.199 | 511.280 | 430.876 | 851.153 | 197.371 | 484.711 | 1 091.539 | 214.613 | 425.034 | 194.512 |
| Chrom-level scaffolds | 27 | 27 | 27 | 30 | 32 | 31 | 27 | 32 | 33 | NA | 32 | NA |
| Total n. scaffolds | 33 | 175 | 451 | 114 | 935 | 1467 | 46 | 1756 | 3,805 | 217 | 823 | 130 |
| N50 (bp) | 6 893 608 | 7 597 912 | 6 781 292 | 16 510 065 | 13 135 153 | 26 426 000 | 6 914 137 | 12 441 740 | 17 863 197 | 2 249 057 | 13 227 088 | 2 524 267 |
| Sex chrom. or U/V-homolog | chr_13 | chr_13 | chr_13 | HiC_scaffold_23 | chr_03 | chr_02 | chr_13 (Ec25_SDR_F) | chr_03 | LG15 | C-linearis_contig12 | chr_03 | NA |
| Sex chrom. or U/V-homolog size (bp) | 7 072 209 | 9 658 235 | 14 770 496 | 27 543 478 | 17 928 803 | 42 672 188 | 7 248 464 | 20 286 227 | 19 924 529 | 3 613 932 | 14 536 050 | NA |
| SDR length (bp) | 923 344 | 1 084 112 | 1 363 928 | 13 409 094 | 6 568 004 | 17 412 526 | 1 551 053 | 8 626 479 | NA | NA | NA | NA |
| N. genes in SDR | 18 | 18 | 17 | 48 | 30 (+ 20 viral-derived genes) | 52 | 18 | 30 | NA | NA | NA | NA |
| N. genes PAR | 421 | 519 | 904 | 306 | 229 | 451 | 421 | 229 | NA | NA | NA | NA |
| Gamy | anisogamous | anisogamous | anisogamous | oogamous | oogamous | oogamous | anisogamous | oogamous | NA | NA | NA | NA |
| CDS density SDR | 2.96% | NA | 2.31% | 0.51% | 0.99% | 0.28% | 1.75% | 0.81% | NA | NA | NA | NA |
| CDS density PAR or U/V-homolog | 8.50% | NA | 6.69% | 7.35% | 1.98% | 1.70% | 8.90% | 1.98% | 2.07% | NA | 2.83% | NA |
| CDS density autosomes | 14.97% | NA | 15.94% | 13.51% | 5.17% | 2.95% | 14.97% | 5.17% | 2.28% | NA | 9.27% | NA |
| Repeat density SDR | 60.64% | NA | 64.29% | 86.94% | 83.82% | 89.19% | 68.88% | 84.53% | NA | NA | NA | NA |
| Repeat density PAR or U/V homolog | 44.31% | NA | 40.19% | 72.51% | 72.92% | 80.88% | 41.46% | 71.76% | 74.26% | NA | 74.68% | NA |
| Repeat density autosomes | 30.12% | NA | 18.92% | 59.24% | 61.53% | 72.10% | 29.38% | 60.97% | 75.16% | NA | 57.63% | NA |
| Genome source | Liu et al.[18] | This study | This study | Shan et al.[19] | This study | This study | Liu et al.[18] | This study | This study | Denoeud et al.[17] | This study | Denoeud et al.[17] |

Note: Note that *Fucus serratus* (F. serr) has a chromosome (LG15) that is homologous to the ancestral U/V sex chromosome because it contains several genes present in the ancestral U/V-SDR. However, none of these genes are sex-linked in *F. serratus* (see text for details). Note that the dioecious species *F. serratus* has male and female (diploid) sexes and no gametophyte generation, that is, an animal-like life cycle. The CDS and repeat densities of the U/V-homologues were placed in the same rows as the PAR values of the dioicous species.

# Reporting Summary

## Statistics

For all statistical analyses, confirm that the following items are present in the figure legend, table legend, main text, or Methods section.

| n/a | Confirmed | |
|---|---|---|
| ☐ | ☒ | The exact sample size (*n*) for each experimental group/condition, given as a discrete number and unit of measurement |
| ☐ | ☒ | A statement on whether measurements were taken from distinct samples or whether the same sample was measured repeatedly |
| ☐ | ☒ | The statistical test(s) used AND whether they are one- or two-sided<br>*Only common tests should be described solely by name; describe more complex techniques in the Methods section.* |
| ☒ | ☐ | A description of all covariates tested |
| ☐ | ☒ | A description of any assumptions or corrections, such as tests of normality and adjustment for multiple comparisons |
| ☐ | ☒ | A full description of the statistical parameters including central tendency (e.g. means) or other basic estimates (e.g. regression coefficient) AND variation (e.g. standard deviation) or associated estimates of uncertainty (e.g. confidence intervals) |
| ☐ | ☒ | For null hypothesis testing, the test statistic (e.g. *F*, *t*, *r*) with confidence intervals, effect sizes, degrees of freedom and *P* value noted<br>*Give P values as exact values whenever suitable.* |
| ☒ | ☐ | For Bayesian analysis, information on the choice of priors and Markov chain Monte Carlo settings |
| ☒ | ☐ | For hierarchical and complex designs, identification of the appropriate level for tests and full reporting of outcomes |
| ☐ | ☒ | Estimates of effect sizes (e.g. Cohen's *d*, Pearson's *r*), indicating how they were calculated |

*Our web collection on statistics for biologists contains articles on many of the points above.*

## Software and code

Policy information about availability of computer code

| Data collection | details are provided in the method section of the manuscript |
|---|---|
| Data analysis | details on data analysis are provided in the methods section of the manuscript |

For manuscripts utilizing custom algorithms or software that are central to the research but not yet described in published literature, software must be made available to editors and reviewers. We strongly encourage code deposition in a community repository (e.g. GitHub). See the Nature Portfolio guidelines for submitting code & software for further information.

## Data

Policy information about availability of data

All manuscripts must include a data availability statement. This statement should provide the following information, where applicable:
- Accession codes, unique identifiers, or web links for publicly available datasets
- A description of any restrictions on data availability
- For clinical datasets or third party data, please ensure that the statement adheres to our policy

All data and accession codes are provided in the supplemental tables of the manuscript. There are no restrictions on data availability

# Research involving human participants, their data, or biological material

Policy information about studies with human participants or human data. See also policy information about sex, gender (identity/presentation), and sexual orientation and race, ethnicity and racism.

| | |
|---|---|
| Reporting on sex and gender | *Use the terms sex (biological attribute) and gender (shaped by social and cultural circumstances) carefully in order to avoid confusing both terms. Indicate if findings apply to only one sex or gender; describe whether sex and gender were considered in study design; whether sex and/or gender was determined based on self-reporting or assigned and methods used.*<br>*Provide in the source data disaggregated sex and gender data, where this information has been collected, and if consent has been obtained for sharing of individual-level data; provide overall numbers in this Reporting Summary. Please state if this information has not been collected.*<br>*Report sex- and gender-based analyses where performed, justify reasons for lack of sex- and gender-based analysis.* |
| Reporting on race, ethnicity, or other socially relevant groupings | *Please specify the socially constructed or socially relevant categorization variable(s) used in your manuscript and explain why they were used. Please note that such variables should not be used as proxies for other socially constructed/relevant variables (for example, race or ethnicity should not be used as a proxy for socioeconomic status).*<br>*Provide clear definitions of the relevant terms used, how they were provided (by the participants/respondents, the researchers, or third parties), and the method(s) used to classify people into the different categories (e.g. self-report, census or administrative data, social media data, etc.)*<br>*Please provide details about how you controlled for confounding variables in your analyses.* |
| Population characteristics | *Describe the covariate-relevant population characteristics of the human research participants (e.g. age, genotypic information, past and current diagnosis and treatment categories). If you filled out the behavioural & social sciences study design questions and have nothing to add here, write "See above."* |
| Recruitment | *Describe how participants were recruited. Outline any potential self-selection bias or other biases that may be present and how these are likely to impact results.* |
| Ethics oversight | *Identify the organization(s) that approved the study protocol.* |

Note that full information on the approval of the study protocol must also be provided in the manuscript.

# Field-specific reporting

Please select the one below that is the best fit for your research. If you are not sure, read the appropriate sections before making your selection.

☐ Life sciences  ☐ Behavioural & social sciences  ☒ Ecological, evolutionary & environmental sciences

For a reference copy of the document with all sections, see nature.com/documents/nr-reporting-summary-flat.pdf

# Ecological, evolutionary & environmental sciences study design

All studies must disclose on these points even when the disclosure is negative.

| | |
|---|---|
| Study description | We identified the sex chromosomes and sex determining regions of several brown algal species represenitng the full phylogeny and level of morphological complexity across the lineage. |
| Research sample | brown algal lines |
| Sampling strategy | samples for each brown algal species used were grown in the laboratory in separate petri dishes as haploids. Sampeles were obtained from RCC (culture collection of Roscoff) and all details of codes and origin are provided in the manuscript. |
| Data collection | na |
| Timing and spatial scale | na |
| Data exclusions | no data was excluded |
| Reproducibility | for expression analysis the experiments were performed at least in triplicate |
| Randomization | samples were grown in the culture room in a randomized fashion. |
| Blinding | n.a |

Did the study involve field work?  ☐ Yes  ☒ No

# Reporting for specific materials, systems and methods

We require information from authors about some types of materials, experimental systems and methods used in many studies. Here, indicate whether each material, system or method listed is relevant to your study. If you are not sure if a list item applies to your research, read the appropriate section before selecting a response.

## Materials & experimental systems

| n/a | Involved in the study |
|-----|----------------------|
| ☒ | ☐ Antibodies |
| ☒ | ☐ Eukaryotic cell lines |
| ☒ | ☐ Palaeontology and archaeology |
| ☒ | ☐ Animals and other organisms |
| ☒ | ☐ Clinical data |
| ☒ | ☐ Dual use research of concern |
| ☒ | ☐ Plants |

## Methods

| n/a | Involved in the study |
|-----|----------------------|
| ☒ | ☐ ChIP-seq |
| ☒ | ☐ Flow cytometry |
| ☒ | ☐ MRI-based neuroimaging |

## Plants

Seed stocks

*Report on the source of all seed stocks or other plant material used. If applicable, state the seed stock centre and catalogue number. If plant specimens were collected from the field, describe the collection location, date and sampling procedures.*

Novel plant genotypes

*Describe the methods by which all novel plant genotypes were produced. This includes those generated by transgenic approaches, gene editing, chemical/radiation-based mutagenesis and hybridization. For transgenic lines, describe the transformation method, the number of independent lines analyzed and the generation upon which experiments were performed. For gene-edited lines, describe the editor used, the endogenous sequence targeted for editing, the targeting guide RNA sequence (if applicable) and how the editor was applied.*

Authentication

*Describe any authentication procedures for each seed stock used or novel genotype generated. Describe any experiments used to assess the effect of a mutation and, where applicable, how potential secondary effects (e.g. second site T-DNA insertions, mosiacism, off-target gene editing) were examined.*

