## [Peer Review File · Nature Ecology & Evolution]

Origin and evolutionary trajectories of brown algal sex chromosomes

Corresponding Author: Dr Susana Coelho

This manuscript has been previously reviewed at another journal. This document only contains information relating to versions considered at Nature Ecology & Evolution.

Version 0:

Decision Letter:

30th September 2024

Dear Susana,

Thank you for submitting your Article entitled "Origin and evolutionary trajectories of brown algal sex chromosomes" for consideration. Following our conversation last week and after careful consideration and discussion with my editorial colleagues, we have decided that, while we find the study of potential interest for Nature Ecology & Evolution, we cannot consider it for publication in its current format. Nevertheless, we would be interested in considering a streamlined version of the study that is no longer than 4000 words and better emphasizes the novel insights of each section.

Link Redacted

Yours sincerely,

[Redacted]

Version 1:

Decision Letter:

7th November 2024

Dear Susana,

Your Article, "Origin and evolutionary trajectories of brown algal sex chromosomes" has now been seen by two experts in sex chromosomes evolution. You will see from their comments copied below that while they find your work of considerable potential interest, they have raised quite substantial concerns that must be addressed. In light of these comments, we cannot accept the manuscript for publication, but would be open to consider a revised version that addresses these serious concerns.

We hope you will find the reviewers' comments useful as you decide how to proceed. If you wish to submit a substantially

revised manuscript, please bear in mind that we will be reluctant to approach the reviewers again in the absence of major revisions.

We are particularly concerned that the reviewers think that you overstate the significance of the findings or support for the conclusions. We encourage you to take a step back and think carefully about which questions can be answered with your system and focus your paper only on those.

If you choose to revise your manuscript taking into account all reviewer and editor comments, please highlight all changes in the manuscript text file in Microsoft Word format.

* Include a "Response to reviewers" document detailing, point-by-point, how you addressed each referee comment. If no action was taken to address a point, you must provide a compelling argument. This response will be sent back to the referees along with the revised manuscript.

* If you have not done so already we suggest that you begin to revise your manuscript so that it conforms to our Article format instructions at <http://www.nature.com/natecolevol/info/final-submission>. Refer also to any guidelines provided in this letter.

Link Redacted

If you wish to submit a suitably revised manuscript we would hope to receive it within 6 months. If you cannot send it within this time, please let us know. We will be happy to consider your revision so long as nothing similar has been accepted for publication at Nature Ecology & Evolution or published elsewhere.

Nature Ecology & Evolution is committed to improving transparency in authorship. As part of our efforts in this direction, we are now requesting that all authors identified as 'corresponding author' on published papers create and link their Open Researcher and Contributor Identifier (ORCID) with their account on the Manuscript Tracking System (MTS), prior to acceptance. This applies to primary research papers only. ORCID helps the scientific community achieve unambiguous attribution of all scholarly contributions. You can create and link your ORCID from the home page of the MTS by clicking on 'Modify my Springer Nature account'. For more information please visit www.springernature.com/orcid.

Thank you for the opportunity to review your work.

Yours sincerely,

[Redacted]

Reviewers' comments:

Reviewer #1 (Remarks to the Author):

General comments

This ms describes some results concerning interesting questions about sex chromosome evolution. The brown algal haploid sex chromosome system is problematic for asking questions about some aspects of sex chromosome evolution (including genetic degeneration and "evolutionary strata, see below), but is excellent for testing the idea that sex-linked regions will show an over-representation of genes with functions in sex determination and the development of male and female structures. Unsurprisingly, therefore, my comments on the problematic aspects are somewhat negative, but more positive about the latter questions. Considerable revision is needed to communicate clearly and accurately, and, most importantly, to clearly describe evidence for the conclusions (or to make clear to readers when these are tentative, and how they can be further studied in the future, if possible).

The main conclusions are the following, not all of which is clearly explained in the present ms:

1. Chromosome homologies can, to some extent, be detected in the different brown algal species studied (Figure 1). As one would expect, homologies are clearest for species that are related in the tree, but, if I understand correctly, one monoicous species, *C. linearis*, is polyploid (perhaps with 2 whole genome duplications) and its chromosomes have many-to-one relationships with those of other species, though 2 have red lines which indicate syntenic blocks with the V sex chromosomes of other species. Another monoicous species, *Desmarestia dudresnaya*, has syntenic blocks with the V sex chromosomes of the other *Desmarestia* species, which is dioicous.
2. All the extant brown algal species studied are stated to share the same ancestral SDR, on the homologous chromosome in all the species, and there is a single male-determining factor, MIN, within all of these SDRs, and therefore probably in the ancestral V-SDR (but see my caveat about a diploid species below). It is not clear whether these are all SDRs, in the sense of controlling sex-determination.
3. Parts of the chromosomes inferred to carry the SDRs of the dioicous species show clear evidence that they do not recombine (usually in the middle region), and that these regions are flanked at both chromosome ends by recombining PARs. This is interesting, as the sex chromosomes of haploid plants (bryophytes) have not been found to have detectable PARs in the genome sequences (though genetic mapping detected PARs in a moss, *C. purpureus*, suggesting that the genome sequencing missed them, for some reason). The difference suggests that those PARs may be smaller than in the brown algae.
4. Figure 2 shows that the SDRs of the 2 *Ectocarpus* species sequenced, and those of the 2 *Desmarestia* species (one of which is dioicous) are in inverted orders in the U and V, with several further rearrangements. It seems unlikely that these inversions were the cause of the lack of recombination (see below).
5. Large parts of the text describe results about gene numbers in the SDRs, on which I comment below.

The analysis infers that there was a common ancestral V-SDR, and that it probably also carried six genes other than MIN. If I understood correctly, these are V gametologs of genes that were also carried on the ancestral U chromosome. Interestingly, all 7 genes appear likely to function in sex determination processes. If the results of this analysis can be considered reliable, this is interesting, as it relates to testing the prediction that sex-linked regions should show an over-representation of genes with functions in sex determination and the development of male and female structures. Probably Bull's predictions should be cited here, and the prediction explained before the data are described, but this work is cited only in a bit of unclear text and not in clear relation to this question.

Other conclusions are either harder to understand as described, or are not evidently reliable or firm conclusions, as detailed below in my comments on each section. It is often unclear which are speculations, or simply consistent with a conclusion, and, among those, which have excluded alternative possibilities, as only the latter can properly be termed conclusions.

There is no need to use a lot of words with emotional connotations, suggesting that the findings are very interesting. The results are indeed interesting, and there is no need to 'advertise' this. Please avoid such writing and also unnecessary text with the same aim, such as the first sentence "Sexual reproduction, present in almost all eukaryotes, allows species to survive environmental challenges by increasing genetic variation". Words like "exploit" are also unsuitable in scientific writing, and (luckily) are not needed, and words like "dynamic" and "insights" are vague and communicate no clear meaning to readers. Other vague words need to be clarified. These are minor comments, but the vagueness is a general problem for understanding the results. Such terms include "gene shuffling", "drive" (and similar words, when a more explicit one is required). Similarly, terminology such as "gene cradle pattern" is obscure. It may be a term used in the authors' lab, but outsiders will need clearer words, and definitions. It is best to avoid coining new terms like this (and "gene nurseries") that require defining, and to use standard word as far as possible. Many readers will not be native English speakers, and it is more considerate to restrict wording to standard scientific and other terms. Finally, it is extremely confusing to use the word "linked" when writing about a genetical topic, when one does not mean genetically linked. Another word should be used if the meaning is something like "is a consequence of" or "is correlated with" (in which case, a statistical test is needed). Please also see the annotated pdf file

The background section

The introductory paragraph fails to distinguish between diploid and haploid-dominant organisms, although the study deals with the latter. Several statements (such as "heterogametic sex") apply only in diploids.

Assuming that recombination has been suppressed (see comments in the next section on strata formation) were inversions involved?

It is interesting to test the widely repeated claim that inversions are involved in the suppression of recombination that characterises many sex-determining regions. The conclusion is stated that "that the U/V sex chromosomes of the brown algae emerged between 450 and 224 Mya, via an inversion that suppressed recombination". The ms states that differences in the sizes of certain regions (which the text does not make clear enough) "coincide with extensive structural rearrangements, particularly inversions, even among closely related taxa. Thus, structural rearrangements may have caused the recombination suppression event around the MIN gene, creating the brown algae U/V sex-linked regions". However, this is not really clearly demonstrated. Readers should be told precisely how they can see that size differences of clearly specified regions indeed coincide with inversions, with a statistical test to show that these changes are associated. Consistency with the idea that inversions may lead to suppressed recombination between sex chromosomes is not evidence that they did indeed do so (as the text states), as inversions are also predicted to arise and spread after recombination becomes suppressed (the argument is very clearly explained in this paper, for a different non-recombining situation: Sun et al. 2017 Large-scale suppression of recombination predates genomic rearrangements in *Neurospora tetrasperma*. *Nature Communications* 8: 1140. doi: 10.1038/s41467-017-01317-6).

If inversions can be inferred to have been involved, or evidence exists that this is very likely, then it should be mentioned whether these can be inferred to have affected the U or V, or both haplotypes (or whether no suitable outgroups are available to make any such inferences). Some models for recombination suppression predict that inversions should affect only Y or W chromosomes. The authors should also consider which of the models can apply in haploids, as not all of them can do so.

This text also needs revising. "Together our results indicate that the U/V sex chromosomes of the brown algae emerged between 450 and 224 Mya, via an inversion that suppressed recombination in a locus that contained the male-determining MIN factor". Here readers need to be told why suppressed recombination might evolve in a locus, or maybe the word "locus" needs to be amplified to say something like "suppressed recombination in the genome region containing the male-determining MIN factor (henceforth "male-determining locus")".

Inferring SDR expansions and gene gain versus degradation, and evolutionary strata

The evolution of U- or V-specific genes is interesting, and might reflect either U/V-SDR divergence or loss from one haplotype. The meaning of the phrase evolved "after U/V-SDR divergence", in the authors' wording "U- or V-specific genes ... either evolved after U/V-SDR divergence..." is not clear to me, as it could either mean that genes duplicated into the U or the V haplotype after recombination ceased, or that U and V sequences of a gametolog pair diverged so much that their allelic nature is no longer detectable, as is the case for some human YX pairs).

The text reads as if there is a contradiction: "Despite differences in SDR gene numbers" versus "U and V-SDR gene counts coincide in both species". Can this be clarified?

The difficulty in detecting strata in U/V systems is because neither of these fully sex-linked regions recombines. In XY diploid systems, for example, the X recombines, and this largely prevents rearrangements, so that the X arrangement can be assumed to be similar to the ancestral one, even if due to gene movements and/or chromosome rearrangements have disrupted colinearity between different species. The ms does not explain the difficulty correctly.

Another difficulty arises because Ks analysis requires many genes to detect strata with different Ks values (because the variance of these estimates is large), but in these species the numbers of genes are very small (Ectocarpus sp. 7 and D. herbacea), and the inference of at least four strata in the latter seems optimistic, as does the conclusion that its U/V-SDR reflects an expansion. It is therefore not clear that one can firmly conclude that the results "indicate that SDR expansions in brown algae are driven by nested inversions". This needs to be clarified, and it should also be made clear what process is suggested as having led to the inferred larger SDR size in this species. It is also important to distinguish between larger SDR size due to accumulation of repetitive sequences, and changes in its boundaries so that genes flanking it in some species (meaning PAR genes) are within the SDR in others.

A related interesting question that was studied is when size differences reflect expansions of the SDR, versus reductions in sizes, and evaluating the possibility of gene loss, which might have reduced SDR sizes, in some cases.

As just mentioned, it is important/valuable to describe the density patterns of repetitive sequences, to test the prediction that these should accumulate when a genome region ceases to recombine. However, the ms deal with this quite sketchily.

Do the sex-linked regions show an over-representation of genes with functions in sex determination and the development of male and female structures?

The authors inference that the ancestral brown algal V-SDR carried seven genes, including the male-determining MIN factor, plus six V gametologs of genes also carried on the U chromosome, and that these genes are all likely to be related to sex determination processes. This would be a valuable conclusion, but needs a statistical test.

Large parts of the text discuss different gene contents of SDRs, and ideas about new genes, and the reasons for the conclusions are not described clearly enough. Specifically, changes such as movement of the PAR boundaries are not clearly distinguished from the ideas about new genes.

Near the end of the text, it says "sexual dimorphism may drive SDR expansion". I think this means that SDR expansion via movement of the PAR boundaries may have occurred as differences in the morphology of the male and female gametophytes evolved, suggesting the possible involvement of trade-offs between male and female functions, leading to sexually antagonistic polymorphisms in partially sex-linked genes and selecting for closer linkage to an existing sex-determining region. The text mentions We expanding this model to UV systems by demonstrating that derived lineages with pronounced gamete sexual dimorphism (oogamy) tend to have changed U/V-SDR boundaries in brown algae. This suggestion seems plausible, but should be more clearly worked out, and the absence of a correlation between levels of sex-biased expression and sexual dimorphism in brown algae is surprising. Statements like "sexually antagonistic selection may drive gene movements into the SDR" are not clear enough (especially if the scenario envisaged does not actually invoke gene movements, but simply changes in the PAR boundaries). This section can be shortened, as essentially the same argument was already made earlier.

The overall conclusion concerning SDR gene gains and losses is not clear. Page 15 mentions "streamlined gene content of brown algae U/V-SDRs", and suggests, plausibly, that (like other sex-determining regions) genes within the region could regulate an autosomal effector gene network controlling sexual dimorphism, with most sex-biased genes influencing the physiology or vegetative development of gametophytes, though it would be surprising (and interesting) if vegetative

development differed greatly between gametophytes of the two sexes. The observed substantial sex-biased gene expression in mature gametophytes, which presumably carry sex organs, is less surprising, and may be similar in haploid plants, but worth reporting. It is not helpful to cite the case of the moss, *C. purpureus*, whose sex chromosomes carry thousands of genes, probably due to fusions with at least two autosomes.

As predicted (probably Bull's predictions should be cited here), all appear intact (non-degenerated) in *Ectocarpus* sp. 7, and are presumably expressed at appropriate developmental times (it should be mentioned that this is tests a bit later in the ms), but in *D. herbacea* one (a casein kinase) was lost from the U-SDR, and one was not found, presumably meaning not found in the genome of either sex. These ancestral V-SDR genes remain present in the sex-homolog of the species that have lost their UV system. Here, it should be explained whether those species still undergo sexual reproduction, for instance having gametophytes that produce archegonia and antheridia (or the relevant brown algal terminology). If so, this is not really remarkable, but it nevertheless suggests that these genes are important for pathways in sex development (though of course not sex determination, as stated on page 5).

It is less clear whether the ancestral brown algal SDRs carried other genes that did not have such functions, and have been lost from these regions. Similarly, it may be interesting to test whether any regions that have become part of a lineage's SDR by changes in the PAR boundary included any genes with functions unrelated to those of the genes just mentioned, and whether these genes have been retained or lost (as might be predicted). If lost, how did this occur? For instance, have any such genes become duplicated to autosomal regions and then lost from SDRs? The text a bit later suggests that such things are also found (in my slightly shorter wording "expression of many U/V-SDR genes is not confined to fertile gametophytes. Thus the SDRs contains both genes involved in sex determination and gametophyte fertility but also genes involved in other functions"). In my opinion, these different results should be better integrated into an overall conclusion.

I believe that Bull's predictions may suggest that the evolution of sexual dimorphism of the gametophytes might evolve through the incorporation of new genes into species' SDRs. Intriguingly, the study found a strong correlation between the V-SDR size and the extent of sexual dimorphism, with anisogamous species retaining the same V-SDR genes as inferred in the ancestral state (with very few gene gains), consistent with this being the V-SDR ancestral state. It would be helpful to mention this when describing the ancestral state inference, as it would seem to validate that inference. Oogamy is found in different lineages, and this is accompanied by convergent gain of ATP-dependent RNA helicase in all cases, and of various new V-SDR genes in different lineages, suggesting possible independent evolution. Autosomal sex-biased gene (SBG) expression was not found to correlate with sexual dimorphism levels.

PAR genes should not be involved in the sex-determination process, and are predicted to be less likely to have reproductive functions, though ones closely linked to the boundary with the SDR might have such functions, and might establish sexually antagonistic polymorphisms that could select for closer linkage to the actual sex-determining gene(s). It is thus interesting to use *D. herbacea*'s expanded SDR (which includes genes in the PAR of *Ectocarpus* sp. 7) to test this. At this point, the text finally makes clear that the expansion involved 2 events, one in which four genes from one *Ectocarpus* sp. 7 PAR (PAR1) stopped recombining, and a second affecting 13 genes in its PAR2, forming two younger SDR strata in *D. herbacea*. These results would be more helpful if explained when the strata are first described. Of these 17 genes, only one gene remained partially sex-linked *D. herbacea*, and most (14?) were retained and became SDR gametologs, supporting the conclusion that U/V degeneration is minor in the time since they formed (however, it is not stated in the text how large Ks values are for these strata, and Fig. 2 shows Ks values on a y axis scale that includes the shared, ancient stratum, so one cannot see the relevant information), while two others are located on this species' autosomes, which suggests that some genes not involved in male or female functions may move off the UV pair, contrary to the authors' conclusions.

The finding that gene ages appear to be younger for "sex-chromosome-enriched" categories than their last common ancestors in the same order or broader taxonomic groups supports the idea that they became sex-linked independently in different species. The pattern is said to be consistent with a predicted retention of young genes with beneficial effects in the sporophyte stage, and to be lacking in species that lack heteromorphic generations (a concept that needs to be explained earlier).

A further complication in inferring the history of brown algal (and other) U/V chromosomes is the possibility of de novo evolution of new genes from previously non-coding regions, and the text on p. 16 mentions that the authors infer that such changes have happened, or at least that "rapid sequence divergence" creates the appearance that it has happened. The wording implies that the authors conclude that new genes evolve from non-coding regions more often in these SDRs than in other genome regions (or possibly they mean in other sex chromosomes), but again this is not supported by a test showing an enrichment of new genes in these regions. Such a conclusion might be interesting, but I suspect that it would require an entire study devoted to testing it explicitly, excluding different possible explanations. It seems premature to discuss what mechanism might be involved. The text speculates that, as U/V chromosomes are enriched in heterochromatic marks, which may involve TE repression, this may contribute to heterochromatic regions' higher mutation rates than other genome regions, which is plausible. It then mentions that mutation rates in U/Vs are high (a reference is needed), and states that this "could facilitate the emergence of de novo loci", which seems too vague, or that their high density of DNA transposons "could promote the co-option of their regulatory motifs and enable de novo transcript birth, as seen in *Drosophila*", and that these new genes "could then be retained in the sex chromosomes by 'generation-antagonistic selection'". These are interesting speculations, but solid evidence is needed that new genes are actually appearing at an unexpected rate. The only evidence described is comparative, ("the gene cradle pattern is unique to U/V systems and gradually disappears when these systems are lost, as in monoicy or diploid sexual species"), but no adequate comparative analysis is described, and other U/V systems are stated to show the same pattern ("this process is pervasive across distant, independently evolved eukaryotic kingdoms"), but no references are given. The wording that "a unique interplay between complex life cycles, heterochromatic

landscape, DNA transposons, and higher mutation rates ... may drive de novo gene birth in UV chromosomes”) relies completely on the word “may” to tell readers that this is speculative, and I feel that this is not clear enough.

Repetitive sequence densities and gene densities.

The Circos plots in Fig. 3 are extremely difficult to read, and I could not find an explanation of some symbols, such as some microscopic ones near the chromosome names. Track 1 shows gene “ages”, which need to be defined. Track 2 shows gene and repeat densities, but it is not possible to understand whether these show any patterns when both are in a single track. Beyond the impression that the gene densities are fairly uniform (on a scale that may not allow any differences to be seen, though they may be lower on the sex chromosomes than elsewhere), and that repeat densities show no obvious pattern other than being high on the sex chromosomes, at least in some cases (as expected if these are mostly non-recombining, though the high densities may extend to the PARs). It would be helpful to plot results for just the UV pair, or even just U or V chromosomes whose other pair member was not sequenced, as the repeat densities can provide valuable supporting evidence for the presence of non-recombining regions (even in monoicous species, as noted below). More detailed figures might also help clarify whether the PAR boundaries differ between the species, as I believe the authors think, versus SDRs gaining new genes in other ways. The lack of clarity about these differences is a major barrier to understanding these results, and the processes that may be involved in creating the situations observed (see comments above).

It is also unclear if these are mean densities for the U and V, or values (presumably in windows of some size) from just one of these. More explanation is needed, as it would be valuable to describe repetitive sequence densities, as these are predicted to be high in non-recombining regions, and can be used to detect such regions in the absence of genetic map results (as in these species). However, the ms deals only sketchily with these analyses.

To understand the “higher Ks values”, and the finding that these are “localized in the PARs”, readers need to be told what Ks is estimated between (mutation rates are mentioned, and this appears to refer to pairwise inter-species divergence between closely related species, but the species pairs are not specified), and what they are higher than.

Loss of U/V systems

Results are mentioned that “show that monoicy arose at least twice from a male ancestor that acquired female genes”, but readers are not told where these can be seen, or which species this refers to. It is valuable to report that previous conclusions may have been incorrect because in brown algae expression of male-biased genes tends to be tissue-specific versus broader expression of female-biased ones, possibly explaining why female and monoicous transcriptomes are similar.

It is plausible that monoicy evolved (not “emerged” in an unspecified manner) via a recombination event that provided essential female genes to a male genotype, as proposed for such a change in a liverwort and in *Volvox* species, but again it is not easy to understand what is meant by “a U-SDR-derived gene present in both monoicous species” (perhaps the authors mean that only a single gene was identified from the ancestral U that was inferred in the inferences already described). With only 2 monoicous species, it is also unclear whether this suggests its fundamental role in the female developmental pathway. However, one can infer that if monoicy can arise by a male genotype acquiring U-linked femaleness factor(s), this has important implications for sex-determination. Specifically, it shows that the MIN factor is not both essential for maleness and also excludes expression of female functions, since it is present in the genome of species that do express those functions. This seems to show that there must also be actively femaleness-promoting functions in the genomes of these organisms. These could, of course, be autosomal, but clear ideas about how these things might work are needed, including how a species expressing both sex functions could re-evolve dioicy.

If genes can move off the sex chromosomes, it is plausible that the change to monoicy might, as in other cases, result in loss of one sex chromosome, or possibly even both. An example in brown algae is worth reporting, adding to the cases already described. However, the results do not actually describe “genetic networks governing sex determination” being “rewired” during the transitions described. Again, therefore, the text, over-states the significance of the findings here.

It would be valuable to describe repetitive sequence densities in the retained SDRs and the other genome regions of the monoicous species, to assess how long these regions continue to be non-recombining regions (presumably, they can recombine in UU or VV sporophytes produced by monoicous gametophytes) and whether repetitive sequence densities soon decline to the genomic background level. However, the ms seems not to mention these questions, and unfortunately Figure 3 shows Circos plots only for dioicous species.

The evolution of a diploid XX/XY system in *Fucus serratus* is stated to be associated with the (permanent) transition to diploidy in the Fucales. Here, it is essential to make clear what evidence exists for genetic sex-determination in these species, and for male heterogamety. The term XX/XY system gives readers the impression that sex chromosomes have been seen in males and females, but this seems not to be the case, as it is stated that no sex chromosome (meaning sex-linked region) was detectable in *F. serratus*, which turns out to mean that the sequences of wild males and females in this study did not identify any sex-linked sequences, suggesting that its SDR is small and/or little differentiated.

At this point I was unsure of what was previously known about this species. Reading the earlier text, I found out that the male-determining gene MIN is present in V-SDRs of all the dioicous species, on the same (homologous) chromosome, including this diploid XY species.

It is not made clear enough whether the MIN gene, or sequences physically near it in *F. serratus*, have been tested for sex linkage (a straightforward matter), but presumably this gene is not sex linked in the species. This suggests that MIN may

simply be essential for male functions, and that a new system may have evolved in *Fucus*. A bit later in the text, this impression is strengthened, as this chromosome is called an 'ex'-sex chromosome, that has lost all the evolutionary vestiges of its past as a U/V chromosome. I suspect that I have not correctly understood the story about this species, and I feel that it can be explained more clearly, including the evidence for a monoicous intermediate.

The new data suggest that the "UV to XY transition" (which is not understood, and is thus potentially interesting, but appears not to have involved evolution of a new non-recombining sex-linked region, other than maybe a very small one) "involved an intermediate monoicous stage, not a direct shift from the U/V system" (I did not understand what the authors have in mind for a "direct shift"). Retention of an ancestral SDR certainly seems unlikely in *F. serratus*, as it has no close UV outgroup. The text on p. 13 suggests that the findings are "consistent with the hypothesis that a young sex chromosome in *F. serratus* arose following the transition to the new life cycle system. However, it is not explain why a monoicous intermediate is indicated, rather than an extension of the diploid stage, which might have reproduced vegetatively and then evolved a new genetic sex-determining system, leaving the MIN gene as a male-essential, but not, sex-determining, gene. Under this scenario, or one involving monoicy, it would still be necessary to explain how two sexes could evolve.

The final Discussion sections appear mainly to repeat the findings and interpretations, and are unnecessary, or could be greatly shortened. It might be better to integrate discussions into the earlier sections, avoiding repetition, and making the ideas clearer. This would allow space for evidence that is not currently included (as outlined above)

Established models of XX/XY and ZW/ZZ systems posit that recombination suppression between sex chromosomes explains Y or W degeneration^{60–62}. Differences in the evolutionary paths of UV compared to XX/XY and ZW/ZZ ones were predicted by Bull, and some of them have been documented in bryophytes²⁸. In brown algae, the U/V structure may undergo successive inversions and TE accumulation, inevitably reducing gene density (this is not remarkable, as the text says). Low gene density may also reflect gene loss by degeneration or movements out of the SDR. The main difference is that, in the haploid stage, deleterious mutations are more efficiently removed by selection than in diploid systems.

The connection with TE enrichment in the Ectocarpales sex-linked regions would be easier to explain in an expanded section on such accumulation. The authors speculate that the U/V-SDR acts as a source of DNA transposons, and the SDRs subsequently expand to the PARs through local movements of such TEs (in larger genomes, this signal is diluted by the increased colonization of LTR elements). They also suggest that the SDRs therefore form 'gene cradles' or 'gene nurseries', perhaps through U/V chromosomes being enriched in heterochromatin, and therefore possibly having high mutation rates (this repeats earlier discussion of this possibility). These ideas are not implausible, but they are presented vaguely, and there is no substantial evidence supporting them, as already noted.

Minor comments

The writing is not always clear enough. Examples include the following:

In the text "Many genes located in the pseudoautosomal region (PAR) of Ectocarpales have moved into the V-SDR of *U. pinnatifida*, *D. herbacea* and *D. dichotoma*", does "have moved into" simply mean "are now within"? And what mechanism causes these movements? Is it through rearrangements, or do the PAR boundaries change?

Genome assemblies are stated to be better quality for males than females, but no details or references are given. In this sentence, it is unclear what is meant by "assuming parallel U/V-SDR evolution". As one of the references cited in Bull's work, possibly this means that processes occurring on the U are assumed to be very similar to those on the V, but this should be made clearer. However, the case of the monoicous liverwort species *Ricciocarpos natans*, another haploid organism, shows that this cannot necessarily be assumed, as the U chromosome has been lost and a V retained. Possibly this assumption is plausible for brown algae, but the meaning of the statement should be explicitly clear, and the assumption explicitly justified. The exact aspects of Figs. 2 E and F that tell us this should also be explicitly pointed out. I think the authors are probably referring to the inferred numbers in Fig. 2E. It is not clear how often genes are gained or lost, and whether these changes can be reliably inferred over the large evolutionary times in the x axis of this figure. It would also be good to discuss whether these are thought to involve individual genes or not; for instance, how many events were involved in gains of > 20 genes, and are single genes gained by duplications or movements from other genome regions, whereas gains of multiple genes in an event occur by a different mechanism (e.g. an inversion changing a PAR boundary, as the text suggests when it says "independently-acquired gametologs in *D. herbacea*")? Again, these points should be communicated more clearly.

I think that genes with sex-limited expression is meant when "sex-limited genes" is used. If so, this should be corrected. In this paragraph, the meaning is not clear of "newly acquired genes on the SDR"; as these "had similar expression levels to their autosomal counterparts, presumably it means genes gained by duplications or movements, but (as already noted) these changes need to be distinguished from other kinds of changes that can lead to a gene becoming sex-linked. The anthropomorphic word "co-option" is also unclear in this text, and an evolutionarily correct form of words should be used.

I could not find a definition of "TGRs".

Reviewer #2 (Remarks to the Author):

Barrera-Redondo et al. generate high quality for several species of brown algae, allowing them to follow the birth and

evolutionary trajectory of UV sex chromosomes in this clade. Key findings include:

- the UV system is ancestral to the clade
- the non-recombining SDRs were expanded in several lineages, in particular those which have oogamy
- V sex chromosomes have high TE contents, mutations rates, and harbour an excess of young genes
- Transitions to other reproductive systems involved the cooption of V chromosomes

Overall this is a very impressive amount of work on a very exciting system, but I have some questions and comments on the analyses:

* How was the phylogeny inferred, and how reliable is it? This is key to the interpretation of the results and some information should be provided.

* The strata analysis is a bit confusing. Using only Ks to infer strata seems limited, given how noisy mutation rates can be. There then seems to be a combination of ks and phylogenetic distribution of sex-linkage, but the results are not clearly presented.

* Some claims do not seem completely supported. For instance, the idea that oogamy is associated with expansions of the PAR is exciting. But from the phylogeny, it seems more that a single branch (which happens to be not have oogamy) is different from the rest, which does not provide much power to make inferences. It does not help that this branch differs in other ways (e.g. genome size), further muddling the picture. I think that either stronger evidence needs to be provided or conclusions need toning down.

* I could not understand from the methods how the ancestral gene content, and the inference of gene losses, were done. I assume that genes were deemed to have been gained if they were sex-specific in one lineage but pseudoautosomal in the others? If we assume that the ancestral SDR was very large, and gene loss was so extensive that each lineage ended up with their own subset of these genes plus a few highly conserved ones, would you detect this as gene loss, or would some of these be considered gene gains? What did it take for a gene to be considered lost from an SDR?

* In the discussion of young genes and mutation rates, predictions and patterns for the PAR and SDR are not clearly distinguished, making it hard to interpret what could be driving these patterns.

* Results of statistical tests should be reported throughout the manuscript to support the conclusions, and not just in the methods and supplementary materials.

* I found the many small circular plots difficult to read. In general I thought the figures were beautiful but not always completely informative as to how the authors came to the conclusions in the text. Maybe again this was partly due to the fact that significant differences were not consistently illustrated.

* It was not clear from reading the text which genome assemblies were new and which were published. Some quality measures should also be provided for new assemblies.

* Similarly it would be helpful to have a (supplementary) figure showing male and female coverage of the SDRs (or the kmer equivalent).

Other / more detailed comments:

* Fig 1: several chromosomes appear inverted (e.g. between *Undaria* and *Fucus*). I think this simply reflects the fact that opposite strands were assembled in their respective genomes, and not a biological difference. Fixing this would make the picture clearer and help highlight true differences.

Also, in panels B and C, could you add a scale for Mbs?

We noticed that many genes located on the pseudoautosomal region (PAR) of Ectocarpales had moved into the V-SDR of *U. pinnatifida*, *D. herbacea* and *D. dichotoma*.

this is a funny way to put it, as it implies gene movement as the mechanism, rather than expansion of recombination suppression. Maybe something like "many genes located in the PAR of Ectocarpales are included in the V-SDR" would be clearer?

suppression event leading to the birth of U/V sex chromosomes occurred between the split of *S. ischiensis* and *D. dichotoma* around 450-224 Mya²⁰ (Fig. 1A).

 I don't think something can occur between the split of two lineages, which is one time point. "after the split"?

The evolution of the SDRs is driven by expansions and gene gain rather than gene degradation

 Is this meant to be "expansions of the non-recombining region"? I would make it explicit, otherwise it could mean gene family expansion through duplications.

Despite differences in SDR gene

numbers, both species show a similar ratio of gametologs (53-61%) and sex-specific genes (47-39%; Table S2).

 It would be helpful to provide numbers in the text, otherwise one must keep going to the supplementary materials to be

able to follow.

Note that U

and V-SDR gene counts coincide in both species (Table S2), supporting parallel evolution of U and V chromo-
some 23–25.

 I assume both species are *Ectocarpus* sp. 7 and *D. herbacea*, which seem to be the ones for which U chromosome assemblies are available? But maybe this could be more explicit.

Ks analysis showed that the U/V-SDRs of *Ectocarpus* sp. 7 have

mostly highly-divergent gametologs in two old strata ($K_s > 1$), with only two gametolog pairs in a recent stra-
tum ($K_s < 1$; Fig. 2B-C, Table S3). In contrast, *D. herbacea* exhibited at least four strata (Fig. 2B-C). The K_s values
were broadly consistent between orthologs in *Ectocarpus* sp. 7 and *D. herbacea* (Table S3).

 This seems like a strangely arbitrary way of classifying things. How did you determine the number of strata? Were any kind of statistics performed? Which of the genes are sex-linked in both clades and which were acquired in specific lineages? Could this be shown in fig 2C?

To examine SDR size dynamics, we reconstructed the ancestral state of SDR gene content. We

focused on the V chromosome (due to better male genome assembly quality, and assuming parallel U/V-SDR
evolution^{23–25}). This analysis revealed that brown algal V-SDR evolution is driven by lineage-specific expan-
sions, rather than reductions, from a common ancestor with few SDR genes (Fig. 2E-F).

 it would be very helpful to see a (supplementary) figure to 2F for all the genes in all the SDRs, not just the ones that were inferred to be ancestral.

Figure S6. Distribution and proportion of classified TEs across the chromosomes of six brown algal species. The sex V chromosome

509 or sex-homolog is highlighted in green. The residuals in the V sex chromosome were used to interpret the enrichment or depletion of

510 Unclassified (Unknown) repetitive elements for each species.

 I found these figures impossible to read and interpret. It may just be my tired eyes, but you maybe you could consider modifying them to more standard plots, and make sure the font size is large enough to be easily read?

Figure S7-S9: can you state what species were used for K_s calculations?

We observed an inverted pattern between V and U-SDR in *Ectocarpus* sp. 7 and *D. herbacea* (Fig. 2A-B), demonstrating that inversions may lead to suppressed recombination between sex chromosomes.

 Inverted pattern is not very clear; maybe something like "inverted gene order" would work?

Consistently, ancestral V-SDR genes were associated with higher K_s values, while independently-acquired gametologs in *D. herbacea* had lower K_s values (Table S2, Fig. 2C).

 I can't see which genes are ancestral vs acquired in fig 2C. Is this information there?

SDR genes remain in the sex-homolog of the species that have lost their UV system (Fig. 2F, Table S4), emphasizing their importance for pathways in sex determination even in absence of sex chromosomes.

 "sex chromosome homolog"?

Intriguingly, we found the V-SDR size to be strongly linked to the extent of sexual dimorphism (Fig. S2A). Spe-
cies with low sexual dimorphism (anisogamous) retained the ancestral V-SDR genes with very few gene gains,
suggesting they may represent the V-SDR ancestral state. Oogamous species each independently gained di-
verse V-SDR genes

 How many datapoints do you have? Can you run statistics on this? Otherwise I would move this to the discussion, as it seems rather speculative.

On a similar note, these species also apparently underwent genome expansions, which could correlate with structural rearrangements that expand the SDR.

Oogamy has been suggested to be the ancestral state in the brown algae¹⁵ or its crown radiation clade²⁰, but
our finding of independent V-SDR expansions positively associated with changes in the level of sexual dimor-
phism (Fig. 2E; Fig. S2A) supports instead the recurrent, independent evolution of oogamy in the brown algal
lineage.

 Again this seems rather speculative

However, we observed an enrichment of male-biased genes on the

sex chromosomes in all species except *D. dichotoma* (Fig. S5B).

 presumably not on the female sex chromosomes?

192 We previously proposed a theoretical model where generation-antagonistic selection drives TRG accumula-
193 tion, favoring the retention of young sporophyte-beneficial loci in the *Ectocarpus* sp. 7³³ UVs.

 Does the model make predictions for the SDR or the PAR (or both)?

 Similarly, can you separate the PAR and SDR in the violin plots in figure 3B? The presence of young genes seems to be driven by the PAR in some cases, but not others.

 Also in panel 3B: I assume that the numbers in parenthesis next to autosome / chromosome V are the number of the chromosome that they correspond to (but please specify this). But how were the autosomes picked? For example, in *D. dichotoma*, the sex chromosome looks extremely similar to autosomes 1 and 3 -- why show autosome 4?

Consistent with this model, sporophyte-biased genes are indeed enriched in the sex chromosomes of *Ectocarpus* sp. 7 and *U. pinnatifida*, but less so in *D. dichotoma* and *S. promiscuus* that lack heteromorphic generations^{15,31} and consequently no scope for generation-antagonism (Table S13).

 Again this seems like a small number of data points

If synonymous mutations behave neutrally^{36,37}, K_s is a proxy for mutation rates^{38,39}. Consistently, we found higher K_s values in the V sex chromosomes across all dioicous brown algae (Fig. 3A; Table S14), suggesting higher mutation rates relative to autosomes. Young gene enrichment and higher K_s values were localized in the PARs of the Ectocarpales (Fig. S7-S8), but this pattern extended to the entire sex chromosome in species with larger V-SDRs (Fig. S9-S11).

 I think that in a fully nonrecombining region, neutral mutations will necessarily be under the effect of linked selection, and will not behave neutrally.

Version 2:

Decision Letter:

28th March 2025

Dear Susana,

Your revised Article, "Origin and evolutionary trajectories of brown algal sex chromosomes" has now been seen by the same reviewers. You will see from their comments copied below that while they find that the paper improved in clarity, they still have significant concerns about the strength of conclusions.

Reviewer #2 requires much more information about how SDRs were determined, a central aspect of the study. More worryingly, Reviewer #1 points out several flaws with interpretation of results and conclusions that are not supported by the data. These concerns would be enough to reject the study for publication in our journal. However, given the interest in the data demonstrated by both reviewers and the detailed comments of Reviewer #2, we have decided to give you one last chance to revise the manuscript into something that communicates conclusions that reflect a fair interpretation of results and are well supported by the data.

If you choose to revise your manuscript taking into account all reviewer and editor comments, please highlight all changes in the manuscript text file in Microsoft Word format.

* Include a "Response to reviewers" document detailing, point-by-point, how you addressed each referee comment. If no action was taken to address a point, you must provide a compelling argument. This response will be sent back to the referees along with the revised manuscript.

* If you have not done so already we suggest that you begin to revise your manuscript so that it conforms to our Article format instructions at <http://www.nature.com/natecolevol/info/final-submission>. Refer also to any guidelines provided in this letter.

Link Redacted

If you wish to submit a suitably revised manuscript we would hope to receive it within 6 months. If you cannot send it within this time, please let us know. We will be happy to consider your revision so long as nothing similar has been accepted for publication at *Nature Ecology & Evolution* or published elsewhere.

Nature Ecology & Evolution is committed to improving transparency in authorship. As part of our efforts in this direction, we are now requesting that all authors identified as 'corresponding author' on published papers create and link their Open Researcher and Contributor Identifier (ORCID) with their account on the Manuscript Tracking System (MTS), prior to acceptance. This applies to primary research papers only. ORCID helps the scientific community achieve unambiguous attribution of all scholarly contributions. You can create and link your ORCID from the home page of the MTS by clicking on 'Modify my Springer Nature account'. For more information please visit www.springernature.com/orcid.

Thank you for the opportunity to review your work.

Yours sincerely,

[Redacted]

Reviewers' comments:

Reviewer #1 (Remarks to the Author):

This ms is now a bit easier to understand, and the material is organised better, although a number of things are still not explained as needed (as detailed in comments below). The new Table 1 is very helpful, and I thank the authors for providing this information in an accessible manner (my small problems with this table are noted below). Now that the main results are clearer, however, it seems clear that the emphasis is excessive on the idea that the sex-linked regions of these chromosomes have undergone changes like those in the sex chromosomes of some diploid species, involving new chromosome regions stopping recombining (and forming new "evolutionary strata", as summarized in Figure 6). Rather, the overall picture appears to be a remarkable conservation of the completely sex-linked region over a vast evolutionary time, with at most rather small numbers of genes having become completely sex-linked after an ancestral completely sex-linked first evolved. This is an important finding, as it may be evidence against one of the ideas that has been suggested as a mechanism for the loss of recombination between members of sex chromosome pairs (Jeffries et al. 2021). This is cited as ref. 61, but not related to the findings.

I next outline and comment on the main interesting results. Please note that some or most of the information I failed to find is probably actually present, but unless it is explained clearly in relation to the obvious questions that readers may ask as they read the text, it is very hard to understand the answers, given the mass of very condensed information. The organization of points below is intended to help the authors think about how to present the chief findings and their implications for sex chromosome evolution.

1. SDR identification, homology, sizes and the role of repetitive sequences

As Table 1 shows, 6 dioicous species were studied (haploids thought to have separate male and female gametophytes), two with both the U- and V- linked regions being ascertained, and four with just the V, plus 2 monoicous species and one species whose sex system is not known, and one species termed dioecious because its diploid sporophyte stage has separate males and females (maybe also the gametophyte stage — this is not made clear). The new haploid dioicous species for which both male and female gametophytes were sequenced, *Desmarestia herbacea*, indeed proves to have a plausible sex-determining region, based on finding male- or female-specific variants that identify inferred male- and female-determining fully sex-linked regions (respectively termed "V-" and "U-" SDRs or "V- and U-linked regions"). The gene content of the chromosome corresponds with that carrying the U- and V-linked regions in the previously sequenced *Ectocarpus* species 7, even though these are very distant relatives, showing that this chromosome homolog has carried the SDRs for a long time. This suggests that this could also be the sex chromosome in other brown algae, inherited from a common dioicous ancestor. Putative SDRs of 3 other species (*Undaria pinnatifida*, *Dictyota dichotoma* and *Scytosiphon promiscuus*) were ascertained without assembling sequences of both sexes (see further comments below about the explanation of the evidence — in another *Ectocarpus* species, this appears to be assumed, which is plausible).

The first main result is that the sex-determining regions in all 6 haploid dioicous species appear to be on a homologous chromosome, and Fig. 1B shows details for 5 of them (*Ectocarpus* species are represented by just the one previously sequenced, as the other is so similar). In 4 of these 5 species, the V-linked regions are in the middle of the chromosome, but in *U. pinnatifida* it is at the left-hand end of the assembly. The sizes of the whole chromosome, the SDR, and the pseudoautosomal regions (PARs), differ between the species. It would be helpful to mention that the two *Ectocarpus* species have the physically smallest sizes of all three types in Table 1 (in the table, the sizes should be shown with commas separating the thousands, so that readers can understand the relative sizes). For the *Ectocarpus* species with both U and V region sequences, both are smaller than the other Vs (or one U) with which they can be compared. In other words, it appears that some species have bigger genomes than others. The table also shows that the smaller genomes have the lowest repeat densities, whether one measures the whole genome, the sex chromosome, or one of the SDRs. As expected, the SDRs had higher repeat densities than autosomes. I find it much more remarkable how little the SDRs appear to have changed in the very long evolutionary time periods involved, given the high repetitive densities in these genomes.

Despite these size differences, the PARs appear largely syntenic, which is also as expected. Even in *U. pinnatifida*, whose SDR is not in the middle of the chromosome, the right-hand PAR is syntenic, and synteny is not greatly disturbed by the change in the left-hand PAR's position. The SDRs are more rearranged between the species, consistent with a long history of evolving without crossing over.

It would be helpful to show the numbers of PAR genes in Table 1, for species where the chromosome assemblies allow estimates, so that readers can see the proportions of genes that are fully versus partially sex-linked. Fig. 1 suggests that the proportions are around 1/3rd or a quarter. Such basic information helps the reader to understand what is going on, and clearly the numbers and types of genes in these regions are central to understanding the authors' thoughts.

Another interesting result (which is not generally expected) the PARs also have higher repeat densities than the autosomes (in all 5 species with V sequences and both species with U sequences). This is not discussed, but a possible explanation that might be worth thinking about is that these species may not reproduce sexually very often. If so, the actual frequency of crossing over in the PARs would be low, and U- and V-specific repeats would be expected to accumulate more than on the autosomes, possibly resulting in a higher density estimate when a V chromosome assembly is analyzed (the authors might think about whether the approach used in the analysis might lead to such an outcome).

Other than this, the overall repetitive sequence results are largely as expected, and confirm that, similar to other systems, repetitive sequence densities can differ greatly between species, but are highest in genome regions that rarely recombine.

More interesting results require analyses of the genes in the SDRs, to ask whether they differ, and, if so, in what way, and I believe that this aspect is the authors' main interest.

2. SDR gene content and strata

The abstract claims that "Over time, nested inversions caused stepwise expansions of the sex locus, independently in each lineage, and concomitant with the increasing morphological complexity and level of sexual differentiation observed in brown seaweeds". In other words, the claim is that the SDR boundaries differ between the species, similar to what is found in sex chromosomes of diploid species in which evolutionary strata have evolved by stepwise suppression of recombination. Other kinds of change in the number of sex-linked genes can also potentially be analysed, including gene movements into the SDR from other chromosomes, or the opposite, or evolution of new genes from non-coding sequences within the SDR, or genetic degeneration involving deletion of genes from it, as is documented in diploid systems.

It would be helpful to mention that the SDRs in all the species include small numbers of genes (the highest number is 50, in *D. herbacea*, but Table 1 states that 30 are 'viral derived', leaving 48 as the highest number, in *S. prom*). If I have understood Fig. 2E correctly, 7 genes are inferred to have been V-linked in the common ancestor, with genes being added independently in different lineages.

It is mentioned that the centromere in the V chromosome of *Ectocarpus* is found within the SDR (Liu et al., Nat. Comm 2024). This may partly explain the low gene density, even in the species in this genus, whose SDRs are smaller than in the other species. Given the similar locations in the V chromosomes, presumably the SDRs include the centromeres in other species also.

The gene contents suggest that one reason for the different gene numbers is that the SDR boundaries differ between the species (in other words, new strata have formed). The evidence is still not very well explained. The text states that many genes located in the PAR of *Ectocarpales* (does this mean the 2 *Ectocarpus* species? — if so, it would be helpful to explain this) are in the V-SDRs of the other 4 species whose inferred V-linked regions were examined. If I have understood Figure 1 correctly, excluding *U. pinnatifida*, whose SDR has apparently moved from the middle of the chromosome to its left-hand end (but even so appears to include mostly the same genes), the SDR boundaries are remarkably similar.

It is strange to compare both the *D. herbacea* U and V with just the *Ectocarpus* species 7 V. As the U and V of any species differ greatly, surely the relevant comparisons needed to test for expansions would be between the two species' Vs and also between their Us. This would be easiest to understand after these had first been described in each species, so that we understand which genes are specific to a species' V or U, versus shared between the two, and potentially also shared by both species, and then which sex-specific genes are shared by the two Us, or the two Vs.

The change appears quite minor, given the distant relationship between the two species, and the roughly six-fold larger SDR size in *D. herbacea* (which is independent of extra genes, and reflects its larger genome size). This is not reflected in the wording in the abstract (quoted above) and elsewhere in the ms, including in Figure 6, that the SDRs have expanded via the evolution of "strata". Figure S8 shows the single more detailed comparison that is possible. It shows that a few genes are just outside the PAR boundaries of *Ectocarpus* species 7 V, and just inside the V- and U-SDRs in *D. herbacea*, an astonishing similarity in the SDR gene contents of such distantly related species. Compared with the *Ectocarpus* species 7 V, the inferred *D. herbacea* V has 2 genes in the left-hand PAR (in the same order in both species), and about 9 in the right-hand PAR (with several rearrangements). The same PAR genes should have changed to U-linked, but this is not mentioned, and it is not mentioned if this has happened, and the two species' Us are not compared (if I have understood Figure S8 correctly, compared with the *Ectocarpus* species 7 U, the inferred *D. herbacea* U does also include the same 2 genes, though this is not the ideal comparison). There are more rearrangements, which I think might reflect changes in the ancestry of either species. Fig. 2E summarizes the inferred changes in the Vs, with 13 changes of this kind in the *D. herbacea* V.

Estimated numbers of changes in the lineages leading to other species are based just on comparisons between the Vs, as U sequences are not available for most of the dioicous species. It should be noted that these inferences rely on perfect assignment of genes to the PARs versus SDRs, and some of these may not be reliable, given the approach used to test for complete sex linkage, and the other possible kinds of changes (see comments below). The evidence for rearrangements in the SDRs is convincing, but it is not clear that they show a significant tendency to be near the PAR boundaries, as appears to be claimed. If they do, of course this suggests that these regions may be difficult to assemble reliably, making the assignment of gene locations less certain, as just suggested.

Fig. 2E also summarizes inferred gene movements into the SDR from other chromosomes, or new genes within the V-SDRs of species at increasing distances from *Ectocarpus* species 7. The text should mention that the numbers of changes are all small, even across the vary large times involved (the highest inferred number of changes is the 22 autosomal duplications into the *U. pinnatifida* V-SDR). Concerning these types of changes, it is important to ask whether they are functional genes in these species, and to exclude genes in transposable elements that may be present in these V-linked regions. The 'viral derived' genes mentioned above are not explained, and the extent to which the other genes have been tested for brown algal function is not clear enough.

3. SDR gene content and gene loss (degeneration)

Losses since the brown algal ancestral state are inferred to be few. This is interpreted as minor degeneration, but this conclusion is not justified, as the ancestral state may already have been highly degenerated or gene-poor, if the centromere was within the ancestral SDR, as in *Ectocarpus* (see above). In addition, the small numbers of SDR genes mean that degeneration can be estimated only roughly, at best. The statements about a difference in degeneration between haploid and diploid species, as a generality, should therefore be considerably toned down.

However, rough degeneration estimates might be possible using the genes that are inferred to have become fully sex-linked most recently, for example those in Fig. S8. These were inferred to be PAR genes until relatively recently in the history of *D. herbacea*, and therefore they represent a set of genes that can be examined for loss from the V or U regions since strata formed by loss of recombination of the left- and right-hand PARs.

I did not understand the statement that "Most of these engulfed genes into the SDR of *D. herbacea* were retained as gametologs", but possibly this is expressing the idea just outlined. If so, it should be explained more clearly, and it presumably suggests that this small sample of genes have not degenerated. If this is correct, please be explicit about the actual numbers ("Most of the genes" is too vague). Also the U-V Ks values for these retained gametologs should be mentioned, probably in a small table, to give some idea of the time(s) during which they have remained present. Also, this text now mentions 17 genes, whereas I think that Fig. S9 describes only 16. If properly explained and confirmed, these observations would indeed support expansions of the SDRs of this brown algal species, and lack of degeneration since that event, or events.

However, the conclusion of less degeneration than in XX/XY and ZW/ZZ chromosome regions must take account of the different gene numbers between the species studied. The 50 genes suggested by the results in Table 1 probably represents a number that is unlikely to lead to considerable gene losses evolving. Therefore the difference in gene losses from highly degenerated sex chromosomes may simply reflect the small gene numbers in U- and V-linked regions. On the other hand, those small gene numbers might have evolved by degeneration processes in the brown algal ancestor that have now stopped because the numbers are too small for them to continue to operate.

An important criticism of the tests for "evolutionary strata" in *Ectocarpus* and *D. herbacea*, however, is that the synonymous site divergence between the gametolog pairs used is almost always extremely high, indicating saturation of the sequences, which implies that the divergence values will not relate to the evolutionary times since recombination stopped. Even if the values were no as high, it is implausible that 11 or 16 gametolog pairs could be assigned reliably to 4 different strata, as claimed in these species. Only in *D. herbacea* are there as many as 4 pairs with values < 0.5 (already too high to be very useful, though it is not made clear whether any correction for saturation was used). If those values refer to the genes that are inferred to have been in the PAR until relatively, but are now SDR genes in *D. herbacea*, it would be helpful to explain this explicitly (see the suggested table above).

4. SDR gene function evolution and its relationship with sexual dimorphism

The Discussion proposes that the evolution of the male-determining gene MIN was involved in the evolution of the ancient U/V system found in most dioicous brown algae, but evidence about how separate sexes evolved in these species will not be easy to obtain, given the huge amounts of time since this event. How strong is the evidence concerning this statement? It would be helpful to make clear whether MIN is the only gene conserved in all 6 V-SDRs, or whether the seven genes indicated in Fig. 2E include MIN plus other shared ones.

The evidence that SDR gene acquisition promoted the formation of new evolutionary strata is also not convincing. Even in XX/XY and ZW/ZZ sex chromosome evolution models, the description in the ms is not quite correct (it has been proposed that sexually antagonistic selection may lead to the establishment of partially sex-linked polymorphism, which selects for closer linkage across the regions. This is not expressed by the text "sexual dimorphism may lead to...". The text about expanding this model to U/V systems expresses the interesting idea that greater sexual dimorphism in gametes, which should often involve sexually antagonistic selection, may be associated with larger U/V-SDRs. It is thus important to document that the fully sex-linked regions of the 3 brown algal species with greater sexual dimorphism (presumably meaning oogamy) have additional genes that are not within the regions of the two studied species with anisogamy. As the other reviewer also pointed out, not enough species have been studied for such an association to be tested for statistical

significance. Results that are merely consistent should not be described in a manner that gives the impression that they constitute evidence supporting a conclusion. A proper comparative test is needed, to take account of the relationships and test whether changes in the degree of sexual dimorphism tend to be accompanied by changes in the number of SDR genes with relevant functions. The wording is still much too strong, as no such test can be done at present.

Finally, the text mentions two chromatin-related transcription factors in the *Ectocarpus* PARs were independently incorporated into the SDRs of four other dioecious species. However, it does not make clear whether these two are the only other SDR expansions claimed. This seems weak evidence for the abstract's claim about stepwise expansions of the sex locus.

5. An abstract

Based on the comments above, a modified abstract might say something like the following, which is intended to highlight what the results and analyses appear to show: Despite their age, the non-recombining regions have changed remarkably little. Increases in size in some lineages mainly reflect genome expansion due to increased repetitive sequence content, but small numbers of genes near PAR borders have also become fully sex-linked. We evaluate whether gene gains of various kinds are associated with increased morphological complexity and sexual differentiation, and whether new genes evolve unexpectedly often in U and V chromosomes' SDRs or PARs. We also study two situations in which UV-linked regions have changed. First, evolution of hermaphrodites occurred by ancestral males acquiring U-specific genes. Second, the *Fucus* dioecious system involves new sex-determining genes, acting upstream of formerly V-specific genes during development.

NOTEs on the above abstract text

- The abstract mentions the transition from a U/V to the *Fucus* dioecious system. However, the later text states that the Fucales probably evolved diploid dioecy from an ancestral monoecious system, which seems inconsistent with the abstract.
- Also relating to *Fucus*, it was unclear what was meant by "moved down the sex-determining developmental hierarchy", but I believe that the wording above expresses the concept.

Some detailed comments

Ascertainment of the U/V sex-determining (or maybe fully sex-linked) regions (SDRs) in the dioecious species should be explained better, as this is crucial to the rest of the results. It is not clear what is meant by saying that *F. serratus* lacks U/V sex chromosomes. I think that readers of Table 1 will need to be told that the phylogeny in Fig. 1 shows that this species evolved from an ancestor that probably has the same U/V system as the dioecious species studied, and they will want to be told whether it has a sex-linked region, and, if so whether the chromosome that carries it is homologous to the one in the haploids with UV chromosomes (which prove to be homologous across the 6 dioecious species studied). The chromosome homologous to the ancestral U/V carries a MIN gene, consistent with the overall conserved gene content of perhaps 7 ancestral genes in V-linked regions (see comment above), but presumably not suggesting that this gene is involved in its sex determining system. This should be made clear.

The sex-linked regions (mostly V-linked, as noted above) were identified "using a combination of bioinformatic and experimental approaches (see methods; Fig. S1-S4)". Presumably, this refers to the Methods section Discovery of the U/V sex determination regions, and it would be helpful to mention this explicitly. The basic principle used should also be outlined. I believe that it involved initial searches for sex-specific k-mers (I think this is the same as 'unmatched k-mers' in Fig. S1.), following the approach of Carvalho and Clark, followed by validation of candidate sequences by their coverage among Illumina short reads of sequenced individuals of both sexes, and then by PCR using 4 individuals of each sex. Presumably, this refers to all the species including those other than the single one in which the U was assembled. This would allow the region to be inferred in those other species, which is otherwise puzzling (see comment above). It would be helpful to explain that both sexes known for these species.

It should also be explained how the chromosomes were numbered. If this used the numbers in *Ectocarpus* (presumably based on homologous gene contents), this should be stated clearly in order to understand Fig. 1 (but Figs. S1-S4 show the sex-linked candidate regions on various chromosomes). The relationships between the numbers, and those in Table 1 should be made clear, and also that Fig. 1 shows only sequences identified as male-specific in the two species in which U-linked regions were studied. The red horizontal line should also be explained. The description in the Methods section needs editing to correct some mistakes in the English and punctuation, and can be made shorter and clearer, but the basic principle should be made clear in the main text.

It is not quite clear how the criterion for V-linkage was applied, especially in species with no female assembly. The text states that such regions were defined as ones within male reference genomes where the coverage in males was between 75% to 125% of the genomic average, while the coverage in females did not exceed 50% of the genomic average). Here, the coverage values are presumably some kind of average over sequences of some kind (for example exons or perhaps some other type) that could be mapped to the genome, and perhaps regions were defined as those in which no such sequence has a value outside those thresholds, or maybe there is some flexibility in applying them (maybe the red horizontal line in Fig. S1 indicates this)?

The statement that the ratios of gametologs, versus U- or V-specific genes, are similar in the distantly related *Ectocarpus* sp. 7 and *D. herbacea*, is not clear. Please can it be stated more clearly. The important question is whether the same genes appear as gametologs, versus U- or V-specific genes, in both these species, as opposed to each species having its own specific set of genes that have become U- or V-specific, which might suggest independent gains of genes by the U or V (but not both), or independent degeneration in these two lineages. A comment on the relevant text is below.

I am afraid that I was unable to understand the meaning of “the number of genes between their U and V-SDRs is symmetric within each species (Table S2), supporting parallel evolution of U and V chromosomes”. Please can it be explained what this means.

Fig. S9 mentions that 11 gametolog pairs were found in *Ectocarpus* sp. 7, and 16 in *D. herbacea*, but the question is whether the 5 extra ones in the latter might represent genes that became fully sex-linked since the lineages split. The main text states (if I understand correctly, as the text implies this, but does not make it completely explicit) that four genes “that form the most recent putative evolutionary strata in the U-SDR of *D. herbacea*” have homologs in the PAR1 of *Ectocarpus* sp. 7 (and that inversions in the U chromosome that included all the U-SDR and segments of the PARs can explain the more extensive U/V-SDR regions in *D. herbacea* shown in Fig. S8; note that one cannot say “expansion of boundaries”), and that 13 PAR2 genes are also in putatively recent strata (Fig. 2D, Table S4, Fig. S8).

I also did not understand the idea that the larger V-SDRs in early-diverging lineages (like *D. dichotoma*) than in later-diverging orders (Ectocarpales) suggest that gene loss might have reduced SDR sizes. Can this be explained better? I should have expected the early-diverging lineages to have greater gene loss, because there has been more time for this, but I may have misunderstood.

Reconstruction of the ancestral V-SDR gene content (with better quality genomic data; I did not understand “parallel U/V-SDR evolution^{24–26} as seen in *Ectocarpus* sp. 7 and *D. herbacea* (Fig. 2D, Table S2)”) is stated to have revealed that the small Ectocarpales SDRs are not caused by gene losses, but that gene gains of various kinds occurred in other lineages, as shown in Fig. 2E, including gains due to PAR genes becoming part of the SDRs, near their boundaries, as in Fig. S8. The key for part F does not make clear whether “autosomal” means translocations of autosomal genes into the SDR, and I did not understand the meaning of “sex homolog”, or how Fig. 2C shows that ancestral V-SDR genes have consistently higher Ks values than independently-acquired gametologs in *D. herbacea*, as the figure does not seem to distinguish these 2 categories of genes.

Information related to these issues is that the sex chromosomes (as a whole) had fewer conserved orthologs between species than the autosomes, and phylostratigraphy analyses is stated to reveal a statistically significant enrichment of taxonomically-restricted genes (TRGs) in the sex chromosomes of all dioicous species (furthermore, it is not clear that phylostratigraphy analysis is capable of extracting reliable conclusions from the rather few changes over the very large times involved, and this is not justified in the ms). This effect was (surprisingly and somewhat confusingly) due to the PARs of the Ectocarpales, but to the entire sex chromosome in species with larger V-SDRs, but it is not clear whether it occurs in the SDRs specifically. The text claims that the sex chromosomes carry several genes that became sex-linked since the last common ancestor of the five dioicous species, based on being shared only by members of taxonomic groups lower than a given rank. This is indeed consistent with independent enrichment in different species, but can losses be firmly excluded?

The reference to higher Ks values in the PARs is unclear. Does this refer to inter-species divergence, and are these values reliable (for U-V divergence, they are not, as explained above, and the values in Table S16 are also often very high). A sample of 6 such sequences from the *Ectocarpus* sp. 7 PAR was examined, and five of them represent potential new genes that may have evolved from noncoding DNA in the other *Ectocarpus* species. In my view, it is premature to claim a ‘gene cradle’ pattern on the basis of these few data, and without evidence concerning these genes’ functions. I would advise doing a much more thorough analysis and writing this up separately if clear evidence can be described. If there is such a thing as a ‘gene cradle’ effect in the V chromosomes of the recently sex-linked *C. purpureum*, and also in *S. angustifolium* (its age is not made clear), but not in *M. polymorpha*, which has an ancient UV system, reasons for the effect, and the differences, should be explained. In the response, the authors argue that mentioning *Ceratodon* is appropriate, but the issue is that it needs to be explained that the genes in this species fully sex-linked region were autosomal until recently. Certainly, this species’ neo-sex chromosomes are suitable for testing whether the newly sex-linked regions have started to evolve sex-specific allele expression, or other differences, but this requires a careful study in its own right, not just a passing remark.

However, the enrichment of genes with sporophyte-biased expression (not “sporophyte-biased genes”) is consistent with the model mentioned, in which generation-antagonistic selection may favor the retention of young sporophyte-beneficial loci in the PARs.

6. Transitions to monoicy

As expected, given that the V probably carries sequences important for male reproduction, and the U for female reproduction. The homologs of U/V genes can be recognised, and it turns out that most are derived from their ancestral Vs, whereas one might have expected also to find ones descended from ancestral U alleles (in *D. dudresnayi*, 20 genes are inferred to be V-SDR-derived and four U-SDR-derived; in *C. linearis*, whose closest dioicous relative is more distant than for *D. dudresnayi*, the numbers are 11 V- and only two U-derived). Presumably this suggests that the retained sex chromosome homolog is the V, while the U has disappeared as an intact chromosome? If so, this should be stated explicitly. As expected, the 7 ancestral V-SDR genes, including MIN, are transcribed during reproductive stages of both monoicous species. I did not understand the meaning of “retained gametolog copies or closely-related autosomal paralogs for most of their lost U/V-SDR orthologs”.

The statement that “the U/V-homolog of *D. dudresnayi* retains some vestiges of its past as a U/V chromosome, such as low coding density, high repeat density” suggests that a chromosome can be recognised as a U/V-homolog, but that it cannot be determined whether it is a U or a V. This could perhaps be explained more clearly. Note also that the suggestion above that the region may have low gene density and high repeat density if it includes the centromere, so these properties do not necessarily indicate a past as a U/V chromosome state, so this statement should be revised.

I did not understand some of the text on p. 12, including the words in upper-case font (and I have a couple of questions, indicated by []). The text is “The only three U-SDR-derived orthologs in *C. linearis* are flanked by PAR orthologs translocated across the CORNERS of the SDR-homolog, suggesting that [which chromosome(s)? of] *C. linearis* acquired U-SDR-derived genes through two ectopic recombination events (Fig. S25A). Three U-SDR-derived genes in *D. dudresnayi* are dispersed across the SDR-homolog [does this mean the V?], suggesting independent U-V [correct?] translocation events, while a fourth was translocated to an autosome”.

I am not sure why ectopic recombination is specifically mentioned as the mechanism, rather than simply mentioning that U genes have been translocated to new genome locations, either on the homolog of the
These events presumably occurred during meiosis in the diploid sporophyte stage, and it might be worth making this clear in the main text.

7. The diploid Fucales

The transition from haploid to diploid sex determination in the diploid *Fucus serratus* remains mysterious, as no sex-linked sequences were detected, which is worth reporting, as it is a striking contrast with the overall conservation of the SDRs. It suggests a fairly recent turnover event, somewhat similar to those detected between diploid species in other taxa, and expression data from three other Fucales species suggests that genes with essential reproductive functions have come under the control of different sex-determining genes from those on the U and V of the haploid species. The failure to detect completely sex-linked sequences in *Fucus*, suggests that recombination has not become suppressed in the genome region carrying the new sex determining locus, a striking similarity to the absence of strata in the haploid species. It is not clear how this relates to the sections about monoicous species, but presumably some relationship is expected, since an intermediate monoicous stage is mentioned for the Fucales.

General problems requiring mostly minor (but still essential) revision

1. When mentioning the “fundamental inheritance differences between U/V and XX/XY or ZW/ZZ systems that have broad evolutionary and genomic implications” (in the introduction), the work of Bull should be cited. It is not clear whether the concepts he developed are made use of in interpreting the findings. If they are, the connection should be clearly explained. At present, Bull is cited only twice, as follows (with my comments)

a. “independent expansions of the U/V-SDRs boundaries and gene content may have occurred, as predicted”.

The first part of this is not a clear or accurate outline of his prediction. The boundaries between SDRs and PARs must be the same in the U and V chromosomes. The PARs must start in the same locations, because they are defined as recombining regions (and the study confirms this by absence of sex-specific variants, and by synteny, along with showing loss of synteny in non-recombining regions). A caveat is that it may be difficult to determine the exact locations where PARs start, because close linkage with the SDR leads to linkage disequilibrium, so that the U and V may show some level of divergence, with sex-associated variants that may appear sex-specific in real-life samples. This is particularly likely if sexual reproduction occurs infrequently, as this could be the case in these algal species. A different comment explains that this may make it difficult to determine precise PAR boundaries.

Concerning the second statement, Bull did predict that U- and V-linked regions should evolve different gene contents, but the statement is much too vague, as he specifically predicted that, although genes with male function should be preserved on the V, they could be lost from the U, provided that their functions are not essential for functions of gametophytes of both sexes. If I understand his paper correctly, he thought that many genes should have such essential functions, and consequently predicted that degeneration should be minor. If this does not apply to brown algae, the case should be made that this predicted difference from the situation in plants such as bryophytes should be briefly explained, along with the evidence about this issue from the new results. If the ms documents the predicted differences in gene contents, and if Bull is cited, these things should be connected clearly.

b. “Anisogamous species only experienced few gene gains in their SDRs”

Presumably, the idea is that genes should be gained when gametophytes of one sex gain sex-specific functions, say female functions in an oogamous lineage. This needs to be explained, as it is not exactly the same as Bull's idea (if Bull also predicted this, of course it can be explained and his paper can be cited for the idea). However, as explained in other comments, the evidence supporting the hypothesis/prediction is not clear.

An alternative would be to mention Bull in the introduction, and explain which of his predictions applies in brown algae, and any hypotheses/predictions that he did NOT make, but which can be made for brown algae, and are tested in the study.

2. Concepts needed to understand the study need to be explained before being used. The core concept of UV systems in haploid organisms is not clearly explained, and many readers may not understand these concepts, and especially (as just mentioned) specific properties of brown algae that relate to U and V chromosomes' special properties. The brief description of dioicy/dioecy mentioned in the responses from the authors probably means the text “...which may not undergo recombination in the heterogametic sex (XY or ZW) of diploid species (dioecious), or in the diploid stage of haploid-dominant (dioicous) species”, but I don't think that readers will understand this as definitions unless they already understand these things (that they refer to species with a genetic sex-determination system controlling males and females, that may not undergo recombination in the heterogametic sex (XY or ZW) of diploid species (termed dioecious), or in the diploid stage of haploid-dominant (dioicous) species”.

It is also not quite correct to say that sex-determining regions encode factors directing sex identity (protein-coding genes encode proteins, but factors are carried in genome regions). This may seem a minor quibble, but communication is important, and it is best to use correct terminology, and to avoid creating an impression that is not correct (here, that protein-coding genes are always responsible, and that regions carrying these factors are always recombinationally inactive).

Related to this problem, it is too vague to say “conventional” to refer to the XX/XY and ZW/ZZ systems. The text probably means “Research on the biology and evolution of sex chromosomes has primarily focused on diploid XX/XY and ZW/ZZ systems in mammals, birds, fish, *Drosophila* and diploid plants.. Haploid U/V sex-determination systems, such as those of bryophytes and algae have been less studied. It might be helpful to say this in the abstract. Then the introductory part could end with stating how many brown algal species (not just “in a range of”) and outgroups with an explicit list of their sexual systems (not just “diverse” ones) were studied. The Results starts by mentioning ten species covering the phylogenetic, morphological and reproductive diversity of the brown algal clade and their closest outgroup, *Schizocladia ischiensis* (not explained, see comment below). Why not combine these two statements and be clear and definite, rather than vague?

The explanation of gametolog gene pairs is not clear enough. Also, the term “female-limited” and similar terms probably mean U-specific, as opposed to simply being on the U-linked member of a U-V gametolog pair.

The MIN gene’s involvement in male reproduction is not clearly enough explained in the background text.

3. Vague or unclear words should be replaced with clear ones, or phrases. Specifically, “emerged” presumably means “evolved”. And the phrase “the demise” of U and V chromosomes (used in several places) should be replaced with something more clear and explicit. Does it refer to a situation in which an entire U or V has been lost and the species reproduces vegetatively or asexually as the remaining sex, or what. Or maybe to degeneration? As written, the meaning is unclear.

4. Some statements are too sweeping, and should be made more cautious. For example, dosage compensation may sometimes evolve, but one cannot simply say that it does so. When it does, this is a remarkable evolutionary change, but the manner of writing gives the (no doubt unintended) impression that it obviously will do so, and that no thought is necessary about it.

Minor comments

The meaning of this sentence is unclear “these studies have provided important findings that help to understand the genetic structure of bryophyte U/Vs”. Does this mean “the genomic structure of bryophyte U- and V-linked regions”?

The final sentence of the introduction section adds nothing and should be deleted. The text has already explained that the study described the sex-linked regions of these algae, and readers are now hoping to read what insights the results yielded.

The closest outgroup is said to be something called *Schizocladia ischiensis* but it is not explained what this is, or the evidence that it is closer than other organisms.

In Table1, the meaning of the column headings “PAR - U/V-homolog” is not clear. Does this simply mean regions assigned as PARs?

The statement that “synteny conservation” is “akin to multicellular Metazoans” is not clear. Presumably it means that some measure of synteny is similar to the one for multicellular Metazoans, not that synteny with the chromosomes of multicellular Metazoans is high.

Figure 5 is difficult to understand, as it contains so many different things. It is not explained what Ks refers to, and the concept of gene ages is non-standard and needs to be explained clearly. The meaning of U/V homolog in “chromosome compartment (autosomes and U/V-homolog)” should be explained — does it mean the homolog of the U and V in other brown algae?

Reviewer #2 (Remarks to the Author):

Thank you for the thorough revisions! I think the manuscript is now easier to follow, the conclusions are better supported, and the more speculative aspects are acknowledged as such.

My only remaining comment is that given how central the correct identification of the SDRs and their boundaries is for the manuscript, a stronger case could in my view still be made:

* Figures S1-S4 are a good start, but they are quite noisy for some of the species, and clearly not the whole story as both coverage analyses and PCR verification were also performed. Would it be possible to add a coverage track to the plots, as well as the genomic locations that were validated, and the boundaries that were consequently identified?

* In the methods, you say that “Genomic regions with a minimum of 70% of unmatched single copy kmers were then retained as candidate male or female SDR sequences”, but this threshold is never reached for *D. dichotoma* and *U. pinnatifida* (at

least this is not visible in Figs S1-S4). Could you clarify this?

* It was also not clear to me how the exact boundaries of the SDR were determined, and this needs to be clarified -- in some figures it seems obvious (but even there there must have been a method to determine specific coordinates), in others not at all.

It also seems pretty standard to provide primer sequences and gel images of the PCR.

Version 3:

Decision Letter:

Dear Susana,

Apologies, the Reviewer #1 Review Attachment #1 was too large to send via our email system. Please download from the below file.

https://springernature-my.sharepoint.com/:f/p/adam_battishill/EnZqFhdhfGdlprvAlZqzsWcBhP0APf3Uv2zCNpBWyb8vQA?email=susana.coelho@tuebingen.mpg.de&e=1xWadG

Let us know if you have any questions.

Kind regards,

[Redacted]

6th June 2025

Dear Susana,

Thank you for submitting your revised manuscript "Origin and evolutionary trajectories of brown algal sex chromosomes" (NATECOLEVOL-24092629C). It has now been seen again by the first reviewer and their comments are below. The reviewers find that the paper has improved in revision, and therefore we'll be happy in principle to publish it in Nature Ecology & Evolution, pending revisions to satisfy the reviewers' final requests and to comply with our editorial and formatting guidelines. We will check your responses to the reviewer's comments in house. Please make sure to make it clear in the text what are strong conclusions vs hypotheses as indicated by the reviewer and focus on the former.

We will send the annotated PDF by the reviewer separately since the file is too large to attach to this email.

Sincerely,

[Redacted]

Reviewer #1 (Remarks to the Author):

This ms is now much improved, though a surprising number of minor corrections of the text are still needed, as indicated in the annotated pdf file and the comments below.

Moreover, several ideas are stated as conclusions despite weak support, and the ms fails to focus on hypotheses that can be tested or on the most interesting and well supported conclusions, which I list below. Ideally, the Discussion should evaluate how the findings that are well supported help reach interesting conclusions. Instead, it unnecessarily repeats much that has now been made clear in the Introduction, Abstract and Results sections. For instance, the emphasis (L. 351 onwards) seems inappropriate on the statistically unsupported appearance that the SDRs of anisogamous (oogamous) species have independently and consistently gained genes (whether by changes in the PAR boundaries as recombination was suppressed, or by creation of new genes or perhaps duplication of genes into non-recombining regions).

Similarly, it is not very clear what is meant by "streamlined gene content of the *M. polymorpha* U/V-SDRs" (have these lost genes, in contrast to the gains inferred in the brown algae? L 398 states that it is degenerated, based on Iwasaki et al 2021, which contradicts the earlier statements about such conclusions in haploid organisms), or "sex-biased gene expression in mature gametophytes carrying sex organs" (does this mean sex-biased expression of autosomal genes, reflecting

differences in somatic development between male and female gametophytes that might be controlled by just a few SDR genes?). The text (L 361) contrasts these cases with that of the moss (it is not explained that the species is a moss until a few lines later, and the genus name is not mentioned) *Ceratodon purpureus*, whose sex chromosomes are not correctly described. They do carry thousands of genes (and only a few autosomal genes have sex-biased expression), but situation of thousands of fully sex-linked genes (based on also being sex-linked in another moss, *Syntrichia* (Silva et al., 2021 doi: 10.1111/tpl.15116), but (as L 397 mentions, albeit rather vaguely, this SDR expanded very recently (by fusions of autosomes with earlier established U and V chromosomes, so that the resemblance to PARs is unsurprising, leaving it unclear whether the new understanding is simply that the brown algal SDRs did not arise in this way, which the synteny results show is correct (and this is interesting).

A shorter Discussion focusing on what is new and interesting (listed under the next set of comments) would be an improvement.

Major comments

L 134. The statement that “the gametolog Ks values are broadly consistent between orthologs in *Ectocarpus* sp. 7 and *D. herbacea*” is vague. Are the values consistent with a great age estimate for U-V divergence, or do the data tell us something else. It is not convincing that such saturated sequence divergence can be used to infer a conclusion, but, if it can, the authors should certainly explain clearly what it is, and the logic for their inference(s).

L 136 may, in fact, be a statement of the conclusion from these results (if so, the text should be revised to avoid the vague earlier statement). This line states (in my wording) that “the position of the gametologs with the lowest Ks values in the U-SDR of *D. herbacea*, relative to the PAR genes in *Ectocarpus* sp. 7 indicate that inversions in the U chromosome that included all the U-SDR and segments of the PARs can explain the expansion of the U/V-SDR boundaries in *D. herbacea*”. This sentence is difficult to understand, but may mean something like “the two [or maybe more than 12 —the number is not stated] gametologs with the lowest Ks values (below a value that is not stated) in the *D. herbacea* U-SDR (but rearranged in its V) are PAR genes in *Ectocarpus* sp. 7, indicating that inversions in the U chromosome that included these U-SDR sequences and parts of the PARs can explain the expansion of the U/V-SDR boundaries in *D. herbacea*, a process that we term ‘engulfment’”.

L 154 In discussing the results from inference of ancestral V-SDR gene contents, the text says that Fig. 2C shows ancestral genes consistently having high inter-gametolog Ks values, while independently acquired gametolog pairs had lower values (in one of the species pair, *D. herbacea*, in which I think the authors conclude part of an ancestral PAR became fully sex linked, though this is not made clear enough in relation to these Ks estimates). However, this is not clearly shown in the figure, as there is no key. I assumed that the smaller symbol (back or blue dots) indicate the ancestral genes, as in the key to the figure part below, and the larger squares indicate the independently acquired genes that presumably mostly evolved when parts of the PAR became fully sex linked and have remained as gametolog pairs (since genes newly evolved within the fully sex linked, or gained by duplications from other genome regions, would presumably rarely be found as gametolog pairs). However, this may be incorrect, as the squares are mostly higher than the dots, and so perhaps the symbols are reversed in the 2 parts of this figure (if so, this is confusing). All of this still needs to be explained clearly, and it currently remains unclear whether there is a consistent and statistically significant difference. It is quite plausible, and indeed expected, that, if a set of genes became fully sex linked recently enough, forming a younger stratum, they will often still be present as gametolog pairs and have low Ks values, so if there is clear evidence for this in this species comparison, it can be described very simply.

The inter-species Ks values discussed later (top of p. 11) are not specified, so it is not clear whether these are also saturated. Looking at Figure 3A, it is difficult to understand whether this is the case, and some of the figures appear to show higher values (yellow) for the autosomes (is this correct?) and for regions in the middle of the sex chromosome. Please can you check this figure, and perhaps mention some values, rather than readers having to look at figures that just show “higher” and “lower” when the values have been estimated quantitatively.

L 171. I have several questions about this text: “Although the number of SDR changes is small, oogamous species each independently gained numerous V-SDR genes, and one gene (ATP-dependent RNA helicase) was convergently acquired in all”.

1. Could this gain have occurred in the common ancestor?
2. How many of these gains occurred when PAR genes ceased recombining? Distinct events that created new, young strata should be described clearly, with the numbers of genes that can be shown to have been gained in each. In connection with these, am I correct in thinking that the V-SDR of *Ectocarpus* sp. 7 is thought not to have undergone any such enlargement, whereas that of *D. herbacea* has done so (this is the same as my query above, as I feel that the text remains unclear, despite this being a rather obvious question).
3. And similarly for translocations of autosomal genes into the V-SDRs of *Ectocarpus* sp. 7 and *D. herbacea*, we should be told how many in each case also happen to be sex-specific genes (Table S2, Table S5), consistent with a model where sexual antagonism in autosomal loci was resolved by the evolution of sex linkage.

The most interesting conclusions, suitable for the Discussion section

1. The observed enrichment of male-biased genes on the PARs in all species except *D. dichotoma* is consistent with the sexual antagonism model just mentioned, with locations in the PARs showing close linkage to the sex-determining region. However, the ms appears not to explain that parts of the PARs may recombine very rarely with the male-determining region, especially if outcrossing occurs rarely in a species.

2. In the two species tested, gametologs had higher expression levels than genes present in only one of the SDRs, even when expression was estimated in the haploid stage. This may be either due to their functions (as the text in lines 183 onwards mentions), or partly to the time during which they have been evolving in the absence of recombination (young

stratum genes would be expected to have expression similar to the values for their autosomal counterparts in other species, as is indeed observed, and to be gametolog pairs in most cases), while older ones (without gametologs) might have lost expression capacity. If this is correct, it would suggest the possibility of genetic degeneration, albeit at a slow rate.

3. It is also interesting that, compared with the autosomes, the sex chromosomes (or perhaps just the SDRs, see my minor comment below on L 210 onwards) had fewer orthologs conserved between species. A clear discussion of what might explain this pattern would be valuable.

The result from phylostratigraphy analyses, that the fully sex-linked regions are enriched for taxonomically restricted genes, is related, and also interesting. It presumably means the appearance of new genes, not a degeneration process leading to loss of some ancestral genes, as PAR genes are stated to show this more than SDR ones (L 224), though the text does not appear to make this argument. Also, I was unable to find a clear description of the meaning of taxonomically-restricted genes should be defined, and I think that it may mean genes not recognisable in any other species, or perhaps in any other brown algal species (suggesting that they are indeed true new genes), but am not sure (it might mean duplicated genes that are not in the fully sex-linked regions of other brown algal species).

I am still unhappy about introducing the term “gene cradle” in addition to TRG. The pattern(s) of overrepresentation of new genes can be described simply and clearly without a new term (as lines 247-8 in fact do) and there are already too many words within biology.

Minor comments

L 88. This states that the SDRs contain a relatively small number of genes overall (between 18 and 52) compared to the PARs (between 229 and 904), but does not relate the numbers to the physical sizes, which would allow one to understand whether the small overall numbers indicate a noteworthy difference. It is mentioned immediately after this that SDR size differences in different species are strongly correlated with the number of genes.

L 112 states that the U/V-SDRs of *Ectocarpus* sp. 7 and *D. herbacea* carry homologous genes (gametolog pairs), indicating descent from a common ancestral region, but this statement is unclear, as it may mean that these two distantly related species carry homologous sets of gametolog gene pairs, indicating that their U regions descend from a common ancestral U region, and similarly for their Vs — or it might mean that, in each of these two species, the U and V carry gametolog pairs, indicating that the U and V are descended from a common ancestral region, or both may be true. From the text in L 115, I think that the latter is meant (“only ten genes share SDR orthologs between both species”), but it would be good to make the meaning unambiguous from the outset, rather than leaving readers puzzled.

Related to the description of gene numbers, the statement is also puzzling/unclear writing that “each species shows an equal number of gametologs and sex-specific genes in its U- and V-SDRs [and this] symmetry supports the idea that the U and V chromosomes may have undergone parallel evolutionary changes within each lineage”. Does “an equal number of gametologs and sex-specific genes” mean the total number of both types (with some ancestral gametolog pairs having lost the U or V copy, as explained just before this, thus becoming sex-specific genes), supporting the idea that the U and V initially had the same gene content, but that some of these have been lost by each lineage?

L 179. What does “U/V-SDR genes” mean? Does it mean genes that are either U- or V-linked, including gametolog pairs (in other words genes present in the U- or V-linked regions, or both of them), or something else?

L 210. The reference for transposable element accumulation in the sex-linked regions is not very appropriate, as it refers to another smut fungi, which have non-recombining regions (which evolved for reasons that are well understood, but not related to sex chromosome, and not sex-linked). This should be replaced by a reference to a review of such accumulation in sex-linked regions.

It also seems strange to describe such results for entire sex chromosomes (including the finding that fewer orthologs are conserved between species than for the autosomes), because it is only the fully sex-linked regions are predicted to show unusual repeat richness and low gene densities, and indeed L. 213 confirms that V-SDRs differ from the PARs and the autosomes in these ways, though the PARs were also enriched in repeats compared with the autosomes, presumably because parts of them recombine rarely with the male-determining region (see above).

Similarly, L 230 compares entire sex chromosomes for sporophyte-biased genes, although (in this case) the prediction is that such genes should be preferentially retained in PARs.

L 310. I don't think that monoecy has been defined.

L. 342. The meaning is unclear of “sequestration ... essential for maintaining the U/V system or incidental during the initial recombination-suppression event”. Please be explicit about the two alternative hypotheses that remain to be tested.

Dear Editor

We sincerely thank you and the reviewers for the overall positive comments and advices to further improve our manuscript. Please find below the point-by-point response to the reviewers' suggestions, which we are confident we have effectively addressed. We also attach a track change version of the manuscript showing the revised sections and also including your edits.

We have extensively revised the text and main figures to accommodate the suggestions from the reviewers, performed further analysis and also included nine further supplemental figures and one main Table. In summary:

A) We now include a detailed analysis of the evolutionary origin for each SDR gene, showing whether they were incorporated through expansions of the SDR borders into the PARs, translocation of autosomal genes or gene birth events within the SDR (**new Fig. 2E** and **new Table S5**). This should clarify the concerns of Reviewer 1.

B) We now present results from the YGS analysis supporting the prediction of the sex-linked regions (**new Fig. S1-S4**) and the lack thereof in *Fucus serratus* (**new Fig. S27**).

C) We have incorporated Table S1 into the main text and included additional information there (**new Table 1**).

D) We performed additional comparative analysis that support our conclusion that SDR borders expand through nested inversions (**new Fig. S8**). We include functional annotation of these genes (**new Table S4**), and show that transcription factors were independently engulfed by the SDR over evolutionary time. Importantly, we acknowledge our limitations to detect the precise evolutionary event behind the initial recombination suppression in the SDR, as Reviewer 1 asked, and we offer alternative scenarios to the inversion hypothesis.

E) We performed a change-point analysis using sorted gametolog Ks values that give statistical support to the putative evolutionary strata in the SDRs (**new Fig. S9**). We also highlight the ancient U/V-SDR gametologs from the ancestral state reconstruction into the figure with the putative evolutionary strata (**new Fig. 2C**). We acknowledge the limitations of predicting strata in U/V systems, so we use the term "putative", and refrained from making any overstatements.

F) We performed additional analyses describing the repeat content in the U/V-SDRs, included permutation tests (**new Fig. S6B-C**, **new Fig. S13**, **S14**).

G) We improved the visualization of the plots and added statistical support in the figures (**new Fig. 3A**, **new Fig. S12**). We also provide violin plots with permutation tests to show that the "gene cradle" pattern is mostly visible on the PARs, but that it extends into the whole chromosome in species with large SDRs, as Rev. 2 requested (**new Fig. 3B**). Permutation tests were also performed in the non-vascular plant species with U/V chromosomes, further supporting our proposal that the "gene cradle" pattern evolved multiple times in eukaryotes (**new Fig. S21-S24**).

H) We have been very careful not to overstate the significance of our findings and clearly separated what is evidence from discussions.

Importantly, the new analyses did not result in any changes to the manuscript's results or conclusions. On the contrary, we believe that our conclusions have been strengthened by the analyses and re-writing, and we are very grateful to you and the reviewers, as the manuscript is much improved. We hope you agree with us and consider our manuscript acceptable for publication.

Best regards, Susana Coelho

Reviewers' comments:

Reviewer #1 (Remarks to the Author):

General comments

This ms describes some results concerning interesting questions about sex chromosome evolution. The brown algal haploid sex chromosome system is problematic for asking questions about some aspects of sex chromosome evolution (including genetic degeneration and "evolutionary strata, see below), but is excellent for testing the idea that sex-linked regions will show an over-representation of genes with functions in sex determination and the development of male and female structures. Unsurprisingly, therefore, my comments on the problematic aspects are somewhat negative, but more positive about the latter questions. Considerable revision is needed to communicate clearly and accurately, and, most importantly, to clearly describe evidence for the conclusions (or to make clear to readers when these are tentative, and how they can be further studied in the future, if possible).

The main conclusions are the following, not all of which is clearly explained in the present ms: 1. Chromosome homologies can, to some extent, be detected in the different brown algal species studied (Figure 1). As one would expect, homologies are clearest for species that are related in the tree, but, if I understand correctly, one monoicous species, *C. linearis*, is polyploid (perhaps with 2 whole genome duplications) and its chromosomes have many-to-one relationships with those of other species, though 2 have red lines which indicate syntenic blocks with the V sex chromosomes of other species. Another monoicous species, *Desmarestia*

dudresnaya, has syntenic blocks with the V sex chromosomes of the other *Desmarestia* species, which is dioicous.

RE: We thank the reviewer for the overall positive comments and suggestions to improve the clarity of our manuscript. As the reviewer acknowledges, we demonstrate chromosome homology across a range of species, particularly the sex chromosome whose homolog was found in all analysed species, including the species that lack a U/V system (termed “sex-homologs” in the previous version, now “U/V-homologs”). We should emphasize that *C. linearis* is *not* a polyploid (based on our analysis, and on Denoeud et al., Cell 2024). It is possible the reviewer was confused because of the large number of contigs inferred for this species (**Fig. 1**) due to the extreme technical difficulty in HMW DNA extractions (see also Denoeud et al, 2024), precluding the achievement of a chromosome-level assembly. We still include it in our analysis because it is an interesting monoicous species, and the quality is more than adequate for our study. Note that the two red lines in **Fig. 1** indicate the two contigs that belong to the U/V-homolog in *C. linearis*, rather than duplicated copies of the same chromosome. We have now specified in the legend that the larger number of contigs in this species is not due to polyploidy, but to the lack of a chromosome level assembly.

2. All the extant brown algal species studied are stated to share the same ancestral SDR, on the homologous chromosome in all the species, and there is a single male-determining factor, MIN, within all of these SDRs, and therefore probably in the ancestral V-SDR (but see my caveat about a diploid species below). It is not clear whether these are all SDRs, in the sense of controlling sex-determination.

RE: The term sex-determining region (SDR) is used here to refer to a region with suppressed recombination that is fully sex-linked based on bioinformatic analysis using k-mers (YGS approach) and read coverage, as well as PCR validation (please see Methods). Moreover, we confirm that *MIN*, the factor that determines male sex in Ectocarpales and Laminariales, is present across all the V-specific regions, indicating that these regions do contain this master sex determinant factor and are, consequently, involved in sex-determination. Note that we can of course change the term to NRR (non-recombining regions), but because *MIN* is a master sex determinant, we believe it is acceptable to keep the term SDR. For completeness, the new version of the manuscript provides additional figures (**new Figures S1-S5**) showing the proportion of unmatched k-mers between the male genome and the female sequence reads, and highlighting the sex-linked regions in the V chromosome of each species.

3. Parts of the chromosomes inferred to carry the SDRs of the dioicous species show clear evidence that they do not recombine (usually in the middle region), and that these regions are flanked at both chromosome ends by recombining PARs. This is interesting, as the sex chromosomes of haploid plants (bryophytes) have not been found to have detectable PARs in the genome sequences (though genetic mapping detected PARs in a moss, *C. purpureus*, suggesting that the genome sequencing missed them, for some reason). The difference suggests that those PARs may be smaller than in the brown algae.

RE: This is correct. We note that the SDR is consistently in the middle of the sex chromosomes and is bordered by PARs, except for *Undaria pinnatifida* whose SDR is found on one of the borders. We have now highlighted this difference between plant and brown algal sex chromosomes in the first paragraph of the discussion.

4. Figure 2 shows that the SDRs of the 2 *Ectocarpus* species sequenced, and those of the 2 *Desmarestia* species (one of which is dioicous) are in inverted orders in the U and V, with several further rearrangements. It seems unlikely that these inversions were the cause of the lack of recombination (see below).

RE: We would like to clarify that **Figure 2** does not compare the U and V chromosomes *between* different species. Instead, it presents the U and V chromosomes *within the same species* for *Ectocarpus* and *D. herbacea*, both of which are dioicous. The synteny plots in **Figure 2A** and **2B** illustrate the collinearity of genes on the PARs of the U and V chromosomes within each species. As shown in the figure, the SDRs on the U and V chromosomes are *clearly inverted in relation to each other*. For clarity, we have now included a new Figure (**Figure S8**) where we present a synteny plot between the V chromosome of *Ectocarpus* and the V and U chromosomes of *D. herbacea*.

Our data suggests that inversions played an important role in the recombination suppression event that created the SDR, although we agree that alternative mechanisms cannot be excluded (see our arguments below). We have taken the reviewer’s comment in consideration, and tried to address these in our response below about inversions and in the revised version of the manuscript.

5. Large parts of the text describe results about gene numbers in the SDRs, on which I comment below.

The analysis infers that there was a common ancestral V-SDR, and that it probably also carried six genes other than *MIN*. If I understood correctly, these are V gametologs of genes that were also carried on the ancestral U chromosome. Interestingly, all 7 genes appear likely to function in sex determination processes. If the results of this analysis can be considered reliable, this is interesting, as it relates to testing the prediction that sex-linked regions should show an over-representation of genes with functions in sex determination and the development of male and female structures. Probably Bull’s predictions should be cited here, and the prediction explained before the data are described, but this work is cited only in a bit of unclear text and not in clear relation to this question.

RE: Yes, the reviewer is correct in their reasoning. We added new supplemental **Tables (S5 and S6)** that summarize the genes in the SDRs/PAR/autosomes in the different species and ancestral gene content, which should help further to convey the information. We cited Bull's predictions as suggested.

Other conclusions are either harder to understand as described, or are not evidently reliable or firm conclusions, as detailed below in my comments on each section. It is often unclear which are speculations, or simply consistent with a conclusion, and, among those, which have excluded alternative possibilities, as only the latter can properly be termed conclusions.

Re: We hope the new version addresses the lack of clarity concerns. Please find each comment addressed below.

There is no need to use a lot of words with emotional connotations, suggesting that the findings are very interesting. The results are indeed interesting, and there is no need to 'advertise' this.

Please avoid such writing and also unnecessary text with the same aim, such as the first sentence "Sexual reproduction, present in almost all eukaryotes, allows species to survive environmental challenges by increasing genetic variation". Words like "exploit" are also unsuitable in scientific writing, and (luckily) are not needed, and words like "dynamic" and "insights" are vague and communicate no clear meaning to readers. Other vague words need to be clarified. These are minor comments, but the vagueness is a general problem for understanding the results. Such terms include "gene shuffling", "drive" (and similar words, when a more explicit one is required). Similarly, terminology such as "gene cradle pattern" is obscure. It may be a term used in the authors' lab, but outsiders will need clearer words, and definitions. It is best to avoid coining new terms like this (and "gene nurseries") that require defining, and to use standard word as far as possible. Many readers will not be native English speakers, and it is more considerate to restrict wording to standard scientific and other terms. Finally, it is extremely confusing to use the word "linked" when writing about a genetical topic, when one does not mean genetically linked. Another word should be used if the meaning is something like "is a consequence of" or "is correlated with" (in which case, a statistical test is needed).

Please also see the annotated pdf file

RE: We thank the reviewer for the helpful advice with wording. We have modified the manuscript and accepted all the suggestions of the annotated pdf file that the reviewer kindly provided.

We would however prefer to conserve the term "gene cradle" in the manuscript. This is a novel pattern in U/V sex chromosome evolution that we describe in detail for the first time across several taxa of brown algae and plants (see below). We tried to find another 'simple and precise' term to describe this pattern, but we still feel that "gene cradle" conveys the meaning of our findings which is that the sex chromosome is indeed a location where taxonomically-restricted genes emerge. To ensure that the term is clearly understood by the readers, we added a precise definition for the gene cradle pattern (*i.e.*, the overrepresentation of taxonomically-restricted genes across U/V sex chromosomes compared to the rest of the genome) in the revised manuscript.

We accepted all the suggestions from the annotated pdf.

The background section

The introductory paragraph fails to distinguish between diploid and haploid-dominant organisms, although the study deals with the latter. Several statements (such as "heterogametic sex") apply only in diploids.

RE: We agree; we now include a brief description of dioicy/dioecy in the introduction.

Assuming that recombination has been suppressed (see comments in the next section on strata formation) were inversions involved? It is interesting to test the widely repeated claim that inversions are involved in the suppression of recombination that characterises many sex-determining regions. The conclusion is stated that "that the U/V sex chromosomes of the brown algae emerged between 450 and 224 Mya, via an inversion that suppressed recombination". The ms states that differences in the sizes of certain regions (which the text does not make clear enough) "coincide with extensive structural rearrangements, particularly inversions, even among closely related taxa. Thus, structural rearrangements may have caused the recombination suppression event around the MIN gene, creating the brown algae U/V sex-linked regions". However, this is not really clearly demonstrated. Readers should be told precisely how they can see that size differences of clearly specified regions indeed coincide with inversions, with a statistical test to show that these changes are associated.

Re: We hope that the reviewer will find our clarifications in the new version of the manuscript helpful, and agree with our conclusions on the role of rearrangements (inversions) in the emergence and expansion of the SDR. We generated a new supplementary figure (**Fig. S8**) that clearly defines the SDR boundary differences between *Ectocarpus* sp. 7 and *D. herbacea*, showing that the PAR genes in *Ectocarpus* entered the SDR in *D. herbacea* likely via inversions on the U-SDR. Unfortunately, we are unaware of statistical tests to support this empirical observation, but we will gladly include any test that the reviewer might suggest for this purpose.

Consistency with the idea that inversions may lead to suppressed recombination between sex chromosomes is not evidence that they did indeed do so (as the text states), as inversions are also predicted to arise and spread after recombination becomes suppressed (the argument is very clearly explained in this paper, for a different non-recombining situation: Sun et al. 2017 Large-scale suppression of recombination predates genomic rearrangements in *Neurospora tetrasperma*. Nature Communications 8: 1140. doi: 10.1038/s41467-017-01317-6).

RE: We hope the new version of the manuscript presents our arguments more clearly.

Our analysis of two specific pairs of U/V-SDRs revealed an inverted gene order between the U and V SDRs within both *Ectocarpus* and *Desmarestia* (see Fig. 2A-B). This pattern strongly suggests that inversions occurred, as it is the most parsimonious explanation for the observed structural differences and the initial suppression of recombination. Additionally, we observe higher gene collinearity between the U-SDRs (or V-SDRs) of *Ectocarpus* and *D. herbacea* than between the U and V SDRs within each species (Fig. 1B,C, new Fig. S8). This observation indicates that the inversion likely predates the divergence of these species and may have occurred in their common ancestor. We hope the addition of the new Fig. S8 helps clarifying our arguments.

Nevertheless, we have followed the reviewer's advice and we removed any overstatement (e.g. we deleted the mention to inversions causing the initial recombination suppression event: "Together our results indicate that the U/V sex chromosomes of the brown algae emerged between 450 and 224 Mya, via suppressed recombination in a genomic region that contained the male-determining factor MIN (henceforth male-determining locus)"). We also suggest the possibility that pericentromeric recombination suppression may have preceded the genomic rearrangements (e.g., Filatov, 2024), given that the centromere in *Ectocarpus* resides within the SDR.

The evidence regarding the increased size in the SDR and its association with inversions is seen in the U-SDR of *D. herbacea*, with several gametolog pairs in this species that show a clear inverted pattern with respect to the PARs in *Ectocarpus*. This evidence is now better highlighted in the new Figure S8.

If inversions can be inferred to have been involved, or evidence exists that this is very likely, then it should be mentioned whether these can be inferred to have affected the U or V, or both haplotypes (or whether no suitable outgroups are available to make any such inferences). Some models for recombination suppression predict that inversions should affect only Y or W chromosomes. The authors should also consider which of the models can apply in haploids, as not all of them can do so.

Re: Without a suitable outgroup to reconstruct the ancestral gene order on the autosome that became the sex chromosome, we cannot determine with certainty whether the initial putative inversion affected the U, the V, or both chromosomes. Unfortunately, there is no such outgroup in extant brown algae. Note that an inversion on one chromosome (U or V) is sufficient to disrupt recombination between the U and V SDRs, similar to how inversions can disrupt recombination between Y and X or W and Z in other systems.

However, the expansions of the SDR in *D. herbacea* can be explained by inversions in the U chromosome, since the syntenic order of the PAR genes in *Ectocarpus* roughly coincide with the V chromosome of *Desmarestia*, but show an inverted pattern in the U (Fig. S8).

Moreover, we do acknowledge that additional mechanisms may explain the initial recombination suppression. For example, the centromere in the V chromosome of *Ectocarpus* is found within the SDR (Liu et al., Nat. Comm 2024), so we cannot exclude that a centromere-related suppression of recombination might have preceded the inversion events found on the SDR. We added this scenario (end of page 3) in the manuscript, and removed any overstatement related to inversions being the cause of the initial recombination suppression event (see above).

Concerning the comment about which model apply in haploids: Some models apply to both haploid and diploid sex determination systems (for example, the hypothesis under which recombination suppression is favored because it permanently links sex-antagonistic loci to the SDR, Rice 1983 Evolution, Rice 1987 Evolution), while others involve gene expression in the diploid phase (for example, the hypothesis that recombination arrest can be selectively maintained due to the early evolution of dosage compensation mechanisms, Lenormand & Roze 2022 Science). However, the latter hypothesis may still apply to UV systems if some genes in the non-recombining region are expressed in diploid sporophytes (as divergence of expression of U and V copies could stably maintain recombination arrest). More work would be needed to determine which mechanism is most likely to explain recombination suppression on the sex chromosomes of brown algae, including running the models in a U/V background. Therefore, and because of space constrains, we would prefer not to develop too much these ideas which would be more adapted to a review or another paper. If however the editor agrees that this is an important point and worth exploring we are of course happy to extend our text into this discussion.

This text also needs revising. "Together our results indicate that the U/V sex chromosomes of the brown algae emerged between 450 and 224 Mya, via an inversion that suppressed recombination in a locus that contained the male-determining MIN factor". Here readers need to be told why suppressed recombination might evolve in a locus, or maybe the word "locus" needs to be amplified to say something like "suppressed recombination in the genome region containing the male-determining MIN factor (henceforth "male-determining locus")".

RE: The text has been revised as suggested.

Inferring SDR expansions and gene gain versus degradation, and evolutionary strata

The evolution of U- or V-specific genes is interesting, and might reflect either U/V-SDR divergence or loss from one haplotype. The meaning of the phrase evolved “after U/V-SDR divergence”, in the authors’ wording “U- or V-specific genes ... either evolved after U/V-SDR divergence...” is not clear to me, as it could either mean that genes duplicated into the U or the V haplotype after recombination ceased, or that U and V sequences of a gametolog pair diverged so much that their allelic nature is no longer detectable, as is the case for some human YX pairs).

RE: We have modified the text to clarify that these genes may have emerged through the duplication of autosomal genes and/or subsequent translocation into the SDR.

The text reads as if there is a contradiction: “Despite differences in SDR gene numbers” versus “U and V-SDR gene counts coincide in both species”. Can this be clarified?

RE: We understand the reviewer’s confusion. What we meant is that the SDR gene number is different between species, but that both species contain similar ratios of genes between the U and the V SDRs within each species. We rephrased this sentence.

The difficulty in detecting strata in U/V systems is because neither of these fully sex-linked regions recombines. In XY diploid systems, for example, the X recombines, and this largely prevents rearrangements, so that the X arrangement can be assumed to be similar to the ancestral one, even if due to gene movements and/or chromosome rearrangements have disrupted colinearity between different species. The ms does not explain the difficulty correctly.

RE: We agree with the reviewer; sentences were modified to emphasise the reasons why it is difficult to detect strata in UV systems.

Another difficulty arises because Ks analysis requires many genes to detect strata with different Ks values (because the variance of these estimates is large), but in these species the numbers of genes are very small (*Ectocarpus* sp. 7 and *D. herbacea*), and the inference of at least four strata in the latter seems optimistic, as does the conclusion that its U/V-SDR reflects an expansion. It is therefore not clear that one can firmly conclude that the results “indicate that SDR expansions in brown algae are driven by nested inversions”. This needs to be clarified, and it should also be made clear what process is suggested as having led to the inferred larger SDR size in this species. It is also important to distinguish between larger SDR size due to accumulation of repetitive sequences, and changes in its boundaries so that genes flanking it in some species (meaning PAR genes) are within the SDR in others.

RE: We performed a change-point analysis to give statistical support to the prediction of four evolutionary strata (Fig. S9). However, we acknowledge that the prediction of the exact number of evolutionary strata remains difficult due to the variance in Ks values, leading to a possible overestimation of strata by the change-point analysis (particularly the oldest stratum that only harbors one gene). Therefore, we emphasize in the text that these strata are merely “putative”.

We have also clarified why we believe inversions are associated with expansion of the SDRs. We provided evidence that the gene content in the SDR increased in different lineages of brown algae, and our synteny analyses indicates that the most recently acquired gametologs in *D. herbacea* are inverted between each other and that they were previously found in the PAR (see Fig. 2, S8 and our reply above), and the gene order unambiguously points towards an inversion having uptaken genes from the PAR into the SDR.

Additionally, we now include a new figure displaying the correlation coefficient between the size of the SDR and both the protein coding content and the repeat content in the SDR across different species, showing how the accumulation of both elements are independent but equally important predictors of SDR size expansion in the brown algae (new Figs. S6B and S6C).

A related interesting question that was studied is when size differences reflect expansions of the SDR, versus reductions in sizes, and evaluating the possibility of gene loss, which might have reduced SDR sizes, in some cases. As just mentioned, it is important/valuable to describe the density patterns of repetitive sequences, to test the prediction that these should accumulate when a genome region ceases to recombine. However, the ms deal with this quite sketchily.

RE: We thank the reviewer for the suggestion. We have incorporated the mean values of repeat density across genomic compartments (SDR, PARs and autosomes) in the new Table 1 as requested. We also incorporated permutation tests of repeat density across compartments, showing that the observed differences in repeat density across compartments are all significant (page 9; Fig. S6).

Do the sex-linked regions show an over-representation of genes with functions in sex determination and the development of male and female structures?

The authors inference that the ancestral brown algal V-SDR carried seven genes, including the male-determining MIN factor, plus six V gametologs of genes also carried on the U chromosome, and that these genes are all likely to be related to sex determination processes. This would be a valuable conclusion, but needs a statistical test.

Re: The question is of course very interesting but addressing it experimentally would require reverse genetic approaches to study the role of each individual gene present on the sex locus, which would take several years to accomplish and is out of the scope of this manuscript.

However, we tried indirectly to address the roles of sex-linked genes by looking at gene expression during the reproductive phase (**Fig 2D**). In terms of statistical tests suggested by the reviewer: there is a statistically significant excess of male biased genes on the PARs of sex chromosome of all species except *D. dichotoma* (page 4), suggesting that the sex chromosome has indeed functions in sexual development. However, we note that any meaningful expression analysis on the SDR genes would be hampered by the lack of statistical power due to having a low number of SDR genes (17 in the case of *Ectocarpus*). Again, we were careful not to make any overstatement.

Large parts of the text discuss different gene contents of SDRs, and ideas about new genes, and the reasons for the conclusions are not described clearly enough. Specifically, changes such as movement of the PAR boundaries are not clearly distinguished from the ideas about new genes.

RE: We understand the confusion when referring to “new genes”, which can be interpreted as TRG emergence or as the incorporation of genes into the SDR. We changed the term “new genes” in the text and modified some sections to improve clarity in distinguishing between these different processes.

Near the end of the text, it says “sexual dimorphism may drive SDR expansion”. I think this means that SDR expansion via movement of the PAR boundaries may have occurred as differences in the morphology of the male and female gametophytes evolved, suggesting the possible involvement of trade-offs between male and female functions, leading to sexually antagonistic polymorphisms in partially sex-linked genes and selecting for closer linkage to an existing sex-determining region.

Re: Yes indeed the reviewer understood correctly. We explain this idea now more clearly in the discussion.

The text mentions We expanding this model to UV systems by demonstrating that derived lineages with pronounced gamete sexual dimorphism (oogamy) tend to have changed U/V-SDR boundaries in brown algae. This suggestion seems plausible, but should be more clearly worked out, and the absence of a correlation between levels of sex-biased expression and sexual dimorphism in brown algae is surprising. Statements like “sexually antagonistic selection may drive gene movements into the SDR” are not clear enough (especially if the scenario envisaged does not actually invoke gene movements, but simply changes in the PAR boundaries). This section can be shortened, as essentially the same argument was already made earlier.

RE: We believe that the lack of correlation between level of sexual dimorphism and sex-biased gene expression is not that surprising, as has also been described in plant species (Scharmann et al. 2021). But we agree that more explanation is needed to conciliate this result with the idea that sexual antagonism may drive the expansion of the SDR. As suggested, we now make a clear distinction between gene movements into the SDR potentially due to sexual antagonism and the expansion of the SDR boundaries into the PAR, showing that oogamous species display both types of gene gains in their V-SDRs (**Fig. 2E**).

The text has also been trimmed according to the suggestion.

The overall conclusion concerning SDR gene gains and losses is not clear. Page 15 mentions “streamlined gene content of brown algae U/V-SDRs”, and suggests, plausibly, that (like other sex-determining regions) genes within the region could regulate an autosomal effector gene network controlling sexual dimorphism, with most sex-biased genes influencing the physiology or vegetative development of gametophytes, though it would be surprising (and interesting) if vegetative development differed greatly between gametophytes of the two sexes.

Re: We hope the inclusion of the new tables and figures increases clarity. We would like to point that vegetative gametophytes also show sex biased gene expression, suggesting that gene expression precedes any visible morphological sexual dimorphic features (Lipinska et al. 2015, MBE). Importantly, sexual dimorphism between male and female vegetative gametophytes have been reported in many brown algae such as kelps (Luthringer et al. 2014, Perspectives in Phycology). We included this information in the revised manuscript.

The observed substantial sex-biased gene expression in mature gametophytes, which presumably carry sex organs, is less surprising, and may be similar in haploid plants, but worth reporting. It is not helpful to cite the case of the moss, *C. purpureus*, whose sex chromosomes carry thousands of genes, probably due to fusions with at least two autosomes.

Re: We also added the information regarding the presence of sex organs in mature gametophytes. We still think it is helpful to include a reference to *Ceratodon*'s sex chromosomes, since it represents a counter-example to our observations.

As predicted (probably Bull's predictions should be cited here), all appear intact (non-degenerated) in *Ectocarpus* sp. 7, and are presumably expressed at appropriate developmental times (it should be mentioned that this is tested a bit later in the ms), but in *D. herbacea* one (a casein kinase) was lost from the U-SDR, and one was not found, presumably meaning not found in the genome of either sex. These ancestral V-SDR genes remain present in the sex-homolog of the species that have lost their UV system. Here, it should be explained whether those species still undergo sexual reproduction, for instance having gametophytes that produce archegonia and antheridia (or the relevant brown algal terminology). If so, this is not really remarkable, but it nevertheless suggests that these genes are important for pathways in sex development (though of course not sex determination, as stated on page 5).

RE: As suggested, the text has been modified and we added the reference to Bull's work.

It is less clear whether the ancestral brown algal SDRs carried other genes that did not have such functions, and have been lost from these regions. Similarly, it may be interesting to test whether any regions that have become part of a lineage's SDR by changes in the PAR boundary included any genes with functions unrelated to those of the genes just mentioned, and whether these genes have been retained or lost (as might be predicted). If lost, how did this occur? For instance, have any such genes become duplicated to autosomal regions and then lost from SDRs?

Re: As mentioned previously, we lack an extant outgroup to infer the genes present in the ancestral SDR. However, we have analyzed in detail the changes of SDR borders into the PAR in *D. herbacea* (Fig. S8). As noted in the text, almost all genes from the PAR that became sex-linked were retained as gametologs, with only one potential instance of gene movement to the autosome and two losses of either of the gametologs.

As mentioned, it is difficult to specifically assess gene functions without reverse genetic analysis, however we have added the predicted functions of these newly sex-linked SDR genes in new Table S4. Notably, two chromatin-related transcription factors have been independently added to the SDR in *D. herbacea* and across all other studied species. We have now included a table summarizing the presence/absence of genes in SDR, PAR and autosomes across all species (Table S5). All the information requested by the reviewer is presented in this table.

The text a bit later suggests that such things are also found (in my slightly shorter wording "expression of many U/V-SDR genes is not confined to fertile gametophytes. Thus the SDRs contains both genes involved in sex determination and gametophyte fertility but also genes involved in other functions"). In my opinion, these different results should be better integrated into an overall conclusion.

Re: These ideas are now integrated in the discussion as proposed.

I believe that Bull's predictions may suggest that the evolution of sexual dimorphism of the gametophytes might evolve through the incorporation of new genes into species' SDRs. Intriguingly, the study found a strong correlation between the V-SDR size and the extent of sexual dimorphism, with anisogamous species retaining the same V-SDR genes as inferred in the ancestral state (with very few gene gains), consistent with this being the V-SDR ancestral state. It would be helpful to mention this when describing the ancestral state inference, as it would seem to validate that inference.

Re: We added a sentence on this as suggested, and cited Bull.

Oogamy is found in different lineages, and this is accompanied by convergent gain of ATP-dependent RNA helicase in all cases, and of various new V-SDR genes in different lineages, suggesting possible independent evolution. Autosomal sex-biased gene (SBG) expression was not found to correlate with sexual dimorphism levels.

PAR genes should not be involved in the sex-determination process, and are predicted to be less likely to have reproductive functions, though ones closely linked to the boundary with the SDR might have such functions, and might establish sexually antagonistic polymorphisms that could select for closer linkage to the actual sex-determining gene(s). It is thus interesting to use *D. herbacea*'s expanded SDR (which includes genes in the PAR of *Ectocarpus* sp. 7) to test this.

RE: The revised manuscript includes the predicted functions of the recently acquired SDR genes in *D. herbacea* (new supplementary Table S4). We noted that two chromatin-related transcription factors have been independently added to the SDR in *D. herbacea* and across all other studied species, suggesting they may be important for sexual reproduction. However, the predicted gene functions are based on the sequence homology and would require experimental validation in the future.

At this point, the text finally makes clear that the expansion involved 2 events, one in which four genes from one *Ectocarpus* sp. 7 PAR (PAR1) stopped recombining, and a second affecting 13 genes in its PAR2, forming two

younger SDR strata in *D. herbacea*. These results would be more helpful if explained when the strata are first described.

Re: We moved these sentences earlier in the manuscript, in the same section as the putative evolutionary strata.

Of these 17 genes, only one gene remained partially sex-linked *D. herbacea*, and most (14?) were retained and became SDR gametologs, supporting the conclusion that U/V degeneration is minor in the time since they formed (however, it is not stated in the text how large Ks values are for these strata, and Fig. 2 shows Ks values on a y axis scale that includes the shared, ancient stratum, so one cannot see the relevant information), while two others are located on this species' autosomes, which suggests that some genes not involved in male or female functions may move off the UV pair, contrary to the authors' conclusions.

RE: The information regarding the Ks values of the different evolutionary strata, including the ancestral SDR genes, was already available in supplemental **Table S3** in the original manuscript. However, we agree with the reviewer that incorporating this information in the main manuscript is useful. Therefore, we modified **Fig. 2** to differentiate between the ancient SDR genes (black squares) and the more recently acquired gametologs (black dots).

The two autosomal genes that the reviewer is referring to are also autosomal in the outgroup *Dictyota dichotoma* (see **Table S5**), suggesting that these genes moved to the PAR later in the phylogenetic tree. Therefore, we believe that our conclusions are not challenged by this observation.

The finding that gene ages appear to be younger for "sex-chromosome-enriched" categories than their last common ancestors in the same order or broader taxonomic groups supports the idea that they became sex-linked independently in different species. The pattern is said to be consistent with a predicted retention of young genes with beneficial effects in the sporophyte stage, and to be lacking in species that lack heteromorphic generations (a concept that needs to be explained earlier).

RE: There may have been a confusion concerning the gene ages in the sex chromosomes, since the "young genes" are not sex-linked but rather represent taxonomically-restricted genes (i.e., these genes do not have traceable homologs in other organisms) that seem to be appearing in the sex chromosomes repeatedly and independently in brown algae and bryophytes. We appreciate the reviewer's suggestion to explain heteromorphic life cycles earlier in the manuscript. We now added this information in the introductory paragraph of the brown algae (lines 59-60).

A further complication in inferring the history of brown algal (and other) U/V chromosomes is the possibility of de novo evolution of new genes from previously non-coding regions, and the text on p. 16 mentions that the authors infer that such changes have happened, or at least that "rapid sequence divergence" creates the appearance that it has happened. The wording implies that the authors conclude that new genes evolve from non-coding regions more often in these SDRs than in other genome regions (or possibly they mean in other sex chromosomes), but again this is not supported by a test showing an enrichment of new genes in these regions.

Re: The reviewer may have missed the information - we do provide statistical support for the enrichment of taxonomically restricted genes in the sex chromosomes (please see **Table S14**). We now also include the statistical comparison between each autosome against the sex chromosome of each species in **Fig. 3A**. We also provide violin plots with permutation tests in **Fig. 3B** showing that the PARs are significantly enriched in TRGs compared to the autosomes.

Such a conclusion might be interesting, but I suspect that it would require an entire study devoted to testing it explicitly, excluding different possible explanations.

Re: Again, we do provide ample evidence that U/V sex chromosomes exhibit an excess of young, taxonomically restricted genes, and comparative genomics across several *Ectocarpus* species demonstrates that at least some of these genes emerged *de novo* from non-coding regions. The reviewer is correct that testing each of the TRGs across all the brown algae would require an entire new study, out of the scope of this manuscript. However, our intention was to provide evidence that at least some of the young genes emerge *de novo*.

It seems premature to discuss what mechanism might be involved. The text speculates that, as U/V chromosomes are enriched in heterochromatic marks, which may involve TE repression, this may contribute to heterochromatic regions' higher mutation rates than other genome regions, which is plausible. It then mentions that mutation rates in U/Vs are high (a reference is needed), and states that this "could facilitate the emergence of *de novo* loci", which seems too vague, or that their high density of DNA transposons "could promote the co-option of their regulatory motifs and enable *de novo* transcript birth, as seen in *Drosophila*", and that these new genes "could then be retained in the sex chromosomes by 'generation-antagonistic selection'". These are interesting speculations, but solid evidence is needed that new genes are actually appearing at an unexpected rate. The only evidence described is comparative, ("the gene cradle pattern is unique to U/V systems and gradually disappears when these systems are lost, as in monoicy or diploid sexual species"), but no adequate comparative analysis is described, and other U/V systems are stated to show the same pattern ("this process is pervasive across distant, independently evolved eukaryotic kingdoms"), but no references are given. The wording that "a unique interplay between complex life cycles, heterochromatic landscape, DNA transposons, and higher

mutation rates ... may drive de novo gene birth in UV chromosomes”) relies completely on the word “may” to tell readers that this is speculative, and I feel that this is not clear enough.

RE: We believe the reviewer might have misunderstood some information in this section of the manuscript, and we hope our explanation below and the new version of the manuscript more clearly conveys the message.

After observing a statistically significant lower number of orthologs in the sex chromosome across species (see **Table S12**), we tested if the genes without detectable orthologs represented taxonomically-restricted genes (TRGs; genes that are only found in a particular clade or species. See Johnson 2018 for a detail explanation of TRGs). We used a method called genomic phylostratigraphy (see Barrera-Redondo et al., 2023) to calculate the relative ages of these TRGs. We performed Wilcoxon rank sum tests for every brown algal species in the study to show that young TRGs (i.e., TRGs found in a very narrow range of species) are significantly enriched in the sex chromosomes when compared to the autosomes (see **Table S14** and **Figures S15-19**). This is what we refer in the manuscript as the “gene cradle pattern”, and it is consistent across all dioicous species of brown algae. We also performed this analysis in plant species with a U/V system (hence, no reference for these results is needed), showing that the gene cradle pattern is not exclusive of brown algae, but is a broader phenomenon of U/V systems (see **Table S14** and **Figures S21-S22**).

To test whether these are bone fide *de novo* gene birth events (and not homology detection failure due to rapid evolutionary rates), we analysed a subset of candidates in the PAR of *Ectocarpus*. We used several *Ectocarpus* species as our focal species because the recent divergence between these species could help us distinguish between all of the possible processes behind TRG emergence. With the exception of one gene, we confirmed that these genes indeed arise ‘de novo’. Therefore, the gene cradle pattern may be driven, at least partially, by the emergence of “de novo” genes (see **Figure S20**).

Evidence of higher mutation rates in the U/V chromosomes is also provided in the manuscript, since we found significantly higher synonymous substitution rates in the PAR genes than in the rest of the genome (see **Table S17**).

In sum, we do provide clear evidence for an enrichment of TRG genes in the sex chromosomes of UV systems (gene cradle pattern), and that a subset of these genes arises *de novo* (this is ‘evidence’). Then, we build on this evidence together with existing evidence for other structural and evolutionary features of U/V sex chromosomes to propose potential scenarios favoring TRG emergence specifically in these regions. These hypothetical scenarios are considered in the ‘discussion’ section of the manuscript.

The new manuscript clearly distinguishes between the evidence provided and the speculative/discussion aspects of our findings to avoid confusions.

Repetitive sequence densities and gene densities.

The Circos plots in Fig. 3 are extremely difficult to read, and I could not find an explanation of some symbols, such as some microscopic ones near the chromosome names. Track 1 shows gene “ages”, which need to be defined. Track 2 shows gene and repeat densities, but it is not possible to understand whether these show any patterns when both are in a single track. Beyond the impression that the gene densities are fairly uniform (on a scale that may not allow any differences to be seen, though they may be lower on the sex chromosomes than elsewhere), and that repeat densities show no obvious pattern other than being high on the sex chromosomes, at least in some cases (as expected if these are mostly non-recombining, though the high densities may extend to the PARs). It would be helpful to plot results for just the UV pair, or even just U or V chromosomes whose other pair member was not sequenced, as the repeat densities can provide valuable supporting evidence for the presence of non-recombining regions (even in monoicous species, as noted below). More detailed figures might also help clarify whether the PAR boundaries differ between the species, as I believe the authors think, versus SDRs gaining new genes in other ways. The lack of clarity about these differences is a major barrier to understanding these results, and the processes that may be involved in creating the situations observed (see comments above).

RE: The circos plots were meant to show that the sex chromosomes: 1) Have younger gene ages than the rest of the genome, 2) display higher Ks values, 3) have a low gene density and higher TE content and 4) display an enrichment of unclassified repeats in the Ectocarpaceae. All these observations are supported by statistical tests (see **Tables S8-S12**). In order to improve clarity, we now use linear plots, which are easier to read. Following the reviewer’s suggestion, we also plotted these values exclusively for the U and V chromosomes as supplemental data (**Fig. S12**). We also include additional supplementary figures to show the differences in repeat density across genomic compartments (autosomes, PARs and SDR) to further illustrate the accumulation of repeats in the sex chromosome (**Figs. S13-S14**). We hope this modification makes the results more clearly visible for the readers. We also added an explicit definition of “gene ages” in the manuscript.

It is also unclear if these are mean densities for the U and V, or values (presumably in windows of some size) from just one of these. More explanation is needed, as it would be valuable to describe repetitive sequence densities, as these are predicted to be high in non-recombining regions, and can be used to detect such regions in the absence of genetic map results (as in these species). However, the ms deals only sketchily with these analyses.

RE: We thank the reviewer for the suggestion. The values in the plots represent percentage of repetitive and coding elements within 100 kb non-overlapping windows (see Methods section). We have now modified **Fig. 3** to plot our results more clearly (see comment above). We also performed additional analysis on repeat content and include a **new Table 1** and new supplementary figures (**Figs. S13-14**) that compare the repeat content of each genomic compartment in the brown algal genomes using permutation tests.

To understand the “higher Ks values”, and the finding that these are “localized in the PARs”, readers need to be told what Ks is estimated between (mutation rates are mentioned, and this appears to refer to pairwise inter-species divergence between closely related species, but the species pairs are not specified), and what they are higher than.

RE: A comprehensive description of this analysis is included in the Methods section, including the species pairs that were used for the calculation, the details concerning the detection of one-to-one orthologues and the calculation of the Ks values. The specific Ks value for each pair of ortholog genes can be found in **Table S16**. We considered including this information about Ks comparisons in the results section, but we have word limitations, so unless the reviewer considers this to be essential, we suggest to leave the information in the methods only. The Ks values in the PAR genes are higher compared to the Ks values in the autosomal genes (see **Fig. S15-S18**).

Loss of U/V systems

Results are mentioned that “show that monoicy arose at least twice from a male ancestor that acquired female genes”, but readers are not told where these can be seen, or which species this refers to. It is valuable to report that previous conclusions may have been incorrect because in brown algae expression of male-biased genes tends to be tissue-specific versus broader expression of female-biased ones, possibly explaining why female and monoicous transcriptomes are similar.

RE: The female genes of the monoicous species were depicted in **Figs. 4A-B** with pink color and the label “Female SDR” and the exact label of these genes are found in **Tables S19-S20**. The names of the two monoicous species are included in the respective result section and the figure (**Fig. 4A-B**). We would like to point out that Cossard et al. explicitly mentioned the possibility that the ancestral state was a male and that the female transcriptomic similarities may have been due to broader expression of female-biased genes, therefore their conclusions were not incorrect. These ideas, including citation to Cossard et al. are included in the discussion section.

It is plausible that monoicy evolved (not “emerged” in an unspecified manner) via a recombination event that provided essential female genes to a male genotype, as proposed for such a change in a liverwort and in *Volvox* species, but again it is not easy to understand what is meant by “a U-SDR-derived gene present in both monoicous species” (perhaps the authors mean that only a single gene was identified from the ancestral U that was inferred in the inferences already described). With only 2 monoicous species, it is also unclear whether this suggests its fundamental role in the female developmental pathway. However, one can infer that if monoicy can arise by a male genotype acquiring U-linked femaleness factor(s), this has important implications for sex-determination. Specifically, it shows that the MIN factor is not both essential for maleness and also excludes expression of female functions, since it is present in the genome of species that do express those functions. This seems to show that there must also be actively femaleness-promoting functions in the genomes of these organisms. These could, of course, be autosomal, but clear ideas about how these things might work are needed, including how a species expressing both sex functions could re-evolve dioicy.

RE: We replaced the word “emerge” for “evolve” as suggested. We rephrased this section to convey more clearly the idea that, from all the female genes that were integrated into the two monoicous genomes, only one of these genes is conserved in both species, suggesting that this gene has important implications in the activation of female-related functions. We also discuss the idea that a femaleness-promoting gene in the U-SDR must exist. Concerning the reviewer’s about “how a species expressing both sex functions could re-evolve dioicy” : in the brown algae there are no cases of re-evolving dioicy from a monoicous lineage.

If genes can move off the sex chromosomes, it is plausible that the change to monoicy might, as in other cases, result in loss of one sex chromosome, or possibly even both. An example in brown algae is worth reporting, adding to the cases already described. However, the results do not actually describe “genetic networks governing sex determination” being “rewired” during the transitions described. Again, therefore, the text, over-states the significance of the findings here.

RE: Our intention was not to over-state the significance of our finding – the sentence about genetic networks was a more general one, not claiming that we are the ones describing this for the first time. We deleted this sentence to avoid further misinterpretations.

It would be valuable to describe repetitive sequence densities in the retained SDRs and the other genome regions of the monoicous species, to assess how long these regions continue to be non-recombining regions (presumably, they can recombine in UU or VV sporophytes produced by monoicous gametophytes) and whether

repetitive sequence densities soon decline to the genomic background level. However, the ms seems not to mention these questions, and unfortunately Figure 3 shows Circos plots only for dioicous species.

RE: We would like to point out that this topic had been addressed in the manuscript: we had generated this plot for the monoicous species *D. dudresnayi* (Fig. 4C) and performed statistical tests that proves a higher repeat density in the U/V-homolog (as opposed to the autosomes) due to its past as a sex chromosome (please see Table S11). In *Fucus serratus* (see Fig. 5B), the repeat density was no longer higher in the U/V-homolog, likely because the U/V system was lost a longer time ago than in *D. dudresnayi* (Table S11). We realize that the circos plots with many chromosomes are difficult to read, so we replaced them by a linear format that is easier to follow by the readers (new Figs. 4 - 5).

The evolution of a diploid XX/XY system in *Fucus serratus* is stated to be associated with the (permanent) transition to diploidy in the Fucales. Here, it is essential to make clear what evidence exists for genetic sex-determination in these species, and for male heterogamety. The term XX/XY system gives readers the impression that sex chromosomes have been seen in males and females, but this seems not to be the case, as it is stated that no sex chromosome (meaning sex-linked region) was detectable in *F. serratus*, which turns out to mean that the sequences of wild males and females in this study did not identify any sex-linked sequences, suggesting that its SDR is small and/or little differentiated.

RE: The existence of XY system in Fucales is inferred based on monoecious vs dioecious natural hybrids (Coyer et al., JEB 2007) but we also understand that the term XX/XY might mislead the readers into thinking that a sex-linked region has been already detected. Thus, we changed the term “XX/XY” for “dioecy” in this section to avoid misinterpretations.

At this point I was unsure of what was previously known about this species. Reading the earlier text, I found out that the male-determining gene MIN is present in V-SDRs of all the dioicous species, on the same (homologous) chromosome, including this diploid XY species.

It is not made clear enough whether the MIN gene, or sequences physically near it in *F. serratus*, have been tested for sex linkage (a straightforward matter), but presumably this gene is not sex linked in the species. This suggests that MIN may simply be essential for male functions, and that a new system may have evolved in *Fucus*. A bit later in the text, this impression is strengthened, as this chromosome is called an ‘ex’-sex chromosome, that has lost all the evolutionary vestiges of its past as a U/V chromosome. I suspect that I have not correctly understood the story about this species, and I feel that it can be explained more clearly, including the evidence for a monoicous intermediate.

RE: The reviewer is correct: we confirm that we did test (PCR and using genomic datasets) if MIN was sex-linked in *Fucus* species, and the result was negative, i.e., MIN is autosomal in Fucales. We modified this section to improve clarity.

The new data suggest that the “UV to XY transition” (which is not understood, and is thus potentially interesting, but appears not to have involved evolution of a new non-recombining sex-linked region, other than maybe a very small one) “involved an intermediate monoicous stage, not a direct shift from the U/V system” (I did not understand what the authors have in mind for a “direct shift”). Retention of an ancestral SDR certainly seems unlikely in in *F. serratus*, as it has no close UV outgroup.

RE: We modified this sentence in the revised manuscript. We would like to point out that all the ancestral SDR genes were found within a syntenic region in the ex-sex chromosome (“U/V-homolog”) of *F. serratus*. We also note that the transition to diploid sex determination in the Fucales may have involved a diploid U/V sporophyte - in this case one would expect to find both U and V SDRs in the Fucales. We are developing verbal and mathematical models for the transitions from U/V to X/Y in collaboration with our colleague Denis Roze, and the possibility of a direct transition from U/V to X/Y (without a monoicous intermediate) is envisaged in the models – however this will require an entire paper on its own, so we refrained from adding this on the present manuscript. We opted to delete the mention to “direct shift” to make the message clearer here.

The text on p. 13 suggests that the findings are “consistent with the hypothesis that a young sex chromosome in *F. serratus* arose following the transition to the new life cycle system. However, it is not explain why a monoicous intermediate is indicated, rather than an extension of the diploid stage, which might have reproduced vegetatively and then evolved a new genetic sex-determining system, leaving the MIN gene as a male-essential, but not, sex-determining, gene. Under this scenario, or one involving monoicy, it would still be necessary to explain how two sexes could evolve.

RE: Previous studies indicate that monoecy is the most likely ancestral state of the Fucales, before dioecy evolved in *F. serratus* (e.g. Heesch et al., 2021). We added this important detail in the discussion of the manuscript. The processes involved in an immediate transition from dioicy (U/V) to dioecy (XX/XY) are complex and would require an entire study devoted to this question alone. As explained above, we are currently working with Denis Roze on a mathematical model to describe the different pathways from UV towards diploid sex determination, but we feel this falls outside the scope of the present manuscript, and we are already above the

word limit. However, if the reviewer and editor agree that we can devote a paragraph to this discussion, we are happy to elaborate on these matters.

The final Discussion sections appear mainly to repeat the findings and interpretations, and are unnecessary, or could be greatly shortened. It might be better to integrate discussions into the earlier sections, avoiding repetition, and making the ideas clearer. This would allow space for evidence that is not currently included (as outlined above)

Re: We considered merging results and discussion but we worry this would lead to confusion (as the reviewer pointed) between what is 'evidence' and what is discussion and inferences from our evidence. Therefore, we would prefer to keep the manuscript with a separate results section. We have however addressed the reviewer comment by carefully trimming the results and discussion sections to avoid repetition.

Established models of XX/XY and ZW/ZZ systems posit that recombination suppression between sex chromosomes explains Y or W degeneration^{60–62}. Differences in the evolutionary paths of UV compared to XX/XY and ZW/ZZ ones were predicted by Bull, and some of them have been documented in bryophytes²⁸. In brown algae, the U/V structure may undergo successive inversions and TE accumulation, inevitably reducing gene density (this is not remarkable, as the text says). Low gene density may also reflect gene loss by degeneration or movements out of the SDR. The main difference is that, in the haploid stage, deleterious mutations are more efficiently removed by selection than in diploid systems.

RE: The section has been re-written. We removed the word “remarkable” from this sentence.

The connection with TE enrichment in the Ectocarpales sex-linked regions would be easier to explain in an expanded section on such accumulation. The authors speculate that the U/V-SDR acts as a source of DNA transposons, and the SDRs subsequently expand to the PARs through local movements of such TEs (in larger genomes, this signal is diluted by the increased colonization of LTR elements). They also suggest that the SDRs therefore form 'gene cradles' or 'gene nurseries', perhaps through U/V chromosomes being enriched in heterochromatin, and therefore possibly having high mutation rates (this repeats earlier discussion of this possibility). These ideas are not implausible, but they are presented vaguely, and there is no substantial evidence supporting them, as already noted.

RE: We understand the importance of expanding on the TE aspect of the paper, so we explore the idea further with new results (see **Figs. S13-14**) and in the discussion. We note that the detailed analysis of *Ectocarpus* TEs and their regulation by sRNA is object of another study, unrelated to this manuscript.

We would like to clarify the following:

- 1) we don't think that the SDR expanded to the PARs through TE movement, but rather that the DNA TEs that accumulated in SDR hopped locally into the PARs, increasing their overall presence in the sex chromosome when compared to the autosomes.
- 2) We don't think that the SDRs act as gene cradles, but rather the entire sex chromosome. We improved **Figure 3** of the manuscript and generated an additional supplementary figure (**Fig. S12**) to show this pattern more clearly.

We strongly believe that there is substantial evidence that allow us to reach these ideas:

- While direct measures of mutation rates in the sex chromosome are not possible to obtain, we did find higher *K_s* values in the sex chromosomes (particularly in the PARs),
- heterochromatinization of the brown algal sex chromosome has been demonstrated (Gueno et al., *NAR*, 2022).
- The link between both of these processes (mutation rate and heterochromatin) has been shown (DOI: [10.1038/s41467-021-26108-y](https://doi.org/10.1038/s41467-021-26108-y)), and the theoretical background on the relation between heterochromatin states and mutation rates are sufficiently documented (Makova & Hradison, 2015)

Therefore, we believe that we have ample evidence to propose a model to explain the gene cradle pattern.

We have nevertheless modified this section to clearly distinguish between the evidence we provide in the results section from our hypothetical model in the discussion and we refrained from overinterpreting results.

Minor comments

The writing is not always clear enough. Examples include the following:

In the text “Many genes located in the pseudoautosomal region (PAR) of Ectocarpales have moved into the V-SDR of *U. pinnatifida*, *D. herbacea* and *D. dichotoma*”, does “have moved into” simply mean “are now within”? And what mechanism causes these movements? Is it through rearrangements, or do the PAR boundaries change?

RE: We agree with the reviewer that genes from the PAR did not actually “move” to the SDR, but were integrated through the extension of the SDR boundaries. We modified this sentence to make the statement clearer. These

genes entered the SDR most likely via inversions (as explained in the following sentence, and now highlighted in new Fig S8).

Genome assemblies are stated to be better quality for males than females, but no details or references are given.

RE: The assembly metrics of all the genomes used in this study can be found on Table S1. We also included a new main table (Table 1) for the information to be easily available for the readers.

In this sentence, it is unclear what is meant by “assuming parallel U/V-SDR evolution”. As one of the references cited in Bull’s work, possibly this means that processes occurring on the U are assumed to be very similar to those on the V, but this should be made clearer. However, the case of the monoicous liverwort species *Ricciocarpos natans*, another haploid organism, shows that this cannot necessarily be assumed, as the U chromosome has been lost and a V retained. Possibly this assumption is plausible for brown algae, but the meaning of the statement should be explicitly clear, and the assumption explicitly justified. The exact aspects of Figs. 2 E and F that tell us this should also be explicitly pointed out. I think the authors are probably referring to the inferred numbers in Fig. 2E.

RE: We agree with the reviewer that, while similar evolutionary processes occurring in the V and U chromosomes cannot be assumed based on haploid plant systems, our data does support this prediction, as can be seen by the similar number of genes between the U and V SDRs of *Ectocarpus* sp. 7 and *Desmarestia herbacea* (Table S2). Rather than Figs. 2E-F, it is Fig. 2D that shows this pattern. We modified this section to illustrate our evidence more explicitly.

It is not clear how often genes are gained or lost, and whether these changes can be reliably inferred over the large evolutionary times in the x axis of this figure. It would also be good to discuss whether these are thought to involve individual genes or not; for instance, how many events were involved in gains of > 20 genes, and are single genes gained by duplications or movements from other genome regions, whereas gains of multiple genes in an event occur by a different mechanism (e.g. an inversion changing a PAR boundary, as the text suggests when it says “independently-acquired gametologs in *D. herbacea*”)? Again, these points should be communicated more clearly.

RE: Gene gain and loss events and ancestral gene content can be analyzed across very broad timescales, including > 450 Million years of evolution (examples of more distant inferences on gene gain/loss events across a phylogeny can be found in <https://www.nature.com/articles/s41588-024-01737-3> and <https://www.nature.com/articles/s41477-018-0188-8>). However, current methods of ancestral state reconstruction are unable to assess the exact number of evolutionary steps that led to the observed gene gains and losses on each branch in the phylogeny, since these methods only calculate the putative changes that occurred between nodes in the tree.

We now expanded the information on gene gains: we modified Fig. 2E to highlight the evolutionary origin of the acquired genes in the SDR of each species, indicating whether they belong to SDR border expansions into de PARs, gene movements into the SDR from other chromosomes or gene birth events within the SDR. We also include a new supplementary table (Table S5) which comprehensively details this information.

I think that genes with sex-limited expression is meant when “sex-limited genes” is used. If so, this should be corrected.

RE: We refer as “sex-limited genes” to all the genes in the SDR of one sex that has no gametolog in the SDR of the other sex (e.g., MIN is a male sex-limited gene). We explicitly state this definition in the revised manuscript.

In this paragraph, the meaning is not clear of “newly acquired genes on the SDR”; as these “had similar expression levels to their autosomal counterparts, presumably it means genes gained by duplications or movements, but (as already noted) these changes need to be distinguished from other kinds of changes that can lead to a gene becoming sex-linked.

RE: We modified “newly acquired” for “engulfed” to distinguish this gene gain as an expansion of the SDR borders in *Desmarestia herbacea*. We also expanded on this topic in the revised manuscript (new Table S5 and Fig. 2E).

The anthropomorphic word “co-option” is also unclear in this text, and an evolutionarily correct form of words should be used.

RE: The term co-option, also known as exaptation, is defined as a trait (e.g., an anatomical structure, a gene or a behavior) that initially evolved a function under certain conditions and later changed to perform a different function under an alternative condition. In this case, we propose that autosomal genes (initial condition of the trait) that are integrated into the SDR (alternative condition) may be co-opted to perform sex-related activities

(novel function of the trait). The term is broadly used in evolutionary biology and has a well-defined meaning in the research field (for an extensive review on this topic, see: <https://doi.org/10.1007/s12052-008-0053-8>).

I could not find a definition of "TGRs".

RE: All the instances of "TGRs" in the manuscript were typos for "TRGs" (taxonomically-restricted genes). This has been corrected.

Reviewer #2 (Remarks to the Author):

Barrera-Redondo et al. generate high quality genomes for several species of brown algae, allowing them to follow the birth and evolutionary trajectory of UV sex chromosomes in this clade. Key findings include:

- the UV system is ancestral to the clade
- the non-recombining SDRs were expanded in several lineages, in particular those which have oogamy
- V sex chromosomes have high TE contents, mutations rates, and harbour an excess of young genes
- Transitions to other reproductive systems involved the cooption of V chromosomes

Overall this is a very impressive amount of work on a very exciting system, but I have some questions and comments on the analyses:

* How was the phylogeny inferred, and how reliable is it? This is key to the interpretation of the results and some information should be provided.

RE: We agree with the reviewer that more information is needed regarding the phylogeny, which we obtained from a parallel study on brown algal genomics (Denoëud et al., 2024). The species tree is based on 32 single-copy nuclear genes whose protein sequences were aligned manually using AliView, and whose best-fit substitution models were assessed independently using the Akaike Information Criterion. The tree was generated using a maximum likelihood approach implemented in RAxML with 10,000 rapid bootstraps and the gamma model. Every node in the phylogeny has 99 to 100% bootstrap support values. Divergence times were subsequently calculated using MCMCtree and three calibration points. The MCMC chains were run for 1.5 million generations and the first 200,000 MCMC chains were discarded as burn-in. We included all of this information in the Methods section of the revised manuscript.

* The strata analysis is a bit confusing. Using only Ks to infer strata seems limited, given how noisy mutation rates can be. There then seems to be a combination of ks and phylogenetic distribution of sex-linkage, but the results are not clearly presented.

RE: The limitations of inferring evolutionary strata in U/V systems were also pointed out by another reviewer (see above). While several genes in the SDR borders of *D. herbacea* were clearly acquired recently (see response above), the other evolutionary strata should be interpreted cautiously, since Ks values can be noisy. We pointed out the limitations of strata inference in the revised manuscript. We also include a change-point analysis to support the results we obtained to infer these putative strata based on Ks values. We added a supplemental figure showing these results in the revised manuscript (Fig. S9). We are happy to further down tone, or even delete, any reference to strata in the manuscript if the editor and reviewer prefer, because these are not essential to our conclusions. However, we think that our new version of the manuscript is sufficiently careful in terms of any conclusions and acknowledges all caveats.

* Some claims do not seem completely supported. For instance, the idea that oogamy is associated with expansions of the PAR is exciting. But from the phylogeny, it seems more that a single branch (which happens to be not have oogamy) is different from the rest, which does not provide much power to make inferences. It does not help that this branch differs in other ways (e.g. genome size), further muddling the picture. I think that either stronger evidence needs to be provided or conclusions need toning down.

RE: We agree that, given the phylogenetic position of anisogamous species, it may seem as if there was just one transition that would limit our inferences. This is further idea is conflated by the colors we added to the phylogeny in Fig. 2E. Nonetheless, our results are consistent with the idea that the isogamous branch retained the ancestral state, rather than being the branch that underwent the evolutionary transition. We show that the SDR border expansions in the oogamous species are independent from each other, which might be regarded as three phylogenetically independent "replicates" of the pattern. We agree that it is challenging to establish a causal link between the SDR expansions and oogamy. Therefore, we have toned down this conclusion in the revised manuscript and we refrain from inferring any causal relationship. We also changed the color scheme in Fig. 2E to avoid further misinterpretations.

* I could not understand from the methods how the ancestral gene content, and the inference of gene losses, were done. I assume that genes were deemed to have been gained if they were sex-specific in one lineage but pseudoautosomal in the others? If we assume that the ancestral SDR was very large, and gene loss was so extensive that each lineage ended up with their own subset of these genes plus a few highly conserved ones, would you detect this as gene loss, or would some of these be considered gene gains? What did it take for a gene to be considered lost from an SDR?

RE: We agree with the reviewer, and have now provided more information for a correct interpretation of our results. We generated a table matrix containing all the orthogroups that are sex-linked in at least one of the dioicous species. For the ancestral state reconstruction, we coded a gene as "present" if the ortholog was sex-linked in the V-SDR of the species, and the gene was coded as "absent" if the ortholog was autosomal in the species or if it was completely absent in its genome. We used this presence/absence matrix as the input file for

the ancestral state reconstruction that is detailed in the Methods section, which searches for the most fitting scenario of gene gain/loss across the phylogeny. We included this crucial information in the Methods section of the revised manuscript. Additionally, we now include the SDR gene presence/absence matrix as a supplementary table (new **Table S5**). Note that unfortunately there is no (recombining) outgroup in the brown algae, precluding the inference of the ancestral gene content, so it is not possible to define how many of the ancestral genes were lost in all species.

In the reviewer's hypothetical scenario of a large ancestral SDR, the observed pattern in the current species would require a large number of independent gene loss events at each branch of the tree and the retention of a few ancestral shared genes. Luckily, the method we used for the ancestral state reconstruction is agnostic to possible scenarios of large vs small ancestral SDRs, and chooses the most fitting scenario based on posterior probabilities. In this case, a small ancestral SDR that expanded independently on each lineage has a higher posterior probability compared to a scenario of large ancestral SDR with massive gene losses. This result is supported by the low gametolog *Ks* values observed in the inferred independently-acquired SDR genes of each species.

* In the discussion of young genes and mutation rates, predictions and patterns for the PAR and SDR are not clearly distinguished, making it hard to interpret what could be driving these patterns.

RE: We agree that the manuscript requires a more detailed discussion regarding the properties of the PARs and the SDR in the context of mutation rates and the gene cradle model. We expanded on this topic in the revised manuscript, but we refrain to extensively speculate because of word limit and comments from reviewer 1.

* Results of statistical tests should be reported throughout the manuscript to support the conclusions, and not just in the methods and supplementary materials.

RE: All results of statistical tests are reported throughout the results section of the revised manuscript.

* I found the many small circular plots difficult to read. In general I thought the figures were beautiful but not always completely informative as to how the authors came to the conclusions in the text. Maybe again this was partly due to the fact that significant differences were not consistently illustrated.

RE: We appreciate the reviewer's feedback on the figures. We made significant changes to all the figures in order to increase clarity. We removed the Circos plots and instead present the data as linear plots, which are hopefully easier to read (**Figs. 3-5**). Statistical analysis are presented in the figures and in the text / tables.

* It was not clear from reading the text which genome assemblies were new and which were published. Some quality measures should also be provided for new assemblies.

RE: The detailed information regarding the source of each genome assembly and their associated assembly metrics can be found on **Table S1**. We agree this information is important, so we now include previous **Table S1** as a main table in the text (**Table 1**).

* Similarly it would be helpful to have a (supplementary) figure showing male and female coverage of the SDRs (or the kmer equivalent).

RE: We agree that this information is relevant. We added supplemental figures with the YGS analysis in the revised manuscript (**Figs. S1-S4**).

Other / more detailed comments:

* Fig 1: several chromosomes appear inverted (e.g. between *Undaria* and *Fucus*). I think this simply reflects the fact that opposite strands were assembled in their respective genomes, and not a biological difference. Fixing this would make the picture clearer and help highlight true differences.

Also, in panels B and C, could you add a scale for Mbs?

RE: We incorporated all the suggested changes to Figure 1.

We noticed that many genes located on the pseudoautosomal region (PAR) of *Ectocarpales* had moved into the V-SDR of *U. pinnatifida*, *D. herbacea* and *D. dichotoma*.

 this is a funny way to put it, as it implies gene movement as the mechanism, rather than expansion of recombination suppression. Maybe something like "many genes located in the PAR of *Ectocarpales* are included in the V-SDR" would be clearer?

RE: We appreciate the reviewer's suggestion; the sentence was modified as advised.

suppression event leading to the birth of U/V sex chromosomes occurred between the split of *S. ischiensis* and

D. dichotoma around 450-224 Mya²⁰ (Fig. 1A).

 I don't think something can occur between the split of two lineages, which is one time point. "after the split"?

RE: The sentence was modified as advised.

The evolution of the SDRs is driven by expansions and gene gain rather than gene degradation

 Is this meant to be "expansions of the non-recombining region"? I would make it explicit, otherwise it could mean gene family expansion through duplications.

RE: We agree; this sentence was modified.

Despite differences in SDR gene numbers, both species show a similar ratio of gametologs (53-61%) and sex-specific genes (47-39%; Table S2).

 It would be helpful to provide numbers in the text, otherwise one must keep going to the supplementary materials to be able to follow.

RE: We added the number comprising the gametolog/sex-specific ratios in the manuscript.

Note that U and V-SDR gene counts coincide in both species (Table S2), supporting parallel evolution of U and V chromosomes  I assume both species are Ectocarpus sp. 7 and D. herbacea, which seem to be the ones for which U chromosome assemblies are available? But maybe this could be more explicit.

RE: The reviewer is correct about the species. We modified this sentence to be more explicit.

Ks analysis showed that the U/V-SDRs of Ectocarpus sp. 7 have mostly highly-divergent gametologs in two old strata ($K_s > 1$), with only two gametolog pairs in a recent stratum ($K_s < 1$; Fig. 2B-C, Table S3). In contrast, D. herbacea exhibited at least four strata (Fig. 2B-C). The K_s values were broadly consistent between orthologs in Ectocarpus sp. 7 and D. herbacea (Table S3).

 This seems like a strangely arbitrary way of classifying things. How did you determine the number of strata? Were any kind of statistics performed? Which of the genes are sex-linked in both clades and which were acquired in specific lineages? Could this be shown in fig 2C?

RE: Our definition of evolutionary strata was based on K_s thresholds ($K_s > 3$, $K_s > 1$, $K_s > 0.1$, $K_s < 0.1$), without any explicit statistical test. We understand these thresholds were arbitrary, and the K_s values across the SDRs is more continuous than discrete, which might be explained by the difficulties of defining evolutionary strata in U/V systems, as pointed out by reviewer 1 (see above). We have now performed a change-point analysis, which shows that the highest density intervals of each change point support our initial classification of evolutionary strata. This information is added as a supplementary figure (Fig. S9) since Figure 2 is already saturated with information. We note, again, that strata in UV systems are very hard to detect so we refrain from over-discussing this aspect on the manuscript.

We also note that, besides the seven genes from the ancestral SDR, all the other genes were independently sex-linked by each species, including those that were linked in parallel in different species. These independently-acquired genes can be seen in Fig. S6D. We also added a new supplementary table with all the relevant information regarding the relative acquisition of each gene into the V-SDR of all the five dioicous species (Table S5).

To examine SDR size dynamics, we reconstructed the ancestral state of SDR gene content. We focused on the V chromosome (due to better male genome assembly quality, and assuming parallel U/V-SDR evolution²³⁻²⁵). This analysis revealed that brown algal V-SDR evolution is driven by lineage-specific expansions, rather than reductions, from a common ancestor with few SDR genes (Fig. 2E-F).

 it would be very helpful to see a (supplementary) figure to 2F for all the genes in all the SDRs, not just the ones that were inferred to be ancestral.

RE: We appreciate the reviewer's suggestion, as this information fits perfectly with the updated assignment of evolutionary strata on the SDR (see above). We included this information as a comprehensive supplementary table (Table S5).

Figure S6. Distribution and proportion of classified TEs across the chromosomes of six brown algal species. The sex V chromosome or sex-homolog is highlighted in green. The residuals in the V sex chromosome were used to interpret the enrichment or depletion of Unclassified (Unknown) repetitive elements for each species.

 I found these figures impossible to read and interpret. It may just be my tired eyes, but you maybe you could consider modifying them to more standard plots, and make sure the font size is large enough to be easily read?

RE: We understand the confusion, so we replaced it for standard boxplots that are easier to read (Fig. S14). We also replaced the Wilcoxon rank sum tests for permutation tests, since this allowed us to compare the unclassified TE content in each genomic compartment (autosomes, PARs and SDR).

Figure S7-S9: can you state what species were used for Ks calculations?

RE: We added this important information on the figure legends of **Figs. S7-S9**. We note that the information is also available in the Methods section of the manuscript.

We observed an inverted pattern between V and U-SDR in *Ectocarpus* sp. 7 and *D. herbacea* (Fig. 2A-B), demonstrating that inversions may lead to suppressed recombination between sex chromosomes.

 Inverted pattern is not very clear; maybe something like "inverted gene order" would work?

RE: We modified this sentence to avoid any potential confusions.

Consistently, ancestral V-SDR genes were associated with higher Ks values, while independently-acquired gametologs in *D. herbacea* had lower Ks values (Table S2, Fig. 2C).

 I can't see which genes are ancestral vs acquired in fig 2C. Is this information there?

RE: We highlighted the ancient SDR gametologs of *Ectocarpus* and *Desmarestia* as black squares on Fig. 2C.

SDR genes remain in the sex-homolog of the species that have lost their UV system (Fig. 2F, Table S4), emphasizing their importance for pathways in sex determination even in absence of sex chromosomes.

 "sex chromosome homolog"?

RE: We modified the term "sex-homolog" for "U/V-homolog" throughout the manuscript, since this term is shorter and conveys the same information as "sex chromosome homolog".

Intriguingly, we found the V-SDR size to be strongly linked to the extent of sexual dimorphism (Fig. S2A).

Species with low sexual dimorphism (anisogamous) retained the ancestral V-SDR genes with very few gene gains, suggesting they may represent the V-SDR ancestral state. Oogamous species each independently gained diverse V-SDR genes

 How many datapoints do you have? Can you run statistics on this? Otherwise I would move this to the discussion, as it seems rather speculative.

RE: We agree that three datapoints (*D. dichotoma*, *D. herbacea* and *U. pinnatifida*) are insufficient to run a meaningful statistical analysis. Nonetheless, this pattern is still observed in our data (see **Fig. 2E** and **Fig. S6**) and we believe it should be mentioned in the results section. We have rephrased the section to be open about the small number of datapoints and its drawbacks. We are still willing to move this section to the discussion if the reviewer has a strong opinion on this issue.

On a similar note, these species also apparently underwent genome expansions, which could correlate with structural rearrangements that expand the SDR.

RE: We appreciate the reviewer's feedback on this possibility. However, the observed pattern of bigger SDR sizes in the oogamous species is not only reflected in terms of absolute SDR size (bp), as expected by overall genome expansions, but also in terms of the total number of protein coding genes contained within each SDR, as well as in the relative ratio of SDR size relative to the total length of the sex chromosome (see **Fig. S6A**). Hence, it's not that the SDR became bigger alongside the entire genome, but that the SDR is bigger with respect to the entire length of the chromosome. Furthermore, we added more information in **Fig. 2E** showing that these species experienced several gene gain events through expansions of the SDR borders, but also through translocation events of autosomal genes, which are better explained through non-neutral processes such as antagonistic selection. Therefore, we don't think this pattern can be explained merely through a conflation of genome expansions in the oogamous species.

Oogamy has been suggested to be the ancestral state in the brown algae¹⁵ or its crown radiation clade, but our finding of independent V-SDR expansions positively associated with changes in the level of sexual dimorphism (Fig. 2E; Fig. S2A) supports instead the recurrent, independent evolution of oogamy in the brown algal lineage.

 Again this seems rather speculative

RE: We agree that this sentence is speculative. We deleted it, since it is also redundant with a section in the discussion.

However, we observed an enrichment of male-biased genes on the sex chromosomes in all species except *D. dichotoma* (Fig. S5B).

 presumably not on the female sex chromosomes?

RE: The observed sex-biased gene enrichment is on the PAR, which are shared between the male and female sex chromosomes. We clarified this misunderstanding in the text.

We previously proposed a theoretical model where generation-antagonistic selection drives TRG

accumulation, favoring the retention of young sporophyte-beneficial loci in the Ectocarpus sp. 733 UVs.
 Does the model make predictions for the SDR or the PAR (or both)?

RE: Yes, the model makes prediction about the retention of these sporophyte-biased genes specifically in the PARs. We added this information in the manuscript.

 Similarly, can you separate the PAR and SDR in the violin plots in figure 3B? The presence of young genes seems to be driven by the PAR in some cases, but not others.

RE: The reviewer is correct in highlighting this the observation of TRGs specifically in the PARs. We replaced the linear plots in Fig. 3B with violin plots that display the distribution of gene ages across the genomic compartments of every species, alongside with permutation tests that show whether these differences are significant or not (new Fig. 3B). We also added violin plots of autosomes against the U/V-homologs in Figs. 4-5 for consistency, as well as in new Figs. S21-S24.

 Also in panel 3B: I assume that the numbers in parenthesis next to autosome / chromosome V are the number of the chromosome that they correspond to (but please specify this).

RE: The reviewer is correct; these numbers represented the chromosome number in the genome assembly. However, we decided to remove previous Fig. 3B from the revised manuscript (see below).

But how were the autosomes picked? For example, in *D. dichotoma*, the sex chromosome looks extremely similar to autosomes 1 and 3 -- why show autosome 4?

Re: We chose autosomes with a similar size as the sex chromosomes of each species to highlight the differences in the genomic composition of the sex chromosomes in a clearer manner. As the reviewer highlights, chromosomes 1 and 3 in *D. dichotoma* are, for reasons that we yet ignore, similar in their TRG composition as the V chromosome. Nonetheless, the TRG content in chromosome 1 is statistically different from the V chromosome in *D. dichotoma* (p -value = $2.85e-5$), although these differences are barely significant in chromosome 2 (p -value = 0.018). Including any of them in the plot would visually undermine the statistical differences that we detected regarding TRG enrichment in the sex chromosomes across all the analyses dioicous species when compared to the rest of their genomes (see Table S12). We decided to remove previous Fig. 3B from the revised manuscript to avoid any suspicion of cherry-picking, and added the p -value significance thresholds to Fig. 3A for transparency in our results.

Consistent with this model, sporophyte-biased genes are indeed enriched in the sex chromosomes of Ectocarpus sp. 7 and *U. pinnatifida*, but less so in *D. dichotoma* and *S. promiscuus* that lack heteromorphic generations^{15,31} and consequently no scope for generation-antagonism (Table S13).

 Again this seems like a small number of data points

RE: We believe that the observation of lack of sporophyte-biased genes in *S. promiscuus* and *D. dichotoma* is worth mentioning in the results, but we have removed the section about heteromorphic generations, due to its hypothetical nature and the lack of statistical power to test this idea.

If synonymous mutations behave neutrally^{36,37}, K_s is a proxy for mutation rates^{38,39}. Consistently, we found higher K_s values in the V sex chromosomes across all dioicous brown algae (Fig. 3A; Table S14), suggesting higher mutation rates relative to autosomes. Young gene enrichment and higher K_s values were localized in the PARs of the Ectocarpales (Fig. S7-S8), but this pattern extended to the entire sex chromosome in species with larger V-SDRs (Fig. S9-S11).

 I think that in a fully nonrecombining region, neutral mutations will necessarily be under the effect of linked selection, and will not behave neutrally.

RE: We appreciate the reviewer's comment. This might actually be a compelling reason of why these mutations are mostly seen on the PAR. Mutation rates should be high on the entire chromosome, but they only behave neutrally on the PAR, and hence might be more prone to fixation. We added this hypothesis in the revised manuscript.

Dear Vera,

Please find below our point-by-point response to the reviewers' comments. We believe we have addressed all the concerns raised. We would like to thank you and the reviewers once again for your valuable insights, which have helped improve our manuscript.

Best wishes,
Susana

Reviewers' comments:

Reviewer #1 (Remarks to the Author):

This ms is now a bit easier to understand, and the material is organised better, although a number of things are still not explained as needed (as detailed in comments below). The new Table 1 is very helpful, and I thank the authors for providing this information in an accessible manner (my small problems with this table are noted below). Now that the main results are clearer, however, it seems clear that the emphasis is excessive on the idea that the sex-linked regions of these chromosomes have undergone changes like those in the sex chromosomes of some diploid species, involving new chromosome regions stopping recombining (and forming new “evolutionary strata”, as summarized in Figure 6). Rather, the overall picture appears to be a remarkable conservation of the completely sex-linked region over a vast evolutionary time, with at most rather small numbers of genes having become completely sex-linked after an ancestral completely sex-linked first evolved. This is an important finding, as it may be evidence against one of the ideas that has been suggested as a mechanism for the loss of recombination between members of sex chromosome pairs (Jeffries et al. 2021). This is cited as ref. 61, but not related to the findings.

Re: We thank the reviewer for highlighting this important finding. As suggested, we have rephrased several sections, including the abstract to move the emphasis to the remarkable conservation of the SDR across evolutionary time.

I next outline and comment on the main interesting results. Please note that some or most of the information I failed to find is probably actually present, but unless it is explained clearly in relation to the obvious questions that readers may ask as they read the text, it is very hard to understand the answers, given the mass of very condensed information. The organization of points below is intended to help the authors think about how to present the chief findings and their implications for sex chromosome evolution.

Re: We thank the reviewer for the suggestions to improve the clarity and highlight the main findings. We have followed the advices and focus more the text of the manuscript by, for example, removing the section on *de novo* gene birth. We hope the manuscript now reads more easily.

1. SDR identification, homology, sizes and the role of repetitive sequences

As Table 1 shows, 6 dioicous species were studied (haploids thought to have separate male and female gametophytes), two with both the U- and V- linked regions being ascertained, and four with just the V, plus 2 monoicous species and one species whose sex system is not known, and one species termed dioecious because its diploid sporophyte stage has separate males and females (maybe also the gametophyte stage — this is not made clear).

Re: The dioecious species *F. serratus* does not have a gametophyte generation (Fucales have diploid life cycles, i.e., the only haploid stage are the gametes). We have made this clear now in the legend of the Figure 1 and Table 1.

The new haploid dioicous species for which both male and female gametophytes were sequenced, *Desmarestia herbacea*, indeed proves to have a plausible sex-determining region, based on finding male- or female-specific variants that identify inferred male- and female-determining fully sex-linked regions (respectively termed “V-” and “U-” SDRs or “V- and U-linked regions”). The gene content of the

chromosome corresponds with that carrying the U- and V-linked regions in the previously sequenced *Ectocarpus* species 7, even though these are very distant relatives, showing that this chromosome homolog has carried the SDRs for a long time. This suggests that this could also be the sex chromosome in other brown algae, inherited from a common dioicous ancestor. Putative SDRs of 3 other species (*Undaria pinnatifida*, *Dictyota dichotoma* and *Scytosiphon promiscuus*) were ascertained without assembling sequences of both sexes (see further comments below about the explanation of the evidence — in another *Ectocarpus* species, this appears to be assumed, which is plausible).

Re: The reviewer is correct in noting that we were unable to obtain chromosome-level assemblies for the females of the three species, as we detailed in the Methods section. Indeed, high molecular weight (HMW) DNA extractions from female samples was particularly challenging. However, it is important to note that chromosome-level assemblies of the female genome *are not required* to identify the male SDR. We used Illumina reads generated from female gametophytes to identify the V-SDR regions.

In response to the reviewer's comment, we have now added further details in the Methods section, including coverage plots and the set of primers used for PCR validations. Additional explanation of how we identified the sex-linked regions is also provided in the Methods.

The first main result is that the sex-determining regions in all 6 haploid dioicous species appear to be on a homologous chromosome, and Fig. 1B shows details for 5 of them (*Ectocarpus* species are represented by just the one previously sequenced, as the other is so similar). In 4 of these 5 species, the V-linked regions are in the middle of the chromosome, but in *U. pinnatifida* it is at the left-hand end of the assembly. The sizes of the whole chromosome, the SDR, and the pseudoautosomal regions (PARs), differ between the species. It would be helpful to mention that the two *Ectocarpus* species have the physically smallest sizes of all three types in Table 1 (in the table, the sizes should be shown with commas separating the thousands, so that readers can understand the relative sizes).

Re: We agree - we have now added spaces to separate each thousand values in Table 1. We also highlighted that the *Ectocarpales* have the smallest SDR sizes.

For the *Ectocarpus* species with both U and V region sequences, both are smaller than the other Vs (or one U) with which they can be compared. In other words, it appears that some species have bigger genomes than others. The table also shows that the smaller genomes have the lowest repeat densities, whether one measures the whole genome, the sex chromosome, or one of the SDRs. As expected, the SDRs had higher repeat densities than autosomes. I find it much more remarkable how little the SDRs appear to have changed in the very long evolutionary time periods involved, given the high repetitive densities in these genomes.

Re: We apologise if we have not been sufficiently clear, and hope that the text now reads more clearly. We now emphasis how little the SDR has changed over evolutionary time (including in the abstract). We also stress that there has been no turnover of sex chromosomes, i.e., all brown algae share the same U/V chromosomes, and although the SDR has expanded, the extent of this expansion is relatively small when considering the large evolutionary timescales in this group of organisms, as suggested.

Despite these size differences, the PARs appear largely syntenic, which is also as expected. Even in *U. pinnatifida*, whose SDR is not in the middle of the chromosome, the right-hand PAR is syntenic, and synteny is not greatly disturbed by the change in the left-hand PAR's position. The SDRs are more rearranged between the species, consistent with a long history of evolving without crossing over.

Re: The reviewer is correct, the PARs are largely syntenic, as are chromosomes in general. We mention that the SDR are rearranged between species in line 77.

It would be helpful to show the numbers of PAR genes in Table 1, for species where the chromosome assemblies allow estimates, so that readers can see the proportions of genes that are fully versus partially sex-linked. Fig. 1 suggests that the proportions are around 1/3rd or a quarter. Such basic information helps the reader to understand what is going on, and clearly the numbers and types of genes in these regions are central to understanding the authors' thoughts.

Re: we added the numbers of PAR genes in Table 1.

Another interesting result (which is not generally expected) the PARs also have higher repeat densities than the autosomes (in all 5 species with V sequences and both species with U sequences). This is not

discussed, but a possible explanation that might be worth thinking about is that these species may not reproduce sexually very often. If so, the actual frequency of crossing over in the PARs would be low, and U- and V-specific repeats would be expected to accumulate more than on the autosomes, possibly resulting in a higher density estimate when a V chromosome assembly is analyzed (the authors might think about whether the approach used in the analysis might lead to such an outcome). Other than this, the overall repetitive sequence results are largely as expected, and confirm that, similar to other systems, repetitive sequence densities can differ greatly between species, but are highest in genome regions that rarely recombine.

Re: Indeed, we noticed that the PARs also have higher repeat density compared to autosomes. While the hypothesis that these patterns may reflect infrequent sexual reproduction sound interesting, we fail to understand how low sexual reproduction would impact the PARs in any different way than the autosomes, since both would have low crossing over frequencies. Furthermore, current evidence suggest that sexual reproduction is common in brown algae. In *Ectocarpus*, sexual reproduction is known to occur regularly in natural populations (Couceiro et al., Evolution, 2015). Moreover, species such as *Undaria pinnatifida*, *Dictyota dichotoma*, and *Scytosiphon promiscuus* are annuals, and—given their haploid–diploid life cycles—they are required to undergo sexual reproduction each year. Fertilization ensures recruitment via the development of a new generation of microscopic diploid sporophytes although parthenogenesis has been sometimes observed in *Scytosiphon* taxa (e.g. Hoshino et al. Nat Ecol Evo 2024). Note that the process of transposon local hopping is a likely mechanism for higher TE content in the PARs, as we discuss in line 670 of the manuscript.

More interesting results require analyses of the genes in the SDRs, to ask whether they differ, and, if so, in what way, and I believe that this aspect is the authors’ main interest.

Re: We already tried to evaluate the function of the U/V-SDR genes in Table S4, S6 and in the main text. As we previously explained, the relatively small number of genes in the SDR and the evolutionary distance between species make the functional annotation and enrichment analysis of these genes challenging, complicating any potential insight about their biological relevance in a statistically meaningful manner. We tried to performed a Gene Ontology enrichment analysis to evaluate possible gene functions that are enriched in the U/V-SDR genes when compared to the rest of the genes in the genome. Nonetheless, we found that the enriched terms in the V-SDR of *Ectocarpus* sp. 7 lack any relevant biological insight. As an example, here are the five most enriched GO terms for the *Ectocarpus* sp. 7 V-SDR:

GO.ID	Term	Classic Fisher p-value
GO:0048148	behavioral response to cocaine	0.0028
GO:0007555	regulation of ecdysteroid secretion	0.0028
GO:2000052	positive regulation of non-canonical Wnt signaling pathway	0.0028
GO:0008062	eclosion rhythm	0.0028
GO:1905426	positive regulation of Wnt-mediated midbrain dopaminergic neuron differentiation	0.0028

We believe that these types of tests are intended only for model organisms, specifically in animals, and are not useful for this manuscript. Any further attempt to study the function of these genes would require knock-out assays, which are out of the scope of this manuscript.

2. SDR gene content and strata

The abstract claims that “Over time, nested inversions caused stepwise expansions of the sex locus, independently in each lineage, and concomitant with the increasing morphological complexity and level of sexual differentiation observed in brown seaweeds”. In other words, the claim is that the SDR boundaries differ between the species, similar to what is found in sex chromosomes of diploid species in which evolutionary strata have evolved by stepwise suppression of recombination. Other kinds of change in the number of sex-linked genes can also potentially be analysed, including gene movements into the SDR

from other chromosomes, or the opposite, or evolution of new genes from non-coding sequences within the SDR, or genetic degeneration involving deletion of genes from it, as is documented in diploid systems.

Re: We have modified the abstract as requested. As explained in the previous round of review, we already explore the different scenarios proposed by the reviewer. These results are shown in **Fig. 2E** and **Table S5**.

It would be helpful to mention that the SDRs in all the species include small numbers of genes (the highest number is 50, in *D. herbacea*, but Table 1 states that 30 are 'viral derived', leaving 48 as the highest number, in *S. prom*). If I have understood Fig. 2E correctly, 7 genes are inferred to have been V-linked in the common ancestor, with genes being added independently in different lineages.

Re: There is a misunderstanding in the reviewer's numbers – the table shows that the SDR sizes vary between 18 (*Ectocarpus*) and 52 genes (*D. dichotoma*). We added a sentence to mention the relatively small sizes of the SDRs across brown algae.

It is mentioned that the centromere in the V chromosome of *Ectocarpus* is found within the SDR (Liu et al., Nat. Comm 2024). This may partly explain the low gene density, even in the species in this genus, whose SDRs are smaller than in the other species. Given the similar locations in the V chromosomes, presumably the SDRs include the centromeres in other species also.

Re: We evaluated the possible influence of the centromere in the gene density of the SDR in *Ectocarpus*. We found that, as the reviewer suggest, the CDS content in the centromere is low (3.51%). However, we observe that the CDS content of the V-SDR that excludes the centromere is even lower (2.85%). We suspect this is due to the relatively small size of the centromere (153 kbp) compared to the overall size of the V-SDR (923 kbp). We mention this discovery in lines 214-215 of the manuscript.

We refrained from expanding the centromere test to other species, as identifying centromeres is not trivial and would deviate from the main scope of this manuscript. Note that we have a manuscript in preparation that investigates the evolutionary history of centromeres in brown algae and we confirm that the centromere in the other brown algae is also inside the SDR.

The gene contents suggest that one reason for the different gene numbers is that the SDR boundaries differ between the species (in other words, new strata have formed). The evidence is still not very well explained.

Re: We are unsure how to explain better, as we have accepted all the changes in the text suggested by this reviewer, who has kindly proposed sentences in our previous pdf file. We hope that our current re-writing is clearer.

The text states that many genes located in the PAR of Ectocarpales (does this mean the 2 *Ectocarpus* species? — if so, it would be helpful to explain this) are in the V-SDRs of the other 4 species whose inferred V-linked regions were examined.

Re: Yes, the reviewer understood correctly. We mean that genes located in the PAR of Ectocarpales species (*Ectocarpus* sp7, *Ectocarpus croanorium*, *S. promiscuous*) are located in the V-SDR of the other species - this is explained in the main text "*Many genes located in the pseudoautosomal region (PAR) of Ectocarpales were engulfed by the V-SDR of U. pinnatifida, D. herbacea and D. dichotoma (...)*". We added a sentence in the manuscript to state the species that encompass the Ectocarpales in our study.

If I have understood Figure 1 correctly, excluding *U. pinnatifida*, whose SDR has apparently moved from the middle of the chromosome to its left-hand end (but even so appears to include mostly the same genes), the SDR boundaries are remarkably similar.

Re: The SDR boundaries are shifting more than it appears at first glance, but we understand that these changes are hard to see due to the color scheme we used to display them. We have now marked the PAR genes of Ectocarpales that became sex linked in other species in the synteny figure (new Figure 1). The "link" between SDR and PAR orthologs are now marked in purple, displaying this result more clearly.

It is strange to compare both the *D. herbacea* U and V with just the *Ectocarpus* species 7 V. As the U and V of any species differ greatly, surely the relevant comparisons needed to test for expansions would be between the two species' Vs and also between their Us. This would be easiest to understand after these

had first been described in each species, so that we understand which genes are specific to a species' V or U, versus shared between the two, and potentially also shared by both species, and then which sex-specific genes are shared by the two Us, or the two Vs.

Re: There is a misunderstanding, as the figure that the reviewer mentions aims to compare the *Desmarestia* U and V to *Ectocarpus* sex chromosome only to show the movement of the PAR genes (in *Ectocarpus*) into the SDRs in *Desmarestia*. Note that it does not matter which *Ectocarpus* chromosome we use here (U or V) because both have the same genes on the PAR.

The change appears quite minor, given the distant relationship between the two species, and the roughly six-fold larger SDR size in *D. herbacea* (which is independent of extra genes, and reflects its larger genome size). This is not reflected in the wording in the abstract (quoted above) and elsewhere in the ms, including in Figure 6, that the SDRs have expanded via the evolution of "strata".

Re: In the first round of review, we had already addressed this point.

We performed a statistical analysis and clearly demonstrated that, contrary to what the reviewer states, the size of the SDR across species is *not* independent of the number of genes within the SDR. Therefore, the increase in SDR size reflects significant increases in gene content and not only genome expansion due to transposable elements. In other words, there is a strong statistically significant positive correlation between the number of genes in the SDR and the size of the SDR in base pairs (see **Fig. S6B**).

Figure S8 shows the single more detailed comparison that is possible. It shows that a few genes are just outside the PAR boundaries of *Ectocarpus* species 7 V, and just inside the V- and U-SDRs in *D. herbacea*, an astonishing similarity in the SDR gene contents of such distantly related species. Compared with the *Ectocarpus* species 7 V, the inferred *D. herbacea* V has 2 genes in the left-hand PAR (in the same order in both species), and about 9 in the right-hand PAR (with several rearrangements). The same PAR genes should have changed to U-linked, but this is not mentioned, and it is not mentioned if this has happened, and the two species' Us are not compared (if I have understood Figure S8 correctly, compared with the *Ectocarpus* species 7 U, the inferred *D. herbacea* U does also include the same 2 genes, though this is not the ideal comparison). There are more rearrangements, which I think might reflect changes in the ancestry of either species. Fig. 2E summarizes the inferred changes in the Vs, with 13 changes of this kind in the *D. herbacea* V.

Re: The point raised about PAR genes inside the SDR of *Desmarestia* is already addressed above. The aim of the figure is to illustrate how the *Ectocarpus* PAR genes are now located inside the *Desmarestia* U and V-SDR. The U chromosome of *D. herbacea* is found on the top part of the figure, and as the reviewer suspected, the PAR genes in *Ectocarpus* are also U-linked, meaning that they are retained as gametologs within the two SDRs of *Desmarestia*. There is, of course, no need for a separate comparison to the *Ectocarpus* U and V chromosomes, as both contain the exact same genes on the PAR.

Estimated numbers of changes in the lineages leading to other species are based just on comparisons between the Vs, as U sequences are not available for most of the dioicous species. It should be noted that these inferences rely on perfect assignment of genes to the PARs versus SDRs, and some of these may not be reliable, given the approach used to test for complete sex linkage, and the other possible kinds of changes (see comments below).

Re: The SDR and PAR genes are clearly and reliably identified, as all identified contigs were systematically tested by bioinformatic approaches (YGS, coverage) and experimental PCR. We have further explained our approach in the methods section and added the primer lists and example gel images to the supplementary materials.

The evidence for rearrangements in the SDRs is convincing, but it is not clear that they show a significant tendency to be near the PAR boundaries, as appears to be claimed. If they do, of course this suggests that these regions may be difficult to assemble reliably, making the assignment of gene locations less certain, as just suggested.

Re: We do not understand the reviewer's comment regarding the rearrangements: we have never claimed that there is any "significant tendency to be near the PAR boundaries" anywhere in the manuscript.

Regarding the concern that rearrangements near the PAR boundaries could indicate assembly difficulties or uncertainty in gene location assignments: we would like to emphasize that all genomes used in this

study were generated from haploid individuals which substantially reduces assembly complexity. Additionally, we generated long-read sequencing data complemented by Hi-C contact maps, which allowed us to assemble the genomes to chromosome-scale resolution with high confidence and without ambiguity at the SDR-PAR boundaries.

Finally, we rigorously validated the sex-linkage of contigs using two bioinformatic methods and included PCR verification (FigS1-5, Table S1). We believe our strategy provides robust support for contig assignment and sex linkage, and in many ways exceeds the standards typically applied in sex chromosome studies. If the reviewer is referring to another specific methodological approach, we would be grateful if they could clarify so we can address it directly.

Fig. 2E also summarizes inferred gene movements into the SDR from other chromosomes, or new genes within the V-SDRs of species at increasing distances from *Ectocarpus* species 7. The text should mention that the numbers of changes are all small, even across the vary large times involved (the highest inferred number of changes is the 22 autosomal duplications into the *U. pinnatifida* V-SDR).

Re: We are uncertain what the reviewer means with the statement that “the numbers of changes are small” in the V-SDR. In order to define “small”, one has to either calculate a basal rate of gene movement per million years in other sex chromosomes, or one could compare the proportion of added sex-linked genes to the number of ancestral sex-linked genes that are shared across species. In the latter case, there are only seven ancestral V-SDR genes that are shared across species. This means that between 10 genes in *Scytosiphon* and 45 genes in *Dictyota* have been sex-linked independently in each lineage. This means that there has been between 2.43x to 7.43x increase in the sex-linked gene content of the V-SDR. Even for the fraction of autosomal translocations, 22 is much higher than 7. Nonetheless, we modified the text as suggested by the reviewer (“*Although the number of SDR changes is small, (...)*”)

Concerning these types of changes, it is important to ask whether they are functional genes in these species, and to exclude genes in transposable elements that may be present in these V-linked regions.

Re: Yes, we confirm that we have excluded any predicted genes that overlap with repeats, and we consider all remaining genes to be functional based on the presence of open reading frames and their expression levels.

Specifically, we would like to emphasise that *all* genes within the SDR were meticulously manually curated to exclude any TE-related genes from the annotation (see Methods section where we include now a sentence to explicitly mention this manual curation). “*All genes within the SDRs in the brown algal species studied (see below) were manually curated to exclude any TE-related genes from the annotation.*”

The majority of these curated genes possess conserved protein domains and are expressed at significant levels during the reproductive gametophyte stages (see **Table S8**). This evidence strongly supports their functionality, but any further study of their biological function would require experimental methods, such as CRISPR knockouts, which fall outside the scope of this manuscript.

The ‘viral derived’ genes mentioned above are not explained, and the extent to which the other genes have been tested for brown algal function is not clear enough.

Re: Functions are based on transcriptional expression, structural annotations and their comparison to other systems. Brown algae have many endogenous viral elements or EVEs (dsDNA Phaeovirus) that are inserted in their genomes, although their biological implications are still being studied (Denoueld et al. Cell, 2024). We understand this information is relevant for the readers, so we briefly mention these EVEs in the manuscript to inform readers about this phenomenon and cite the relevant paper.

Note that a substantial proportion of brown algal genomes correspond to “unknown proteins” (Denoueld et al, Cell, 2024). As explained above, the biological function of each of these genes would take years to perform (would need reverse genetic approaches) and fall outside the scope of this manuscript.

3. SDR gene content and gene loss (degeneration)

Losses since the brown algal ancestral state are inferred to be few. This is interpreted as minor degeneration, but this conclusion is not justified, as the ancestral state may already have been highly degenerated or gene-poor, if the centromere was within the ancestral SDR, as in *Ectocarpus* (see above). In addition, the small numbers of SDR genes mean that degeneration can be estimated only roughly, at

best. The statements about a difference in degeneration between haploid and diploid species, as a generality, should therefore be considerably toned down.

Re: We agree with the reviewer that the small number of genes and the absence of an extant outgroup with the ancestral autosome makes measuring degeneration more challenging from the origin of the U/V-SDRs to the last common ancestor of the studied brown algae, between 450-224 Mya. However, regardless of the ancestral state at the origin of the U/V-SDRs, there has been very few gene losses in the last 224 million years of evolution between the last common ancestor of our studied species and the extant genomes that have been analyzed, strongly supporting a scenario of low degeneration. What we mean is that, based on our information for these species and their last common ancestor, the ancestral reconstruction clearly demonstrates more gene gains than losses (whereas in most diploid systems it is the opposite). However, we have toned down mentions to 'degeneration' or degradation in the manuscript.

However, rough degeneration estimates might be possible using the genes that are inferred to have become fully sex-linked most recently, for example those in Fig. S8. These were inferred to be PAR genes until relatively recently in the history of *D. herbacea*, and therefore they represent a set of genes that can be examined for loss from the V or U regions since strata formed by loss of recombination of the left- and right-hand PARs.

Re: This is exactly what we do - since these SDR genes are retained in both *Desmarestia* U and V, we infer that degeneration is minimal. This information is already presented in the paper, and the reviewer actually cites it in the very next sentence.

I did not understand the statement that "Most of these engulfed genes into the SDR of *D. herbacea* were retained as gametologs", but possibly this is expressing the idea just outlined.

Re: Yes, this is correct.

If so, it should be explained more clearly, and it presumably suggests that this small sample of genes have not degenerated. If this is correct, please be explicit about the actual numbers ("Most of the genes" is too vague).

Re: We changed the text to clarify that twelve genes were retained as gametologs.

Also the U-V Ks values for these retained gametologs should be mentioned, probably in a small table, to give some idea of the time(s) during which they have remained present. Also, this text now mentions 17 genes, whereas I think that Fig. S9 describes only 16. If properly explained and confirmed, these observations would indeed support expansions of the SDRs of this brown algal species, and lack of degeneration since that event, or events.

Re: We would like to clarify that 17 genes were inferred to have been incorporated into the SDR of *Desmarestia* from the ancestral PAR. Of these, 12 retained both U and V copies (i.e., gametologs), whereas the rest became U or V-specific. In addition, 4 gametologs were already present in the SDR prior to this expansion, resulting in a total of 16 gametolog pairs used for the Ks-based analysis of evolutionary strata (shown in the updated Fig. S10, previously Fig. S9, as well as Fig. 2C).

The Ks values for all gametolog pairs are provided in Table S3, and we now indicate which of these genes are PAR-derived by marking them with an asterisk, as suggested. These values give an estimate of the divergence times between gametologs and support the interpretation of ancient incorporation of PAR genes into the SDR without subsequent degeneration. We hope this resolves the confusion and helps reinforce the significance of the observed SDR expansion.

However, the conclusion of less degeneration than in XX/XY and ZW/ZZ chromosome regions must take account of the different gene numbers between the species studied. The 50 genes suggested by the results in Table 1 probably represents a number that is unlikely to lead to considerable gene losses evolving. Therefore the difference in gene losses from highly degenerated sex chromosomes may simply reflect the small gene numbers in U- and V-linked regions. On the other hand, those small gene numbers might have evolved by degeneration processes in the brown algal ancestor that have now stopped because the numbers are too small for them to continue to operate.

Re: We note that it is not expected for the sex-linked regions of the U and V chromosomes to undergo significant degeneration or lose many genes, due to their inheritance patterns, as Bull remarked (and as the reviewer also points out). Again, without an outgroup, we cannot determine how many genes were present in the ancestral autosome or how many were lost when the SDR ceased recombination. We have revised the statements regarding the comparison of degeneration levels between diploid systems and U/V degeneration levels, as suggested. We have deleted from the abstract and results mention to the degeneration levels too.

An important criticism of the tests for “evolutionary strata” in *Ectocarpus* and *D. herbacea*, however, is that the synonymous site divergence between the gametolog pairs used is almost always extremely high, indicating saturation of the sequences, which implies that the divergence values will not relate to the evolutionary times since recombination stopped. Even if the values were no as high, it is implausible that 11 or 16 gametolog pairs could be assigned reliably to 4 different strata, as claimed in these species. Only in *D. herbacea* are there as many as 4 pairs with values < 0.5 (already too high to be very useful, though it is not made clear whether any correction for saturation was used).

Re: We agree that *Ks* is saturated for several gametolog pairs, which is not surprising given the long evolutionary times since recombination suppression. Note that we previously referred to strata as “putative” in order to tone down any statement. Following the first round of reviews we also did more robust statistical analysis concerning the putative strata, as proposed by R2, and this analysis supported our conclusions. However, we agree that the criticism of the reviewer is valid, and, as mentioned in our previous response, the precise estimation of evolutionary strata is not a central focus of our paper. Therefore, we decided to remove the change-point analysis, modify figure 2C and delete any statement regarding evolutionary strata from the manuscript. We still kept the *Ks* value analyses in the manuscript to disclose all this information to the readers.

If those values refer to the genes that are inferred to have been in the PAR until relatively, but are now SDR genes in *D. herbacea*, it would be helpful to explain this explicitly (see the suggested table above).

Re: Note that the *Ks* values and details of the genes requested by the reviewer had been presented already in **Table S3** (as explained above). We now indicate which of these genes are PAR-derived in *Desmarestia* by marking them with an asterisk, as suggested.

4. SDR gene function evolution and its relationship with sexual dimorphism

The Discussion proposes that the evolution of the male-determining gene *MIN* was involved in the evolution of the ancient U/V system found in most dioicous brown algae, but evidence about how separate sexes evolved in these species will not be easy to obtain, given the huge amounts of time since this event. How strong is the evidence concerning this statement?

Re: Yes we agree it is currently impossible to access the ancestral proto sex chromosome, i.e., to the ancestral hermaphrodite species which would have the proto-U and proto-V. We can only state that *all* brown algae with separate sexes and UV systems have consistently *MIN* specifically in their V-linked regions, and this is the **ONLY** gene that is male linked across all the dioicous brown algae. We have been very careful not to make over statements concerning pushing back the age of the UV system (“*The presence of MIN in distantly related lineages could push the age of the U/V chromosomes further back in time, but more evidence would be required to establish that dioicy existed in these organisms.*”).

It would be helpful to make clear whether *MIN* is the only gene conserved in all 6 V-SDRs, or whether the seven genes indicated in Fig. 2E include *MIN* plus other shared ones.

Re: We have strengthened the statements about *MIN* being the only conserved male-limited, sex-linked gene. We note that *MIN* is the male master determinant gene in brown algae (Luthringer et al., Science 2024). As we mention in the manuscript there are seven conserved male genes, six of which are gametologs, with *MIN* being the only male-linked gene. This has been clearly stated in the text (“*The male-determining gene MIN is the only V-specific gene consistently present in all V-SDRs of the dioicous species.*”). While we can add a few sentences in the discussion to re-explain the results of the *MIN* paper and its role in sex determination, we believe this may be excessive, superfluous information in the context of the present manuscript.

The evidence that SDR gene acquisition promoted the formation of new evolutionary strata is also not convincing.

Re: We thank the reviewer for this comment and would like to clarify our interpretation. We show clear evidence that the boundaries of the SDR have expanded into the ancestral PAR in *Desmarestia*, leading to the stable incorporation of previously pseudoautosomal genes into the non-recombining region. These genes are now fully sex-linked and exhibit divergence between the U and V copies (see Table S3 for Ks values), consistent with the formation of new evolutionary strata as classically defined.

We did not intend to suggest that the *function* or *identity* of the acquired genes directly ‘*promoted*’ strata formation. Rather, we describe a pattern in which the physical expansion of the SDR—whether through suppression of recombination or chromosomal rearrangement—has incorporated PAR genes, producing new strata. This interpretation aligns with widely accepted models of sex chromosome evolution. Note that we removed the reference to evolutionary strata in the new version of the manuscript (see above).

Even in XX/XY and ZW/ZZ sex chromosome evolution models, the description in the ms is not quite correct (it has been proposed that sexually antagonistic selection may lead to the establishment of partially sex-linked polymorphism, which selects for closer linkage across the regions. This is not expressed by the text “sexual dimorphism may lead to...”).

Re: The sentence has been changed according to the suggestion.

The text about expanding this model to U/V systems expresses the interesting idea that greater sexual dimorphism in gametes, which should often involve sexually antagonistic selection, may be associated with larger U/V-SDRs. It is thus important to document that the fully sex-linked regions of the 3 brown algal species with greater sexual dimorphism (presumably meaning oogamy) have additional genes that are not within the regions of the two studied species with anisogamy. As the other reviewer also pointed out, not enough species have been studied for such an association to be tested for statistical significance. Results that are merely consistent should not be described in a manner that gives the impression that they constitute evidence supporting a conclusion. A proper comparative test is needed, to take account of the relationships and test whether changes in the degree of sexual dimorphism tend to be accompanied by changes in the number of SDR genes with relevant functions. The wording is still much too strong, as no such test can be done at present.

Re: Indeed, for the samples studied here, there is an association between oogamy and larger SDRs; however, the small sample size prevents a robust statistical analysis. We have toned down this sentence and emphasized that statistical analysis is not feasible due to the small sample size and have revised the wording to reflect this more accurately. We also clarified in the text that we are referring to oogamy to improve clarity. Additionally, we added in lines 172-173 that all the autosomal translocations into the SDR are also sex-specific genes in *Ectocarpus* and *Desmarestia* (Table S2, Table S5), supporting the idea that antagonistic sexual selection in autosomal loci may be solved through sex linkage.

Finally, the text mentions two chromatin-related transcription factors in the *Ectocarpus* PARs were independently incorporated into the SDRs of four other dioicous species. However, it does not make clear whether these two are the only other SDR expansions claimed. This seems weak evidence for the abstract’s claim about stepwise expansions of the sex locus.

Re: We have removed the sentence referring to the chromatin-related transcription factors in order to streamline the text, and we have revised the abstract to better align with the reviewer’s suggestions. Regarding the comment on “weak evidence,” we respectfully believe there may have been a misunderstanding. The manuscript presents multiple lines of evidence supporting the expansions of the sex locus across different brown algal species. We hope the revised version more clearly communicates this.

5. An abstract

Based on the comments above, a modified abstract might say something like the following, which is intended to highlight what the results and analyses appear to show: Despite their age, the non-recombining regions have changed remarkably little. Increases in size in some lineages mainly reflect genome expansion due to increased repetitive sequence content, but small numbers of genes near PAR borders

have also become fully sex-linked. We evaluate whether gene gains of various kinds are associated with increased morphological complexity and sexual differentiation, and whether new genes evolve unexpectedly often in U and V chromosomes' SDRs or PARs. We also study two situations in which UV-linked regions have changed. First, evolution of hermaphrodites occurred by ancestral males acquiring U-specific genes. Second, the *Fucus* dioecious system involves new sex-determining genes, acting upstream of formerly V-specific genes during development.

Re: We have revised the abstract following the reviewer suggestions.

However, we note that our data demonstrates that there is a two-fold increase in the number of SDR genes between the ancestral state (7 genes) and the current state in *Ectocarpus* (14 genes), and this increase is even more pronounced in other species with larger SDRs, whose size, as we have shown, is strongly statistically correlated with the number of genes they contain—and not just the TE content, as the reviewer states.

NOTEs on the above abstract text

- The abstract mentions the transition from a U/V to the *Fucus* dioecious system. However, the later text states that the *Fucales* probably evolved diploid dioecy from an ancestral monoicous system, which seems inconsistent with the abstract.

Re: The abstract has been changed as suggested above. Ultimately the ancestral system for the all browns is the UV system, but we have rephrased to emphasize that the *Fucus* XY likely evolved from a co-sexual ancestor.

- Also relating to *Fucus*, it was unclear what was meant by “moved down the sex-determining developmental hierarchy”, but I believe that the wording above expresses the concept.

Re: We mean that the master sex determinant gene *MIN* is no longer the ‘master’ sex determinant gene but is still part of the cascade of effector genes involved in sex determination/differentiation. We modified this sentence to be clearer in our statement.

Some detailed comments

Ascertainment of the U/V sex-determining (or maybe fully sex-linked) regions (SDRs) in the dioicous species should be explained better, as this is crucial to the rest of the results.

Re: We have now explained the methods section on how sex-linkage was determined. In addition, we included new supplementary figures showing read coverage distribution and kmer coverage for all dioicous species (Fig. S1-S5) and provided a primer list and gel images used to validate sex-linkage (Table S1).

It is not clear what is meant by saying that *F. serratus* lacks U/V sex chromosomes. I think that readers of Table 1 will need to be told that the phylogeny in Fig. 1 shows that this species evolved from an ancestor that probably has the same U/V system as the dioicous species studied,

Re: We agree, we now added that the ancestral state for the brown algae is UV system.

and they will want to be told whether it has a sex-linked region, and, if so whether the chromosome that carries it is homologous to the one in the haploids with UV chromosomes (which prove to be homologous across the 6 dioicous species studied). The chromosome homologous to the ancestral U/V carries a *MIN* gene, consistent with the overall conserved gene content of perhaps 7 ancestral genes in V-linked regions (see comment above), but presumably not suggesting that this gene is involved in its sex determining system. This should be made clear.

Re: We have added in the legend of Table 1 more details as suggested and changed the text to clarify further.

The sex-linked regions (mostly V-linked, as noted above) were identified “using a combination of bioinformatic and experimental approaches (see methods; Fig. S1-S4)”. Presumably, this refers to the Methods section Discovery of the U/V sex determination regions, and it would be helpful to mention this explicitly. The basic principle used should also be outlined. I believe that it involved initial searches for sex-specific k-mers (I think this is the same as ‘unmatched k-mers’ in Fig. S1.), following the approach of Carvalho and Clark, followed by validation of candidate sequences by their coverage among Illumina short reads of sequenced individuals of both sexes, and then by PCR using 4 individuals of each sex. Presumably, this refers to all the species including those other than the single one in which the U was

assembled. This would allow the region to be inferred in those other species, which is otherwise puzzling (see comment above). It would be helpful to explain that both sexes known for these species.

Re: We have revised the main text to direct the reader more clearly to the “Discovery of the U/V sex determination region” section of the Methods, as recommended. We have also substantially expanded this section to include additional methodological details.

Furthermore, we have added new supplementary figures presenting the coverage and k-mer analysis results for each species, which support the identification of sex-linked regions. In addition, we now include a table listing the primers used to define SDR boundaries, along with gel images that validate the sex linkage of these regions. We hope these additions clarify our approach.

It should also be explained how the chromosomes were numbered. If this used the numbers in *Ectocarpus* (presumably based on homologous gene contents), this should be stated clearly in order to understand Fig. 1 (but Figs. S1-S4 show the sex-linked candidate regions on various chromosomes).

Re: chromosomes in *Ectocarpus* sp. 7 were numbered according to their physical size in a previous version of the *Ectocarpus* genome (Cormier et al., 2017). The chromosome sizes changed in the latest version of the assembly (Liu et al., 2024), but the chromosome labeling was maintained to conserve the information of the previous assembly. The chromosome numbers of *Scytosiphon promiscuus* were numbered according to their homology to the chromosomes of *Ectocarpus* sp. 7, given that their chromosomes are so similar. However, the evolutionary distance between *Ectocarpus* and the rest of the species made this task difficult. We kept the chromosome numbers in *Undaria pinnatifida* that were used by the original authors of that genome assembly, presumably by chromosome size (Shan et al., 2020). The rest of the species were numbered according to their chromosome sizes, from largest to smallest. This information was added in the legends of Fig. 1 and Fig. S1-S5.

The relationships between the numbers, and those in Table 1 should be made clear, and also that Fig. 1 shows only sequences identified as male-specific in the two species in which U-linked regions were studied. The red horizontal line should also be explained.

Re: Given the moderate number of chromosomal rearrangements that occurred between species, it is not trivial to give the relationship between the chromosome labeling in Fig. 1 and those in Table 1. However, we added this information in the legends of Figs. S1-S5, and we linked this information by citing those figures in Table 1. We also added a sentence explaining that Fig. 1A only contains male genomes.

The reviewer surely missed the information within the legend of Fig. 1, where it is explained what the red lines corresponded to: “*Syntenic blocks of the V sex chromosome are highlighted in red*”.

The description in the Methods section needs editing to correct some mistakes in the English and punctuation, and can be made shorter and clearer, but the basic principle should be made clear in the main text. It is not quite clear how the criterion for V-linkage was applied, especially in species with no female assembly.

Re: We would like to clarify that we do have female genome sequencing data and draft Illumina assemblies for all species included in the study. However, chromosome-level assemblies of the U chromosomes are not available for all species, which is why our comparative analyses focused primarily on the V chromosome. Importantly, chromosome-level assemblies of female genomes are not required to detect sex-linkage in male genomes. For this purpose, we relied on read coverage and k-mer analyses to identify male-specific sequences, supported by the presence or absence of these sequences in female genomic data. However, fully assembled SDRs are indeed necessary for analyses involving gene order and structural comparisons and this is why we focused the analysis on the V sex chromosomes.

To address this more clearly, we have expanded the “Discovery of the U/V sex determination region” section in the Methods to detail how sex-linkage was assessed across species. We have also added the accession numbers for all female sequencing data to Table S1, along with a list of primers and example gel images from PCR validations confirming male specificity of the V-linked SDR scaffolds.

The text states that such regions were defined as ones within male reference genomes where the coverage in males was between 75% to 125% of the genomic average, while the coverage in females did not exceed 50% of the genomic average). Here, the coverage values are presumably some kind of average over

sequences of some kind (for example exons or perhaps some other type) that could be mapped to the genome, and perhaps regions were defined as those in which no such sequence has a value outside those thresholds, or maybe there is some flexibility in applying them (maybe the red horizontal line in Fig. S1 indicates this)?

Re: We appreciate the reviewer's careful reading and insightful question. We have expanded the "Discovery of the U/V sex determination region" section in the Methods to clarify the coverage-based approach used to identify sex-linked regions. As the reviewer inferred, we used Illumina whole-genome sequencing reads from both males and females, mapped to the corresponding male reference genome. We calculated normalized read coverage in 10 kb non-overlapping windows along the genome for each sex, by dividing the raw coverage in each window by the genome-wide average for that individual.

To identify V-linked SDR regions, we applied the following thresholds: windows were considered candidate male-specific regions if normalized male coverage was between 75% and 125% of the male genome average (i.e., consistent with single-copy autosomal-like coverage), and if normalized female coverage in the same window was below 50% of the female genome average. These thresholds allowed us to identify regions present in males but absent or substantially depleted in females. The same principle was also applied for the female genome assembly of *D. herbacea*. Regions were considered sex-linked if they showed consistent differences across multiple adjacent windows. The results were further validated with kmer coverage analysis and PCR. We have added the results of coverage analysis to new Figures S1-S5.

The statement that the ratios of gametologs, versus U- or V-specific genes, are similar in the distantly related *Ectocarpus* sp. 7 and *D. herbacea*, is not clear. Please can it be stated more clearly. The important question is whether the same genes appear as gametologs, versus U- or V-specific genes, in both these species, as opposed to each species having its own specific set of genes that have become U- or V-specific, which might suggest independent gains of genes by the U or V (but not both), or independent degeneration in these two lineages. A comment on the relevant text is below.

Re: We have clarified this section the manuscripts which reads now: "*The U/V-SDRs of Ectocarpus sp. 7 and D. herbacea carry homologous genes (gametolog pairs), indicating descent from a common ancestral region (Table S2). Both species show similar ratios of gametologs and U- or V-specific genes (16/14 and 11/7 gametolog/sex-specific ratios in Ectocarpus and D. herbacea, respectively; Table S2). From these, only ten genes share SDR orthologs between both species, while the rest were mostly acquired independently in the SDR of each species, with one gene that was retained as a gametolog in Ectocarpus sp. 7 but lost both copies in D. herbacea (Table S2). Five gametolog pairs conserved both copies in the two species, while another three gametolog pairs lost either the male or the female copy in D. herbacea, turning into U- or V-specific genes (Table S2). Additionally, MIN and a U-specific gene are also conserved between species (Table S2)*".

I am afraid that I was unable to understand the meaning of "the number of genes between their U and V-SDRs is symmetric within each species (Table S2), supporting parallel evolution of U and V chromosomes". Please can it be explained what this means.

Re: We understand that this sentence may lead to confusions. It means that *Ectocarpus* U-SDR and *Ectocarpus* V-SDR have the same number of gametologs and sex-specific genes, and that the same happens with *D. herbacea*. We modified this sentence for clarity: "*Although the total number of U/V-SDR genes differs between Ectocarpus sp. 7 (18 genes) and D. herbacea (30 genes), each species shows an equal number of gametologs and sex-specific genes in its U- and V-SDRs (Table S2). This intra-species symmetry supports the idea that the U and V chromosomes have undergone parallel evolutionary changes within each lineage*".

Fig. S9 mentions that 11 gametolog pairs were found in *Ectocarpus* sp. 7, and 16 in *D. herbacea*, but the question is whether the 5 extra ones in the latter might represent genes that became fully sex-linked since the lineages split. The main text states (if I understand correctly, as the text implies this, but does not make it completely explicit) that four genes "that form the most recent putative evolutionary strata in the U-SDR of *D. herbacea*" have homologs in the PAR1 of *Ectocarpus* sp. 7 (and that inversions in the U chromosome that included all the U-SDR and segments of the PARs can explain the more extensive U/V-SDR regions

in *D. herbacea* shown in Fig. S8; note that one cannot say “expansion of boundaries”), and that 13 PAR2 genes are also in putatively recent strata (Fig. 2D, Table S4, Fig. S8).

Re: Yes indeed, this is shown in Fig S9 (previously Fig. S8). We believe that “expansion of boundaries” in this context is grammatically correct and scientifically appropriate. The word “expand” describes a region growing to include new content – in this case the PAR genes in *Ectocarpus* that are now contained within the SDR of *Desmarestia*. Hence the SDR in *Desmarestia* expanded into the PAR and created new gametologs.

I also did not understand the idea that the larger V-SDRs in early-diverging lineages (like *D. dichotoma*) than in later-diverging orders (*Ectocarpales*) suggest that gene loss might have reduced SDR sizes. Can this be explained better? I should have expected the early-diverging lineages to have greater gene loss, because there has been more time for this, but I may have misunderstood.

Re: Thank you for this comment, we understand that the sentence is misleading as it is currently written. Our point was not to state that the early-diverging lineages have had more time to lose genes, because all lineages diverged from a common ancestor and have evolved for the same amount of time since. Rather, what we meant is that the presence of more SDR genes in early-diverging lineages like *Dictyota* compared to the fewer SDR genes in more recently diverged groups such as the *Ectocarpales* could be explained by either independent gene gain events in the early-diverging lineages or gene losses in the *Ectocarpales* lineage.

We modified this section to avoid misunderstandings: “*The observation of greater V-SDR gene content in early-diverging lineages (like D. dichotoma) and reduced V-SDR gene content in the later-diverging Ectocarpales (Table 1) could reflect either gene loss in the V-SDRs of Ectocarpales or independent gene gains in the V-SDRs of each lineage (as predicted by Bull¹⁰), from an ancestral state with low V-SDR gene content that is retained in Ectocarpales*”.

Reconstruction of the ancestral V-SDR gene content (with better quality genomic data; I did not understand “parallel U/V-SDR evolution^{24–26} as seen in *Ectocarpus* sp. 7 and *D. herbacea* (Fig. 2D, Table S2)”) is stated to have revealed that the small *Ectocarpales* SDRs are not caused by gene losses, but that gene gains of various kinds occurred in other lineages, as shown in Fig. 2E, including gains due to PAR genes becoming part of the SDRs, near their boundaries, as in Fig. S8. The key for part F does not make clear whether “autosomal” means translocations of autosomal genes into the SDR, and I did not understand the meaning of “sex homolog”, or how Fig. 2C shows that ancestral V-SDR genes have consistently higher *Ks* values than independently-acquired gametologs in *D. herbacea*, as the figure does not seem to distinguish these 2 categories of genes.

Re: We appreciate the reviewer’s effort to highlighting all these oversights. The section regarding the “parallel U/V-SDR evolution” refers to a previous sentence that was ambiguous and has now been clarified in the manuscript (see above).

We understand the confusion in the term “Autosomal” of Fig. 2F, since it is not related to the autosomal translocations that are described in Fig. 2E or in that section of the manuscript. We refer to “autosomal” in Fig. 2F as a gene that was sex-linked in the last common ancestor of the brown algae, but whose ortholog is found within an autosome in that extant species. We didn’t go into detail on this in the main text, since this is mostly inconsequential for the conclusions of the manuscript. We modified the figure and its legend to explain this and the other labels more explicitly.

We apologize for retaining the term “sex homolog” in Fig. 2F, since this was a remnant of a term that we used in the original version of the manuscript, which we now refer to as the “U/V-homolog” as proposed by reviewer 2 in the previous round of review. The U/V-homolog refers to the chromosome that corresponds to the sex chromosome in the dioicous species, which is explained in the main text.

We now distinguish the ancestral V-SDR gametologs in Fig. 2C as squares, which we now also highlight in red so they are more visible. This will show how most of the recently-acquired gametologs in *D. herbacea* (black dots) have lower *Ks* values than the ancestral V-SDR gametologs. We eliminated the evolutionary strata from this figure, given the valid criticism of saturated *Ks* values pointed out by the reviewer.

Information related to these issues is that the sex chromosomes (as a whole) had fewer conserved orthologs between species than the autosomes, and phylostratigraphy analyses is stated to reveal a statistically significant enrichment of taxonomically-restricted genes (TRGs) in the sex chromosomes of all dioicous species (furthermore, it is not clear that phylostratigraphy analysis is capable of extracting reliable

conclusions from the rather few changes over the very large times involved, and this is not justified in the ms).

Re: We understand that the reviewer may not be familiar with phylostratigraphy approaches, which may also be the case for any potential reader of the manuscript. Phylostratigraphy is a genetic statistical method developed in order to date the putative origin of all the genes contained in the genome of a target species by detecting homologs across species at different evolutionary distances (all the way from species within the same genus to species from different domains of life). Finding the most distant homologs of each gene can link them to their founder events (i.e., the first instance where a gene homolog is found in the history of that lineage), allowing us to then determine their relative ages, coded as the taxonomic group where that gene is detected. We have added this sentence in the Methods section to briefly explain the principles behind the detection of TRGs.

This effect was (surprisingly and somewhat confusingly) due to the PARs of the Ectocarpales, but to the entire sex chromosome in species with larger V-SDRs, but it is not clear whether it occurs in the SDRs specifically.

Re: Our argument is that the TRG enrichment is a feature of the entire sex chromosome, but that the Ectocarpales only display it in the PARs because the SDR is small in proportion to the rest of the chromosome, so it also occurs on the SDRs, we added “including the SDRs” in the text to make this clear.

The text claims that the sex chromosomes carry several genes that became sex-linked since the last common ancestor of the five dioicous species, based on being shared only by members of taxonomic groups lower than a given rank. This is indeed consistent with independent enrichment in different species, but can losses be firmly excluded?

Re: We suspect that the reviewer is confusing the topic that is being discussed in this section of the manuscript. The previous section entitled “The evolution of the SDRs involved boundary expansions and gene gains” deals with the evolution of genes that are sex-linked across species and the extent at which they are conserved or gained through expansions of the SDR borders and autosomal translocations.

However, this section of the paper, entitled “Structural features and evolutionary dynamics of brown algal U/V sex chromosomes”, does not deal with the evolution of sex-linked genes anymore, and rather discusses the structural feature that differentiate the entire U/V sex chromosomes (PARs + SDR) from the autosomes.

We further emphasize this in the revised manuscript to avoid confusions. In this context, the text is not claiming anything regarding “genes that became sex-linked since the last common ancestor of the five dioicous species”, as the reviewer interpreted. Rather, the manuscript claims that taxonomically-restricted genes (not sex-linked genes) are enriched in gene ages that are not shared between the five dioicous species (see above for an explanation of gene ages through phylostratigraphy).

For example, the U/V chromosomes of *Ectocarpus* sp. 7 are enriched in TRGs at the species (*Ectocarpus* sp. 7 genes), genus (genes only found across species of the *Ectocarpus* genus) and family levels (genes only found across species of the family Ectocarpaceae). This means that the most distant retrievable homologs of these genes are only found within taxonomic groups that are not shared with other brown algal species (such as the five dioicous species that were analyzed in the paper) or any other organism in the tree of life (e.g., diatoms, plants, animals, bacteria, etc.). We know this because the detectable gene homologs of these TRGs are only found in a handful of brown algal species and nowhere else in the entire genomic databases of PhaeoExplorer and the NR (that is, including every sequenced organism that has been uploaded to the NCBI). It is important to note that roughly two thirds of the genes in the brown algal genomes can be traced back to homologs that are shared with bacteria and distantly-related eukaryotes (Denoeud et al., Cell 2024), so this is not necessarily a problem of sensitivity or lack of outgroup species.

While extensive loss of these genes across the tree of life cannot be firmly excluded, since one cannot prove that these genes were lost in every species that has been sequenced aside from the Ectocarpaceae family, it is still highly unlikely that gene loss can explain a pattern where genes are consistently restricted to a very small group of brown algal species and are found nowhere else in nature. Furthermore, the method we developed to implement genomic phylostratigraphy takes into consideration the possibility of gene loss through the calculation of a metric we named “taxonomic representativeness”, giving more robustness to the gene age calculations that are presented in the manuscript (see Barrera-Redondo et al., 2023 for an in-depth explanation of this metric).

We discuss the three possible scenarios that could explain this pattern: gene untraceability (i.e., that these genes evolve too few conserved sites or their protein sequences are too short for pairwise sequence methods to detect homology in distantly related species), fast selection-driven substitution rates that make these genes unrecognizable to their homologs in other species, or *de novo* gene birth events.

The reference to higher K_s values in the PARs is unclear. Does this refer to inter-species divergence, and are these values reliable (for U-V divergence, they are not, as explained above, and the values in Table S16 are also often very high). A sample of 6 such sequences from the *Ectocarpus* sp. 7 PAR was examined, and five of them represent potential new genes that may have evolved from noncoding DNA in the other *Ectocarpus* species. In my view, it is premature to claim a 'gene cradle' pattern on the basis of these few data, and without evidence concerning these genes' functions.

Re: We confirm that the K_s values in this section of the manuscript refer to inter-species divergence estimates. We now added this information in the main text to distinguish between "gametolog K_s values" and "inter-species K_s values".

While we agree that the gametolog K_s values are unreliable due to the distant evolutionary times since the divergence of the U/V-SDRs, the same cannot be said for the inter-species K_s values in Table S16. While a few of these values are saturated, the average inter-species K_s values in *Ectocarpus*, sp. 7 (average K_s = 0.123), *Undaria pinnatifida* (average K_s = 0.872), *Desmarestia herbacea* (average K_s = 0.248), *Desmarestia dudresnayi* (average K_s = 0.205) and *Fucus serratus* (average K_s = 0.166) are all lower than 1, meaning that most of these values are below the point of saturation and can be reliably used for our analyses (see Fig. S15, S17, S18, S25 and S27 to observe the exact distribution of inter-species K_s values for each of these species). The same cannot be said for *D. dichotoma*, whose values are highly saturated (average K_s = 3.25), which is the reason why we refrained from using them in the analysis. The values in *Scytosiphon promiscuus* are also saturated (average K_s = 2.24), but still used them since they still display a distribution of values where information could still be extracted (see Fig. S16). If the reviewer finds this unreasonable, we can remove the inter-species K_s analysis in *S. promiscuus*, but this won't change the main conclusion derived from this section of the manuscript.

There appears to be a misunderstanding regarding the gene cradle pattern, with the reviewer interpreting it as being based solely on six '*de novo*' genes. We would like to clarify that the gene cradle pattern pertains to the enrichment of young taxonomically-restricted genes in the U/V sex chromosomes, irrespective of the inter-species K_s values or whether they are *de novo* or arise through other mechanisms.

We understand this selection of six genes can lead to confusions regarding the validity of the cradle pattern, since this test was used to evaluate the processes behind the pattern, rather than being evidence of the pattern itself. However, we agree that this test would need a more robust examination with additional experiments, so we decided to remove the section describing these six *de novo* genes in the revised manuscript, and we modified the discussion so we don't argue in favor of *de novo* gene birth events over any other potential mechanism (see above) behind the gene cradle pattern.

It is important to note that the specific functions of these genes are not the primary focus of this discussion. The key observation is that there is a higher abundance of younger TGR genes within these regions. While future research may focus on elucidating the functional roles of these genes, the presence of this clear pattern cannot be overlooked. It represents one of the central findings of our study, and thus, it cannot be omitted.

I would advise doing a much more thorough analysis and writing this up separately if clear evidence can be described.

Re: We agree that our results concerning *de novo* gene birth events may be premature and should be done thoroughly in another manuscript. Following the reviewer's advice, we therefore removed the *de novo* gene birth section alongside Fig. 3C, and just include the parts of the manuscript that are well supported by the evidence such as the young TRG enrichment pattern.

If there is such a thing as a 'gene cradle' effect in the V chromosomes of the recently sex-linked *C. purpureum*, and also in *S. angustifolium* (its age is not made clear), but not in *M. polymorpha*, which has

an ancient UV system, reasons for the effect, and the differences, should be explained. In the response, the authors argue that mentioning *Ceratodon* is appropriate, but the issue is that it needs to be explained that the genes in this species fully sex-linked region were autosomal until recently. Certainly, this species' neo-sex chromosomes are suitable for testing whether the newly sex-linked regions have started to evolve sex-specific allele expression, or other differences, but this requires a careful study in its own right, not just a passing remark. However, the enrichment of genes with sporophyte-biased expression (not "sporophyte-biased genes") is consistent with the model mentioned, in which generation-antagonistic selection may favor the retention of young sporophyte-beneficial loci in the PARs.

Re: We included this analysis to highlight that the pattern we observe extends beyond brown algae and into other haploid sex-determination systems, finding compelling evidence that this is the case for at least some bryophyte species. While the pattern itself is hardly debatable, we agree with the reviewer that we should try to explain the differences between the species where the pattern is seen and those where the pattern is absent. But we can only speculate about the reasons why this is the case in bryophytes, since most of our data (inter-species *Ks*, RNA-seq) are focused on the brown algae. We speculate that mosses such as *C. purpureus* and *S. angustifolium* display relatively more complex sporophyte body plans than liverworts such as *M. polymorpha* (Ligrone et al., 2012), which could underly stronger generation-antagonistic selection (Luthringer et al, MBE, 2015). Unlike brown algae, most of the sex chromosomes in bryophytes are sex-linked, with minimal space for PARs (Charlesworth, 2022). In this context, the sex chromosome of *C. purpureus* expanded its SDR very recently in evolutionary time (McDaniel et al., 2013), retaining its evolutionary footprints as PARs, while the sex-linked region in *M. polymorpha* is much older and degenerated (Iwasaki et al., 2021), which could also limit the formation of TRGs that are predominantly observed in the PARs of the brown algae. The gene cradle pattern appears thus to be specific to U/V systems where chromosomal degeneration is mild and linked to haploid-diploid life cycles where the sporophyte stage is sufficiently complex, highlighting a key role for generation-antagonistic selection. In this sense, we confirm that the sporophyte-biased expression consistently supports a mathematical model developed earlier (Luthringer et al, MBE, 2015).

As requested by the reviewer, we included these hypotheses in the discussion section of the manuscript.

6. Transitions to monoicy

As expected, given that the V probably carries sequences important for male reproduction, and the U for female reproduction. The homologs of U/V genes can be recognised, and it turns out that most are derived from their ancestral Vs, whereas one might have expected also to find ones descended from ancestral U alleles (in *D. dudresnayi*, 20 genes are inferred to be V-SDR-derived and four U-SDR-derived; in *C. linearis*, whose closest dioicous relative is more distant than for *D. dudresnayi*, the numbers are 11 V- and only two U-derived). Presumably this suggests that the retained sex chromosome homologue is the V, while the U has disappeared as an intact chromosome? If so, this should be stated explicitly.

Re: We suggest a scenario where a haploid individual with a V-SDR got some U genes via ectopic recombination in a (diploid sporophyte). The U chromosome did not strictly 'disappear' but after meiosis and likely ectopic recombination we get a haploid individual with a V chromosome plus a few female genes. Since the resulting monoicous individual possessed the capacity to generate both male and female structures, the individuals with U chromosomes stopped being essential for sexual reproduction, leading to the eventual loss of the U-SDR in the monoicous species. We added a sentence in the manuscript to be explicit about the fate of the U chromosome in the monoicous taxa.

As expected, the 7 ancestral V-SDR genes, including MIN, are transcribed during reproductive stages of both monoicous species. I did not understand the meaning of "retained gametolog copies or closely-related autosomal paralogs for most of their lost U/V-SDR orthologs".

Re: The confusion regarding this sentence is understandable, since these ideas are better conveyed in two separate sentences. Monoicous species retained a male or a female copy for most of the U/V-SDR-derived gametologs (91% in *C. linearis* and 100% in *D. dudresnayi*), whereas several U and V-specific orthologs were lost in these species (10 and 17 sex-limited genes in *C. linearis* and *D. dudresnayi*, respectively). From these lost orthologs, 60% and 43% of them present closely-related autosomal paralogs that may be compensating for these gene losses in *C. linearis* and *D. dudresnayi*, respectively, whereas the other lost genes lack any close paralogs and their putative functions are therefore lost in these species. We added

this information in the manuscript to avoid further confusions. Note that detailed information on the exact genes that were retained or lost can be found in Tables S18-S19.

The statement that “the U/V-homolog of *D. dudresnayi* retains some vestiges of its past as a U/V chromosome, such as low coding density, high repeat density” suggests that a chromosome can be recognised as a U/V-homolog, but that it cannot be determined whether it is a U or a V. This could perhaps be explained more clearly. Note also that the suggestion above that the region may have low gene density and high repeat density if it includes the centromere, so these properties do not necessarily indicate a past as a U/V chromosome state, so this statement should be revised.

Re: We believe we have clearly explained that the hermaphrodites evolved from an individual with a V that gained few U-SDR genes, as determined by the number of V-derived genes that are retained in both species (see text lines 263-273 and Fig. 4).

Concerning the vestiges of the sex-chromosome, we already shown in a previous comment that the *Ectocarpus* sp. 7 SDR has high repeat density regardless of the centromere (see above). Also, consider that the centromere spans only 153 kbp, whereas the ex-SDR region in *D. dudresnayi* that is enriched in TE within the U/V-homolog is much larger (roughly 6 Mbp, see Fig. 4B) and clearly stands out in TE content as can be seen in Figure 4C (and please see statistical analysis in the text). Finally, if the centromeres were responsible for this signal, we would see a similar TE enrichment pattern in the autosomes, which is not the case (see Fig. 4C and the statistical support in Tables S10-S11). Also, note that the full sentence that the reviewer highlighted also mentions young (taxonomically-restricted) genes, not only genes and TEs, as enriched in the U/V homolog.

I did not understand some of the text on p. 12, including the words in upper-case font (and I have a couple of questions, indicated by []). The text is “The only three U-SDR-derived orthologs in *C. linearis* are flanked by PAR orthologs translocated across the CORNERS of the SDR-homolog, suggesting that [which chromosome(s)? of] *C. linearis* acquired U-SDR-derived genes through two ectopic recombination events (Fig. S25A). Three U-SDR-derived genes in *D. dudresnayi* are dispersed across the SDR-homolog [does this mean the V?], suggesting independent U-V [correct?] translocation events, while a fourth was translocated to an autosome”.

Re: We rephrased this section to improve clarity

I am not sure why ectopic recombination is specifically mentioned as the mechanism, rather than simply mentioning that U genes have been translocated to new genome locations, either on the homolog of the V. These events presumably occurred during meiosis in the diploid sporophyte stage, and it might be worth making this clear in the main text.

Re: We think that ectopic recombination may explain this process, at least in *C. linearis*, since the two blocks of U-SDR-related genes in this species are flanked by PAR-derived genes, suggesting that the PAR sections that are shared between the U and the V chromosomes underwent a crossing over event that accidentally carried a few U-SDR genes into the V chromosome (see Fig. S24 for a hypothetical scenario of this process). Nonetheless, we agree that other potential mechanisms could have been at play, such as TE mediated translocation events, so we modified the phrasing.

7. The diploid Fucales

The transition from haploid to diploid sex determination in the diploid *Fucus serratus* remains mysterious, as no sex-linked sequences were detected, which is worth reporting, as it is a striking contrast with the overall conservation of the SDRs. It suggests a fairly recent turnover event, somewhat similar to those detected between diploid species in other taxa, and expression data from three other Fucales species suggests that genes with essential reproductive functions have come under the control of different sex-determining genes from those on the U and V of the haploid species. The failure to detect completely sex-linked sequences in *Fucus*, suggests that recombination has not become suppressed in the genome region carrying the new sex determining locus, a striking similarity to the absence of strata in the haploid species.

Re: The lack of recombination in the U/V species is clear and in *Fucus* we think that we are most likely missing the (surely tiny) SDR region in the assembly. Indeed, we are currently sequencing more individuals and performing HiC to improve the *Fucus* assembly but this study will be out of the scope of the present manuscript.

It is not clear how this relates to the sections about monoicous species, but presumably some relationship is expected, since an intermediate monoicous stage is mentioned for the Fucales.

Re: In both system the ancestral U/V chromosomes are no longer sex chromosomes, and indeed an intermediate monoicous stage is expected during the transition to diploidy.

General problems requiring mostly minor (but still essential) revision

1. When mentioning the “fundamental inheritance differences between U/V and XX/XY or ZW/ZZ systems that have broad evolutionary and genomic implications” (in the introduction), the work of Bull should be cited. It is not clear whether the concepts he developed are made use of in interpreting the findings. If they are, the connection should be clearly explained. At present, Bull is cited only twice, as follows (with my comments)

Re: We incorporated Bull as a reference in this section of the manuscript. (done)

a. “independent expansions of the U/V-SDRs boundaries and gene content may have occurred, as predicted”.

The first part of this is not a clear or accurate outline of his prediction. The boundaries between SDRs and PARs must be the same in the U and V chromosomes. The PARs must start in the same locations, because they are defined as recombining regions (and the study confirms this by absence of sex-specific variants, and by synteny, along with showing loss of synteny in non-recombining regions).

Re: We agree with the reviewer. What we meant is that the SDR/PAR boundaries changed in different species independently after the divergence from the common ancestor. We corrected the sentence in the text.

A caveat is that it may be difficult to determine the exact locations where PARs start, because close linkage with the SDR leads to linkage disequilibrium, so that the U and V may show some level of divergence, with sex-associated variants that may appear sex-specific in real-life samples. This is particularly likely if sexual reproduction occurs infrequently, as this could be the case in these algal species. A different comment explains that this may make it difficult to determine precise PAR boundaries.

Re: as we said before, sexual reproduction occurs frequently in Ectocarpus, and in most algal populations that have been studied mostly sporophytes are found. We added the evidence in Fig. S1-S5 to support the distinction between the PARs and the SDRs for each species.

Concerning the second statement, Bull did predict that U- and V-linked regions should evolve different gene contents, but the statement is much too vague, as he specifically predicted that, although genes with male function should be preserved on the V, they could be lost from the U, provided that their functions are not essential for functions of gametophytes of both sexes. If I understand his paper correctly, he thought that many genes should have such essential functions, and consequently predicted that degeneration should be minor. If this does not apply to brown algae, the case should be made that this predicted difference from the situation in plants such as bryophytes should be briefly explained, along with the evidence about this issue from the new results. If the ms documents the predicted differences in gene contents, and if Bull is cited, these things should be connected clearly.

Re: We hope that the modifications in the text and citation to Bull’s work improve the clarity of the revised version.

b. “Anisogamous species only experienced few gene gains in their SDRs”

Presumably, the idea is that genes should be gained when gametophytes of one sex gain sex-specific functions, say female functions in an oogamous lineage. This needs to be explained, as it is not exactly the same as Bull’s idea (if Bull also predicted this, of course it can be explained and his paper can be cited for the idea). However, as explained in other comments, the evidence supporting the hypothesis/prediction is not clear.

Re: We hope that the modifications in the text and citation to Bull’s work improve the clarity of the revised version. As we explain above, it is difficult to make conclusions about female functions in oogamous lineages, we prefer not to over interpret our results.

An alternative would be to mention Bull in the introduction, and explain which of his predictions applies in brown algae, and any hypotheses/predictions that he did NOT make, but which can be made for brown algae, and are tested in the study.

We have added a citation to Bull's work in the introduction. Due to word count limitations, it is challenging to include additional text at this stage. However, if the editor permits, we would be happy to expand on Bull's predictions in the introduction.

2. Concepts needed to understand the study need to be explained before being used. The core concept of UV systems in haploid organisms is not clearly explained, and many readers may not understand these concepts, and especially (as just mentioned) specific properties of brown algae that relate to U and V chromosomes' special properties. The brief description of dioicy/dioecy mentioned in the responses from the authors probably means the text "...which may not undergo recombination in the heterogametic sex (XY or ZW) of diploid species (dioecious), or in the diploid stage of haploid-dominant (dioicous) species", but I don't think that readers will understand this as definitions unless they already understand these things (that they refer to species with a genetic sex-determination system controlling males and females, that may not undergo recombination in the heterogametic sex (XY or ZW) of diploid species (termed dioecious), or in the diploid stage of haploid-dominant (dioicous) species". It is also not quite correct to say that sex-determining regions encode factors directing sex identity (protein-coding genes encode proteins, but factors are carried in genome regions). This may seem a minor quibble, but communication is important, and it is best to use correct terminology, and to avoid creating an impression that is not correct (here, that protein-coding genes are always responsible, and that regions carrying these factors are always recombinationally inactive).

Re: We have rephrased the sentences and modified parts of the introduction. As explained above, it is challenging to include more details due to word limits.

Related to this problem, it is too vague to say "conventional" to refer to the XX/XY and ZW/ZZ systems. The text probably means "Research on the biology and evolution of sex chromosomes has primarily focused on diploid XX/XY and ZW/ZZ systems in mammals, birds, fish, Drosophila and diploid plants.. Haploid U/V sex-determination systems, such as those of bryophytes and algae have been less studied. It might be helpful to say this in the abstract.

Then the introductory part could end with stating how many brown algal species (not just "in a range of") and outgroups with an explicit list of their sexual systems (not just "diverse" ones) were studied. The Results starts by mentioning ten species covering the phylogenetic, morphological and reproductive diversity of the brown algal clade and their closest outgroup, Schizocladia ischiensis (not explained, see comment below). Why not combine these two statements and be clear and definite, rather than vague?

Re: we changed the text according to the suggestions, and modified the abstract.

The explanation of gametolog gene pairs is not clear enough. Also, the term "female-limited" and similar terms probably mean U-specific, as opposed to simply being on the U-linked member of a U-V gametolog pair.

The MIN gene's involvement in male reproduction is not clearly enough explained in the background text.

Re: We have made changes in the text to clarify these points.

3. Vague or unclear words should be replaced with clear ones, or phrases. Specifically, "emerged" presumably means "evolved". And the phrase "the demise" of U and V chromosomes (used in several places) should be replaced with something more clear and explicit. Does it refer to a situation in which an entire U or V has been lost and the species reproduces vegetatively or asexually as the remaining sex, or what. Or maybe to degeneration? As written, the meaning is unclear.

Re: yes, emerged does mean 'evolved'; in our text, but we have changed the wording according to the suggestions.

4. Some statements are too sweeping, and should be made more cautious. For example, dosage compensation may sometimes evolve, but one cannot simply say that it does so. When it does, this is a

remarkable evolutionary change, but the manner of writing gives the (no doubt unintended) impression that it obviously will do so, and that no thought is necessary about it.

Re: we are not sure what statement the reviewer is refereeing to – the only moment we mention dosage compensation is to say that sex chromosomes have specific evolutionary dynamics. In any case we have rephrased the sentence. Now it reads: “*Sex chromosomes have independently evolved from autosomes multiple times and may be subject to specific evolutionary dynamics, including differential selection between sexes, asymmetrical expression of deleterious mutations and hemizyosity, meiotic silencing and dosage compensation*”.

Minor comments

The meaning of this sentence is unclear “these studies have provided important findings that help to understand the genetic structure of bryophyte U/Vs”. Does this mean “the genomic structure of bryophyte U- and V-linked regions”?

Re: this has been rephrased

The final sentence of the introduction section adds nothing and should be deleted. The text has already explained that the study described the sex-linked regions of these algae, and readers are now hoping to read what insights the results yielded.

Re: we rephrased this section

The closest outgroup is said to be something called *Schizocladia ischiensis* but it is not explained what this is, or the evidence that it is closer than other organisms.

Re: We are not sure to understand what the reviewer wants us to explain about the alga *Schizocladia ischiensis*, except saying it is closest extant outgroup. A detailed description of this organism would be an unnecessary deviation from the main focus of the manuscript, which is already packed with information. We now cite Denoued et al, Cell, 2024 and Bringloe et al 2023 where the phylogeny of brown algae and its relationship with *Schizocladia ischiensis* is explained in detail.

In Table1, the meaning of the column headings “PAR - U/V-homolog” is not clear. Does this simply mean regions assigned as PARs?

Re: The metrics mentioned by the reviewer in Table 1 correspond to the CDS and repeat densities in the PARs of the dioicous species. In the case of monoicous and dioecious species, there is no distinction between the SDR and the PARs, since they don't have U/V chromosomes. However, we still wanted to show the differences in these metrics between the U/V-homolog, that is, the chromosome in these species that is homologous to the U/V sex chromosomes in the dioicous taxa, and the rest of the chromosomes in those species. So given the similarities between the PARs and the U/V-homologs (they both recombine and still display different metrics from the rest of the genome), we decided to place these values in the same rows as the PARs to reduce the number of rows in the table. We added a sentence in the table legend to explain this.

The statement that “synteny conservation” is “akin to multicellular Metazoans” is not clear. Presumably it means that some measure of synteny is similar to the one for multicellular Metazoans, not that synteny with the chromosomes of multicellular Metazoans is high.

Re: This statement meant that brown algae display a strong synteny conservation right after evolving complex multicellularity, as seen by the disruption of such synteny in the outgroup *Schizocladia ischiensis*. A similar phenomenon was observed in Metazoans, where macrosynteny is constrained allegedly because genes need to be activated in a concerted manner during development (Lv et al., BMC Bioinformatics, 2011). We understand that this observation can lead to confusions and could rather be the focus of another manuscript, so we opted to delete this sentence.

Figure 5 is difficult to understand, as it contains so many different things. It is not explained what Ks refers to, and the concept of gene ages is non-standard and needs to be explained clearly. The meaning of U/V homolog in “chromosome compartment (autosomes and U/V-homolog)” should be explained — does it mean the homolog of the U and V in other brown algae?

Re: We understand that figures should be self-explanatory to be useful, and these additions would greatly help the readers interpret our results. We added more details on this and other figure legends that contain *Ks* values, gene ages, or U/V-homologs to avoid confusions. These descriptions were extended to a minimum to avoid dense legends.

Reviewer #2 (Remarks to the Author):

Thank you for the thorough revisions! I think the manuscript is now easier to follow, the conclusions are better supported, and the more speculative aspects are acknowledged as such.

My only remaining comment is that given how central the correct identification of the SDRs and their boundaries is for the manuscript, a stronger case could in my view still be made:

* Figures S1-S4 are a good start, but they are quite noisy for some of the species, and clearly not the whole story as both coverage analyses and PCR verification were also performed. Would it be possible to add a coverage track to the plots, as well as the genomic locations that were validated, and the boundaries that were consequently identified?

Re: We thank the reviewer for the suggestion, and we added the information as requested. We also expanded the methods section and added the coverage tracks for each species as requested. We included more details about primers and PCR tests in Table S1 indicating the position of the validated genes on the SDR. Note that the precise locations of the borders (at the nucleotide level) are challenging to obtain because we would need natural population data, therefore, we focus our analysis on genes. Both, coverage and kmer analysis, identified identical genomic regions, providing high-confidence SDRs (Table 1, Fig. S1–S5) which were further validated by PCR. In *Desmarestia herbacea*, where both male and female chromosome-level genome assemblies were available, we directly compared U and V chromosomes to further confirm the SDR borders by analyzing the collinearity of pseudoautosomal regions flanking the SDRs (as shown in Fig. 2A-B). This has been clarified in the text.

* In the methods, you say that "Genomic regions with a minimum of 70% of unmatched single copy kmers were then retained as candidate male or female SDR sequences", but this threshold is never reached for *D. dichotoma* and *U. pinnatifida* (at least this is not visible in Figs S1-S4). Could you clarify this?

Re: Indeed, 70% is arbitrary, it depends on the divergence between the male and female individuals used for kmer analysis. To standardize the value we used >50% of unmatched kmers to find the candidate sex linked regions, this has been corrected on the text and in figure legends. The identified regions were cross-validated with the coverage analysis and by PCR (see above).

* It was also not clear to me how the exact boundaries of the SDR were determined, and this needs to be clarified -- in some figures it seems obvious (but even there there must have been a method to determine specific coordinates), in others not at all. It also seems pretty standard to provide primer sequences and gel images of the PCR.

Re: We have now expanded the Methods section to clarify how SDR boundaries were defined. Specifically, we first used coverage analysis and k-mer profiling from male and female Illumina reads to identify regions with sex-specific signatures. These two approaches consistently identified the same genomic intervals, providing high-confidence candidate SDRs (see Table 1, Fig. S1–S5). Next, we examined the gene content within these intervals and designed PCR markers targeting candidate SDR genes to validate sex linkage and more precisely define SDR boundaries. For species with chromosome-level assemblies of both sexes (e.g., *Desmarestia herbacea*), we were additionally able to directly compare the U and V chromosomes, using synteny and collinearity of the flanking pseudoautosomal regions to refine SDR boundaries further (see Fig. 2A-B). As suggested, we now include a list of primers and gel images, along with genomic coordinates of the validated SDR genes in Table S1.

Reviewer #1 (Remarks to the Author):

This ms is now much improved, though a surprising number of minor corrections of the text are still needed, as indicated in the annotated pdf file and the comments below.

Moreover, several ideas are stated as conclusions despite weak support, and the ms fails to focus on hypotheses that can be tested or on the most interesting and well supported conclusions, which I list below. Ideally, the Discussion should evaluate how the findings that are well supported help reach interesting conclusions. Instead, it unnecessarily repeats much that has now been made clear in the Introduction, Abstract and Results sections. For instance, the emphasis (L. 351 onwards) seems inappropriate on the statistically unsupported appearance that the SDRs of anisogamous (oogamous) species have independently and consistently gained genes (whether by changes in the PAR boundaries as recombination was suppressed, or by creation of new genes or perhaps duplication of genes into non-recombining regions).

Re: This point has been raised since the first round of reviews, and it has been consistently addressed. We clearly separated our results in the “Results” section from the proposed hypotheses in the “Discussion” section. Furthermore, we’ve been very careful to highlight the degree of support for each conclusion using phrases such as “we speculate” for weakly supported ideas, and “our results indicate” for strongly supported ideas. We agree that we contemplate both types of hypotheses in the discussion, but believe both are worth exploring in the manuscript, since both can lead to further research in the future.

Similarly, it is not very clear what is meant by “streamlined gene content of the *M. polymorpha* U/V-SDRs” (have these lost genes, in contrast to the gains inferred in the brown algae? L 398 states that it is degenerated, based on Iwasaki et al 2021, which contradicts the earlier statements about such conclusions in haploid organisms),

Re: We meant that the SDR gene content in both *Marchantia* and brown algae is relatively simple (i.e., with a small number of conserved genes) compared to other U/V systems, such as *Ceratodon* or *Bostrychia* that have hundreds of genes on their SDRs. We replaced “streamlined” with “relatively simple”. We thank the reviewer for the observation regarding the degeneration in *Marchantia*. We agree that the statement about degeneration is unclear so we decided to remove the term “degenerated” in this section of the manuscript.

or “sex-biased gene expression in mature gametophytes carrying sex organs” (does this mean sex-biased expression of autosomal genes, reflecting differences in somatic development between male and female gametophytes that might be controlled by just a few SDR genes?).

Re: The reviewer is correct in her interpretation. We added the reviewer’s suggestion to this sentence to make the statement clearer.

The text (L 361) contrasts these cases with that of the moss (it is not explained that the species is a moss until a few lines later, and the genus name is not mentioned) *Ceratodon purpureus*, whose sex chromosomes are not correctly described. They do carry thousands of genes (and only a few autosomal genes have sex-biased expression), but situation of thousands of fully sex-linked genes (based on also being sex-linked in another moss, *Syntrichia* (Silva et al., 2021 doi: 10.1111/tpj.15116), but (as L 397 mentions, albeit rather vaguely, this SDR expanded very recently (by fusions of autosomes with earlier established U and V chromosomes, so that the resemblance to PARs is unsurprising, leaving it unclear whether the new understanding is simply that that the brown algal SDRs did not arise in this way, which the synteny results show is correct (and this is interesting).

Re: The genus name for *Ceratodon* is mentioned in a previous section of the manuscript (line 245), so the abbreviation is appropriate in this section. We understand that the reviewer wants us to point out precisely why *Ceratodon*'s SDR has so many genes (autosomal fusion), in order to have an adequate discussion regarding why it contrasts with the evolutionary process observed in the SDRs of brown algae (engulfment of PARs and small-scale autosomal translocations). We therefore briefly mention the autosomal fusion in *Ceratodon* in the revised manuscript. While we agree that the difference in SDR gene acquisition between *Ceratodon* and brown algae is interesting, do bear in mind this section (now L393-394) deals with the enrichment of TRGs, rather than the expansion of the SDR, so the process described by the reviewer bears little consequence for the topic that is being discussed in this context. *Ceratodon*'s sex chromosome resemblance to PARs is very relevant, since the TRGs are mostly found in the PARs of brown algae (see Fig. 3B). This would be the "new understanding" conveyed in this section of the manuscript.

A shorter Discussion focusing on what is new and interesting (listed under the next set of comments) would be an improvement.

Re: We have slightly reduced the discussion, as proposed in the marked document the reviewer sent

Major comments

L 134. The statement that "the gametolog Ks values are broadly consistent between orthologs in *Ectocarpus* sp. 7 and *D. herbacea*" is vague. Are the values consistent with a great age estimate for U-V divergence, or do the data tell us something else. It is not convincing that such saturated sequence divergence can be used to infer a conclusion, but, if it can, the authors should certainly explain clearly what it is, and the logic for their inference(s). L 136 may, in fact, be a statement of the conclusion from these results (if so, the text should be revised to avoid the vague earlier statement).

Re: We agree that this sentence is vague, and actually redundant with the next sentence that explains this "consistency" in a better manner (ancestral SDR gametologs have higher Ks values than independently-acquired gametologs). We modified this sentence.

This line states (in my wording) that "the position of the gametologs with the lowest Ks values in the U-SDR of *D. herbacea*, relative to the PAR genes in *Ectocarpus* sp. 7 indicate that inversions in the U chromosome that included all the U-SDR and segments of the PARs can explain the expansion of the U/V-SDR boundaries in *D. herbacea*". This sentence is difficult to understand, but may mean something like "the two [or maybe more than 12 —the number is not stated] gametologs with the lowest Ks values (below a value that is not stated) in the *D. herbacea* U-SDR (but rearranged in its V) are PAR genes in *Ectocarpus* sp. 7, indicating that inversions in the U chromosome that included these U-SDR sequences and parts of the PARs can explain the expansion of the U/V-SDR boundaries in *D. herbacea*, a process that we term 'engulfment' ".

Re: We replaced the sentence with the following: "The location of gametologs with the lowest Ks values in the U-SDR of *D. herbacea*, relative to the PAR genes in *Ectocarpus* sp. 7, suggests that inversions involving the entire U-SDR and adjacent PAR segments likely contributed to the expansion of the U/V-SDR boundaries in *D. herbacea*, in a process we term 'engulfment' ". We hope this modification conveys this idea clearer to the readers.

L 154 In discussing the results from inference of ancestral V-SDR gene contents, the text says that Fig. 2C shows ancestral genes consistently having high inter-gametolog Ks values, while independently acquired gametolog pairs had lower values (in one of the species pair, *D. herbacea*, in which I think the authors conclude part of an ancestral PAR became fully sex linked, though this is not made clear enough in relation

to these K_s estimates). However, this is not clearly shown in the figure, as there is no key. I assumed that the smaller symbol (back or blue dots) indicate the ancestral genes, as in the key to the figure part below, and the larger squares indicate the independently acquired genes that presumably mostly evolved when parts of the PAR became fully sex linked and have remained as gametolog pairs (since genes newly evolved within the fully sex linked, or gained by duplications from other genome regions, would presumably rarely be found as gametolog pairs). However, this may be incorrect, as the squares are mostly higher than the dots, and so perhaps the symbols are reversed in the 2 parts of this figure (if so, this is confusing).

Re: We modified Figure 2 to show the key displaying that the black dots represent the independently-acquired gametologs in the SDR of each species, and the red squares represent the ancestral SDR gametologs. While we cannot explicitly state the relative acquisition of each of these independently-acquired gametologs, since we previously agreed that the strata estimations were unreliable, it makes sense that the squares are higher in Fig. 2C because they have accumulated more synonymous substitutions per site, which reflects their ancient divergence as gametologs.

All of this still needs to be explained clearly, and it currently remains unclear whether there is a consistent and statistically significant difference.

Re: We would like to remind the reviewer that we have removed all references to strata in brown algae. To clarify, yes indeed, we did present statistical evidence (change-point analysis) for differences in an earlier version of the manuscript. However, in response to continued criticism from the reviewer, we have since completely removed any mention of strata in brown algae, including the statistical test.

It is quite plausible, and indeed expected, that, if a set of genes became fully sex linked recently enough, forming a younger stratum, they will often still be present as gametolog pairs and have low K_s values, so if there is clear evidence for this in this species comparison, it can be described very simply.

Re: We have avoided any allusion to ‘strata’ in these species, in response to this same reviewer. We had presented statistical evidence that the reviewer asked for in a previous response – but these sections about ‘strata’ have been removed from the manuscript to avoid any kind of speculation. Nonetheless, this can be confirmed by looking at Fig. 2C, where *Desmarestia*'s recently-acquired gametologs (black dots) display lower K_s values than the ancestral SDR gametologs (red squares). Also note we explicitly mention this in the second paragraph of the section “The evolution of the SDRs involved boundary expansions and gene gains” as follows: “the location of gametologs with the lowest K_s values in the U-SDR of *D. herbacea*, relative to the PAR genes in *Ectocarpus* sp. 7, suggests that inversions involving the entire U-SDR and adjacent PAR segments likely contributed to the expansion of the U/V-SDR boundaries in *D. herbacea*, in a process we term ‘engulfment’”.

The inter-species K_s values discussed later (top of p. 11) are not specified, so it is not clear whether these are also saturated. Looking at Figure 3A, it is difficult to understand whether this is the case, and some of the figures appear to show higher values (yellow) for the autosomes (is this correct?) and for regions in the middle of the sex chromosome. Please can you check this figure, and perhaps mention some values, rather than readers having to look at figures that just show “higher” and “lower” when the values have been estimated quantitatively.

Re: This was already discussed in the previous round of review. We confirm that the inter-species K_s values are mostly non-saturated, particularly for *Ectocarpus* and *Desmarestia*, where we have species from the same genus with whom to calculate these values (see Supplementary Table 16). The range of inter-species K_s per genomic compartment are displayed as frequency plots in Supplementary Figures 7 – 11. Unfortunately, the range of inter-species K_s values are different in each species in Figure 3, so there is no straightforward way to visually convey this in a systematic manner without messing up the plots. We understand that disclosing the quantitative values of this rates is important, so we now mention

Supplementary Figures 7-11 within the legend of Figure 3, so the readers know where to find this information.

L 171. I have several questions about this text: “Although the number of SDR changes is small, oogamous species each independently gained numerous V-SDR genes, and one gene (ATP-dependent RNA helicase) was convergently acquired in all”.

1. Could this gain have occurred in the common ancestor?

Re: We can confirm that this gene was independently acquired by all the oogamous species. This gene (orthogroup OG0003211) was initially inferred as ancestral though the ancestral state reconstruction analysis. We later performed a gene tree analysis, discovering that OG0003211 was independently acquired by each species, since the gametolog substitution values are lower than the inter-species substitution values (Extended Data Figure 2D). We added this information in the text.

2. How many of these gains occurred when PAR genes ceased recombining? Distinct events that created new, young strata should be described clearly, with the numbers of genes that can be shown to have been gained in each. In connection with these, am I correct in thinking that the V-SDR of *Ectocarpus* sp. 7 is thought not to have undergone any such enlargement, whereas that of *D. herbacea* has done so (this is the same as my query above, as I feel that the text remains unclear, despite this being a rather obvious question).

Re: Each species independently incorporated genes into their SDRs, including *Ectocarpus* sp. 7, since all the species have additional genes from the ancestral cassette of 7 V-SDR genes. This is clearly explained in the third and fourth paragraphs of the section “The evolution of the SDRs involved boundary expansions and gene gains”. The exact number of genes that were acquired by each species, as well as the events leading to the acquisition of each of these genes are available in Figure 2E and Supplementary Table 5.

3. And similarly for translocations of autosomal genes into the V-SDRs of *Ectocarpus* sp. 7 and *D. herbacea*, we should be told how many in each case also happen to be sex-specific genes (Table S2, Table S5), consistent with a model where sexual antagonism in autosomal loci was resolved by the evolution of sex linkage.

Re: All this information is in the Tables, and we would prefer to avoid overloading the text with numbers.

The most interesting conclusions, suitable for the Discussion section

1. The observed enrichment of male-biased genes on the PARs in all species except *D. dichotoma* is consistent with the sexual antagonism model just mentioned, with locations in the PARs showing close linkage to the sex-determining region. However, the ms appears not to explain that parts of the PARs may recombine very rarely with the male-determining region, especially if outcrossing occurs rarely in a species.

Re: While we appreciate the reviewer’s suggestion, the claim that the PAR may recombine very rarely with the SDR is purely speculative. There is no empirical basis to assert whether recombination between the PARs and the SDR occurs rarely or not at all. As we previously stated in response to this same reviewer, there is robust evidence for sexual reproduction and outcrossing in natural populations, as documented in the study by Couceiro et al., which we have consistently cited.

2. In the two species tested, gametologs had higher expression levels than genes present in only one of the SDRs, even when expression was estimated in the haploid stage. This may be either due to their functions (as the text in lines 183 onwards mentions), or partly to the time during which they have been

evolving in the absence of recombination (young stratum genes would be expected to have expression similar to the values for their autosomal counterparts in other species, as is indeed observed, and to be gametolog pairs in most cases), while older ones (without gametologs) might have lost expression capacity. If this is correct, it would suggest the possibility of genetic degeneration, albeit at a slow rate.

Re: We have addressed this point and slightly modified the wording to indicate potential slow rate of degeneration

3. It is also interesting that, compared with the autosomes, the sex chromosomes (or perhaps just the SDRs, see my minor comment below on L 210 onwards) had fewer orthologs conserved between species. A clear discussion of what might explain this pattern would be valuable.

Re: Yes, this is an interesting point, and we do address it in the manuscript when discussing the taxonomically restricted genes enriched on the sex chromosomes—what we refer to as the ‘gene cradle’. The reviewer may have missed this important connection, so we added a few words in the manuscript to better connect these two concepts (line 221-222). We confirm that the lack of detectable orthologs is observed in the entire sex chromosome, particularly in the PARs.

The result from phylostratigraphy analyses, that the fully sex-linked regions are enriched for taxonomically restricted genes, is related, and also interesting. It presumably means the appearance of new genes, not a degeneration process leading to loss of some ancestral genes, as PAR genes are stated to show this more than SDR ones (L 224), though the text does not appear to make this argument.

Re: Please see our response above. The reduced number of orthologs is linked to the enrichment of taxonomically restricted genes, which likely reflects the emergence of new genes rather than gene degeneration or loss. In response to earlier criticisms from this same reviewer in previous versions of the manuscript, we removed the discussion suggesting these may have evolved “de novo” and instead adopted a more cautious phrasing, referring to them simply as “taxonomically restricted.”

Also, I was unable to find a clear description of the meaning of taxonomically-restricted genes should be defined, and I think that it may mean genes not recognisable in any other species, or perhaps in any other brown algal species (suggesting that they are indeed true new genes), but am not sure (it might mean duplicated genes that are not in the fully sex-linked regions of other brown algal species).

Re: The reviewer is correct with her interpretation of taxonomically-restricted genes: genes that are not detectable outside of a defined taxonomic group, which can be a small group of species within the brown algae. We included this definition of “taxonomically-restricted genes” in the manuscript (line 222). This indeed suggest that TRGs are truly new genes evolved within a specific evolutionary lineage, and this is a topic we will analyze in more detail in another manuscript.

I am still unhappy about introducing the term “gene cradle” in addition to TRG. The pattern(s) of overrepresentation of new genes can be described simply and clearly without a new term (as lines 247-8 in fact do) and there are already too many words within biology.

Re: We believe the term should be retained, as it is clear, easily understood, and already in use among colleagues in the field. However, if the editor has strong reservations about it, we are of course willing to replace by “taxonomically restricted genes enrichment”

Minor comments

L 88. This states that the SDRs contain a relatively small number of genes overall (between 18 and 52) compared to the PARs (between 229 and 904), but does not relate the numbers to the physical sizes,

which would allow one to understand whether the small overall numbers indicate a noteworthy difference. It is mentioned immediately after this that SDR size differences in different species are strongly correlated with the number of genes.

Re: We relate these numbers later on, first with the analysis that the reviewer mentions (Extended Data Figure 2), but also by showing that the gene density is significantly lower in the SDRs than in the PARs across sliding windows, which counts as size-corrected measurements (Figure 3).

L 112 states that the U/V-SDRs of *Ectocarpus* sp. 7 and *D. herbacea* carry homologous genes (gametolog pairs), indicating descent from a common ancestral region, but this statement is unclear, as it may mean that these two distantly related species carry homologous sets of gametolog gene pairs, indicating that their U regions descend from a common ancestral U region, and similarly for their Vs — or it might mean that, in each of these two species, the U and V carry gametolog pairs, indicating that the U and V are descended from a common ancestral region, or both may be true. From the text in L 115, I think that the latter is meant (“only ten genes share SDR orthologs between both species”), but it would be good to make the meaning unambiguous from the outset, rather than leaving readers puzzled.

Re: We understand the reviewer’s concern, but we believe it is important not to prematurely state that the shared genes represent the ancient SDR gene set before introducing the ancestral state reconstruction analysis, which follows immediately afterward. Doing so would risk confusing readers, as they would encounter references to “ancestral U/V-SDR genes” without yet understanding the basis for this classification. We feel that the current structure presents a clearer narrative, guiding the reader through the rationale for each analysis and showing how we gradually arrive at the conclusion that these shared genes are indeed ancestral SDR genes.

Related to the description of gene numbers, the statement is also puzzling/unclear writing that “each species shows an equal number of gametologs and sex-specific genes in its U- and V-SDRs [and this] symmetry supports the idea that the U and V chromosomes may have undergone parallel evolutionary changes within each lineage”. Does “an equal number of gametologs and sex-specific genes” mean the total number of both types (with some ancestral gametolog pairs having lost the U or V copy, as explained just before this, thus becoming sex-specific genes), supporting the idea that the U and V initially had the same gene content, but that some of these have been lost by each lineage?

Re: We had rephrased this sentence in the previous round of reviews, and we confirm the reviewer understood correctly the concept.

L 179. What does “U/V-SDR genes” mean? Does it mean genes that are either U- or V-linked, including gametolog pairs (in other words genes present in the U- or V-linked regions, or both of them), or something else?

Re: The reviewer is correct; “U/V-SDR genes” refers to genes that are sex-linked on the U- and V-SDRs, including gametologs and sex-specific genes.

L 210. The reference for transposable element accumulation in the sex-linked regions is not very appropriate, as it refers to anther-smut fungi, which have non-recombining regions (which evolved for reasons that are well understood, but not related to sex chromosome, and not sex-linked). This should be replaced by a reference to a review of such accumulation in sex-linked regions.

Re: We replaced this reference for papers focused on Y chromosomes (Bachtrog, 2003) and U/V chromosomes (Ahmed *et al.*, 2014).

It also seems strange to describe such results for entire sex chromosomes (including the finding that fewer orthologs are conserved between species than for the autosomes), because it is only the fully sex-linked regions are predicted to show unusual repeat richness and low gene densities, and indeed L. 213 confirms that V-SDRs differ from the PARs and the autosomes in these ways, though the PARs were also enriched in repeats compared with the autosomes, presumably because parts of them recombine rarely with the male-determining region (see above). Similarly, I 230 compares entire sex chromosomes for sporophyte-biased genes, although (in this case) the prediction is that such genes should be preferentially retained in PARs.

Re: We appreciate the reviewer's thorough evaluation and recognize their strong style of writing. However, we respectfully ask that our choices in structuring the manuscript also be acknowledged. We believe it is important to retain our current narrative structure, which moves from broad-scale analyses (e.g., inter-chromosomal comparisons) to more focused compartment-level analyses (autosomes, PARs, SDR), for the following reasons:

(1) Narrative clarity: This structure supports a logical flow of exploration, guiding readers from general observations to more specific findings. It allows conclusions to build progressively and be better understood in context. (2) Statistical rationale: Starting with chromosome-level comparisons is statistically more sound due to the substantial differences in size and gene content across genomic compartments. For this reason, we employed permutation tests to compare SDRs, PARs, and autosomes. In contrast, the sex chromosome and autosomes are of similar size and gene number (within the same order of magnitude), making direct statistical comparisons between them more robust. (3) Complementary analyses: Presenting both types of statistical tests (Wilcoxon rank-sum test and permutation tests) adds strength to our conclusions by addressing the limitations and advantages of each method and demonstrating consistency in our analytical approaches.

Regarding the presence of TEs in the PARs, we have already addressed this point by explaining that the unique characteristics of the PAR likely result from an excess of DNA transposons and local hopping events (see Supplementary Figure 6).

L 310. I don't think that monoecy has been defined.

Re: We added the definition (both sexes in the same diploid individual).

L. 342. The meaning is unclear of "sequestration ... essential for maintaining the U/V system or incidental during the initial recombination-suppression event". Please be explicit about the two alternative hypotheses that remain to be tested.

Re: We removed the sentence, as we were asked to reduce the length of the discussion.

Origin and evolutionary trajectories of brown algal sex chromosomes

Authors: Josué Barrera-Redondo^{1,‡}, Agnieszka P. Lipinska^{1,‡}, Pengfei Liu¹, Erica Dinatale¹, Guillaume Cossard¹, Kenny Bogaert¹, Masakazu Hoshino^{1,2}, Rory J. Craig¹, Komlan Avia³, Goncalo Leiria¹, Elena Avdievich¹, Daniel Liesner¹, Rémy Luthringer¹, Olivier Godfroy⁴, Svenja Heesch⁴, Zofia Nehr⁴, Loraine Brillet-Guéguen^{4,6}, Akira F. Peters⁵, Galice Hoarau⁷, Gareth Pearson⁶, Jean-Marc Aury⁸, Patrick Wincker⁸, France Denoëud^{8,‡}, J Mark Cock^{4,‡}, Fabian B. Haas^{1,‡}, Susana M Coelho^{1,‡*}

¹Department of Algal Development and Evolution, Max Planck Institute for Biology Tübingen, 72076 Tübingen, Germany, ²Current address: Research Center for Inland Seas, Kobe University, Rokkodai 1-1, Nada-ku, Kobe 657-8501, Japan; ³INRAE, Université de Strasbourg, UMR SVQV, 68000 Colmar, France; ⁴Sorbonne Université, CNRS, Algal Genetics Group, Integrative Biology of Marine Models Laboratory, Station Biologique de Roscoff, Roscoff, France; ⁵Bezhin Rosko, 29250 Santeg, France; ⁶Universidade do Algarve, UALG - Centro de Ciências do Mar (CCMAR); ⁷Faculty of Biosciences and Aquaculture, Nord University, 8026 Bodø, Norway; ⁸Génomique Métabolique, Genoscope, Institut François Jacob, CEA, CNRS, Univ Evry, Université Paris-Saclay, Evry, 91057, France. [‡]CNRS, Sorbonne Université, FR2424, ABiMS-IFB, Station Biologique, Roscoff, France

[#]Equal contribution

[‡]senior authors

*Lead author: susana.coelho@tuebingen.mpg.de

Running title: Tracing the complex evolutionary history of sex chromosomes across brown seaweeds

ABSTRACT

Sex chromosomes fall into three classes: XX/XY, ZW/ZZ and U/V systems. The rise, evolution and demise of U/V systems has remained an evolutionary enigma. Here, we analyse genomes spanning the entire brown algal phylogeny to determine the evolutionary history of their sex-determination. U/V sex chromosomes emerged between 450 and 224 million years ago, when a region containing the male-determinant *MIN* ceased recombining why? Over time, nested inversions caused stepwise expansion of the sex locus, independently in each lineage, and concomitant with the increasing morphological complexity and level of sexual differentiation observed in brown seaweeds. Unlike XX/XY and ZW/ZZ chromosome pairs, brown algal U/V pairs evolved mainly by gene gains, showing minimal degeneration. They evolve structurally and act as genomic 'cradles' fostering the birth of new genes, potentially from ancestrally non coding sequences. Our analyses demonstrate that hermaphrodites have sometimes arisen from ancestral males that acquired U-specific genes by ectopic recombination, and that, in the transition from a U/V to an XX/XY system in what group?, V-specific genes moved down the sex determining developmental hierarchy. Both such events have led to the loss of U and V chromosomes and erosion ?? of their specific genomic characteristics.

Deleted: Retracing

Deleted: decipher

Deleted: their sex-determination

Deleted: pivotal

Formatted: Highlight

Deleted: drove

Deleted: s

Deleted: are

Deleted: dynamic

Formatted: Highlight

Deleted: arose

Formatted: Highlight

Deleted: genetic hierarchy of

Deleted: determination

Deleted: a

Deleted: demise

Formatted: Highlight

Deleted: Taken together, our findings offer a comprehensive model of U/V sex chromosome evolution.

Formatted: Font colour: Text 1

INTRODUCTION

The mechanisms controlling the development of male or female identities, or cosexuality, when individuals express both sex functions, vary widely across different organisms^{2,3}. In non-cosexual species, sex chromosomes may be present, carrying a sex-determining region (SDR)⁴ with one or more protein-coding and non-coding factors, and which does not undergo recombination in the heterogametic sex (XY or ZW) of diploid species¹, or in the in diploid stage of haploid-dominant species. Sex chromosomes have independently evolved multiple times and are subject to specific evolutionary dynamics, including differential selection between sexes, asymmetrical expression of deleterious mutations and hemizyosity, meiotic silencing and dosage compensation⁴.

Research on the biology and evolution of sex chromosomes has primarily focused on the conventional XX/XY and ZW/ZZ systems of diploid mammals, birds, fish and *Drosophila*^{1,5}, and diploid plants. In contrast, U/V haploid sex-determination systems, such as those of bryophytes and algae (brown, red and green lineages)^{6,7} have been less explored. In U/V systems, sex is not determined at fertilization, but during meiosis, when haploid spores inherit either a U chromosome, and will develop into a female gametophyte, or a V chromosome, controlling male gametophyte formation⁸. These fundamental inheritance differences between U/V and XX/XY or ZW/ZZ systems have broad evolutionary and genomic implications⁹. However, to date, only the U/V systems of the brown alga *Ectocarpus* and four distantly related bryophyte taxa¹⁰⁻¹³ have been fully sequenced and assembled into chromosomes. While these studies have provided important insights into bryophyte U/Vs, the species involved diverged around 500 million years ago (Mya), and do not share homologous U/V chromosomes¹⁴. As a result, we still lack a broad comparative view across multiple U/V systems that would inform a reconstruction of their evolutionary history. Brown algae (Phaeophyceae) represent exceptional models for studying sex chromosome evolution because they display diverse reproductive systems, life cycles and sex chromosome systems in a single lineage¹⁵. Their ancestral state likely involved separate sexes¹⁵, suggesting that their sex chromosomes could share a common origin.

Here, we study the origin, evolution and demise of U/V sex chromosomes in a range of brown algal species and outgroups with diverse sexual systems. Our findings provide new insights into eukaryote sex chromosome evolution how, precisely? and improve our understanding of the biology of this ecologically important lineage.

RESULTS

The origin of brown algal sex chromosomes

We focused on ten species covering the phylogenetic, morphological and reproductive diversity of the brown algal clade¹⁹ and their closest outgroup, *Schizocladia ischiensis*. We substantially improved the brown algal genome datasets available¹⁶⁻¹⁸, using PacBio and extensive Nanopore sequencing, supplemented with genetic and Hi-C maps, to reach chromosome or near-chromosome level genome assemblies for these representative species (see methods). This revealed that brown algae have a relatively stable karyotype (27-33 chromosomes) and largely conserved macrosynteny (Fig. 1A).

Deleted: Sexual reproduction, present in almost all eukaryotes, allows species to survive environmental challenges by increasing genetic variation¹. Although the core processes of meiosis and fertilization are largely conserved, t

Deleted: guiding

Deleted: species

Deleted: s

Deleted: contain

Deleted: that encodes

Deleted: directing sex identity and

Deleted: .

Deleted: from autosomes

Deleted: masking

Deleted: ,

Deleted:

Deleted: found in

Deleted: remain largely

Deleted: un

Deleted: not

Deleted: forming

Deleted: forming a

Deleted: the U/Vs of

Deleted: latter

Deleted: the features of

Formatted: Highlight

Deleted: In this context, b

Deleted: exploit

Deleted: a range of brown algal species and outgroups with diverse types of sexual systems to study

Formatted: Highlight

Formatted: Highlight

Deleted: across eukaryotes

Deleted: relevant

Deleted: eukaryotic

Deleted: included

Deleted: effort

We identified the female (U) and male (V) sex-determining regions (SDRs) in dioicous species **define dioicous and monoicous** (see methods), showing that all U/V species share the same **(highly rearranged)** ancestral sex chromosome (Fig. 1B-C). Thus, the recombination suppression event leading to the birth of U/V sex chromosomes occurred between the split of *S. ischiensis* and *D. dichotoma* around 450-224 Mya²⁰ (Fig. 1A). The male-determining gene *MIN*²¹ is consistently present in all V-SDRs of the dioicous species, and we note that **one** dioecious **species** (*F. serratus*) and **two** monoicous (*C. linearis* and *D. dudresnayi*) species lack U/V sex chromosomes but still retain *MIN* on a chromosome homologous to the ancestral U/V ('sex-homolog' hereafter). The outgroup *S. ischiensis* had remarkably low synteny with the brown algae, and exhibited extensive and **irreversible**²², fusion-with-mixing events²² (Fig. 1A; Fig. S1). The presence of *MIN*²¹ in *S. ischiensis* suggests that the sex-related role of *MIN* may pre-date the emergence of the Phaeophyceae.

We next examined the SDRs of U and V chromosomes by comparing male and female genome **assemblies** (see methods). The SDRs showed considerable variation in gene content and size across species, the smallest being found in the Ectocarpales (Fig. 1B, 1C, Fig. S2A, Table S1). **Many genes located in the pseudoautosomal region (PAR) of Ectocarpales have moved into the V-SDR of *U. pinnatifida*, *D. herbacea* and *D. dichotoma* — by what mechanism — or does this simply mean “are within” (as a result of either rearrangements or changes in the PAR boundaries?)**. These size differences coincide with extensive structural rearrangements, particularly inversions, even among closely related taxa (Fig. 1B-C, Fig. S3). Thus, structural rearrangements may **have caused** the recombination suppression event around the *MIN* gene, **creating the brown algae U/V sex-linked regions,**

Together our results indicate that the U/V sex chromosomes of the brown algae emerged between 450 and 224 Mya, via an inversion that suppressed recombination in a locus that contained the male-determining *MIN* factor. The presence of *MIN* in distantly related lineages could push the age of the U/V chromosomes further back in time, but more evidence would be required to establish that dioicy existed in these organisms.

Formatted: Highlight

Formatted: Highlight

Formatted: Highlight

Formatted: Highlight

Deleted: We noticed that m

Deleted: on

Deleted: had

Formatted: Highlight

Formatted: Highlight

Formatted: Highlight

Formatted: Highlight

Deleted: underlie

Deleted: that gave rise to

Deleted: chromosomes

Deleted: in brown algae

Deleted: determinant

Figure 1. Origins of U/V sex chromosomes in brown algae. (A) Macrosynteny plot comparing genomes of six dioicous (green), two monoicous (red), one dioecious (blue) and one outgroup species (yellow). Syntenic blocks of the V sex chromosome are highlighted in red, with the emergence of U/V chromosomes shown in the phylogeny. Genome sizes are indicated in brackets. **(B)** Microsynteny plot of V chromosomes in five dioicous species, highlighting the male sex-determining regions (blue) and the PARs (green). **(C)** Microsynteny plot of U chromosomes in two dioicous species, highlighting the female sex-determining regions (peach) and the PARs (green).

The evolution of the SDRs involved expansions and gene gains rather than gene degradation

The U/V-SDRs of *Ectocarpus* sp. 7 and *D. herbacea* carry homologous genes (gametolog pairs), indicating descent from a common ancestral region (Table S2). Despite differences in SDR gene numbers, both species show similar ratios of gametologs (53-61%) and U- or V-specific genes (47-39%; Table S2). The latter either evolved after U/V-SDR divergence or were lost from the counterpart haplotype. Note that U and V-SDR gene counts coincide in both species (Table S2), supporting parallel evolution of U and V chromosomes²³⁻²⁵.

Diploid sex chromosome in animals and plants exhibit evolutionary strata representing different recombination suppression events over time. Strata are identified by analyzing synonymous substitutions (K_s) between male/female gametolog pairs²⁶ but detecting strata in U/V systems is difficult due to gene movements and chromosome rearrangements disrupting collinearity between species^{23,27,28}. In both *Ectocarpus* sp. 7 and *D. herbacea*, the V and U-SDR arrangements are inverted (Fig. 2A-B), consistent with the idea that inversions may lead to suppressed recombination between sex chromosomes. K_s analysis showed that gametologs in the U/V-SDRs of *Ectocarpus* sp. 7 are mostly highly diverged, in two very old strata, with only two gametolog pairs in a recent stratum ($K_s < 1$; Fig. 2B-C, Table S3). In contrast, *D. herbacea* exhibited at least four strata (Fig. 2B-C). The K_s values are broadly consistent between orthologs in *Ectocarpus* sp. 7 and *D. herbacea* (Table S3). The relative position of the most recent evolutionary strata in *D. herbacea* indicate that U/V-SDRs expansions in brown algae are driven by nested inversions.

Larger V-SDRs in early-diverging lineages (like *D. dichotoma*) and smaller V-SDRs in later-diverging orders (Ectocarpales) suggest that gene loss might have reduced SDR sizes. Alternatively, independent expansions of U/V-SDRs may have occurred across lineages, with the smaller U/V-SDR size in Ectocarpales reflecting an ancestral state. To analyse SDR size evolution, we reconstructed the ancestral SDR gene content. We focused on the V chromosome (as the genome assemblies are better quality for males than females, and assuming parallel U/V-SDR evolution²³⁻²⁵). This analysis revealed that brown algal V-SDR evolution involves lineage-specific expansions from the state in a common ancestor with seven SDR genes, rather than reductions (Fig. 2E-F). Consistently, ancestral V-SDR genes were associated with higher K_s values, while independently-acquired gametologs in *D. herbacea* had lower K_s values (Table S2, Fig. 2C).

The seven genes in the ancestral V-SDR (Fig. 2F, S2B) include the male-determinant *MIN*²¹ and six V gametologs of genes also carried on the U chromosome (Fig. 2F, Table S4). Interestingly, all likely related to sex determination processes. Gametolog pairs include putative transmembrane proteins that may play a role in gamete recognition²⁹, STE20 serine/threonine kinase gametologs likely involved in pheromone pathways³⁰, and a casein kinase, a MEMO-like domain protein and a GTPase-activating proteins which may act in signal transduction (Table S4). None of these genes show signs of sequence degeneration in *Ectocarpus* sp. 7, but in *D. herbacea* the casein kinase was lost from the U-SDR and the putative transmembrane receptor was not found. Remarkably, we noticed that these ancestral V-SDR genes remain in the sex-homolog of the species that have lost their UV system (Fig. 2F, Table S4), emphasizing their importance for pathways in sex determination even in absence of sex chromosomes.

Deleted: s driven by

Deleted: contain

Deleted: between the U/V-SDRs

Deleted: autosomal

Deleted: a

Deleted: sex

Deleted: emerged

Deleted: by

Deleted: shuffling and dynamic

Deleted: -

Deleted: We observed an inverted pattern between V and U-SDR in...

Deleted: demonstrating

Deleted: have

Deleted: -

Deleted: divergent

Deleted: gametologs

Deleted: ($K_s > 1$)

Deleted: were

Deleted: examine

Deleted: dynamics

Deleted: state of

Deleted: due to

Deleted: better male

Deleted: assembly

Deleted: quality

Deleted: driven by

Deleted: , rather than reductions,

Deleted: few

Formatted: Font: Not Bold

Intriguingly, we found the V-SDR size to be strongly linked to the extent of sexual dimorphism (Fig. S2A). Species with low sexual dimorphism (anisogamous) retained the ancestral V-SDR genes with very few gene gains, suggesting they may represent the V-SDR ancestral state. Oogamous species each independently gained diverse V-SDR genes, except for one gene (ATP-dependent RNA helicase) that was convergently acquired in all. Oogamy has been suggested to be the ancestral state in the brown algae¹⁵ or its crown radiation clade²⁰, but our finding of independent V-SDR expansions positively associated with changes in the level of sexual dimorphism (Fig. 2E; Fig. S2A) supports instead the recurrent, independent evolution of oogamy in the brown algal lineage.

Next, we investigated the fate of the PAR genes of *Ectocarpus* sp. 7 that became sex-linked following the SDR expansion in *D. herbacea*. This expansion led to engulfment of a PAR1 region of *Ectocarpus* sp. 7 with four genes, and a second expansion in the PAR2 with 13 genes (Fig. 2D, Table S5). Most genes in these younger SDR strata of *D. herbacea* were retained as gametologs, further supporting low levels of U/V degeneration. Note that the PAR2 of *Ectocarpus* sp. 7 also contains six genes that are not found in *D. herbacea* but they represent *Ectocarpus* taxonomically restricted genes (TRGs). Another gene remained as a PAR gene in *D. herbacea*, and the final two were located on the *D. herbacea* autosomes.

In contrast to the positive association between SDR size and gamete dimorphism described above, we found no correlation between autosomal sex-biased gene (SBG) expression and sexual dimorphism level (Table S7), supporting previous studies³¹ (Fig. S5A). However, we observed an enrichment of male-biased genes on the sex chromosomes in all species except *D. dichotoma* (Fig. S5B).

Most U/V-SDR genes were prominently expressed in fertile haploid gametophytes, consistent with gene preservation via haploid purifying selection (Table S6). Gametologs had consistently higher expression levels than genes with sex-limited expression (Wilcoxon test, p-value=0.00075 in *D. herbacea*; p-value=0.08843 in *Ectocarpus* sp. 7) (Fig. 2D). A comparative analysis in fertile gametophytes between SDR genes and their autosomal counterparts in other species showed that newly acquired genes on the SDR had similar expression levels to their autosomal counterparts (Fig. S4), suggesting either a co-option of autosomal biological activity into male-specific functions in the V-SDR or the general importance of these genes for gametophyte development. Examining expression levels across multiple tissues in *Ectocarpus* sp. 7 revealed that activity of many U/V-SDR genes is not confined to fertile gametophytes (Fig. 2D). Therefore, the SDRs contain both genes involved in sex determination and gametophyte fertility but also genes playing a broader role in development.

Altogether, our analyses illustrate how brown algal U/V-SDRs are structurally dynamic, evolving mainly by lineage-specific gene gain via recurrent inversions, and SDR expansions are associated with increasing levels of sexual dimorphism. We uncovered a set of genes that has remained conservatively sex-linked across the evolution of dioicous brown algae, suggesting that these genes may have a key role in sex determination and/or differentiation, but the SDR also contain genes that may be involved in other developmental pathways.

Deleted: genes

Figure 2. Lineage-specific U/V-SDR expansion from an ancestral SDR and its association with sexual dimorphism. (A) Microsynteny plot between the U and V chromosomes of *Ectocarpus* sp. 7 and *D. herbacea*. (B) Synteny between the U and V gametologs within the SDRs of both species, colored by synonymous substitutions per site (K_s). (C) Identification of evolutionary strata in the male SDR of both species based on K_s values and the position of male gametologs. (D) Circos plot linking gametolog pairs in each species with expression levels of all SDR genes ($\log_2(\text{TPM}+1)$) across different life stages in *Ectocarpus* sp.7 and mature gametophytes (*matGA*) in *D. herbacea*. Gametologs are highlighted in dark colors, sex-limited genes are highlighted in light colors, insertions are marked in grey, and stars denote conserved SDR genes (also in panel F). (E) Ancestral state reconstruction of V-SDR gene content across brown algae, showing the expected number of genes in the SDR (white circles), gene retention (blue), gain (green) and loss (red) along with changes in gamete dimorphism¹⁵. (F) Schematic of the seven ancestral V-SDR genes, with genomic locations marked: V-SDR (blue), pseudo-autosomal region/sex-homolog (green), autosomal (yellow), unknown (red), and lost (grey). Bold: MIN. See Table S4.

Structural features and evolutionary dynamics of brown algal UV sex chromosomes

We next examined the structural features that differentiate V-SDR and PAR regions from the rest of the genome. As expected for non-recombining regions³², all the V-SDRs are repeat rich and gene poor (Fig. 3A; Table S8, S9). Among repetitive elements, ‘unclassified’ transposable elements (TEs) were enriched in the sex chromosome of the Ectocarpales, whilst the V-SDRs of species that underwent genome expansion (e.g. *U. pinnatifida*, *D. herbacea*, *D. dichotoma*) predominantly accumulated LTR elements (Fig. 3A; Fig. S6; Table S10).

Moreover, the sex chromosomes had fewer conserved orthologs between species compared to the autosomes (Chi-squared test, p -value $< 10^{-4}$, Table S11), and phylostratigraphy analyses^{34,35} revealed an enrichment of taxonomically-enriched (TGRs) in the sex chromosomes of all dioicous species (Fig. 3B; Table S12). Importantly, the sex-chromosome-enriched gene-age categories are significantly younger (species, genus and family levels) than their last common ancestors (same Order or broader taxonomic groups), indicating that the TRG enrichment in the sex chromosomes arose independently in each species (Figs. S7-S11).

We previously proposed a theoretical model where generation-antagonistic selection drives TRG accumulation, favoring the retention of young sporophyte-beneficial loci in the *Ectocarpus* sp. 7³³ UVs. Consistent with this model, sporophyte-biased genes are indeed enriched in the sex chromosomes of *Ectocarpus* sp. 7 and *U. pinnatifida*, but less so in *D. dichotoma* and *S. promiscuus* that lack heteromorphic generations^{15,31} and consequently no scope for generation-antagonism (Table S13). Moreover, we explored additional mechanisms underlying TRG emergence by estimating K_s values between orthologs in closely-related species and comparing these values between chromosomes and genomic compartments (V-SDR, PAR, autosomes). If synonymous mutations behave neutrally^{36,37}, K_s is a proxy for mutation rates^{38,39}. Consistently, we found higher K_s values in the V sex chromosomes across all dioicous brown algae (Fig. 3A; Table S14), suggesting higher mutation rates relative to autosomes. Young gene enrichment and higher K_s values were localized in the PARs of the Ectocarpales (Fig. S7-S8), but this pattern extended to the entire sex chromosome in species with larger V-SDRs (Fig. S9-S11). Therefore, the enrichment of TGRs in the U and V is associated with both enrichment of sporophyte-biased genes and to higher mutation rates.

TRGs can either emerge *de novo* from ancestral non-genic regions⁴³, through pronounced selection-driven sequence divergence⁴⁴ or through neutral mutations leading to untraceable homology⁴⁵. To evaluate if the

Deleted: were

Deleted: were

Deleted: colonized

Deleted: by

Deleted: we noticed that

Deleted: to

TRGs in the sex chromosomes represent putative *de novo* gene birth events, we analyzed six TGRs from the PAR of *Ectocarpus* sp. 7 and compared them across ten *Ectocarpus* species¹⁶. We detected orthologous regions that corresponded to protein-coding genes and noncoding DNA in the other *Ectocarpus* species (**Table S15**). We performed ancestral sequence reconstructions, showing that five out of the six analyzed genes represent potential *de novo* gene birth events^{43,46} from ancestral sequences lacking open reading frames (**Fig. 3C; Fig. S12**). Four of the putative *de novo* genes showed transcriptional activity ($\log_2(\text{TPM}+1) > 1$ at any life stage; **Table S13**), as well as 85.5% of the TRGs in the V chromosome of *Ectocarpus* sp. 7 within the two youngest gene ages. Nonetheless, these putative *de novo* TRGs might still represent noncoding transcripts containing an ORF. We propose that brown algal U/V sex chromosomes act as ‘gene cradles’, fostering novelty through *de novo* birth of coding or non-coding loci.

To test the generality of this pattern, we applied the same approach in other organisms with haploid sex determination systems, the plants *Ceratodon purpureum*, *Sphagnum angustifolium*, *Marchantia polymorpha*^{11,40,41}, and the fungus *Cryptococcus neoformans*⁴². We observed a clear enrichment of TRGs in the V chromosomes of *C. purpureum* and *S. angustifolium* (**Figures S13-S14, Table S12**), but found no such pattern in the highly degenerated U/V chromosomes of *M. polymorpha* or the mating-type chromosome of *C. neoformans* (**Figures S15-S16, Table S12**).

Together, our analyses show that brown algal U/V chromosomes are TE rich, have decreased gene density, higher mutation rates, and are a hotspot for the birth of genetic novelty. The gene cradle pattern appears to be a widespread feature of mildly degenerated UV, but not mating type, chromosomes.

Figure 3. The U/V sex chromosomes act as cradles for putative de novo gene birth. (A) Circos plots for five dioicous species showing: 0) chromosome compartment (autosomes, PARs and SDR); 1) relative gene ages, 2) *Ks* values, 3) proportion of gene (red) and repeat (blue) density; 4) repetitive element classification. (B) Violin plots and physical distribution of relative gene ages across one autosome and the V chromosomes of five dioicous species. The SDR of the V chromosomes are shaded in grey. (C) Identification of putative de novo gene birth events in the PAR of *Ectocarpus* sp. 7 across the *Ectocarpus* phylogeny. The inferred status of pseudogene or noncoding DNA in the orthologous regions is based on maximum likelihood ancestral state reconstruction.

Fate of U/V sex chromosomes following loss of dioicy

We explored the evolutionary trajectory of brown algal genomes after the loss of the U/V system, by exploring two independent transitions from dioicy to monoicy. Most genes in the 'ex'-sex chromosomes (sex-homologs) of both independently derived monoicous species *C. linearis* and *D. dudresnayi* are male-derived, indicating that monoicy emerged from a male background (Fig. 4A-B). The sex-homolog of *C. linearis* contains several rearrangements spanning the regions that are homologous to the PAR and SDR (SDR-homolog), with 11 V-SDR-derived and two U-SDR-derived orthologs located within the SDR-homolog, with an additional V-SDR-derived gene that was translocated elsewhere in the sex-homolog (Fig. 4A; Table S16). Likewise, *D. dudresnayi* underwent at least two inversion events within the SDR-homolog after splitting from *D. herbacea* (Fig. 4B), containing 20 V-SDR-derived genes and four U-SDR-derived genes (Table S17).

Both monoicous species retained gametolog copies or closely-related autosomal paralogs for most of their lost U/V-SDR orthologs (Table S16-S17), though it is unclear if the expression of these autosomal paralogs is compensating the activity of the lost genes. The only three U-SDR-derived orthologs in *C. linearis* are flanked by PAR orthologs translocated across the corners of the SDR-homolog, suggesting that *C. linearis* acquired its U-SDR-derived genes through two ectopic recombination events (Fig. S17A). Three U-SDR-derived genes in *D. dudresnayi* are dispersed across the SDR-homolog, suggesting independent translocation events, while the fourth U-SDR gene was translocated to an autosome (Fig. S17B).

The seven ancestral V-SDR genes are transcriptionally active during reproductive stages of both monoicous species (Table S18), indicating that the ancestral V-SDR genes are important in reproduction despite the absence of a U/V system, particularly *MIN*²¹ which is retained in both species. While most U-SDR-derived genes are absent in the monoicous species, an intracellular cholesterol transporter gene is present in both species (Table S18) and it is actively expressed during fertility in both *Ectocarpus* sp. 7 and *D. herbacea* (Table S6), suggesting its importance in the emergence of monoicy and in the female developmental pathway.

Lastly, we observed that the sex-homolog of *D. dudresnayi* retains some evolutionary vestiges of its past as a U/V chromosome, such as lower gene density, higher TE density and the gene cradle pattern, although the evolutionary mechanisms driving these patterns are gradually being lost, as shown by non-significant differences in *Ks* values across the genome (Fig. 4C; Fig. S18; Tables S8-S12, S14).

Figure 4. Fate of sex chromosomes during transitions from dioecy to co-sexuality (monoecy). (A) Comparison of the sex-homolog in *C. linearis* against the U and V chromosomes of *Ectocarpus* sp. 7. (B) Comparison of the sex-homolog in *D. dudresnayi* against the U and V chromosomes of *D. herbacea*. The color code represents the identity of the genes alongside the chromosomes, while the shapes represent the evolutionary fate of each SDR gene in the monoecious genome. The matching shades between the SDRs and the sex homolog are either color-coded by their ancestral background or they appear as transparent dotted shades if the gametolog of the other sex was retained. (C) Circos plot of *D. dudresnayi* showing: 0) chromosome compartment (autosomes and sex-homolog); 1) relative gene ages, 2) Ks values, 3) proportion of gene (red) and repeat (blue) density; 4) repetitive element classification.

Finally, we examined the transition from haploid to diploid sex determination, which has remained undescribed in Eukaryotes because it typically occurred in deep evolutionary times. The Fucales recently transitioned to a diploid life cycle⁴⁷, with many species, such as *Fucus serratus*⁴⁹ exhibiting diploid separate sexes (dioecy)⁴⁸. Ancestral state reconstruction suggests that dioecy emerged in the last common ancestor of the *Fucus* genus⁵⁰ around 25 to 5 Mya¹⁶, consistent with a young sex chromosome in *F. serratus* that arose following the transition to a diplontic life cycle¹⁵. Accordingly, our high-coverage genome sequencing and RAD-seq data of males and females from a field population failed to identify sex-linked sequences in *F. serratus*, likely due to a small and undifferentiated SDR.

Male *F. serratus* conserves all the ancestral V-SDR genes in its sex-homolog (**Fig. 2F**), although some of these genes likely arose from the ancestral U-SDR (**Fig. 5A**). Importantly, we found that none of the U/V-SDR-derived genes are sex-linked in *F. serratus*, yet *MIN* and four other ancestral V-SDR genes are fully silenced in the females and exhibit strong male-biased expression (**Fig. 5A, Table S19**). This pattern is consistent across three other Fucales species, namely *Ascophyllum nodosum*, *Fucus ceranoides* and *Fucus vesiculosus* (**Table S20**). The only U-SDR-limited gene found in *F. serratus* shows no differential expression between males and females (**Fig. 5A**). Therefore, we conclude that whilst the ancestral V-SDR genes are no longer sex-linked in the Fucales, they likely still play a role in male-sex determination or differentiation pathways.

Interestingly, the sex-homolog of *F. serratus* lacks the gene cradle pattern and all the other features that distinguished it from autosomes. Thus, this 'ex'-sex chromosome has lost all the evolutionary vestiges of its past as a U/V chromosome (**Fig. 5B; Fig. S19; Tables S8-S12, S14**).

Deleted: ,

Figure 5. Transition from haploid to diploid sex determination. (A) Expression of ancestral U/V-SDR genes in the diplontic species *F. serratus*. Gene expression of mature algae (using 3 males and 3 females, see methods) is given as log₂(TPM+1) and bars represent standard deviation of the mean. Bold text represents whether the gene in *F. serratus* corresponds to an ancestral male or the female gametolog. (B) Circos plot of *F. serratus* showing: 0) chromosome compartment (autosomes and sex-homolog); 1) relative gene ages, 2) Ks values, 3) proportion of gene (red) and repeat (blue) density; 4) repetitive element classification.

DISCUSSION

The rise of brown algal sex chromosomes

Here, we characterized the evolutionary trajectory of the sex chromosomes and sex-homologs using representative brown algal species and one outgroup (Fig. 6). Brown algal sex chromosomes date back 450 to 244 Mya²⁰, at the origin of multicellular brown algae. We propose that *MIN* is the main driver behind the birth of this U/V system, due to its conserved and crucial role in male sex determination²¹. The ancestral cassette with seven V-SDR genes is highly conserved across the brown algae, suggesting a very early sequestration of these genes into a non-recombining region during the emergence of the U/V-SDRs. The other six ancestral V-SDR genes likely contribute to reproduction, but not necessarily to male-sex-determination, since they share gametologs with the U-SDR. Whether their sequestration was essential for maintaining the U/V system or incidental during the initial recombination-suppression event remains to be determined.

Brown algal genomes have a high degree of synteny conservation, akin to the synteny conservation associated with multicellularity in Metazoans⁵¹. Despite conserved synteny in brown algal autosomes, the U/V-SDR in the sex chromosomes is structurally highly dynamic, and our results demonstrate the key role for inversions in the

evolution of brown algal U/V-SDRs, driving both the initial recombination suppression in proto-sex chromosomes and the later expansion of the U/V-SDRs into the PARs. Similar to other haploid systems^{32,52}, TEs accumulated in the SDRs following recombination suppression⁵³, possibly causing further rearrangements through TE-mediated inversions⁵⁴.

Independent SDR expansions are linked to sexual dimorphism and rise of morphological complexity

Models of XX/XY and ZW/ZZ sex chromosome evolution suggest that sexual dimorphism may drive SDR expansion, promoting the formation of new evolutionary strata^{55,56}. We expand this model to UV systems by demonstrating that greater sexual dimorphism in gametes is linked to changed U/V-SDR boundaries in brown algae. This association suggests an interplay between morphological complexity, retention/acquisition of sex-specific genes in the SDR, and the emergence of sexual dimorphic traits. Whilst sexually antagonistic selection⁵⁷ may drive gene movements into the SDR, further analyses will be needed to demonstrate that sequestered genes in the U/V-SDRs are important for reproduction. Strong gamete size differences (oogamy) were proposed to be ancestral in the brown algae, but this trait seems to be highly labile^{15,20} and our analyses support multiple independent transitions to oogamy from a less dimorphic ancestor, accompanied by sex locus expansion. Similar to *M. polymorpha*, the streamlined gene content of the U/V-SDRs in the brown algae could regulate an autosomal effector gene network controlling sexual dimorphism. Accordingly, we observed substantial sex-biased gene expression in mature gametophytes. This observation contrasts with *C. purpureus*, where sex chromosomes carry thousands of genes and few sex-biased autosomal genes⁴⁰. Consistent with other studies^{31,58}, we found no correlation between levels of sex-biased expression and sexual dimorphism in brown algae. We propose that sexual dimorphism in this lineage may be controlled by a relatively small subset of genes, and most sex-biased genes may influence gametophyte physiology or vegetative development.

Established models of XX/XY and ZW/ZZ systems posit that recombination suppression between sex chromosomes explains Y and W degeneration⁵⁹ (but see ⁶⁰⁻⁶²). Differences in the evolution of UV systems are predicted, since sex-determination occurs in the haploid stage, in which deleterious mutations are more efficiently removed by selection than in diploid systems^{28,63} and some of the predicted differences have been documented in bryophytes²⁸. In brown algae, as in diploid systems, the SDRs appear to have lost recombination, possibly by successive inversions, TEs accumulation and consequent reduction of gene density. Remarkably, though, while TE accumulation is linked to recombination suppression, low gene density is not necessarily due to gene loss by degeneration or movements out of the SDR. Rather, the SDRs expanded by an accumulation of TEs in noncoding regions, leading indirectly to the overall gene density reduction.

Unlike the degeneration documented in some XX/XY and ZW/ZZ systems, our ancestral state reconstruction revealed more gene gains than losses or gene movements out of the V-SDR in all brown algal lineages. Enrichment of unclassified TEs in the sex-linked regions of Ectocarpales, which have small genomes, but not in species with larger genomes, where LTR retroelements become dominant. DNA transposons are overrepresented among unclassified repeats⁶⁵, and they insert near the progenitor locus, in a process termed 'local hopping'⁶⁴. The U/V-SDR may thus act as a source of DNA transposons that hop to the PARs, whereas, in larger genomes, increased colonization of LTR elements obscures this.

- Deleted: expanded
- Deleted: rather
- Deleted: leads to mutation accumulation and
- Deleted:
- Deleted: of the Y or W
- Formatted: Highlight
- Formatted: Highlight
- Deleted: Our analyses reveal key d
- Deleted: ary paths
- Deleted: compared to XX/XY and ZW/ZZ
- Formatted: Highlight
- Formatted: Highlight
- Formatted: Highlight
- Deleted: them
- Formatted: Highlight
- Deleted: previously no
- Formatted: Highlight
- Deleted: U/V
- Deleted: structure changes through
- Deleted: Since sex-determination occurs in the haploid stage, deleterious mutations are more efficiently purged than in diploid systems^{28,63}. We observed unclassified TEs e
- Deleted: sex chromosomes
- Deleted: er
- Deleted: whereas this pattern disappear
- Deleted: s
- Deleted: exhibit local hopping,
- Deleted: ing themselves in the vicinity of
- Deleted: donor
- Deleted: locus
- Deleted: We propose that t
- Deleted: s
- Deleted: and
- Deleted: ey subsequently expan
- Deleted: d
- Deleted: through local hopping. I
- Deleted: this signal is diluted by the

U/V sex chromosomes are gene nurseries

Brown algae U/V chromosomes display an excess of TRGs, and our analysis indicates that some of these TRGs may have evolved *de novo* from previously non-coding regions. Thus, U/V sex chromosomes function as ‘gene cradles’, likely via a combination of rapid sequence divergence and *de novo* gene birth. The transcriptional activity of these genes indicates that TRG enrichment is not driven by annotation artifacts, although we cannot exclude that some TRGs may be non-coding transcripts with an ORF. What mechanism underlie the ‘cradle’ pattern? U/V chromosomes are enriched in heterochromatin⁶⁶, likely involved in repressing TEs⁶⁷, and heterochromatic regions tend to have higher mutation rates due to reduced access of the DNA repair machinery during replication⁶⁸. Accordingly, we consistently observe higher mutation rates in U/Vs, which could facilitate the emergence of *de novo* loci. Alternatively, the high density of DNA transposons within the U/V could promote the co-option of their regulatory motifs and enable *de novo* transcript birth, as seen in *Drosophila*⁶⁹. These TRGs could then be retained in the sex chromosomes by ‘generation-antagonistic selection’³³, a mechanism requiring distinct selective pressures between gametophyte and sporophyte generations, implying a heteromorphic life cycle. Accordingly, species with no generation dimorphism or reduced sporophytes have a weaker gene cradle pattern. Note that the gene cradle pattern could be reinforced through generation-antagonistic selection, but DNA transposons and higher mutation rates may be sufficient to initiate this pattern in species lacking sporophyte-biased gene expression. Importantly, the gene cradle pattern is unique to U/V systems and gradually disappears when these systems are lost, as in monoicy or diploid sexual species. This pattern extends beyond brown algae to other eukaryotic with U/V systems, mild chromosomal degeneration, and complex haploid-diploid life cycles where sporophytes are well-developed. Therefore, our study uncovers a unique interplay between complex life cycles, heterochromatic landscape, DNA transposons, and higher mutation rates that may drive *de novo* gene birth in UV chromosomes, and this process is pervasive across distant, independently evolved eukaryotic kingdoms.

Deleted: putative

Loss of U/V systems

Previous reports suggested that monoicous brown algae have transcriptomic profiles resembling ancestral females³¹. However, our results show that monoicy arose at least twice from a male ancestor that acquired female genes. The expression of male-biased genes tends to be tissue-specific while female-biased genes tend to be broadly expressed⁷², which may explain the observed similarity between female and monoicous transcriptomes in brown algae^{31,72}. The male pathway may require more elements from the V-SDR such as *MIN*^{21,23,73}, which could have facilitated the emergence of monoicy from males, as also seen in the green lineage^{74,75}. Notably, we identified a U-SDR-derived gene present in both monoicous species, suggesting its fundamental role in the female developmental pathway. We thus propose that monoicy emerged via ectopic recombination adding essential female genes to a male genetic background. While combining crucial female and male genes is necessary for this evolutionary transition, the retention of a sex chromosome should not be. For example, in *Volvox africanus* monoicy required the retention of female SDR-like regions, while most male SDR genes were lost except for a multicopy array of the male-determining gene *MID*⁷⁶. Taken together,

Deleted: The demise

Deleted: o times

Formatted: Highlight

Formatted: Highlight

Deleted: specific

Deleted: appears less important

these findings emphasize how genetic networks governing sex determination can be rewired during transitions to monoicy. Essentially, an event combining crucial U and V genes is necessary, but the mechanisms and genetic background may vary between systems.

The evolution of a diploid XX/XY system in *F. serratus* is associated with a transition to diploidy in Fucales, which has not subsequently reversed^{15,50}. The UV to XY transition has remained elusive^{3,50}, but our data imply that in brown algae it involved an intermediate monoicous stage rather than a direct shift from the U/V to a diploid system. The diploid XX/XY system in *Fucus* is younger than the Y chromosome in mammals (180 Mya) or the W chromosome in birds (140 Mya)⁷⁷. A small, undifferentiated Y-specific region may explain why the sex chromosome in *F. serratus* was undetectable. Nonetheless, we found all ancestral V-SDR genes in the sex-homolog of *F. serratus*, several showing a male-biased gene expression across Fucales species, particularly *MIN*²¹. Our findings imply that *MIN* and possibly other ancestral V-SDR genes are still involved in male differentiation, but shifted downwards in the sex determination cascade. These results thus support and extend the “bottom-up” hypothesis of sex determination, where downstream components of sex differentiation are conserved across taxa, and new master-sex regulators can replace older ones⁷⁸.

- Deleted: ¶
- Deleted: the
- Deleted: n
- Deleted: irreversible
- Deleted: wards
- Deleted: s
- Deleted: transition
- Deleted: wards
- Formatted: Highlight
- Deleted: implies

Figure 6. Hypothetical model for U/V sex chromosome evolution. U/V sex chromosomes arose from an ancestral autosome, via suppression of recombination that likely occurred via an inversion. The SDR expanded into neighboring pseudoautosomal regions (PAR) via inversions, but also by recruitment of genes from autosomes; expansion occurred in a lineage-specific fashion, concomitant with increased sexual dimorphism of the different species. SDR genes are maintained within the SDR if they have roles in sex, whereas genes with no role in sex are lost. Faster substitution rates, likely driven by the heterochromatic context of the sex chromosome may promote the rise of young genes, which are selectively maintained on the sex chromosome if they have advantages to the sporophyte generation. In species that switch to a diploid life cycle, the U/V system disappears, but the genes that are in the V-specific region retain roles in sex, although they are no longer masters. Transition from U/V separate sexes to co-sexuality (hermaphroditism) occurred when a male haploid individual acquired female-specific genes via ectopic recombination. During the demise of the U/V sex chromosomes, their structural and evolutionary footprints slowly erase.

ACKNOWLEDGEMENTS

This work was supported by the MPG, the CNRS, Sorbonne University, the ERC (grant n. 864038 and 638240 to SMC), the France Génomique National infrastructure project Phaeoexplorer (ANR-10-INBS-09), the JSPS Overseas Research Fellowships (to MH), the BMBF-funded de.NBI Cloud within the German Network for Bioinformatics Infrastructure (de.NBI) (031A532B, 031A533A, 031A533B, 031A534A, 031A535A, 031A537A, 031A537B, 031A537C, 031A537D, 031A538A), the Investissements d'Avenir project Idealg (ANR-10-BTBR-04-01), the European BG-01 BlueGrowth H2020 project Genialg (727892) and the ANR project Epicyle (ANR-19-CE20-0028-01). SMC is supported by the Moore Foundation (GBMF11489) and the Bettencourt-Schuller Foundation. JBR is supported by a Humboldt Research Fellowship for postdoctoral researchers from the Alexander von Humboldt Foundation. We thank the members of the Phaeoexplorer consortium, in particular Chloe Jolivet, Leticia Mest and Delphine Scornet for assistance with algae cultures, Corinne Cruaud for help with sequencing libraries preparation, Erwan Corre and Arthur Le Bars for support with the Phaeoexplorer database and Arnaud Couloux for the genome assemblies and annotations. We are grateful to the Roscoff Bioinformatics platform ABiMS (<http://abims.sb-roscoff.fr>), part of the Institut Français de Bioinformatique (ANR-11-INBS-0013) and BioGenouest network, for providing computing and storage resources.

AUTHOR CONTRIBUTIONS

JBR, APL: Investigation (equal); Formal analysis (equal); Methodology (equal); Visualization (equal), Writing – original draft (equal); Writing – review and editing (equal).

PL: Investigation (supporting); Formal analysis (supporting)

ED, GC, OG, KB, MH, KA, GL, EA, DL, RL, OG, SH, ZN, LG, AFP: Investigation (supporting)

GH, JMA, GP, PW, FD, JMC: Data curation (supporting); Data acquisition (supporting)

FBH: Investigation (supporting); Methodology (equal); Data curation (equal); Formal analysis (supporting)

SMC: Conceptualization (lead); Funding acquisition (lead); Methodology (equal); Project administration (lead); Supervision (lead); Visualization (supporting); Writing – original draft (equal); Writing – review and editing (lead).

DECLARATION OF INTEREST

The authors declare no competing interests

SUPPLEMENTAL INFORMATION

Table S1. General characteristics of the genomes and UV sex chromosomes in five dioicous brown algal species.

Table S2. *Ks* values and evolutionary strata of the V-SDR gametologs of *Ectocarpus* sp. 7 and *Desmarestia herbacea*.

Table S3. Gametologs and sex-specific genes for the male and female SDR in *Ectocarpus* sp. 7 and *Desmarestia herbacea*.

Table S4. Ortholog table of the 7 ancestral male SDR genes in 10 brown algal species.

Table S5. SDR expansion into the PAR in *Desmarestia herbacea* and the fate of the newly sex-linked genes.

Table S6. Expression ($\log_2(\text{TPM}+1)$) of male and female SDR genes during fertile gametophyte stage.

Table S7. Sex-biased gene expression between mature male and female gametophytes per species using DESeq2.

Table S8. FDR-corrected p-values for the pairwise Wilcoxon rank sum tests used to evaluate differences in the distribution of protein-coding gene content in 100kb windows across the chromosomes of seven brown algal species.

Table S9. FDR-corrected p-values for the pairwise Wilcoxon rank sum tests used to evaluate differences in the distribution of repetitive genomic elements in 100kb windows across the chromosomes of seven brown algal species.

Table S10. FDR-corrected p-values for the pairwise Wilcoxon rank sum tests used to evaluate differences in the percentage of unclassified transposons relative to the totality of repetitive elements in 100kb windows across the chromosomes of seven brown algal species.

Table S11. Chi-square test between the observed and expected value of orthologs for each chromosome in seven brown algal species.

Table S12. FDR-corrected p-values for the pairwise Wilcoxon rank sum tests used to evaluate differences in the distribution of gene age categories across the chromosomes of seven brown algal species, three plant species and one fungal species.

Table S13. Gene expression measured as $\log_2(\text{TPM}+1)$ in gametophytes and sporophytes of different algal species.

Table S14. FDR-corrected p-values for the pairwise Wilcoxon rank sum tests used to evaluate differences in the distribution of synonymous substitutions per site (*Ks*) across the chromosomes of six brown algal species.

Table S15. Detection of orthologous sequences for six *Ectocarpus de novo* candidate genes.

Table S16. Evolutionary “fate” of the *Ectocarpus* sp. 7 sex-determining genes in *Chordaria linearis*.

Table S17. Evolutionary “fate” of the *Desmarestia herbacea* sex-determining genes in *Desmarestia dudresnayi*.

Table S18. Gene expression of the seven ancestral V-SDR genes in the reproductive stages of *Chordaria linearis* and *Desmarestia dudresnayi*.

Table S19. Expression of conserved VSDR genes (*Ectocarpus* sp. 7 SDR genes as reference) in *Fucus serratus* male and female mature receptacles.

Table S20. Transcriptional activity for V-SDR homologs in three Fucales species.

Table S21. Genomic data used in this study with metrics and accession numbers.

Table S22. Pairs of orthologous genes used to calculate the expected against observed number of orthologs per chromosome and to calculate rates of synonymous substitutions per site (K_s) for seven brown algal species.

Table S23. Gene age assignments based on phylostratigraphy as implemented in GenEra (<https://github.com/josuebarrera/GenEra>) for seven brown algal species, three plant species and one fungal species.

Table S24. Gene and repeat content for seven brown algal species over 100 kb sliding windows.

SUPPLEMENTAL FIGURES

Figure S1. Macrosynteny plot between *S. ischiensis* and *D. dichotoma* using 1,828 orthologs. We highlight two fusion-with-mixing events (red squares) between chromosomes 4 and 9, and between chromosomes 23 and 24 in *D. dichotoma*.

Figure S2. Detection of independently-acquired V-SDR genes across species. (A) Differences in the size of the male SDR between brown algal species based on the total sequence length, the relative size of the SDR compared to the length of the V chromosome and the number of protein-coding genes retained within the SDR. The bars are colored according to the level of gamete dimorphism in each species (based on ¹⁵). **(B)** Gene trees showing the independent acquisition of SDR gametologs across species that were previously interpreted as part of the ancestral male SDR genes.

Figure S3. Male SDR synteny between *Ectocarpus sp. 7* and *Ectocarpus crouaniorum*. One of the species underwent a recent inversion event within the SDR. The arrows in the boxes represent the orientation of each gene within the chromosome.

Figure S4. Expression of genes ($\log_2(\text{TPM}+1)$) that entered the SDR independently in different species. Expression is measured in mature male and female gametophytes, hashing marks missing orthologs, stars inside the cells indicate that the gene is inside the male non-recombining region (V-SDR). Orthogroups containing orthologs in less than three species or with multicopy genes were excluded from this analysis. M: male; F: female.

Figure 55. Sex-biased gene expression per dioicous species. (A) Proportion of sex biased genes in each of the five dioicous species. MBG: male-biased genes; FBG: female biased genes. (B) Number of sex-biased genes in the pseudoautosomal regions of sex chromosomes (U-V-SDRs excluded), male-biased genes are shown in blue and female-biased genes in red. Stars above the bars mark significant enrichment of the sex-biased genes on the PAR (Chi-square test, ** $p < 0.01$, *** $p < 0.001$).

Figure S6. Distribution and proportion of classified TEs across the chromosomes of six brown algal species. The sex V chromosome or sex-homolog is highlighted in green. The residuals in the V sex chromosome were used to interpret the enrichment or depletion of Unclassified (Unknown) repetitive elements for each species.

Figure S7. Gene ages across the *Ectocarpus* sp. 7 genome. (A) Distribution of relative gene ages across the chromosomes of *Ectocarpus* sp. 7. The SDR of the V sex chromosome (chr 13) is highlighted with a red box. (B) The sex chromosome (red) has a significantly higher proportion of young genes and a lower proportion of old genes when compared to the autosomes (green; see Table S12). (C) Mosaic plot showing that the species-level (rank 15) and the genus-level (rank 14) genes are responsible for the enrichment of young genes in the sex chromosome. (D) The K_s values are significantly higher in the PARs of the sex chromosome when compared to the autosomes or the SDR (see Table S14).

Figure S8. Gene ages across the *S. promiscuus* genome. A) Distribution of relative gene ages across the chromosomes of *Scytosiphon promiscuus*. The SDR of the V sex chromosome (chr 13) is highlighted with a red box. (B) The sex chromosome (red) has a significantly higher proportion of young genes and a lower proportion of old genes when compared to most of the autosomes (green; see Table S12). (C) Mosaic plot showing that the species-level (rank 14) genes are responsible for the enrichment of young genes in the sex chromosome. (D) The K_s values are significantly higher in the PARs of the sex chromosome when compared to the autosomes or the SDR (see Table S14).

Figure S9. Gene ages across the *U. pinnatifida* genome. (A) Distribution of relative gene ages across the chromosomes of *Undaria pinnatifida*. The SDR of the V sex chromosome (chr 23) is highlighted with a red box. (B) The sex chromosome (red) has a significantly higher proportion of young genes and a lower proportion of old genes when compared to most of the autosomes (green; see Table S12). (C) Mosaic plot showing that the ALE-clade genes (rank 11) are responsible for the enrichment of young genes in the sex chromosome. (D) The K_s values are significantly higher in the sex chromosome when compared to the autosomes (see Table S14), showing similar values in the PARs and in the SDR.

Figure S10. Gene ages across the *D. herbacea* genome. (A) Distribution of relative gene ages across the chromosomes of *Desmarestia herbacea*. The SDR of the V sex chromosome (chr 03) is highlighted with a red box. (B) The sex chromosome (red) has a significantly higher proportion of young genes and a lower proportion of old genes when compared to most of the autosomes (green; see Table S12). (C) Mosaic plot showing that the genus-level (rank 11) genes are responsible for the enrichment of young genes in the sex chromosome. (D) The K_s values are significantly higher in the sex chromosome when compared to half of the autosomes (see Table S14). Non-significance of K_s values across chromosomes may be driven by the conflation with the K_s values in *Desmarestia dudresnayi*. The SDR displays higher K_s values compared to the PARs or the autosomes.

Figure S11. Gene ages across the *D. dichotoma* genome. (A) Distribution of relative gene ages across the chromosomes of *Dictyota dichotoma*. The SDR of the V sex chromosome (chr 02) is highlighted with a red box. (B) The sex chromosome (red) has a significantly higher proportion of young genes and a lower proportion of old genes when compared to most of the autosomes (green; see Table S12). (C) Mosaic plot showing that the species-level (rank 11) and the DFI-clade-level (rank 9) genes are responsible for the enrichment of young genes in the sex chromosome. *Ks* values were not calculated for *D. dichotoma*, due to a saturation of synonymous mutations with the closest species in the PhaeoExplorer database (*Halopteris paniculata*).

emergence of gene Ec-13_002510. The first step is a big insertion (pos. 117-269) in the last common ancestor between *E. sp. 7* and *E. sp. 9* (node #3#). Later, on the common ancestor between *E. sp. 7* and *E. sp. 6* (node #2#), a G-to-A substitution led to the emergence of a start codon, an insertion on the 3' end (pos. 270-276) introduced a stop codon and four-nucleotide deletion (pos. 24-27) led to a triadic sequence (i.e., a multiple-of-three nucleotide sequence). (B) Evolutionary steps leading to the emergence of gene Ec-13_002490. A two-nucleotide deletion (pos. 93-94) led to a triadic sequence with an open reading frame (ORF) in the common ancestor of *E. sp. 7* and *E. sp. 6* (node #3#). A later insertion (pos. 72) led to the disruption of this gene in *E. sp. 6*. (C) Evolutionary steps leading to the emergence of gene Ec-13_002420. The last common ancestor between *E. sp. 7* and *E. crovaniorum* (node #9#) experienced six deletions (pos. 21-22, 46-72, 129, 271-274, 284-288 and 305-306) that led to the emergence of a triadic sequence with an ORF. Later deletions led to the independent disruption of this gene in *E. siliculosus* (pos. 143-392), *E. sp. 3* (pos. 382-393) and *E. crovaniorum* (pos. 206-207). (D) Evolutionary steps leading to the emergence of gene Ec-13_002840. The last common ancestor between *E. sp. 7* and *E. crovaniorum* (node #9#) experienced three deletions (pos. 22-34, 114-117 and 183) that led to the emergence of a triadic sequence with an ORF. Later mutations led to the independent disruption of this gene in *E. sp. 5* (pos. 81), *E. sp. 6* (pos. 1-2) and *E. sp. 3* (pos. 90-107). (E) Evolutionary steps leading to the emergence of gene Ec-13_003160. The common ancestor of *E. sp. 7* and *E. sp. 9* (node #4#) experienced a T-to-A substitution (pos. 165) that led to the emergence of a stop codon. The common ancestor between *E. sp. 7* and *E. sp. 6* (node #3#) later experience a deletion (pos. 132) that shifted the sequence towards a triadic pattern. Finally, an additional insertion (117) established the triadic pattern within the sequence, leading to the emergence of an ORF in *E. sp. 7*.

Figure S13. Gene ages across the *C. purpureum* genome. (A) Distribution of relative gene ages across the chromosomes of *C. purpureum*. (B) The V sex chromosome (V; red) has a significantly higher proportion of young genes and a lower proportion of old genes when compared to the autosomes (green; see Table S12). (C) Mosaic plot showing that the species-level genes (rank 9) are responsible for the enrichment of young genes in the sex chromosome.

Figure S14. Gene ages across the *Sphagnum angustifolium* genome. (A) Distribution of relative gene ages across the chromosomes of *Sphagnum angustifolium*. (B) The V sex chromosome (LG20; red) has a significantly higher proportion of young genes and a lower proportion of old genes when compared to the autosomes (green; see Table S12). (C) Mosaic plot showing that the species-level genes (rank 8) are responsible for the enrichment of young genes in the sex chromosome.

Figure S15. Gene ages across the *Marchantia polymorpha* genome. (A) Distribution of relative gene ages across the chromosomes of *Marchantia polymorpha*. (B) The U/V sex chromosomes (chrU and chrV; red) show non-significant differences in gene age distribution when compared to the rest of the chromosomes (green; see Table S12). (C) Mosaic plot showing non-significant differences between the sex chromosomes and the autosomes.

Figure S16. Gene ages across the *Cryptococcus neoformans* var. *neoformans* JEC21 genome. (A) Distribution of relative gene ages across the chromosomes of *Cryptococcus neoformans*. (B) The mating-type chromosome (NC_006686.1; red) shows non-significant differences in gene age distribution when compared to the rest of the chromosomes (green; see Table S12). (C) Mosaic plot showing no discernible pattern of gene age distribution in any of the chromosomes.

Figure S17. Proposed scenarios for the transition from dioicy to monoicy in *Chordaria linearis* and *Desmarestia dudresnayi*. (A) The ancestor of *Chordaria linearis* likely underwent an initial translocation event from the U chromosome to the V chromosome, inserting part of the U-SDR and a piece of the 3' PAR towards the 5' end of the V-SDR through an ectopic recombination event. A subsequent inversion within this translocation spread the 3' PAR genes to both sides of the U-SDR insertion. Finally, a second ectopic recombination event inserted an additional piece of the U-SDR within the 3' PAR translocation. (B) The ancestor of *Desmarestia dudresnayi* underwent three translocations of U-SDR genes into the V-SDR. Additionally, a fourth translocation event happened between the U-SDR and an autosome (chr_04).

Figure S18. Gene ages across the *D. dudresnayi* genome. (A) Distribution of relative gene ages across the chromosomes of *Desmarestia dudresnayi*. (B) The sex-homolog in *D. dudresnayi* (chr 03; red) has a significantly higher proportion of young genes and a lower proportion of old genes when compared to most of the other chromosomes (green; see **Table S12**). (C) Mosaic plot showing that the species-level genes (rank 12) are responsible for the enrichment of young genes in the sex-homolog. (D) The K_s values are similar in the sex-homolog when compared to the other chromosomes (see **Table S14**).

Figure S19. Gene ages across the *F. serratus* genome. (A) Distribution of relative gene ages across the chromosomes of *Fucus serratus*. (B) The sex-homolog in *F. serratus* (LG15; red) shows no significant differences in gene age distribution when compared to the rest of the chromosomes (green; see **Table S12**). (C) Mosaic plot showing no discernible pattern of gene age distribution in any of the chromosomes. (D) The Ks values are similar in the sex-homolog when compared to the other chromosomes (see **Table S14**).

METHODS

Resource availability

Lead contact

Further information and requests for resources and reagents should be directed to and will be fulfilled by the lead contact, Susana M. Coelho (susana.coelho@tuebingen.mpg.de).

Data and code availability

- The accession numbers and download links for all the genomic data that was generated and used in this study are available on **Table S21**.
- This paper does not report original code.

Biological material

Scytosiphon promiscuus, *Dictyota dichotoma*, *Undaria pinnatifida* and *Desmarestia dudresnayi* haploid gametophytes were cultivated in the laboratory conditions as in ⁷⁹. We cultivated the gametophytes at 14°C with a photoperiod of 12:12 h light:dark an irradiance of 25µmol photons.m⁻².s⁻¹. The media consisted of filtered natural seawater (NSW), which was autoclaved and enriched with half-strength Provasoli nutrient solution (Provasoli-enriched seawater; PES)⁷⁹. We grew the first biomass in 140mm Petri dishes and the gametophytes were later transferred to 1L flask with gentle aeration. The gametophytes were fragmented once a month and the media were changed every two weeks to promote biomass production. Prior to freezing, gametophytes were treated with antibiotics for 3 days with a gentle agitation and under the same culture conditions. The first day, gametophytes were treated with a mix Streptomycin (2g/L of PES), Penicillin G (0.5g/L of PES) and Chloramphenicol (0.1g/L of PES); the next day with Ampicilin (1g/L of PES) and finally the last day with Kanamycin (1g/L of PES). Between each day of treatment and before freezing, gametophytes were rinsed with 500mL of NSW to remove the traces of antibiotic.

Samples for fucoid algae sexual and vegetative tissue were collected in the intertidal zone during low tides in June 2012 from Viana do Castelo (*F. vesiculosus*, *A. nodosum*) and Caminha (Rio Minho; *F. ceranoides*), northern Portugal. Sexual phenotypes were verified in the field by sectioning and observing receptacles under a field microscope. Tissue samples were flash-frozen in liquid nitrogen on the shore and transported to the laboratory in a cryoshipper, after which they were lyophilized and stored dry at room temperature on silica crystals. See **Table S21** for list of strains used in this study.

DNA and RNA extraction and sequencing

Genomic DNA was isolated from algal tissue (~100mg) by grinding into fine powder under liquid nitrogen and subsequent cell lysis in 500µL of Genomic Lysis Buffer (OMNIPREP for plant kit) for 1 hour at 60°C. The lysate was cleaned up with 200µL of chloroform and DNA was precipitated in EtOH. The DNA pellet was digested in CF buffer (Macherey-Nagel) for 45 min at 65°C and purified using NucleoBond AXG20 Mini columns according to the user manual (Macherey-Nagel). Final high molecular weight gDNA was quantified (Qubit), analyzed for purity (Nanodrop) and checked for size distribution (Femto Pulse System) before preparing the

sequencing libraries. We sequenced the libraries using an Oxford Nanopore Technologies (ONT) MinION Mk1B. We prepared the ONT libraries using an SQK-LSK110 library preparation kit for R9.4.1 flow cells and an SQK-LSK114 library preparation kit for R10.4.1 flow cells. Two libraries were sequenced for *Desmarestia dudresnayi* on R9.4.1 flowcells and a third library was sequenced on a R10.4.1 flowcell.

RNA was isolated from mature gametophytes of *Undaria pinnatifida* and *Scytosiphon promiscuus* following modified procedure of Qiagen RNAeasy kit and the TruSeq RNA Library Prep Kit v2 was used to sequence the transcriptomes in an Illumina NextSeq 2000 platform (150bp, PE reads). Extraction of total RNA from fucoidal algae (*F. vesiculosus*, *A. nodosum* and *F. ceranoides*) was performed following 65 and RNA libraries were sequenced on Illumina HiSeq 2000 machine (100 bp, PE reads).

Genome assembly and annotation

High quality, chromosome level assemblies of brown algae genomes have been notoriously difficult to obtain due to technical challenges in extracting nucleic acids. Whole-genome assemblies and annotations of *S. promiscuus* male, *D. dichotoma* male, *D. herbacea* male and female, *E. crouanarium* male, *C. linearis*, *S. ischiensis* and *F. serratus* male were obtained from Denoed et al.¹⁵. We also downloaded the genome of *Ectocarpus* sp. ⁷¹⁷ and the male genome of *Undaria pinnatifida*¹⁸ which were already assembled at a chromosome level. For *Desmarestia dudresnayi*, we performed genome sequencing, *de novo* genome assembly and *ab initio* gene annotation. Base calling was done using ONT Guppy⁸⁰ with the configuration files `dna_r9.4.1_450bps_sup.cfg` and `dna_r10.4.1_e8.2_400bps_sup.cfg` and the options `--trim_adapters --trim_primers`, yielding 17.4 Gbp of data in 2,871,152 reads. We merged all the reads and analyzed them using Kraken v2.1.2⁸¹ and the bacteria database (downloaded 08-2022) to remove potential contaminant sequences. All data classified as bacterial reads by Kraken were screened using blastN v2.13.0+⁸² (-evalue 0.001 -num_alignments 20) against the NCBI genbank bacterial database (downloaded 11-2023). The blastN output was visualized in MEGAN v6.23.4⁸³, and all the reads that were declared as bacterial were extracted and removed from further analyses. We obtained 1,908,772 decontaminated reads with an average length of 5.1Kbp (9.8 Gbp of data, 20x coverage), which were deposited on the NCBI Sequence Read Archive (see **Table S21**).

The decontaminated reads were assembled *de novo* using flye v2.9.1-b1780⁸⁴ with the options `'--nano-raw -g 450m -t 28 -i 3 --scaffold'`. The draft assembly consisted of 1,032 contigs with a total size of 425 Mbp, an N50 of 4.6 Mbp and an L50 of 29 contigs. We used TransposonPSI (<http://transposonpsi.sourceforge.net/>) to predict the TEs and RepeatScout v1.0.6⁸⁵ to predict the simple repeats in the genome assembly. Both predictions were combined to soft-mask the repetitive content in the genome assembly using bedtools maskfasta v2.27.1⁸⁶. We mapped the RNA-seq data of *Desmarestia dudresnayi* from the PhaeoExplorer database¹⁶ to the soft-masked genome assembly using STAR v2.7.1a⁸⁷. We used BRAKER v2.1.6 alongside the RNA-seq data⁸⁸ to predict the protein-coding genes in the soft-masked genome assembly.

Hi-C library preparation and sequencing for chromosome-level assemblies

We generated Hi-C libraries for three male genomes (*Scytosiphon promiscuus*, *Desmarestia herbacea* and *Dictyota dichotoma*) and two female genomes (*Ectocarpus* sp. 7 and *Desmarestia herbacea*). Fresh algal tissue was cross-linked for 20 minutes at room temperature in a solution of 2% formaldehyde with filtered

natural sea water (NSW) and then transferred into a 400 mM Glycine solution with filtered NSW for five minutes to quench the formaldehyde. The samples were then stored at -80°C until use. The Hi-C libraries were prepared as follows. The samples were de-frosted in 1 mL of 1x *DpnII* buffer with protease inhibitors (Roche cOmplete™), transferred to Precellys VK05 lysis tubes (Bertin Technologies, Rockville, MD) and disrupted using the Precellys apparatus with five grinding cycles of 30 seconds at 7,800 rpm followed by 20 second pauses. SDS was added to the lysate at 0.5% final concentration and samples were incubated for 10 minutes at 62°C, followed by the addition of Triton-X100 to a final concentration of 1% and 10 minutes of incubation at 37°C under gentle shaking. We added 500 U of *DpnII* to 4.6 mL of the digestion mixture and incubated the samples for two hours at 37°C under gentle shaking (180 rpm in an inclined rack to prevent sedimentation), followed by the addition of another 500 U of *DpnII* and an overnight incubation under the same conditions. The digested samples were centrifuged at 4°C for 20 minutes at 16,000×g. The supernatant was discarded and the pellet was incubated for biotinylation at 37°C for an hour under a constant shaking (300 rpm) in a 500 ml biotinylation mix with a concentration of 1x ligation buffer, 0.09 mM of dATP-dGTP-dTTP, 0.03 mM of Biotin-14-dCTP and 0.64 U/mL of Klenow fragments. After biotinylation, the samples were incubated for three hours at room temperature in a 1.2 mL ligation reaction with a concentration of 1x ligation buffer, 100 mg/mL of BSA, 1 mM of ATP and 0.4 U/mL of T4 DNA Ligase. The samples were then incubated overnight at 65°C after adding 20μl of 0.5M EDTA, 80μl of 10% SDS and 1.6 mg of Proteinase K. DNA was extracted with 1 volume of phenol/cholor-form/isoamyl (24:24:1) alcohol, followed by 30 seconds of vortex at top speed and a five-minute centrifugation at top speed. We precipitated the DNA by adding 1/10 volume of 3M NaAc pH5 and two volumes of cold EtOH 100%, followed by a 30-minute incubation at -80°C and a 20-minute centrifugation at 14,000×g and 4°C. The DNA pellet was washed with 1mL of EtOH 70%, then dried at 37°C for 10 minutes and resuspended in 100μl 1x TE buffer with 1mg/ml of RNase. DNA was sheared to 250-500bp fragments using Covaris S220, purified with AMPure beads (0.6X) (Beckman) and eluted in 20μl 10mM Tris pH8.0. Biotinylated but not ligated DNA fragments were first removed by T4 DNA polymerase treatment (final concentration=300 U/pellet; NEB), and the biotin-labeled fragments were selectively captured by Dynabeads MyOne Streptavidin C1 (Invitrogen). The libraries were prepared using NEB Ultra II library preparation system and sequenced on the NextSeq2000 Illumina platform (2x150 bp) (Table S21).

We scaffolded the genomes from Denoeud et al.¹⁶ into chromosome-level assemblies using the Hi-C data. We filtered the low-quality Hi-C reads using Trimmomatic v0.39⁸⁹ (ILLUMINACLIP:2:30:10 LEADING:25 TRAILING:25 SLIDINGWINDOW:4:15 MINLEN:75 AVGQUAL:28). We mapped the Hi-C reads against each genome assembly using BWA-mem v0.7.17-r1188⁹⁰ as implemented in the Juicer v1.6 pipeline⁹¹ to generate a contact map, which was then fed to 3D-DNA v190716⁹² to scaffold the genomes into chromosomes. The obtained scaffolds were manually inspected against the contact maps to solve the limits of each chromosome using Juicebox v1.11.08⁹³. The PhaeoExplorer gene annotations¹⁶ were lifted into the new assemblies using LiftOff v1.6.1⁹⁴, while the annotation of TEs was performed using RepeatModeler2⁹⁵. We scaffolded the genomes of *Ectocarpus crouaniorum* and *Desmarestia dudresnayi* into chromosomes using a reference-guided assembly with RagTag v2.0.1⁹⁶ against the chromosome-level assemblies of *Ectocarpus* sp. 7 and *Desmarestia herbacea*, respectively.

Discovery of the UV sex determination regions

Male sex determining regions (V-SDR) in *S. promiscuus*, *U. pinnatifida*, *D. herbacea* and *D. dichotoma*, as well as female sex determining region (U-SDR) in *D. herbacea* were analyzed following a YGS approach developed by Carvalho and Clark⁹⁷ and coverage analysis described previously⁹⁸. The YGS method principle is to identify male or female sex-linked scaffolds by comparing kmer frequencies between reference genome assembly and kmers generated from DNAseq reads of the opposite sex. Regions in the male reference genome with low density coverage of female kmers will indicate candidate male SDR sequences, similarly, female genomic scaffolds with low coverage in male kmers will denote female SDR region. First, fifteen base pair kmer sequences were generated from respective Illumina reads (**Table S21**) using Jellyfish v2.3.0 count (-m 15 -s 10G -C --quality-start=33 --min-quality=20) and converted to fasta format with Jellyfish dump (--lower-count=5)⁹⁹. Next, non-overlapping 100kb sliding windows of the reference chromosome genome assemblies were created using seqkit v2.3.1¹⁰⁰ and used as input for the YGS.pl script together with the fasta kmer files produced in the previous step. Genomic windows with a minimum of 70% of unmatched single copy kmers were then retained as candidate male or female SDR sequences. These regions were further validated by the coverage analysis. In detail, the short Illumina reads coming from males and females of each investigated species were trimmed with Trimmomatic⁸⁹ (see above) and mapped to the reference genome, for which the SDR was to be studied, using HISAT2¹⁰¹ (default settings). Bam files produced by HISAT2 were used as input for Mosdepth¹⁰² to calculate coverage in 100kb windows along the genome sequence (-m -n -b 100000 --fast-mode -Q 30). Read mapping depth in genomic windows was normalized by the genome-wide mean for each sex and the coverage in genomic intervals was then compared between males and females. Because V-SDR-linked sequences are present only in males, we expect them to have similar read coverage as autosomal regions in males, but little or no coverage in females (and conversely for the U-SDR sequences). The comparison focused on regions within male reference genomes where the coverage in males fell within the range of 75% to 125% of the genome average, while the coverage in females remained below 50% of the genome average. These findings were then cross-referenced with the results obtained from the YGS analysis. The reverse strategy was applied to female U-SDR regions for a comprehensive evaluation. Both, coverage and kmer analysis, identified identical genomic regions (**Table S1**). Candidate sex-specific scaffolds were further verified by PCR on at least 4 
[revised manuscript text omitted]
 by PhyML v3.1¹²⁴ with default model and visualized in TreeDyn v198.3¹²⁵. Approximate Likelihood-Ratio test (aLRT) was chosen as statistical test for branch support. We inferred the function of the ancestral V-SDR genes through the annotation of genes in *Ectocarpus* sp. 7 belonging to that orthogroup.

Genomic content across chromosomes

We used closely-related genome assemblies available in the PhaeoExplorer database¹⁶ to assess the depletion of orthologs in the sex chromosome. We predicted one-to-one orthologs using OrthoFinder¹¹⁵ between the following species pairs: *Ectocarpus* sp. 7 with *Ectocarpus siliculosus*, *Scytosiphon promiscuus* with *Chordaria linearis*, *Undaria pinnatifida* with *Saccharina japonica*, *Fucus serratus* with *Fucus distichus*, *Desmarestia herbacea* with *Desmarestia dudresnayi*, and *Dictyota dichotoma* with *Halopteris paniculata* (**Table S22**). We calculated the expected number of detectable orthologs for each chromosome and compared it against the observed number of detected orthologs using chi-squared tests. We performed Benjamini-Hochberg corrections to the p -values of the chi-squared tests to control the false discovery rate (FDR) in the analysis¹²⁶.

GenEra³⁵ was used by running DIAMOND in ultra-sensitive mode¹¹⁶ against the NCBI NR database and all the PhaeoExplorer proteins¹⁶ to perform a phylostratigraphic analysis (e-value threshold of 10^{-5}) and calculate the relative ages of each gene in each genome (**Table S23**). The gene age categories outside of the brown algae and *Schizocladia ischiensis* were based on the taxonomic classification of each species within the NCBI Taxonomy database¹²⁷, while the gene ages within the brown algae were manually assessed to reflect the evolutionary relationships obtained in the PhaeoExplorer maximum likelihood tree¹⁶. We performed Wilcoxon rank-sum tests in R v4.3.1¹²⁸ to assess nonrandom differences in gene age distributions between pairs of chromosomes (**Table S12**). We performed Benjamini-Hochberg corrections to the p -values of the Wilcoxon rank-sum tests to control the FDR in the analysis¹²⁶. The gene ages responsible for these differences were found by evaluating the standardized residuals using mosaic plots (**Figs. S7-S11, S13-S16, S18-S19**). The relative gene ages in **Fig. 3B** and in **Figs. S7-S11, S13-S16 and S18-S19** were plotted against the chromosome-level assemblies using karyoploteR v1.20.3¹²⁹.

We used the K_s values between pairs of species as a proxy for neutral mutation rates across six of the seven chromosome-level assemblies by using the most closely related genome assemblies available in the PhaeoExplorer database¹⁶. We used the same set of one-to-one orthologs detected between species pairs as for the ortholog-depletion test (**Table S22**). However, the evolutionary distance between *Dictyota dichotoma* and *Halopteris paniculata* prevented us from calculating reliable K_s values for this species since synonymous substitutions reached the point of saturation. The amino acid sequences of each pair of orthologs were aligned with MAFFT¹¹⁷ and subsequently aligned into codons using pal2nal¹¹⁸. The K_s values were calculated using the model by Yang & Nielsen¹¹⁹ as implemented in KaKs_calculator v2.0¹²⁰. We also evaluated the difference in K_s values between the autosomes and the sex chromosomes through FDR-corrected Wilcoxon rank sum tests (**Table S14**). We calculated the protein-coding density, the density of TEs and the taxonomic identity of these TEs within 100 kb non-overlapping windows across each chromosome using bedtools⁸⁶ (**Table S24**). The differences in protein-coding space, TE content and TE classification between the autosomes and the sex chromosomes were also performed using FDR-corrected Wilcoxon rank sum tests (**Tables S8-S10**). All the genomic features were plotted using shinyCircos-V2.0¹³⁰.

We tested for *de novo* gene birth events in six monoexonic genes contained within the positions 3886599 to 4923391 in chromosome 13 of the *Ectocarpus* sp. 7 genome assembly¹⁷. We focused on monoexonic genes within the two youngest gene ages (genus and species-level genes) to facilitate the testing procedure of *de*

de novo birth events⁴⁶. We searched for protein-coding homologs of these genes by performing a BLASTp⁸² search with an e-value threshold of 10^{-3} against the annotated proteins of ten additional *Ectocarpus* species¹⁶ (see **Table S21**). We subsequently searched for noncoding regions that were homologous to the candidate genes by performing a tBLASTn⁸² search with an e-value threshold of 10^{-3} against the genome assemblies of the ten *Ectocarpus* species. Once we established the coordinates for the orthologous proteins and noncoding regions in the other *Ectocarpus* species, we extracted the nucleotide sequences from each genome assembly while adding 100 bp upstream and downstream from the BLAST coordinates to encompass possible start and stop codons that might be found in the vicinity of the matched sequences. The nucleotide sequences were initially aligned with MAFFT¹¹⁷. We subsequently used the evolutionary relationships between *Ectocarpus* species inferred by Akita et al¹³¹ to re-align the sequences in a phylogenetically-aware fashion and infer the ancestral state of the sequence at each node in the phylogeny through the maximum likelihood approach implemented in PRANK v170427¹³². We manually assessed the coding potential of the ancestral and extant sequences in the alignment using Aliview v1.27¹³³ to infer the enabling mutations that led to the *de novo* birth of these genes from a non-coding background, and whether the non-annotated orthologous sequences found with tBLASTn represent unannotated sequences with coding potential, pseudogenization events or non-coding DNA sequences⁴⁶.

Gene expression analysis

We used kallisto v.0.44.0¹³⁴ to calculate gene expression levels using 31-base-pair-long k-mers and 1000 bootstraps. Transcript abundances were then summed within genes using the tximport v3.19 package¹³⁵ to obtain the expression level for each gene in TPM. Differential expression analysis was done in DESeq2 v3.19 package¹³⁶ in R v.4.3.1, applying $FC \geq 2$ and $Padj < 0.05$ 
[revised manuscript text omitted]

43. Van Oss, S. B. & Carvunis, A.-R. De novo gene birth. *PLoS Genet* **15**, e1008160 (2019).
44. Tautz, D. & Domazet-Lošo, T. The evolutionary origin of orphan genes. *Nat Rev Genet* **12**, 692–702 (2011).
45. Weisman, C. M., Murray, A. W. & Eddy, S. R. Many, but not all, lineage-specific genes can be explained by homology detection failure. *PLoS Biol* **18**, e3000862 (2020).
46. Vakirlis, N. & McLysaght, A. Computational Prediction of De Novo Emerged Protein-Coding Genes. in *Computational Methods in Protein Evolution* (ed. Sikosek, T.) vol. 1851 63–81 (Humana New York, NY, 2019).
47. Clayton, M. N. Isogamy and a Fuclean type of life history in the Antarctic brown alga *Ascoseira mirabilis* (Ascoseirales, Phaeophyta). *Botanica Marina* **30**, 447–454 (1987).
48. Hatchett, W. J. *et al.* Evolutionary dynamics of sex-biased gene expression in a young <scp>XY</scp> system: insights from the brown alga genus *Fucus*. *New Phytologist* **238**, 422–437 (2023).

49. Coyer, J. A., Peters, A. F., Hoarau, G., Stam, W. T. & Olsen, J. L. Hybridization of the marine seaweeds, *Fucus serratus* and *Fucus evanescens* (Heterokontophyta: Phaeophyceae) in a 100-year-old zone of secondary contact. *Proc R Soc Lond B Biol Sci* **269**, 1829–1834 (2002).
50. Cánovas, F. G., Mota, C. F., Serrão, E. A. & Pearson, G. A. Driving south: a multi-gene phylogeny of the brown algal family Fucaceae reveals relationships and recent drivers of a marine radiation. *BMC Evol Biol* **11**, 371 (2011).
51. Lv, J., Havlak, P. & Putnam, N. H. Constraints on genes shape long-term conservation of macro-synteny in metazoan genomes. *BMC Bioinformatics* **12**, S11 (2011).
52. Lee, S. C., Ni, M., Li, W., Shertz, C. & Heitman, J. The Evolution of Sex: a Perspective from the Fungal Kingdom. *Microbiology and Molecular Biology Reviews* **74**, 298–340 (2010).
53. Dolgin, E. S. & Charlesworth, B. The Effects of Recombination Rate on the Distribution and Abundance of Transposable Elements. *Genetics* **178**, 2169–2177 (2008).
54. Gray, Y. H. M. It takes two transposons to tango: transposable-element-mediated chromosomal rearrangements. *Trends in Genetics* **16**, 461–468 (2000).
55. Charlesworth, D. Evolution of recombination rates between sex chromosomes. *Philosophical Transactions of the Royal Society B: Biological Sciences* **372**, 20160456 (2017).
56. Jordan, C. Y. & Charlesworth, D. The potential for sexually antagonistic polymorphism in different genome regions. *Evolution (N Y)* **66**, 505–516 (2012).
57. Cox, R. M. & Calsbeek, R. Sexually Antagonistic Selection, Sexual Dimorphism, and the Resolution of Intralocus Sexual Conflict. *Am Nat* **173**, 176–187 (2009).
58. Scharmann, M., Rebelo, A. G. & Pannell, J. R. High rates of evolution preceded shifts to sex-biased gene expression in *Leucadendron*, the most sexually dimorphic angiosperms. *Elife* **10**, (2021).
59. Charlesworth, B. Model for evolution of Y chromosomes and dosage compensation. *Proceedings of the National Academy of Sciences* **75**, 5618–5622 (1978).
60. Lenormand, T., Fyon, F., Sun, E. & Roze, D. Sex Chromosome Degeneration by Regulatory Evolution. *Current Biology* **30**, 3001-3006.e5 (2020).
61. Jeffries, D. L., Gerchen, J. F., Scharmann, M. & Pannell, J. R. A neutral model for the loss of recombination on sex chromosomes. *Philosophical Transactions of the Royal Society B: Biological Sciences* **376**, 20200096 (2021).
62. Jay, P., Tezenas, E., Véber, A. & Giraud, T. Sheltering of deleterious mutations explains the stepwise extension of recombination suppression on sex chromosomes and other supergenes. *PLoS Biol* **20**, e3001698 (2022).
63. Coelho, S. M., Gueno, J., Lipinska, A. P., Cock, J. M. & Umen, J. G. UV Chromosomes and Haploid Sexual Systems. *Trends Plant Sci* **23**, 794–807 (2018).

64. Ivics, Z. & Izsvák, Z. The expanding universe of transposon technologies for gene and cell engineering. *Mob DNA* **1**, 25 (2010).
65. Peona, V. *et al.* Teaching transposon classification as a means to crowd source the curation of repeat annotation – a tardigrade perspective. *Mob DNA* **15**, 10 (2024).
66. Gueno, J. *et al.* Chromatin landscape associated with sexual differentiation in a UV sex determination system. *Nucleic Acids Res* **50**, 3307–3322 (2022).
67. Bourdareau, S. *et al.* Histone modifications during the life cycle of the brown alga *Ectocarpus*. *Genome Biol* **22**, 12 (2021).
68. Makova, K. D. & Hardison, R. C. The effects of chromatin organization on variation in mutation rates in the genome. *Nat Rev Genet* **16**, 213–223 (2015).
69. Lebherz, M. K., Fouks, B., Schmidt, J., Bornberg-Bauer, E. & Grandchamp, A. DNA Transposons favour *de novo* transcript emergence through enrichment of transcription factor binding motifs. *Genome Biol Evol* (2024) doi:10.1093/gbe/evae134.
70. McDaniel, S. F. Divergent outcomes of genetic conflict on the UV sex chromosomes of *Marchantia polymorpha* and *Ceratodon purpureus*. *Curr Opin Genet Dev* **83**, 102129 (2023).
71. Iwasaki, M. *et al.* Identification of the sex-determining factor in the liverwort *Marchantia polymorpha* reveals unique evolution of sex chromosomes in a haploid system. *Current Biology* **31**, 5522–5532.e7 (2021).
72. Liesner, D. *et al.* Developmental pathways underlying sexual differentiation in a U/V sex chromosome system. *bioRxiv* 2024.02.09.579736 (2024) doi:10.1101/2024.02.09.579736.
73. Vigneau, J. *et al.* Sex chromosome dominance in a UV sexual system. *bioRxiv* 1–31 (2023).
74. Singh, S., Davies, K. M., Chagné, D. & Bowman, J. L. The fate of sex chromosomes during the evolution of monoicy from dioicy in liverworts. *Current Biology* **33**, 3597–3609.e3 (2023).
75. Takahashi, K. *et al.* Reorganization of the ancestral sex-determining regions during the evolution of trioecy in *Pleodorina starrii*. *Commun Biol* **6**, 590 (2023).
76. Yamamoto, K. *et al.* Three genomes in the algal genus *Volvox* reveal the fate of a haploid sex-determining region after a transition to homothallism. *Proceedings of the National Academy of Sciences* **118**, (2021).
77. Cortez, D. *et al.* Origins and functional evolution of Y chromosomes across mammals. *Nature* **508**, 488–493 (2014).

[revised manuscript text omitted]

130. Wang, Y. *et al.* shinyCircos-V2.0: Leveraging the creation of Circos plot with enhanced usability and advanced features. *iMeta* **2**, (2023).
131. Akita, S. *et al.* Providing a phylogenetic framework for trait-based analyses in brown algae: Phylogenomic tree inferred from 32 nuclear protein-coding sequences. *Mol Phylogenet Evol* **168**, 107408 (2022).
132. Löytynoja, A. & Goldman, N. Phylogeny-Aware Gap Placement Prevents Errors in Sequence Alignment and Evolutionary Analysis. *Science (1979)* **320**, 1632–1635 (2008).
133. Larsson, A. AliView: a fast and lightweight alignment viewer and editor for large datasets. *Bioinformatics* **30**, 3276–3278 (2014).
134. Bray, N. L., Pimentel, H., Melsted, P. & Pachter, L. Near-optimal probabilistic RNA-seq quantification. *Nat Biotechnol* **34**, 525–527 (2016).
135. Sonesson, C., Love, M. I. & Robinson, M. D. Differential analyses for RNA-seq: transcript-level estimates improve gene-level inferences. *F1000Res* **4**, 1521 (2015).
136. Love, M. I., Huber, W. & Anders, S. Moderated estimation of fold change and dispersion for RNA-seq data with DESeq2. *Genome Biol* **15**, 550 (2014).
137. Bushmanova, E., Antipov, D., Lapidus, A. & Prjibelski, A. D. rnaSPAdes: a de novo transcriptome assembler and its application to RNA-Seq data. *Gigascience* **8**, (2019).
138. Smith-Unna, R., Boursnell, C., Patro, R., Hibberd, J. M. & Kelly, S. TransRate: reference-free quality assessment of de novo transcriptome assemblies. *Genome Res* **26**, 1134–1144 (2016).
139. Langmead, B. & Salzberg, S. L. Fast gapped-read alignment with Bowtie 2. *Nat Methods* **9**, 357–359 (2012).
140. Leng, N. *et al.* EBSeq: an empirical Bayes hierarchical model for inference in RNA-seq experiments. *Bioinformatics* **29**, 1035–1043 (2013).

Origin and evolutionary trajectories of brown algal sex chromosomes

Authors: Josué Barrera-Redondo^{1,2#}, Agnieszka P. Lipinska^{1,5#}, Pengfei Liu¹, Erica Dinatale¹, Guillaume
Cossard¹, Kenny Bogaert¹, Masakazu Hoshino^{1,3}, Rory J. Craig¹, Komlan Avia⁴, Goncalo Leiria¹, Elena Avdie-
vich¹, Daniel Liesner¹, Rémy Luthringer¹, Olivier Godfroy⁵, Svenja Heesch⁵, Zofia Nehr⁵, Loraine Brillet-
Guéguen^{5,10}, Akira F. Peters⁶, Galice Hoarau⁸, Gareth Pearson⁷, Jean-Marc Aury⁹, Patrick Wincker⁹, France
Denoed^{9,Φ}, J Mark Cock^{5,Φ}, Fabian B. Haas^{1,Φ}, Susana M Coelho^{1,Φ*}

¹Department of Algal Development and Evolution, Max Planck Institute for Biology Tübingen, 72076 Tübingen, Germany; ²Current
address: ¹Department of Biotechnology and Biochemistry, Center for Research and Advanced Studies (Cinvestav), 36824 Irapuato, Gto.,
Mexico; ³Current address: Research Center for Inland Seas, Kobe University, Rokkodai 1-1, Nada-ku, Kobe 657-8501, Japan; ⁴INRAE,
Université de Strasbourg, UMR SVQV, 68000 Colmar, France; ⁵Sorbonne Université, CNRS, Integrative Biology of Marine Models
Laboratory, Station Biologique de Roscoff, Roscoff, France; ⁶Bezhin Rosko, 29250 Santeg, France; ⁷Universidade do Algarve, UALG
· Centro de Ciências do Mar (CCMAR); ⁸Faculty of Biosciences and Aquaculture, Nord University, 8026 Bodø, Norway; ⁹Génomique
Métabolique, Genoscope, Institut François Jacob, CEA, CNRS, Univ Evry, Université Paris-Saclay, Evry, 91057, France. ¹⁰CNRS,
Sorbonne Université, FR2424, ABiMS-IFB, Station Biologique, Roscoff, France

#Equal contribution

16 ^Φsenior authors

*Lead author: susana.coelho@tuebingen.mpg.de

*Running title: Tracing the complex evolutionary history of sex chromosomes across brown seaweeds*

ABSTRACT

Sex chromosomes fall into three classes: XX/XY, ZW/ZZ and U/V systems. Research on the biology and evo-
lution of sex chromosomes has primarily focused on diploid XX/XY and ZW/ZZ systems. In contrast, the rise,
evolution and demise of U/V systems has remained an evolutionary enigma. Here, we analyse genomes span-
ning the entire brown algal phylogeny to determine the history of their sex-determination. U/V sex chromo-
somes emerged between 450 and 224 million years ago, when a region containing the pivotal male-determinant
*MIN* ceased recombining. Seven ancestral genes within the sex determining region show remarkable conserva-
tion over this vast evolutionary time, although nested inversions **near the boundaries -determining**
**caused expansions** of the sex locus, inde-
pendently in each lineage. We evaluate whether these expansions are associated with increased morphological
complexity and sexual differentiation, and show that taxonomically-restricted genes evolve unexpectedly often
in U and V chromosomes. We also investigate two situations in which UV-linked regions have changed. First,
we demonstrate that evolution of hermaphrodites occurred by ancestral males acquiring U-specific genes. Sec-
ond, the *Fucus* dioecious system involves new sex-determining gene(s), acting upstream of formerly V-specific
genes during development. Both such situations have led to the demise of U and V chromosomes and erosion
of their specific genomic characteristics.

INTRODUCTION

[revised manuscript text omitted]

evolution

*ischiensis* and *D. dichotoma*, around 450-224 Mya²¹ (**Fig. 1A**). The male-determining gene *MIN*²² is the only
V-specific gene consistently present in all V-SDRs of the dioicous species. We note that one dioecious (*F.*
*serratus*) and two monoicous (haploid, co-sexual *C. linearis* and *D. dudresnayi*) species lack U/V sex chromo-
somes but still retain *MIN* on a chromosome homologous to the ancestral U/V ('U/V-homolog' hereafter). The
outgroup *S. ischiensis* has low synteny with the brown algae, and exhibits putative fusion-with-mixing events²³
(**Fig. 1A; Fig. S6**).

We next examined the U/V-SDRs by comparing male and female genome assemblies (see methods).
The SDRs contain a relatively small number of genes overall (between 18 and 52), and
between 229 and 904), but considerable variation in gene content and size across species, the smallest being found
in the Ectocarpales (*Ectocarpus* sp. 7, *E. crouanorium*, *S. promiscuous*; **Fig. 1B, 1C, Fig. S7A, Table 1**). SDR
size differences across species are strongly correlated with the number of genes ($R^2 = 0.97$; **Fig. S7B**) and the
repeat content ($R^2 = 0.99$; **Fig. S7C**), inside these regions. Many genes located in the pseudoautosomal region
(PAR) of Ectocarpales were engulfed by the V-SDR of *U. pinnatifida*, *D. herbacea* and *D. dichotoma*, indicating
that the SDR boundaries have changed across species. The boundary differences coincide with extensive struc-
tural rearrangements, particularly inversions, even among closely related taxa (**Fig. 1B-C, Fig. S8**). Note that
the centromere in the V chromosome of *Ectocarpus* is found within the SDR¹⁹, so we cannot exclude that a
centromere-related suppression of recombination may have preceded the inversion events found on the SDR²⁴.

Together, our results indicate that the brown algal U/V sex chromosomes evolved
Mya, via suppressed recombination in a genomic region that contained *MIN* (henceforth male-determining lo-
cus). The presence of *MIN* in distantly related lineages could push the age of the U/V chromosomes further back
in time, but more evidence would be required to establish that dioicy existed in these organisms.

**Figure 1. Origins of U/V sex chromosomes in brown algae.** (A) Macrosynteny plot comparing genomes of six dioicous (green), two
 monoicous (red), one dioecious (blue) and one outgroup species (yellow). The chromosomes were originally numbered by their physical
 size in the *Ectocarpus* v2 genome²⁵. Note that the dioecious species *F. serratus* has a fully diploid life cycle (without gametophytes⁷).
 Syntenic blocks of the V sex chromosome are highlighted in red, with the emergence of U/V chromosomes shown in the phylogeny.
 Genome sizes are indicated in brackets. (B) Microsynteny plot of V chromosomes in five dioicous species, highlighting the male sex-
 determining regions (blue) and the PARs (green). The PAR genes whose orthologs are found within the SDR of other species are high-
 lighted in purple. (C) Microsynteny plot of U chromosomes in two dioicous species, highlighting the female sex-determining regions
 (peach) and the PARs (green). The PAR genes whose orthologs are found within the SDR of other species are highlighted in purple.
 Note that the genome assemblies for *C. linearis* and *S. ischiensis* are not chromosome level, leading to a high number of contigs.

**The evolution of the SDRs involved boundary expansions and gene gains**

The U/V-SDRs of *Ectocarpus* sp. 7 and *D. herbacea* carry homologous genes (gametolog pairs), indicating
descent from a common ancestral region (Table S2). Both species show ~~similar ratios of gametologs and U- or~~ **more pairs than**
~~V-specific genes (16/14 and 11/7 gametolog/sex-specific ratios in *Ectocarpus* and *D. herbacea*, respectively;~~ **genes**
**Table S2)**. ~~From these, only ten genes share SDR orthologs between both species, while the rest were mostly~~
~~acquired independently in the SDR of each species, with one gene that was retained as a gametolog in *Ectocar-*~~ **pair**
~~*pus* sp. 7 but lost both copies in *D. herbacea* (Table S2). Five gametolog pairs conserved both copies in the two~~
~~species, while another three gametolog pairs lost either the male or the female copy in *D. herbacea* (Table S2).~~
~~Additionally, *MIN* and a U-specific gene are also conserved between species (Table S2). Although the total~~ **these two**
~~number of U/V-SDR genes differs between *Ectocarpus* sp. 7 (18 genes) and *D. herbacea* (30 genes), each~~
~~species shows an equal number of gametologs and sex-specific genes in its U- and V-SDRs (Table S2). This~~
~~intra-species symmetry supports the idea that the U and V chromosomes may have undergone parallel evolu-~~
~~tionary changes within each lineage^{10,26,27}. The V-SDR of *D. herbacea* contains 20 additional genes that belong~~
~~to endogenous viral elements, which are common across brown algal genomes¹⁸.~~

Diploid sex chromosome in animals and plants exhibit evolutionary strata representing different recom-
bination suppression events over time. Strata are identified by analyzing synonymous substitutions (*K_s*) between
male/female gametolog pairs²⁸ but detecting evolutionary strata in U/V systems is difficult because neither of
~~these fully sex-linked regions recombines and gene movements and chromosome rearrangements disrupt col-~~ **whose locations in fully X or Z-linked regions are known**
~~linearity between species^{26,29,30}. Moreover, in absence of a recombining outgroup (which does not exist in brown~~ **of both chromosomes**
~~algae) it is challenging to infer the ancestral gene order. In both *Ectocarpus* sp. 7 and *D. herbacea*, the V- and~~ **cannot be reliably inferred**
~~U-SDR rearrangements are inverted (Fig. 2A-B, Fig. S9) consistent with the idea that inversions may lead to~~ **differ by inversions**
~~suppressed recombination between sex chromosomes. An analysis of gametolog pair divergence revealed satu-~~
~~rated levels of *K_s* values (Fig. 2B-C, Table S3), further limiting the inference of evolutionary strata across~~
~~brown algal SDRs. Nonetheless, the gametolog *K_s* values are broadly consistent between orthologs in *Ectocar-*~~
~~*pus* sp. 7 and *D. herbacea* (Table S3). Furthermore, the relative position of the gametologs with the lowest *K_s*~~ **the**
~~values in the U-SDR of *D. herbacea* when compared to the PAR genes in *Ectocarpus* sp. 7 indicate that inver-~~ **relative**
~~sions in the U chromosome that included all the U-SDR and segments of the PARs can explain the expansion~~
~~of the U/V-SDR boundaries in *D. herbacea* (Fig. S9). The expansion of the SDR boundaries in *D. herbacea* led~~
~~to the engulfment of a region containing four genes in the PAR1 of *Ectocarpus* sp. 7, and a second region with~~
~~13 genes located on the PAR2 (Fig. 2D, Table S4, Fig. S9). Twelve of these engulfed genes into the SDR of~~
~~*D. herbacea* were retained as gametologs. These observations support a scenario where expansions in the SDR~~
~~boundaries of brown algae occur through nested inversions. Two chromatin-related transcription factors in the~~
~~*Ectocarpus* PARs were independently incorporated into the SDRs of four other dioicous species (Tables S4-~~
~~S5).~~

The observation of greater V-SDR gene content in early-diverging lineages (like *D. dichotoma*) ~~and~~ **than**
~~reduced V-SDR gene content in the later-diverging Ectocarpales (Table 1) could reflect either gene loss in the~~
~~V-SDRs of Ectocarpales or independent gene gains in the V-SDRs of each lineage (as predicted by Bull¹⁰, from~~ **distinguish between possibilities**
~~an ancestral state with low V-SDR gene content that is retained in Ectocarpales. To test these hypotheses, we~~
~~reconstructed the ancestral SDR gene content (Table S5), focusing on the V chromosome, as the genomic data~~
~~is of better quality (Table S1), and assuming parallel U/V-SDR evolution^{10,26,27} as seen in *Ectocarpus* sp. 7 and~~ **are**

*D. herbacea* (Fig. 2D, Table S2). This analysis revealed that brown algal V-SDR evolution occurred via line-
age-specific gene gains rather than gene loss in the Ectocarpales (Fig. 2E-F). Gene gains were caused by a
combination of three processes: expansions of the SDR boundaries into the PARs, translocation of autosomal
genes into the SDR and lineage-specific gene birth events within the SDR (Fig. 2E, Table S5). Consistently,
ancestral V-SDR genes were associated with higher gametolog K_s values, while independently-acquired game-
tologs in *D. herbacea* had lower K_s values (Table S2, Fig. 2C).

The seven genes in the ancestral V-SDR (Fig. 2F, S7D) include the male-determinant *MIN*²² and six V
gametologs of genes; also carried on the U chromosome (Fig. 2F, Table S6). As predicted by early models of
U/V-SDR evolution¹⁰, all seven genes are likely related to sex determination processes. Gametolog pairs include
putative transmembrane proteins that may play a role in gamete recognition³¹, STE20 serine/threonine kinase
gametologs likely involved in pheromone pathways³², and a casein kinase, a MEMO-like domain protein and a
GTPase-activating proteins which may act in signal transduction (Table S6). None of these genes show signs
of sequence degeneration in *Ectocarpus* sp. 7, but in *D. herbacea* the casein kinase was lost from the U-SDR
and the putative transmembrane receptor was lost in both sexes. We noticed that these ancestral V-SDR genes
remain in the U/V-homolog of the species that have lost their U/V system (Fig. 2F, Table S6), emphasizing
their importance for pathways in sex determination even in absence of sex chromosomes.

The V-SDR size is associated with the level of sexual dimorphism (Fig. 2E; Fig. S7A), but the small
sample size is insufficient for formal statistical analysis. Species with low sexual dimorphism (anisogamous)
retained the ancestral V-SDR genes with very few gene gains, further suggesting that they may represent the V-SDR
ancestral state. Although the number of SDR changes is small, oogamous species each independently gained
diverse V-SDR genes, and one gene (ATP-dependent RNA helicase) was convergently acquired in all. All the
detected autosomal translocations into the V-SDRs of *Ectocarpus* sp. 7 and *D. herbacea* (Fig. 2E) also happen
to be sex-specific genes (Table S2, Table S5), consistent with a model where sexual antagonism in autosomal
loci may be solved through sex linkage. In contrast to the positive association between SDR size and gamete
dimorphism described above, we found no correlation between autosomal sex-biased gene (SBG) expression
and sexual dimorphism level (FDR corrected $p_{\text{ues}} > 0.01$; Table S7), supporting previous studies³³ (Fig. S10A).
However, we observed an enrichment of male-biased genes on the PARs in all species (Chi-square test p -values
< 0.01) except *D. dichotoma* (Fig. S10B).

Most U/V-SDR genes were prominently expressed in fertile haploid gametophytes, consistent with gene
preservation via haploid purifying selection (Table S8). Gametologs had typically higher expression levels than
sex-limited genes (present in only one of the SDRs) (Wilcoxon test, p -value=0.00075 in *D. herbacea*; p -
value=0.08843 in *Ectocarpus* sp. 7) (Fig. 2D). A comparative analysis in fertile gametophytes between SDR
genes and their autosomal counterparts in other species showed that newly acquired genes on the SDR had
similar expression levels to their autosomal counterparts (Fig. S11), suggesting either a co-option of autosomal
biological activity into male-specific functions in the V-SDR or the general importance of these genes for ga-
metophyte development. Examining expression levels across multiple tissues in *Ectocarpus* sp. 7 revealed that
activity of U/V-SDR genes is not confined to fertile gametophytes (Fig. 2D). Therefore, the SDRs contains both
genes involved in sex determination and gametophyte fertility but also genes playing a broader role in develop-
ment.

Altogether, our analyses illustrate how brown algal U/V-SDRs undergo structural changes, evolving
mainly by lineage-specific gene gains associated with increasing levels of sexual dimorphism. We identified a
set of conservatively sex-linked genes in dioicous brown algae, suggesting their role in sex determination and/or
differentiation, along with genes potentially involved in other developmental pathways.

**Figure 2. Lineage-specific U/V-SDR expansion from an ancestral SDR and its association with sexual dimorphism.** (A) Microsynteny
plot between the U and V chromosomes of *Ectocarpus* sp. 7 and *D. herbacea*. (B) Synteny between the U and V gametologs within the
V-SDRs of both species, colored by synonymous substitutions per site (K_s). (C) Identification of ancestral SDR gametologs (red squares)
and independently acquired gametologs (black dots) with respect to their gametolog K_s values and their position in the V-SDR. (D)
Circos plot linking gametolog pairs in each species with expression levels of all SDR genes ($\log_2(\text{TPM}+1)$) across different life stages
in *Ectocarpus* sp.7 and mature gametophytes (*matGA*) in *D. herbacea*. Gametologs are highlighted in dark colors, sex-limited genes are
highlighted in light colors, insertions are marked in grey, and stars denote conserved SDR genes (also in panel F). (E) The ancestral
state reconstruction of V-SDR gene content across brown algae, showing the expected number of genes in the SDR (white circles), gene
retention (blue), gene gain through expansion of the SDR boundaries (orange), gene gain through autosomal translocation (purple),
gene birth event inside the SDR (yellow) and gene loss (red) along with changes in gamete dimorphism¹⁶. (F) Schematic of the seven
ancestral V-SDR genes, with genomic locations marked: retained in the V-SDR (blue), found in the U/V-homolog of non-dioicous species
(green), translocated from the V-SDR to an autosome (yellow), present in a non-scaffolded contig (red), and lost (grey). Bold: MIN. See
Tables S5 and S6.

Structural features and evolutionary dynamics of brown algal U/V sex chromosomes

We next examined the structural features that differentiate the entire U/V sex chromosomes (V-SDR and PARs)
from the rest of the genome (**Fig. 3A; Fig. S12A**). As expected for non-recombining regions³⁴, all V sex chro-
mosomes are repeat rich and gene poor (Wilcoxon rank-sum test, FDR corrected p -values < 0.01 ; **Fig. 3A; Fig.**
**S12A; Tables S9-S11**). V-SDRs have significantly higher repeat density than the PARs or the autosomes (per-
mutation test, FDR corrected p -values < 0.001 ; **Fig. S13**). This low gene density is not influenced by the pres-
ence of centromeres within the SDRs, as the coding density in the *Ectocarpus* sp. 7 V centromere (3.51%) is
slightly higher than in the rest of the V-SDR (2.85%), presumably due to the small size of the centromere (153
kbp)¹⁹. The PARs were also significantly enriched in repeats when compared to the autosomes, although less so
than the V-SDRs (permutation test, FDR corrected p -values < 0.001 ; **Fig. S13**). Among repetitive elements,
‘unclassified’ transposable elements (TEs) were enriched in the PARs and SDRs of the Ectocarpales (permuta-
tion test, FDR corrected p -values < 0.01), whilst the V-SDRs of species that underwent genome expansion (e.g.
*U. pinnatifida*, *D. herbacea*, *D. dichotoma*) predominantly accumulated LTR elements (**Fig. S14; Table S9**).

Moreover, sex chromosomes had fewer conserved orthologs between species compared to the auto-
somes (Chi-squared test, p -value $< 10^{-4}$, **Table S12**), and phylostratigraphy analyses^{35,36} revealed an enrichment
of taxonomically-restricted genes (TRGs) in the sex chromosomes of all dioicous species (Wilcoxon rank-sum
test, FDR corrected p -values < 0.01 ; **Fig. 3B; Tables S13-S14**). TRG enrichment was localized in the PARs of
the Ectocarpales, but this pattern extended to the entire sex chromosome, including the SDRs, in species with
larger V-SDRs (permutation test, FDR corrected p -values < 0.001 ; **Fig. 3B**). Importantly, sex chromosomes
have statistically younger genes than the last common ancestor of the five dioicous species (same Order or
broader taxonomic groups), indicating that TRG enrichment arose independently in each species (Pearson stand-
arized residuals > 2.4 ; **Figs. S15-S19**).

We previously proposed a theoretical model where generation-antagonistic selection may favor the re-
tention of young sporophyte-beneficial loci in the PARs of the *Ectocarpus* sp. 7 U/Vs³⁷. Consistent with this
model, sporophyte-biased genes are indeed enriched in the sex chromosomes of *Ectocarpus* sp. 7 and *U. pin-*
*natifida* (fold change > 2 , adjusted p -values < 0.05), but less so in *D. dichotoma* and *S. promiscuus* (**Table S15**).
Moreover, we explored additional mechanisms underlying TRG emergence by estimating inter-species K_s val-
ues between orthologs in closely-related species (**Table S16**) and comparing these values between chromo-
somes and genomic compartments (V-SDR, PAR, autosomes). If synonymous mutations behave neutrally^{38,39},

then inter-species K_s can be used as a proxy for mutation rates^{40,41}. Consistently, we found higher inter-species
K_s values in the V sex chromosomes compared to autosomes across all dioicous species (Wilcoxon rank-sum
test, FDR corrected p -values < 0.01; **Fig. 3A; Tables S16-S17**), suggesting higher mutation rates relative to
autosomes. Higher inter-species K_s values are also localized in the PARs, mirroring the pattern observed with
the TRGs (**Fig. S15-S18**). Therefore, the enrichment of TRGs in the U and V is associated with both enrichment
of sporophyte-biased genes and higher synonymous substitution rates. We propose the term ‘gene cradle’ to
describe the overrepresentation of TRGs across U/V chromosomes compared to autosomes.

To test the generality of the ‘gene cradle’ pattern, we applied the same approach in other organisms
with haploid sex determination, the plants *Ceratodon purpureum*, *Sphagnum angustifolium*, *Marchantia poly-*
*morpha*^{11,12,42}, and the fungus *Cryptococcus neoformans*⁴³. We observed a clear enrichment of TRGs in the V
chromosomes of *C. purpureum* and *S. angustifolium* (Wilcoxon rank-sum test, FDR corrected p -values < 0.01;
**Figures S20-S21, Tables S13-S14**), but not in the highly degenerated U/V chromosomes of *M. polymorpha* nor
the mating-type chromosome of *C. neoformans* (**Figures S22-S23, Tables S13-S14**).

~~Together, our analyses show that brown algal U/V chromosomes are TE rich, have decreased gene~~
~~density, higher mutation rates, and are a hotspot for the emergence of TRGs. The gene cradle pattern appears to~~
~~be a widespread feature of mildly degenerated U/V, but not mating type, chromosomes.~~

Figure 3. The U/V sex chromosomes act as cradles for taxonomically-restricted genes. (A) Karyoplots for five dioicous species showing the following features from bottom to top: chromosome compartments (autosomes, PARs and SDR); relative gene ages, inter-species Ks values, and proportion of coding (CDS, red) and repeat (TEs, blue) density. Statistically significant differences for each feature between each autosome and the V chromosome are depicted on top of the track for that autosome (FDR-corrected Wilcoxon rank-sum test; NS = non-significant, * $p < 0.05$, ** $p < 0.01$, *** $p < 0.001$). (B) Violin plots for five dioicous species showing the relative gene age ranks (higher ranks equate to younger ages) of the TRGs across chromosome compartments (autosomes, PARs and SDR). Statistically significant differences in mean values of gene ages were assessed using FDR-corrected permutation tests (NS = non-significant, ** $p < 0.01$, *** $p < 0.001$).

Fate of U/V sex chromosomes following loss of dioicy

We studied the evolutionary trajectory of brown algal genomes after the loss of the U/V system, by exploring
two independent transitions to monoicy in *C. linearis* and *D. dudresnayi* that undergo sexual reproduction and
develop male and female gametangia³³. Most genes in the ‘ex’-sex chromosomes (U/V-homologs) of both mo-
noicous species are male-derived, indicating that monoicy emerged from a male background (Fig. 4A-B). The

U/V-homolog of *C. linearis* contains several rearrangements spanning the regions that are homologous to the
PAR and SDR (SDR-homolog), with 11 V-SDR-derived and two U-SDR-derived orthologs located within the
SDR-homolog, with an additional V-SDR-derived gene that was translocated elsewhere in the U/V-homolog
(**Fig. 4A; Table S18**). Likewise, *D. dudresnayi* underwent at least two inversion events within the SDR-homolog
after splitting from *D. herbacea* (**Fig. 4B**), containing 20 V-SDR-derived genes and four U-SDR-derived
genes (**Table S19**).

Both monoicous species retained mostly male and a few female copies for most of the U/V-SDR-derived
gametologs (91% in *C. linearis* and 100% in *D. dudresnayi*), whereas several U and V-specific orthologs
were lost in these species (10 and 17 sex-limited genes in *C. linearis* and *D. dudresnayi*, respectively). From
these lost orthologs, 60% and 43% of them present closely-related autosomal paralogs in *C. linearis* and *D.*
*dudresnayi*, respectively (**Tables S18-S19**), though it is unclear if the expression of these autosomal paralogs is
compensating the activity of the lost genes. The only three U-SDR-derived orthologs in *C. linearis* are flanked
by PAR orthologs translocated at the end of the V-SDR-derived region, suggesting that the U/V-homolog (con-
tigu 12) of *C. linearis* acquired its U-SDR-derived genes through two translocations (**Fig. S24A**). Three U-SDR-
derived genes in *D. dudresnayi* are dispersed across the V-SDR-derived region of the U/V-homolog, suggesting
independent U-SDR translocations into the V-SDR, while the fourth U-SDR gene was translocated to an auto-
some (**Fig. S24B**).

The seven ancestral V-SDR genes are transcriptionally active ($\log_2(\text{TPM}+1) > 2$) during reproductive
stages of both monoicous species (**Table S20**), emphasizing their role in reproduction despite the absence of a
U/V system, particularly *MIN*²² which is retained in both species. While most U-SDR-derived genes are absent
in monoicous species, a single intracellular cholesterol transporter gene was convergently preserved in both
monoicous genomes (**Table S20**) and actively expressed during fertility in both *Ectocarpus* sp. 7 and *D. her-*
*bacea* (**Table S8**).

The U/V-homolog of *D. dudresnayi* retains some vestiges of its past as a U/V chromosome, such as low
coding density, high repeat density and enrichment of TRGs (Wilcoxon rank-sum test, FDR corrected *p*-values
< 0.01), although we found non-significant differences in inter-species *Ks* values across the genome (**Fig. 4C;**
**Fig. S12B, S25; Tables S9-S14, S16-S17**).

Figure 4. Fate of sex chromosomes during transitions from dioicy to co-sexuality (monoicy). (A) Comparison of the U/V-homolog in *C. linearis* against the U and V chromosomes of *Ectocarpus* sp. 7. (B) Comparison of the U/V-homolog in *D. dudresnayi* against the U

and V chromosomes of *D. herbacea*. The color code represents the identity of the genes alongside the chromosomes, while the shapes
represent the evolutionary fate of each SDR gene in the monoicous genome. The matching shades between the SDRs and the sex homolog
are either color-coded by their ancestral background or they appear as transparent dotted shades if the gametolog of the other sex was
retained. (C) Karyoplot of *D. dudresnayi* showing the following features from bottom to top: chromosome compartment (autosomes and
U/V-homolog), relative gene ages, inter-species *Ks* values, proportion of coding (CDS, red) and repeat (TEs, blue) density. Statistically
significant differences for each feature between each autosome and the U/V-homolog are depicted on top of the track for that autosome
(FDR-corrected Wilcoxon rank-sum test; NS = non-significant, * $p < 0.05$, ** $p < 0.01$, *** $p < 0.001$). (D) Violin plot showing the relative
gene age ranks (higher ranks equate to younger ages) of the TRGs between the autosomes and the U/V-homolog of *D. dudresnayi*.
Statistically significant difference in mean values of gene ages was assessed using an FDR-corrected permutation test (*** = $p < 0.001$).

Finally, we examined the transition from haploid to diploid sex determination, which has remained
~~undescribed~~ ^{unstudied} in Eukaryotes. Although the ancestral state for the brown algae is a U/V sexual system, the Fucales
recently transitioned to a diploid life cycle⁴⁴, with many species, such as *Fucus serratus*⁴⁵ exhibiting diploid
separate sexes (dioecy)⁴⁶. Dioecy likely evolved from monoecy in the last common ancestor of the *Fucus* ge-
nus⁴⁷ (25 to 5 Mya¹⁸) consistent with a young sex chromosome in *F. serratus*. Our extensive bioinformatic
analysis and PCR sex-linkage testing for candidate genes such as *MIN* (see methods) failed to identify sex-
linked sequences in *F. serratus* (Fig. S26), likely due to a small and undifferentiated ~~SDR~~ ^{suggesting that the SDR is}. However, male *F.*
*serratus* conserves the *MIN* gene and all the ancestral V-SDR genes in its U/V-homolog (see Fig. 2F; Fig. 5A).
Importantly, although none of the U/V-SDR-derived genes are sex-linked in *F. serratus*, *MIN* and four other
ancestral V-SDR genes are exclusively expressed in males (fully silenced in females) (FC > 2, adjusted p -values
< 0.05; Fig. 5A, Table S21). This pattern is consistent ~~in~~ ⁱⁿ three other Fucales species (Table S22). There-
fore, ~~whilst~~ ^{probably} the ancestral V-SDR genes are no longer sex-linked in the Fucales, they still play ~~a~~ ^a role in male-sex
determination or differentiation pathways.

Contrary to the observations in *D. dudresnayi*, the U/V-homolog of *F. serratus* lacks the gene cradle
pattern and all the other distinctive features of the U/V chromosomes (Wilcoxon rank-sum test, FDR corrected
p -values > 0.01). Thus, this 'ex'-sex chromosome has lost all the evolutionary vestiges of its past as a U/V
chromosome (Fig. 5B; Fig. S12B, S27; Tables S9-S14, S16-S17).

**Figure 5. Transition from haploid to diploid sex determination.** (A) Expression of ancestral U/V-SDR genes in the diplontic species *F.*
 *serratus*. Gene expression of mature algae (using 3 males and 3 females, see methods) is given as log₂(TPM+1) and bars represent
 standard deviation of the mean. Bold text represents whether the gene in *F. serratus* corresponds to an ancestral male or the female
 gametolog. (B) Karyoplot of *F. serratus* showing the following features from bottom to top: chromosome compartment (autosomes and
 U/V-homolog), relative gene ages, inter-species Ks values (between 0.00079 and 6.838, with an average value of 0.148), proportion of
 coding (CDS, red) and repeat (TEs, blue) density. Statistically significant differences for each feature between each autosome and the
 U/V-homolog are depicted on top of the track for that autosome (FDR-corrected Wilcoxon rank-sum test; NS = non-significant, **p*<0.05,
 ***p*<0.01, ****p*<0.001). (C) Violin plot showing the relative gene age ranks (higher ranks equate to younger ages) of the TRGs between
 the autosomes and the U/V-homolog of *F. serratus*. The mean values of gene ages are not significantly different (FDR-corrected permu-
 tation test; NS = non-significant).

DISCUSSION

Here, we characterized the evolutionary trajectory of brown algal sex chromosomes (**Fig. 6**). Brown algal sex
 chromosomes date back 450 to 244 Mya²¹, at the origin of brown algae. We propose that the male-determining
 gene *MIN* underlies the birth of this U/V system²². The ancestral cassette with seven V-SDR genes suggests a
 very early ~~sequen~~ ^{evolu} sequestration of these genes into a non-recombining locus during the evolution of the U/V-SDRs.

16

The ancestral V-SDR genes likely contribute to reproduction, but may also be involved in broader developmen-
tal functions. Whether their sequestration was essential for maintaining the U/V system or incidental during the
initial recombination-suppression event remains to be determined. Despite their old age, brown algal U/V chro-
mosomes retain large PARs bordering the SDR, unlike haploid systems in non-vascular plants that mostly lack
detectable PARs¹¹⁻¹⁴.

[revised manuscript text omitted]

**ACKNOWLEDGEMENTS**

This work was supported by the MPG, the CNRS, Sorbonne University, the ERC (grant n. 864038 and 638240
to SMC), the France Génomique National infrastructure project Phaeoexplorer (ANR-10-INBS-09), the JSPS
Overseas Research Fellowships (to MH), the BMBF-funded de.NBI Cloud within the German Network for
Bioinformatics Infrastructure (de.NBI) (031A532B, 031A533A, 031A533B, 031A534A, 031A535A,
031A537A, 031A537B, 031A537C, 031A537D, 031A538A), the Investissements d'Avenir project Idealg
(ANR-10-BTBR-04-01), the European BG-01 BlueGrowth H2020 project Genialg (727892) and the ANR pro-
ject Epicycle (ANR-19-CE20-0028-01). SMC is supported by the Moore Foundation (GBMF11489) and the
Bettencourt-Schuller Foundation. JBR was supported by a Humboldt Research Fellowship for postdoctoral re-
searchers from the Alexander von Humboldt Foundation. We thank the members of the Phaeoexplorer consor-
tium, in particular Chloe Jolivet, Leticia Mest and Delphine Scornet for assistance with algae cultures, Corinne
Cruaud for help with sequencing libraries preparation, Erwan Corre and Arthur Le Bars for support with the
Phaeoexplorer database and Arnaud Couloux for the genome assemblies and annotations. We are grateful to the
Roscoff Bioinformatics platform ABiMS (<http://abims.sb-roscoff.fr>), part of the Institut Français de Bioin-
formatique (ANR-11-INBS-0013) and BioGenouest network, for providing computing and storage resources.

**AUTHOR CONTRIBUTIONS**

JBR, APL: Investigation (equal); Formal analysis (equal); Methodology (equal); Visualization (equal), Writing
– original draft (equal); Writing – review and editing (equal).

PL: Investigation (supporting); Formal analysis (supporting)

ED, GC, OG, KB, MH, KA, GL, EA, DL, RL, OG, SH, ZN, LG, AFP: Investigation (supporting)

GH, JMA, GP, PW, FD, JMC: Data curation (supporting); Data acquisition (supporting)

FBH: Investigation (supporting); Methodology (equal); Data curation (equal); Formal analysis (supporting)

SMC: Conceptualization (lead); Funding acquisition (lead); Methodology (equal); Project administration (lead);
Supervision (lead); Visualization (supporting); Writing – original draft (equal); Writing – review and editing
(lead).

**DECLARATION OF INTEREST**

The authors declare no competing interests

**SUPPLEMENTAL INFORMATION**

Table S1. Genomic data and PCR sex markers used in this study, with metrics and accession numbers.

Table S2. Gametologs and sex-specific genes for the male and female SDR in *Ectocarpus* sp. 7 and *Desmarestia*
*herbacea*.

Table S3. *Ks* values of the V-SDR gametologs of *Ectocarpus* sp. 7 and *Desmarestia herbacea*, and their inferred
acquisition into the SDR based on ancestral state reconstruction.

TableS4. SDR expansion into the PAR in *Desmarestia herbacea* and the fate of the recently-acquired sex-linked
genes.

Table S5. Orthogroup table for every V-SDR gene in the 5 dioicous species.

Table S6. Ortholog table of the 7 ancestral male SDR genes in 10 brown algal species.

Table S7. Sex-biased gene expression between mature male and female gametophytes per species using
DESeq2.

Table S8. Expression ($\log_2(\text{TPM}+1)$) of male and female SDR genes during fertile gametophyte stage.

Table S9. Protein-coding (CDS) and repeat content for seven brown algal species over 100 kb sliding windows.

Table S10. FDR-corrected p-values for the pairwise Wilcoxon rank-sum tests used to evaluate differences in
the distribution of protein-coding gene content in 100kb windows across the chromosomes of seven brown algal
species.

Table S11. FDR-corrected p-values for the pairwise Wilcoxon rank-sum tests used to evaluate differences in
the distribution of repetitive genomic elements in 100kb windows across the chromosomes of seven brown algal
species.

Table S12. Chi-square test between the observed and expected value of orthologs for each chromosome in seven
brown algal species.

Table S13. Gene age assignments based on phylostratigraphy as implemented in GenEra
(<https://github.com/josuebarrera/GenEra>) for seven brown algal species, three plant species and one fungal spe-
cies.

Table S14. FDR-corrected p-values for the pairwise Wilcoxon rank-sum tests used to evaluate differences in
the distribution of gene age categories across the chromosomes of seven brown algal species, three plant species
and one fungal species.

Table S15. Gene expression measured as $\log_2(\text{TPM}+1)$ in gametophytes and sporophytes of different algal
species.

Table S16. Pairs of orthologous genes used to calculate the expected against observed number of orthologs per
chromosome and to calculate inter-species rates of synonymous substitutions per site (K_s) for seven brown algal
species.

Table S17. FDR-corrected p-values for the pairwise Wilcoxon rank-sum tests used to evaluate differences in
the distribution of inter-species synonymous substitutions per site (K_s) across the chromosomes of six brown
algal species.

Table S18. Evolutionary “fate” of the *Ectocarpus* sp. 7 sex-determining genes in *Chordaria linearis*.

Table S19. Evolutionary “fate” of the *Desmarestia herbacea* sex-determining genes in *Desmarestia dudresnayi*.

Table S20. Gene expression of the seven ancestral V-SDR genes in the reproductive stages of *Chordaria linearis*
and *Desmarestia dudresnayi*.

Table S21. Expression of conserved VSDR genes (*Ectocarpus* sp. 7 SDR genes as reference) in *Fucus serratus*
male and female mature receptacles.

Table S22. Transcriptional activity for V-SDR homologs in three Fucales species.

**SUPPLEMENTAL FIGURES**

*Figure S1. Detection of the sex-determining region (SDR) in the V chromosome of Scytosiphon promiscuus. Results from the k-mer–*
*based YGS analysis (top panels) and genomic read coverage normalized by the genome-wide mean (bottom panels) are shown for each*
*chromosome of the S. promiscuus male genome. Chromosome 13 (chr_13, highlighted with a black header), is homologous to the V*
*chromosome of Ectocarpus sp. 7 and was identified as the sex chromosome. The SDR region, delimited by vertical dashed lines (positions*
*4,795,133 to 6,159,060), displays a high proportion of unmatched k-mers in the YGS analysis and a clear difference in sequencing read*
*coverage between males (blue) and females (red). The horizontal red line in the YGS plots indicates the threshold of 70% unmatched k-*
*mers used to define putative sex-linked regions.*

Figure S2. Detection of the sex-determining region (SDR) in the V chromosome of *Undaria pinnatifida*. Results from the *k*-mer-based YGS analysis (top panels) and genomic read coverage normalized by the genome-wide mean (bottom panels) are shown for each chromosome of the *U. pinnatifida* male genome. Chromosome 23 (chr_23, highlighted with a black header), is homologous to the V chromosome of *Ectocarpus* sp. 7 and was identified as the sex chromosome. The SDR region, delimited by vertical dashed lines (positions 13,867,553 to 27,276,646), displays a high proportion of unmatched *k*-mers in the YGS analysis and a clear difference in sequencing read coverage between males (blue) and females (red). The horizontal red line in the YGS plots indicates the threshold of 50% unmatched *k*-mers used to define putative sex-linked regions.

Figure S3. Detection of the sex-determining region (SDR) in the V chromosome of *Desmarestia herbacea*. Results from the *k*-mer-based YGS analysis (top panels) and genomic read coverage normalized by the genome-wide mean (bottom panels) are shown for each chromosome of the *D. herbacea* male genome. Chromosome 3 (chr_03, highlighted with a black header), is homologous to the V chromosome of *Ectocarpus* sp. 7 and was identified as the sex chromosome. The SDR region, delimited by vertical dashed lines (positions 5,427,118 to 11,995,121), displays a high proportion of unmatched *k*-mers in the YGS analysis and a clear difference in sequencing read coverage between males (blue) and females (red). The horizontal red line in the YGS plots indicates the threshold of 70% unmatched *k*-mers used to define putative sex-linked regions.

Figure S4. Detection of the sex-determining region (SDR) in the U chromosome of *Desmarestia herbacea*. Results from the *k*-mer-based YGS analysis (top panels) and genomic read coverage normalized by the genome-wide mean (bottom panels) are shown for each chromosome of the *D. herbacea* female genome. Chromosome 3 (*chr_03*, highlighted with a black header), is homologous to the V chromosome of *D. herbacea* and was identified as the sex chromosome. The SDR region, delimited by vertical dashed lines (positions 5,479,174 to 14,105,652), displays a high proportion of unmatched *k*-mers in the YGS analysis and a clear difference in sequencing read coverage between males (blue) and females (red). The horizontal red line in the YGS plots indicates the threshold of 50% unmatched *k*-mers used to define putative sex-linked regions.

*Figure S5. Detection of the sex-determining region (SDR) in the V chromosome of Dictyota dichotoma. Results from the k-mer-based*
 *YGS analysis (top panels) and genomic read coverage normalized by the genome-wide mean (bottom panels) are shown for each chro-*
 *mosome of the D. dichotoma male genome. Chromosome 2 (chr_02, highlighted with a black header), is homologous to the V chromo-*
 *some of Ectocarpus sp. 7 and was identified as the sex chromosome. The SDR region, delimited by vertical dashed lines (positions*
 *14,808,243 to 32,220,768), displays a high proportion of unmatched k-mers in the YGS analysis and a clear difference in sequenc-*
 *ing read coverage between males (blue) and females (red). The horizontal red line in the YGS plots indicates the threshold of 70% un-*
 *matched k-mers used to define putative sex-linked regions.*

Figure S6. Macro-synteny plot between *S. ischiensis* and *D. dichotoma* using 1,828 orthologs. We highlight two fusion-with-mixing events (red squares) between chromosomes 4 and 9, and between chromosomes 23 and 24 in *D. dichotoma*.

Figure S7. SDR size differences and detection of independently-acquired V-SDR genes across species. (A) Differences in the size of the male SDR between brown algal species based on the total sequence length, the relative size of the SDR compared to the length of the V chromosome and the number of protein-coding genes retained within the SDR. The bars are colored according to the level of gamete dimorphism in each species (based on¹⁶). (B) Correlation between the V (blue) and U (pink) SDR sizes and the SDR gene content across species. (C) Correlation between the V (blue) and U (pink) SDR sizes and the SDR repeat content across species. (D) Gene trees showing the independent acquisition of SDR gametologs across species that were previously interpreted as part of the ancestral male SDR genes.

*Figure S8. Male SDR synteny between Ectocarpus sp. 7 and Ectocarpus crouaniorum. One of the species underwent a recent inversion*
 *event within the SDR. The arrows in the boxes represent the orientation of each gene within the chromosome.*

Figure S9. Synteny analysis plot illustrating the expansion of the *Desmarestia herbacea* U- and V-sex-determining regions (SDRs) into the surrounding pseudoautosomal region (PAR). The *Ectocarpus* sp. 7 sex chromosome is shown in the middle as a reference, with the SDR regions outlined by grey boxes. Green lines trace syntenic relationships between *Ectocarpus* PAR genes and the recently-acquired *Desmarestia* SDR genes, with each gene pair represented in a distinct shade of green. This demonstrates that nearly all PAR genes from *Ectocarpus*, which have entered the expanded SDR in *Desmarestia*, are retained as gametologs. Orange lines highlight the PAR boundary genes in *Desmarestia*, which remain within the PAR of *Ectocarpus*.

*Figure S10. Sex-biased gene expression per dioicous species. (A) Proportion of sex biased genes in each of the five dioicous species.*
 *MBG: male-biased genes; FBG: female biased genes. (B) Number of sex-biased genes in the pseudoautosomal regions of sex chromo-*
 *somes (U-V-SDRs excluded), male-biased genes are shown in blue and female-biased genes in red. Stars above the bars mark significant*
 *enrichment of the sex-biased genes on the PAR (Chi-square test, ** $p < 0.01$, *** $p < 0.001$).*

*Figure S11. Expression of genes ($\log_2(\text{TPM}+1)$) that entered the SDR independently in different species. Expression is measured in*
 *mature male and female gametophytes, hashing marks missing orthologs, stars inside the cells indicate that the gene is inside the male*
 *non-recombining region (V-SDR). Orthogroups containing orthologs in less than three species or with multicopy genes were excluded*
 *from this analysis. M: male; F: female.*

Figure S12. Structural features across the sex chromosomes and U/V-homologs of brown algae. (A) V and U chromosomes of *Ectocarpus* sp. 7, *Scytosiphon promiscuus*, *Undaria pinnatifida*, *Desmarestia herbacea* and *Dictyota dichotoma*. (B) U/V-homologs of *Desmarestia dudresnayi* and *Fucus serratus*. Features displayed from bottom to top: chromosome compartments (PARs, SDR, U/V-homolog); relative gene ages, inter-species Ks values, and proportion of coding (CDS, red) and repeat (TEs, blue) density.

Figure S13. Accumulation of repetitive elements in the V-SDRs and PARs of five dioicous species. Statistically significant differences in median values of repeat density were assessed using FDR-corrected permutation tests (* $p < 0.05$, ** $p < 0.01$, *** $p < 0.001$).

Figure S14. Enrichment in unclassified repeats in the V-SDRs and PARs of *Ectocarpus sp. 7* and *Scytosiphon promiscuus*. Statistically significant differences in mean values of repeat density were assessed using permutation tests (* $p < 0.05$, ** $p < 0.01$, *** $p < 0.001$).

*Figure S15. Gene ages across the Ectocarpus sp.7 genome. (A) Distribution of relative gene ages across the chromosomes of Ectocarpus*
 *sp. 7. The SDR of the V sex chromosome (chr 13) is highlighted with a red box. (B) The sex chromosome (red) has a significantly higher*
 *proportion of young genes and a lower proportion of old genes when compared to the autosomes (green; see Table S14). (C) Mosaic*
 *plot showing that the species-level (rank 15) and the genus-level (rank 14) genes are responsible for the enrichment of young genes in*
 *the sex chromosome. (D) Inter-species Ks values across genomic compartments (autosomes, PARs and SDR) obtained from one-to-one*
 *orthologs between Ectocarpus sp. 7 and Ectocarpus siliculosus. The inter-species Ks values are significantly higher in the PARs of the*
 *sex chromosome when compared to the autosomes or the SDR (see Table S17).*

*Figure S16. Gene ages across the S. promiscuus genome. A) Distribution of relative gene ages across the chromosomes of Scytosiphon*
 *promiscuus. The SDR of the V sex chromosome (chr 13) is highlighted with a red box. (B) The sex chromosome (red) has a significantly*
 *higher proportion of young genes and a lower proportion of old genes when compared to most of the autosomes (green; see Table S14).*
 *(C) Mosaic plot showing that the species-level (rank 14) genes are responsible for the enrichment of young genes in the sex chromosome.*
 *(D) Inter-species Ks values across genomic compartments (autosomes, PARs and SDR) obtained from one-to-one orthologs between S.*
 *promiscuus and Chordaria linearis. The inter-species Ks values are significantly higher in the PARs of the sex chromosome when*
 *compared to the autosomes or the SDR, although most values already reached saturation (see Table S17).*

*Figure S17. Gene ages across the U. pinnatifida genome. (A) Distribution of relative gene ages across the chromosomes of Undaria*
 *pinnatifida. The SDR of the V sex chromosome (chr 23) is highlighted with a red box. (B) The sex chromosome (red) has a significantly*
 *higher proportion of young genes and a lower proportion of old genes when compared to most of the autosomes (green; see Table S14).*
 *(C) Mosaic plot showing that the ALE-clade genes (rank 11) are responsible for the enrichment of young genes in the sex chromosome.*
 *(D) Inter-species Ks values across genomic compartments (autosomes, PARs and SDR) obtained from one-to-one orthologs between*
 *U. pinnatifida and Saccharina japonica. The inter-species Ks values are significantly higher in the sex chromosome when compared to*
 *the autosomes (see Table S17), showing similar values in the PARs and in the SDR.*

Figure S18. Gene ages across the *D. herbacea* genome. (A) Distribution of relative gene ages across the chromosomes of *Desmarestia*
 *herbacea*. The SDR of the V sex chromosome (chr 03) is highlighted with a red box. (B) The sex chromosome (red) has a significantly
 higher proportion of young genes and a lower proportion of old genes when compared to most of the autosomes (green; see Table S14).
 (C) Mosaic plot showing that the genus-level (rank 11) genes are responsible for the enrichment of young genes in the sex chromosome.
 (D) Inter-species Ks values across genomic compartments (autosomes, PARs and SDR) obtained from one-to-one orthologs between *D.*
 *herbacea* and *D. dudresnayi*. The inter-species Ks values are significantly higher in the sex chromosome when compared to half of the
 autosomes (see Table S17). Non-significance of inter-species Ks values across chromosomes may be a consequence of the conflation
 with the Ks values in *Desmarestia dudresnayi*. The SDR displays higher inter-species Ks values compared to the PARs or the autosomes.

*Figure S19. Gene ages across the D. dichotoma genome. (A) Distribution of relative gene ages across the chromosomes of Dictyota*
 *dichotoma. The SDR of the V sex chromosome (chr 02) is highlighted with a red box. (B) The sex chromosome (red) has a significantly*
 *higher proportion of young genes and a lower proportion of old genes when compared to most of the autosomes (green; see Table S14).*
 *(C) Mosaic plot showing that the species-level (rank 11) and the DFI-clade-level (rank 9) genes are responsible for the enrichment of*
 *young genes in the sex chromosome. Inter-species Ks values were not analyzed for D. dichotoma, due to a saturation of synonymous*
 *mutations with the closest species in the PhaeoExplorer database (Halopteris paniculata).*

Figure S20. Gene ages across the *C. purpureum* genome. (A) Distribution of relative gene ages across the chromosomes of *C. purpureum*. (B) The V sex chromosome (V; red) has a significantly higher proportion of young genes and a lower proportion of old genes when compared to the autosomes (green; see Table S14). (C) Violin plot showing the relative gene age ranks (higher ranks equate to younger ages) of the TRGs between the autosomes and the V chromosome. Statistically significant difference in mean values of gene ages was assessed using an FDR-corrected permutation test (***) = $p < 0.001$). (D) Mosaic plot showing that the species-level genes (rank 9) are responsible for the enrichment of young genes in the sex chromosome.

Figure S21. Gene ages across the *Sphagnum angustifolium* genome. (A) Distribution of relative gene ages across the chromosomes of *Sphagnum angustifolium*. (B) The V sex chromosome (LG20; red) has a significantly higher proportion of young genes and a lower proportion of old genes when compared to the autosomes (green; see Table S14). (C) Violin plot showing the relative gene age ranks (higher ranks equate to younger ages) of the TRGs between the autosomes and the V chromosome. Statistically significant difference in mean values of gene ages was assessed using an FDR-corrected permutation test (** = $p < 0.01$). (D) Mosaic plot showing that the species-level genes (rank 8) are responsible for the enrichment of young genes in the sex chromosome.

Figure S22. Gene ages across the *Marchantia polymorpha* genome. (A) Distribution of relative gene ages across the chromosomes of *Marchantia polymorpha*. (B) The U/V sex chromosomes (chrU and chrV; red) show non-significant differences in gene age distribution when compared to the rest of the chromosomes (green; see Table S14). (C) Violin plot showing the relative gene age ranks (higher ranks equate to younger ages) of the TRGs between the autosomes and the sex chromosomes. No statistically significant differences were found in mean values of gene ages (FDR-corrected permutation tests; NS = non-significant). (D) Mosaic plot showing non-significant differences between the sex chromosomes and the autosomes.

 *Figure S23. Gene ages across the Cryptococcus neoformans var. neoformans JEC21 genome. (A) Distribution of relative gene ages*
 *across the chromosomes of Cryptococcus neoformans. (B) The mating-type chromosome (NC_006686.1; red) shows non-significant*
 *differences in gene age distribution when compared to the rest of the chromosomes (green; see Table S14). (C) Violin plot showing the*
 *relative gene age ranks (higher ranks equate to younger ages) of the TRGs between the autosomes and the mating-type chromosome.*
 *No statistically significant difference was found in the mean values of gene ages (FDR-corrected permutation tests; NS = non-signifi-*
 *cant). (D) Mosaic plot showing no discernible pattern of gene age distribution in any of the chromosomes.*

*Figure S24. Proposed scenarios for the transition from dioicy to monoicy in Chordaria linearis and Desmarestia dudresnayi. (A) The*
 *ancestor of Chordaria linearis likely underwent an initial translocation event from the U chromosome to the V chromosome, inserting*
 *part of the U-SDR and a piece of the 3' PAR towards the 5' end of the V-SDR potentially through an ectopic recombination event. A*
 *subsequent inversion within this translocation spread the 3' PAR genes to both sides of the U-SDR insertion. Finally, a second translo-*
 *cation led to the insertion of an additional piece of the U-SDR within the 3' PAR translocation. (B) The ancestor of Desmarestia dudres-*
 *nayi underwent three translocations of U-SDR genes into the V-SDR. Additionally, a fourth translocation event happened between the*
 *U-SDR and an autosome (chr_04).*

Figure S25. Gene ages across the *D. dudresnayi* genome. (A) Distribution of relative gene ages across the chromosomes of *Desmareestia*
 *dudresnayi*. (B) The U/V-homolog in *D. dudresnayi* (chr 03; red) has a significantly higher proportion of young genes and a lower
 proportion of old genes when compared to most of the other chromosomes (green; see Table S14). (C) Mosaic plot showing that the
 species-level genes (rank 12) are responsible for the enrichment of young genes in the U/V-homolog. (D) The inter-species Ks values
 are similar in the U/V-homolog when compared to the other chromosomes (see Table S17).

*Figure S26. No detectable sex-determining region in the male genome of Fucus serratus. None of the chromosome-level scaffolds display*
 *differences in k-mer coverage between the male and the female genomes of F. serratus.*

Figure S27. Gene ages across the *F. serratus* genome. (A) Distribution of relative gene ages across the chromosomes of *Fucus serratus*.
 (B) The U/V-homolog in *F. serratus* (LG15; red) shows no significant differences in gene age distribution when compared to the rest of
 the chromosomes (green; see Table S14). (C) Mosaic plot showing no discernible pattern of gene age distribution in any of the chromo-
 somes. (D) The inter-species K_s values are similar in the U/V-homolog when compared to the other chromosomes (see Table S17).

**METHODS**

Resource availability

**Lead contact**

Further information and requests for resources and reagents should be directed to and will be fulfilled by the
lead contact, Susana M. Coelho (susana.coelho@tuebingen.mpg.de).

**Data and code availability**

- • The accession numbers and download links for all the genomic data that was generated and used in this
study are available on Table S1.
• This paper does not report original code.

**Biological material**

*Scytosiphon promiscuus*, *Dictyota dichotoma*, *Undaria pinnatifida* and *Desmarestia dudresnayi* haploid game-
topophytes were cultivated in the laboratory conditions as in ⁷⁸. We cultivated the gametophytes at 14°C with a
photoperiod of 12:12 h light:dark an irradiance of 25µmol photons.m⁻².s⁻¹. The media consisted of filtered
natural seawater (NSW), which was autoclaved and enriched with half-strength Provasoli nutrient solution
(Provasoli-enriched seawater; PES)⁷⁸. We grew the first biomass in 140mm Petri dishes and the gametophytes
were later transferred to 1L flask with gentle aeration. The gametophytes were fragmented once a month and
the media were changed every two weeks to promote biomass production. Prior to freezing, gametophytes were
treated with antibiotics for 3 days with a gentle agitation and under the same culture conditions. The first day,
gametophytes were treated with a mix Streptomycin (2g/L of PES), Penicillin G (0.5g/L of PES) and Chloram-
phenicol (0.1g/L of PES); the next day with Ampicilin (1g/L of PES) and finally the last day with Kanamycin
(1g/L of PES). Between each day of treatment and before freezing, gametophytes were rinsed with 500mL of
NSW to remove the traces of antibiotic.

Samples for fucoid algae sexual and vegetative tissue were collected in the intertidal zone during low tides in
June 2012 from Viana do Castelo (*F. vesiculosus*, *A. nodosum*) and Caminha (Rio Minho; *F. ceranoides*), north-
ern Portugal. Sexual phenotypes were verified in the field by sectioning and observing receptacles under a field
microscope. Tissue samples were flash-frozen in liquid nitrogen on the shore and transported to the laboratory
in a cryoshipper, after which they were lyophilized and stored dry at room temperature on silica crystals. See
Table S1 for list of strains used in this study.

**DNA and RNA extraction and sequencing**

Genomic DNA was isolated from algal tissue (~100mg) by grinding into fine powder under liquid nitrogen and
subsequent cell lysis in 500µL of Genomic Lysis Buffer (OMNIPREP for plant kit) for 1 hour at 60°C. The
lysate was cleaned up with 200µL of chloroform and DNA was precipitated in EtOH. The DNA pellet was
digested in CF buffer (Macherey-Nagel) for 45 min at 65°C and purified using NucleoBond AXG20 Mini col-
umns according to the user manual (Macherey-Nagel). Final high molecular weight gDNA was quantified

(Qubit), analyzed for purity (Nanodrop) and checked for size distribution (Femto Pulse System) before prepar-
ing the sequencing libraries. We sequenced the libraries using an Oxford Nanopore Technologies (ONT) Min-
ION Mk1B. We prepared the ONT libraries using an SQK-LSK110 library preparation kit for R9.4.1 flow cells
and an SQK-LSK114 library preparation kit for R10.4.1 flow cells. Two libraries were sequenced for
*Desmarestia dudresnayi* on R9.4.1 flowcells and a third library was sequenced on a R10.4.1 flowcell.

RNA was isolated from mature gametophytes of *Undaria pinnatifida* and *Scytosiphon promiscuus* following
modified procedure of Qiagen RNAeasy kit and the TruSeq RNA Library Prep Kit v2 was used to sequence the
transcriptomes in an Illumina NextSeq 2000 platform (150bp, PE reads). Extraction of total RNA from fucoid
algae (*F. vesiculosus*, *A. nodosum* and *F. ceranoides*) was performed following 65 and RNA libraries were
sequenced on Illumina HiSeq 2000 machine (100 bp, PE reads).

**Genome assembly and annotation**

High quality, chromosome level assemblies of brown algae genomes have been notoriously difficult to obtain
due to technical challenges in extracting nucleic acids. Whole-genome assemblies and annotations of *S. promiscuus*
male, *D. dichotoma* male, *D. herbacea* male and female, *E. crouanorium* male, *C. linearis*, *S. ischiensis*
and *F. serratus* male were obtained from Denoeud et al.¹⁸. We also downloaded the genome of *Ectocarpus* sp.
7¹⁹ and the male genome of *Undaria pinnatifida*²⁰ which were already assembled at a chromosome level. For
*Desmarestia dudresnayi*, we performed genome sequencing, *de novo* genome assembly and *ab initio* gene an-
notation. Base calling was done using ONT Guppy⁷⁹ with the configuration files dna_r9.4.1_450bps_sup.cfg
and dna_r10.4.1_e8.2_400bps_sup.cfg and the options --trim_adapters --trim_primers, yielding 17.4 Gbp of data
in 2,871,152 reads. We merged all the reads and analyzed them using Kraken v2.1.2⁸⁰ and the bacteria database
(downloaded 08-2022) to remove potential contaminant sequences. All data classified as bacterial reads by
Kraken were screened using blastN v2.13.0+⁸¹ (-evalue 0.001 -num_alignments 20) against the NCBI genbank
bacterial database (downloaded 11-2023). The blastN output was visualized in MEGAN v6.23.4⁸², and all the
reads that were declared as bacterial were extracted and removed from further analyses. We obtained 1,908,772
decontaminated reads with an average length of 5.1Kbp (9.8 Gbp of data, 20x coverage), which were deposited
on the NCBI Sequence Read Archive (see Table S1).

The decontaminated reads were assembled *de novo* using flye v2.9.1-b1780⁸³ with the options ‘--nano-raw -g
450m -t 28 -i 3 --scaffold’. The draft assembly consisted of 1,032 contigs with a total size of 425 Mbp, an N50
of 4.6 Mbp and an L50 of 29 contigs. We used TransposonPSI (<http://transposonpsi.sourceforge.net/>) to predict
the TEs and RepeatScout v1.0.6⁸⁴ to predict the simple repeats in the genome assembly. Both predictions were
combined to soft-mask the repetitive content in the genome assembly using bedtools maskfasta v2.27.1⁸⁵. We
mapped the RNA-seq data of *Desmarestia dudresnayi* from the PhaeoExplorer database¹⁸ to the soft-masked
genome assembly using STAR v2.7.1a⁸⁶. We used BRAKER v2.1.6 alongside the RNA-seq data⁸⁷ to predict
the protein-coding genes in the soft-masked genome assembly.

**Hi-C library preparation and sequencing for chromosome-level assemblies**

We generated Hi-C libraries for three male genomes (*Scytosiphon promiscuus*, *Desmarestia herbacea* and *Dic-*
*tyota dichotoma*) and two female genomes (*Ectocarpus* sp. 7 and *Desmarestia herbacea*). Fresh algal tissue was
cross-linked for 20 minutes at room temperature in a solution of 2% formaldehyde with filtered natural sea

water (NSW) and then transferred into a 400 mM Glycine solution with filtered NSW for five minutes to quench
the formaldehyde. The samples were then stored at -80°C until use. The Hi-C libraries were prepared as follows.
The samples were de-frosted in 1 mL of 1x *DpnII* buffer with protease inhibitors (Roche cOmplete™), trans-
ferred to Precellys VK05 lysis tubes (Bertin Technologies, Rockville, MD) and disrupted using the Precellys
apparatus with five grinding cycles of 30 seconds at 7,800 rpm followed by 20 second pauses. SDS was added
to the lysate at 0.5% final concentration and samples were incubated for 10 minutes at 62°C, followed by the
addition of Triton-X100 to a final concentration of 1% and 10 minutes of incubation at 37°C under gentle
shaking. We added 500 U of *DpnII* to 4.6 mL of the digestion mixture and incubated the samples for two hours
at 37°C under gentle shaking (180 rpm in an inclined rack to prevent sedimentation), followed by the addition
of another 500 U of *DpnII* and an overnight incubation under the same conditions. The digested samples were
centrifuged at 4°C for 20 minutes at 16,000×g. The supernatant was discarded and the pellet was incubated for
biotinylation at 37°C for an hour under a constant shaking (300 rpm) in a 500 ml biotinylation mix with a
concentration of 1x ligation buffer, 0.09 mM of dATP-dGTP-dTTP, 0.03 mM of Biotin-14-dCTP and 0.64
U/mL of Klenow fragments. After biotinylation, the samples were incubated for three hours at room temperature
in a 1.2 mL ligation reaction with a concentration of 1x ligation buffer, 100 mg/mL of BSA, 1 mM of ATP and
0.4 U/mL of T4 DNA Ligase. The samples were then incubated overnight at 65°C after adding 20µl of 0.5M
EDTA, 80µl of 10% SDS and 1.6 mg of Proteinase K. DNA was extracted with 1 volume of phenol/cholor-
form/isoamyl alcohol (24:24:1), followed by 30 seconds of vortex at top speed and a five-minute centrifugation
at top speed. We precipitated the DNA by adding 1/10 volume of 3M NaAC pH5 and two volumes of cold
EtOH 100%, followed by a 30-minute incubation at -80°C and a 20-minute centrifugation at 14,000×g and 4°C.
The DNA pellet was washed with 1mL of EtOH 70%, then dried at 37°C for 10 minutes and resuspended in
100µl 1x TE buffer with 1mg/ml of RNase. DNA was sheared to 250-500bp fragments using Covaris S220,
purified with AMPure beads (0.6X) (Beckman) and eluted in 20µl 10mM Tris pH8.0. Biotinylated but not
ligated DNA fragments were first removed by T4 DNA polymerase treatment (final concentration=300 U/pellet;
NEB), and the biotin-labeled fragments were selectively captured by Dynabeads MyOne Streptavidin C1 (Invi-
trogen). The libraries were prepared using NEB Ultra II library preparation system and sequenced on the
NextSeq2000 Illumina platform (2x150 bp) (Table S1).

We scaffolded the genomes from Denoeud et al.¹⁸ into chromosome-level assemblies using the Hi-C data. We
filtered the low-quality Hi-C reads using Trimmomatic v0.39⁸⁸ (ILLUMINACLIP:2:30:10 LEADING:25
TRAILING:25 SLIDINGWINDOW:4:15 MINLEN:75 AVGQUAL:28). We mapped the Hi-C reads against
each genome assembly using BWA-mem v0.7.17-r1188d in the Juicer v1.6 pipeline⁸⁹ to generate a contact map,
which was then fed to 3D-DNA v190716⁹⁰ to scaffold the genomes into chromosomes. The obtained scaffolds
were manually inspected against the contact maps to solve the limits of each chromosome using Juicebox
v1.11.08⁹¹. The PhaeoExplorer gene annotations¹⁸ were lifted into the new assemblies using LiftOff v1.6.1⁹²,
while the annotation of TEs was performed using RepeatModeler2⁹³. We scaffolded the genomes of *Ectocarpus*
*crouaniorum* and *Desmarestia dudresnayi* into chromosomes using a reference-guided assembly with RagTag
v2.0.1⁹⁴ against the chromosome-level assemblies of *Ectocarpus* sp. 7 and *Desmarestia herbacea*, respectively.
All genes within the SDRs in the brown algal species studied (see below) were manually curated to exclude any
TE-related genes from the annotation.

Discovery of the U/V sex determination regions

Male sex determining regions (V-SDR) in *S. promiscuus*, *U. pinnatifida*, *D. herbacea* and *D. dichotoma*, as well
as female sex determining region (U-SDR) in *D. herbacea* were analyzed following two complementary meth-
ods: (1) a kmer-based YGS approach, originally designed to detect Y-linked sequences in heterogametic sys-
tems, developed by Carvalho and Clark⁹⁵, and (2) genomic coverage analysis, designed to identify sex-linked
regions through differences in read depth between male and female individuals⁹⁶. These methods are well suited
for organisms with divergent sex chromosomes, such as brown algae, where U and V haplotypes have diverged
over extended evolutionary time.

The YGS method principle is to identify male or female sex-linked scaffolds by comparing kmer frequencies
between reference genome assembly and kmers generated from DNaseq reads of the opposite sex. Regions in
the male reference genome that contain kmers that are absent in female reads will indicate candidate male SDR
sequences, similarly, female genomic scaffolds with low coverage in male kmers will denote female SDR re-
gion. For each species, fifteen base pair kmer sequences were generated separately from male and female Illu-
mina reads (see Table S1 for data accession numbers) using Jellyfish v2.3.0 count (-m 15 -s 10G -C --quality-
start=33 --min-quality=20) and converted to fasta format with Jellyfish dump (--lower-count=5)⁹⁷. Next, non-
overlapping 500kb sliding windows (*Desmarestia*, *Dictyota* and *Undaria*) or 200kb sliding windows (*Scytosi-*
*phon*) of the reference chromosome genomes (from the sex whose SDR was to be identified) were created using
seqkit v2.3.1⁹⁸ and used as input for the YGS.pl script⁹⁵ together with the fasta kmer files produced in the
previous step. Each window was then analyzed to calculate the proportion of k-mers in the reference window
that are not present in the opposite-sex kmer database. Genomic windows with a minimum of $\geq 50\%$ of un-
matched single-copy kmers were then retained as candidate male or female SDR sequences. Because the borders
of the SDRs cannot be precisely defined at the single-nucleotide level with the available data, we focused on
genes within these regions and defined the SDR boundaries based on the flanking genes located at the transition
to pseudoautosomal regions (PARs).

Candidate SDR regions identified by YGS were further validated by analyzing sex-specific differences in read
coverage. In detail, the short Illumina reads coming from males and females of each investigated species were
trimmed with Trimmomatic⁸⁸ (see above) and mapped to the reference genome, for which the SDR was to be
studied, using HISAT2⁹⁹ (default settings). Bam files produced by HISAT2 were used as input for Mosdepth¹⁰⁰
to calculate coverage in 10kb windows along the genome sequence (-m -n -b 10000 -fast-mode -Q 30). Read
mapping depth in genomic windows was normalized by the genome-wide mean for each sex and the coverage
in genomic intervals was then compared between males and females. Because V-SDR-linked sequences are
present only in males, we expect them to have similar read coverage as autosomal regions in males, but little or
no coverage in females (and conversely for the U-SDR sequences in *Desmarestia herbacea*). The comparison
focused on regions within male reference genomes where the coverage in males fell within the range of 75% to
125% of the genome average, while the coverage in females remained below 50% of the genome average.

Both, coverage and kmer analysis, identified identical genomic regions, providing high-confidence candidate
SDRs (Table 1, Fig. S1–S5). In *Desmarestia herbacea*, where both male and female chromosome-level genome
assemblies were available, we directly compared U and V chromosomes to further confirm the SDR borders by
analyzing the collinearity of pseudoautosomal regions flanking the SDRs. The SDR scaffolds for all studied
species were further validated by PCR amplification (see **Table S1**) using 4 males and 4 females.

**Genetic mapping and search for the sex chromosome in *Fucus serratus***

Three different sets of materials were used in this study: (i) twelve male and twelve female field samples here-
after denoted the 24-individual natural population; (ii) 157 sporophyte progeny population derived from a cross
between one male sample and one female sample collected from the field and (iii) three male and three female
samples collected from the field for whole-genome sequencing. The 157-progeny population and 24-individual
natural population were genotyped by double digest RAD sequencing approach (ddRAD-seq). Briefly, individ-
ual genomic DNA was digested with the restricted enzymes PstI and HhaI to obtain fragments that were size
selected between 400 and 800 bp before sequencing on in Illumina HiSeq 2500 platform (paired-end 2 x 125
852 bp). See ¹⁰¹ for detailed protocol of the ddRAD-seq.

We performed whole-genome sequencing on Illumina HiSeq 2500 (2x 150 bp paired-end) for the three
male and three female samples. For ddRAD-seq data, raw reads were cleaned and trimmed with Trimmomatic
as above and mapped to the draft genome of *Fucus serratus* male. For the progeny population, genotypes were
called from the obtained bam files, using the Stacks pipeline (v2.5)¹⁰². The obtained vcf files were filtered with
VCFtools v0.1.16¹⁰³ and bcftools¹⁰⁴ (max missing per locus:30%, max missing per sample:40%, max mean
coverage:30, minQG:20).

The filtered vcf file of the progeny population was used to construct a genetic map with Lep-MAP3¹⁰⁵.
Briefly, ParentCall2 module was used to call parental genotypes, SeparateChromosomes2 module was used to
split the markers into linkage groups and OrderMarkers2 module was used to order the markers within each
linkage group using 30 iterations per group and finally computing genetic distances. Phased data were converted
to informative genotypes with the script map2genotypes.awk.

We used different approaches to identify the SDR in *Fucus serratus*:

*Coverage analysis.* We combined whole-genome sequence data from the three males and three females along-
side the ddRAD-seq data of the 24-individual natural population, mapping both datasets to the *F. serratus* male
genome assembly using bwa-mem¹⁰⁶. Coverage analyses have been done in several ways:

- Using SATC (sex assignment through coverage)¹⁰⁷, a method that uses sequencing depth distribution across
scaffolds to jointly identify: (i) male and female individuals, and (ii) sex-linked scaffolds. This identification is
achieved by projecting the scaffold depths into a low-dimensional space using principal component analysis
and subsequent Gaussian mixture clustering. Male and female whole genome sequences were used for this
analysis.

- Using the method SexChrCov described in ¹⁰⁸ with the 24-individual natural population.

- Using the method DifCover¹⁰⁹ which identifies regions in a reference genome for which the read coverage of
one sample is significantly different from the read coverage of another sample when aligned to a common
reference genome. The 24-individual natural population was used for this analysis.

- Using soap.coverage v2.7.9¹¹⁰ to calculate the coverage (number of times each site was sequenced divided by
the total number of sequenced sites) of each scaffold in each sample. For each scaffold, the male to female
(M:F) fold change coverage was calculated as $\log_2(\text{average male coverage}) - \log_2(\text{average female coverage})$.
The 24-individual natural population was used for this analysis.

*FST and sex-biased heterozygosity.* This approach has been previously used to find sex linked genomic regions
in several studies^{111,112}. Using the 24-individual natural population, *FST* was calculated using vcfTools¹⁰³. Sex-
biased heterozygosity was defined as the log10 of the male heterozygosity:female heterozygosity, where hetero-
zygosity is measured as the fraction of sites that are heterozygous. This ratio is expected to be zero for autosomal
scaffolds and elevated on young sex scaffolds due to excess heterozygosity in males.

*Identification of eventual female scaffolds that failed to map to the male reference genome.* VcfTools and
bedtools were used to extract female regions that did not map to the reference genome, consistently in the three
re-sequenced female samples.

All candidate contigs were tested by PCR in 4 males and 4 females.

**Synteny analyses, Ks analysis and transitions to co-sexuality**

Whole-genome synteny comparisons were performed for each pair of chromosome-level assemblies using
MCscan v1.2.14¹¹³, both between different species, between sex chromosomes in the same species and between
hermaphrodites and their closest relatives with U/V chromosomes. The putative gametologs between sex chro-
mosomes that were predicted with MCscan were reassessed using OrthoFinder v2.5.4¹¹⁴ and best reciprocal
DIAMOND v2.1.8.162¹¹⁵ hits.

We calculated the number of synonymous substitutions per synonymous site (*Ks*) for each pair of male
and female gametologs as a proxy to assess the relative time at which both genes diverged from each other. The
amino acid sequences of each pair of gametologs were aligned with MAFFT v7.520¹¹⁶ and subsequently aligned
into codons using pal2nal v14¹¹⁷. The gametolog *Ks* values were calculated using the model by Yang & Niel-
sen¹¹⁸ as implemented in KaKs_calculator v2.0¹¹⁹.

We evaluated the male or female identity of the genes in the co-sexual species whose orthologs were
found within the SDR in their closest non-co-sexual relatives. For this, we compared the results obtained with
MCscan¹¹³ against the orthogroup prediction performed with OrthoFinder¹¹⁴, with best reciprocal DIAMOND¹¹⁵
hits and by calculating gene trees for each orthogroup using an amino acid alignment with MAFFT¹¹⁶ and gene
tree reconstructions using FastTree v2.1.11¹²⁰.

**Ancestral reconstruction of the male SDR**

The brown algal phylogeny was obtained from Denoed *et al.*¹⁸. The species tree is based on 32 single-copy
nuclear genes whose protein sequences were aligned manually using AliView¹²¹, and whose best-fit substitution
models were assessed independently using the Akaike Information Criterion. The tree was generated using a
maximum likelihood approach implemented in RAxML bootstraps and the gamma model. Every node in the
phylogeny has 99 to 100% bootstrap support values. Divergence times were subsequently calculated using
MCMCtree¹²² and three calibration points. The MCMC chains were run for 1.5 million generations and the first
200,000 MCMC chains were discarded as burn-in.

We searched for ortholog genes within the V-SDR of five species (*Ectocarpus* sp. 7, *Scytosiphon*
*promiscuus*, *Undaria pinnatifida*, *Desmarestia herbacea* and *Dictyota dichotoma*) in our OrthoFinder results.
For each V-SDR gene, we coded its ortholog in the other species as “present” (1) if it is also sex-linked in the

V-SDR, whereas it was coded as “absent” (0) if the ortholog resides in the PARs, in an autosome or if there is
no detectable ortholog in that species. Once we generated this presence/absence matrix with the evolutionary
relationship of the genes within the V-SDR (Table S5), we used it as the input file for the software Count
v10.04¹²³ to estimate the ancestral content of the V-SDR throughout a phylogeny and determine the most likely
scenario of V-SDR evolution in the brown algae. We employed posterior probabilities under a phylogenetic
birth-and-death model with independent gain and loss rates across each branch in the phylogeny. We modeled
the independent gain and loss rates through 10 gamma categories and performed 1000 optimization rounds with
a convergence threshold on the likelihood > 0.1 to find the most fitting model for the data. The branch lengths
in the tree that were used for the ancestral state reconstruction were retrieved from the molecular clock analysis
performed by Heesch et al.¹⁶. We distinguished between conserved V-SDR genes that are ancestral and parallel
acquisitions of the same gene in the V-SDR by analyzing gene trees between male and female genomes, in
addition to female transcriptome assemblies of *Dictyota dichotoma* and *Undaria pinnatifida*. Sequence align-
ments were done using MAFFT¹¹⁶ with default settings and uploaded to <http://www.phylogeny.fr/> platform.
Alignments were further curated using Gblocks v0.91b¹²⁴ (Min. seq. for flank pos.: 85%, Max. contig. noncon-
served pos.: 8, Min. block length: 10). Trees were produced by PhyML v3.11¹²⁵ with default model and visual-
ized in TreeDyn v198.3¹²⁶. Approximate Likelihood-Ratio test (aLRT) was chosen as statistical test for branch
support. We inferred the function of the ancestral V-SDR genes through the annotation of genes in *Ectocarpus*
sp. 7 belonging to that orthogroup. The most likely acquisition mechanism of each SDR gene in each species
was assessed based on the position of each ortholog in the other species (pseudautosomic, autosomic or miss-
ing; Table S5).

**Genomic content across chromosomes**

We used closely-related genome assemblies available in the PhaeoExplorer database¹⁸ to assess the depletion
of orthologs in the sex chromosome. We predicted one-to-one orthologs using OrthoFinder¹¹⁴ between the fol-
lowing species pairs: *Ectocarpus* sp. 7 with *Ectocarpus siliculosus*, *Scytosiphon promiscuus* with *Chordaria*
*linearis*, *Undaria pinnatifida* with *Saccharina japonica*, *Fucus serratus* with *Fucus distichus*, *Desmarestia her-*
*bacea* with *Desmarestia dudresnayi*, and *Dictyota dichotoma* with *Halopteris paniculata* (Table S16). We cal-
culated the expected number of detectable orthologs for each chromosome and compared it against the observed
number of detected orthologs using chi-squared tests. We performed Benjamini-Hochberg corrections to the *p*-
values of the chi-squared tests to control the false discovery rate (FDR) in the analysis¹²⁷.

GenEra³⁶ was used by running DIAMOND in ultra-sensitive mode¹¹⁵ against the NCBI NR database
and all the PhaeoExplorer proteins¹⁸ to perform a phylostratigraphic analysis (e-value threshold of 10⁻⁵) and
calculate the relative ages of each gene in each genome (**Table S13**). Phylostratigraphy is a genetic statistical
method developed in order to date the putative origin of all the genes contained in the genome of a target species
by detecting homologs across species at different evolutionary distances (all the way from species within the
same genus to species from different domains of life). Finding the most distant homologs of each gene can link
them to their founder events (i.e., the first instance where a gene homolog is found in the history of that lineage),
allowing us to then determine their relative ages, coded as the taxonomic group where that gene is de-
tected^{35,36,128}. The gene age categories outside of the brown algae and *Schizocladia ischiensis* were based on the
taxonomic classification of each species within the NCBI Taxonomy database¹²⁹, while the gene ages within
the brown algae were manually assessed to reflect the evolutionary relationships obtained in the PhaeoExplorer

maximum likelihood tree¹⁸. We performed Wilcoxon rank-sum tests in R v4.3.1¹³⁰ to assess nonrandom differ-
ences in gene age distributions between pairs of chromosomes (Table S14). We performed Benjamini-Hochberg
corrections to the *p*-values of the Wilcoxon rank-sum tests to control the FDR in the analysis¹²⁷. The gene ages
responsible for these differences were found by evaluating the standardized residuals using mosaic plots (Figs.
S15-S19, S21-S24, S26, S28).

We used the inter-species *Ks* values between pairs of species as a proxy for neutral mutation rates across six of
the seven chromosome-level assemblies by using the most closely related genome assemblies available in the
PhaeoExplorer database¹⁸. We used the same set of one-to-one orthologs detected between species pairs as for
the ortholog-depletion test (Table S16). However, the evolutionary distance between *Dictyota dichotoma* and
*Halopteris paniculata* prevented us from calculating reliable inter-species *Ks* values for this species since syn-
onymous substitutions reached the point of saturation. The amino acid sequences of each pair of orthologs were
aligned with MAFFT¹¹⁶ and subsequently aligned into codons using pal2nal¹¹⁷. The inter-species *Ks* values were
calculated using the model by Yang & Nielsen¹¹⁸ as implemented in KaKs_calculator v2.0¹¹⁹. We also evaluated
the difference in inter-species *Ks* values between the autosomes and the sex chromosomes through FDR-cor-
rected Wilcoxon rank-sum tests (Table S17). We calculated the protein-coding density, the density of TEs and
the taxonomic identity of these TEs within 100 kb non-overlapping windows across each chromosome using
bedtools⁸⁵ (Table S9). The differences in protein-coding space and repeat content between the autosomes and
the sex chromosomes were also performed using FDR-corrected Wilcoxon rank-sum tests (Tables S10-S11).
The differences in repeat density, percentage of unclassified repeats and gene ages across genomic compart-
ments (SDR, PARs and autosomes) was tested using FDR-corrected permutation tests with 10,000 permuta-
tions. All the genomic features were plotted using karyoploteR v1.20.3¹³¹.

Gene expression analysis

We used kallisto v.0.44.0¹³² to calculate gene expression levels using 31-base-pair-long k-mers and 1000 boot-
straps. Transcript abundances were then summed within genes using the tximport v3.19 package¹³³ to obtain the
expression level for each gene in TPM. Differential expression analysis was done in DESeq2 v3.19 package¹³⁴

[revised manuscript text omitted]

*matics* **29**, 1035–1043 (2013).
- 1268
- 1269

A**B****C**
A**B**
■ Male SDR
■ Female SDR
■ PAR

Ks value
5
1
0.1
0

**C****D****F****E****V-SDR gene dynamics**

Conserved genes
 Expansion of SDR boundaries into the PAR
 Autosomal duplication/translocation into the SDR
 Gene birth within SDR (duplication or TRG)
 Gene losses

Num. genes in V-SDR

Gamete dimorphism

■ Anisogamy
 ■ Oogamy

A**B****C**
A**B****C****D**
A

B

Fucus serratus

Tracks

C

proto-V

V

V

V

V

V

V

U/V-homologs

U/V-homologs

new master gene?

Suppression of recombination

Expansion of the non recombining region to neighboring regions

Acquisition of genes from autosomes, enlargement of SDR with TE accumulation

SDR genes are selectively maintained if they play active roles in reproduction

Genes with no function in sex degenerate

DNA transposons and faster substitution rates lead to taxonomically restricted gene evolution

Evolution of an X/Y system with male-biased gene expression

male gene expression

female gene expression

Gene identity

- Pseudoautosomal gene
- Master sex-determining gene
- Male sex-determining gene
- Female sex-determining gene
- Autosomal gene
- Taxonomically restricted gene

Scytosiphon Male YGS analysis

Scytosiphon Male genome coverage

Undaria Male YGS analysis

Undaria Male genome coverage

Desmarestia Male YGS analysis

Desmarestia Male genome coverage

Desmarestia Female YGS analysis

Desmarestia Female genome coverage

Dictyota Male YGS analysis

Dictyota Male genome coverage

Inter-genomic comparison: *Dictyota dichotoma* vs *Schizocladia ischiensis* (1,828 gene pairs)

Ectocarpus sp. 7

1.84-3.26Mb

Compartment

- Male SDR
- PAR

Ectocarpus crouaniorum

2.34-4.07Mb

Desmarestia herbacea
Female chr3 (14.56-5.49 Mb)

Ectocarpus sp.7
chr13 (2.10-3.45 Mb)

Desmarestia herbacea
Male chr3 (5.19-12.15 Mb)

A**B**
A

Ectocarpus sp. 7

Scytosiphon promiscuus

Undaria pinnatifida

Desmarestia herbacea

Dictyota dichotoma

Tracks

Genome content

Ks values

Gene ages

Compartment

B

Desmarestia dudresnayi

Fucus serratus

Ectocarpus sp. 7

Scytosiphon promiscuus

Undaria pinnatifida

Desmarestia herbacea

Dictyota dichotoma

Ectocarpus sp. 7

Scytosiphon promiscuus

Undaria pinnatifida

Desmarestia herbacea

Dictyota dichotoma

A**B****C****D**
A**B****C****D**
A**B****C****D**
A**B****C****D**
A**B****C**
A**B****C****D**
A**B****C****D**
A**B****C****D**
A**B****C****D**
A**B****Genomic identity**

- Male SDR (V-SDR)
- Female SDR (U-SDR)
- 3' PAR
- 5' PAR
- Autosome

A**B****C****D**
Fucus YGS analysis

A**B****C****D**